CPHT-RR001.012020

# 1/8-BPS Couplings and Exceptional Automorphic Functions

Guillaume Bossard[1], Axel Kleinschmidt[2,3] and Boris Pioline[4]

[1] *Centre de Physique Théorique, CNRS, Institut Polytechnique de Paris*
*91128 Palaiseau cedex, France*

[2] *Max-Planck-Institut für Gravitationsphysik (Albert-Einstein-Institut)*
*Am Mühlenberg 1, DE-14476 Potsdam, Germany*

[3] *International Solvay Institutes*
*ULB-Campus Plaine CP231, BE-1050 Brussels, Belgium*

[4] *Laboratoire de Physique Théorique et Hautes Energies (LPTHE),*
*Sorbonne Université and CNRS UMR 7589, Campus Pierre et Marie Curie,*
*4 place Jussieu, 75252 Paris cedex 05, France*

Unlike the $\mathcal{R}^4$ and $\nabla^4\mathcal{R}^4$ couplings, whose coefficients are Langlands–Eisenstein series of the U-duality group, the coefficient $\mathcal{E}_{(0,1)}^{(d)}$ of the $\nabla^6\mathcal{R}^4$ interaction in the low-energy effective action of type II strings compactified on a torus $T^d$ belongs to a more general class of automorphic functions, which satisfy Poisson rather than Laplace-type equations. In earlier work [1], it was proposed that the exact coefficient is given by a two-loop integral in exceptional field theory, with the full spectrum of mutually 1/2-BPS states running in the loops, up to the addition of a particular Langlands–Eisenstein series. Here we compute the weak coupling and large radius expansions of these automorphic functions for any $d$. We find perfect agreement with perturbative string theory up to genus three, along with non-perturbative corrections which have the expected form for 1/8-BPS instantons and bound states of 1/2-BPS instantons and anti-instantons. The additional Langlands–Eisenstein series arises from a subtle cancellation between the two-loop amplitude with 1/4-BPS states running in the loops, and the three-loop amplitude with mutually 1/2-BPS states in the loops. For $d = 4$, the result is shown to coincide with an alternative proposal [2] in terms of a covariantised genus-two string amplitude, due to interesting identities between the Kawazumi–Zhang invariant of genus-two curves and its tropical limit, and between double lattice sums for the particle and string multiplets, which may be of independent mathematical interest.

# 1 Introduction and summary

The scattering of massless excitations of type II superstrings compactified on a torus $T^d$ is described at low energy by maximally supersymmetric supergravity in dimension $D = 10 - d$. The effective action consists of the classical supersymmetric two-derivative action plus an infinite tower of cut-off dependent higher derivative interactions, which conspire to ensure ultraviolet finiteness. The coefficients of these interactions are strongly constrained by non-perturbative dualities [3,4], which tie together perturbative and non-perturbative (instanton and anti-instanton) contributions. Combined with supersymmetry, duality invariance sometimes allows determining the exact coefficient of these interactions in terms of automorphic functions under the U-duality group. Such results offer an invaluable window into the non-perturbative regime of string theory, including the full spectrum of D-branes, membranes or supersymmetric black holes, despite the absence of a first principle, non-perturbative formulation of string theory or of its eleven-dimensional parent.

This programme has been pursued most extensively for four-graviton scattering, generalising the celebrated lowest order result [5] in ten-dimensional type IIB string theory to lower dimension $D = 10 - d$ and to higher orders in the derivative expansion. Schematically, these effective interactions take the form $\mathcal{E}^{(d)}_{(p,q)}(\phi) \, \nabla^{4p+6q} \mathcal{R}^4$, where $p$ and $q$ denote powers of the Mandelstam invariants $\sigma_2 = s^2 + t^2 + u^2$ and $\sigma_3 = s^3 + t^3 + u^3$ of the external momenta in the corresponding amplitude and $\mathcal{R}^4$ denotes the fourth order polynomial $t_8 t_8 \mathcal{R}^4$ in the Riemann tensor that generalises the square of the Bel–Robinson tensor in $D$ dimensions [6,7]. The coordinates $\phi$ on the classical moduli space $\mathcal{M}_D = E_{d+1}/K_{d+1}$ include the constant metric and gauge potentials on $T^d$, as well as the string coupling constant $g_D$. At weak coupling $g_D \to 0$, $\mathcal{E}^{(d)}_{(p,q)}$ admits an asymptotic expansion of the form

$$\mathcal{E}^{(d)}_{(p,q)} = \mathcal{E}^{(d),\text{n.an.}}_{(p,q)} + g_D^{\frac{2d+8p+12q-4}{d-8}} \sum_{h=0}^{\infty} g_D^{-2+2h} \, \mathcal{E}^{(d,h)}_{(p,q)} + \mathcal{O}(e^{-2\pi/g_D}) + \mathcal{O}(e^{-2\pi/g_D^2}) \qquad (1.1)$$

where $\mathcal{E}^{(d,h)}_{(p,q)}$ arises at genus $h$ in the perturbative string expansion, while the last two terms originate from Euclidean D-branes and NS-branes wrapped on $T^d$. The term $\mathcal{E}^{(d),\text{n.an.}}_{(p,q)}$ is a non-

analytic function of the string coupling $g_D$, which arises in the process of translating from string frame to Einstein frame [8]. U-duality requires that $\mathcal{E}^{(d)}_{(p,q)}$ should be automorphic, *i.e.* invariant under the left-action of an arithmetic subgroup $E_{d+1}(\mathbb{Z}) \subset E_{d+1}$ on $\mathcal{M}_D$, while supersymmetry imposes further differential constraints when $4p + 6q < 8$ (the so-called F-terms) [9–15].

At leading and subleading order, the coefficients $\mathcal{E}^{(d)}_{(0,0)}$, $\mathcal{E}^{(d)}_{(1,0)}$ are known exactly in all dimensions $D \geq 3$, in terms of a special type of automorphic functions known as Langlands–Eisenstein series [16–23]:

$$\mathcal{E}^{(d)}_{(0,0)} \underset{d \geq 1}{=} 4\pi \xi(d-2)\, E^{E_{d+1}}_{\frac{d-2}{2}\Lambda_{d+1}} \,, \qquad \mathcal{E}^{(d)}_{(1,0)} \underset{\substack{d \geq 2 \\ d \neq 4}}{=} 8\pi \xi(d-3)\, \xi(d-4)\, E^{E_{d+1}}_{\frac{d-3}{2}\Lambda_d} \,, \tag{1.2}$$

where $E^G_{s\Lambda_k}$ is the (regularised) Langlands–Eisenstein series associated to the maximal parabolic subgroup $P_k \subset G$, in Langlands' normalisation (in particular, $E^G_{0\Lambda_k} = 1$), and we denote by $\xi(s) = \pi^{-s/2}\Gamma(s/2)\zeta(s)$ the completed Riemann zeta function. Here and elsewhere in this work, we denote by $\Lambda_k$ the $k^{\text{th}}$ fundamental weight of the algebra $\mathfrak{g}$ according to Bourbaki's labelling[1], and $P_k$ the associated maximal parabolic subgroup. We recall (see e.g. [24]) that maximal parabolic Langlands–Eisenstein series are defined for $\text{Re}(s)$ large enough as the Poincaré sum of the canonical multiplicative character $y_k$ of $P_k$,[2]

$$E^G_{s\Lambda_k} \equiv \sum_{\gamma \in P_k(\mathbb{Z}) \backslash G(\mathbb{Z})} y_k^{-2s}\Big|_\gamma = \frac{1}{2\zeta(2s)} {\sum_{\substack{\mathcal{Q} \in M^G_{\Lambda_k} \\ \mathcal{Q} \times \mathcal{Q} = 0}}}' G(\mathcal{Q},\mathcal{Q})^{-s} \,, \tag{1.3}$$

and admit a meromorphic continuation to the full complex $s$-plane. As exhibited in (1.3), they can be written as a constrained sum over the lattice $M^G_{\Lambda_k}$ transforming in the irreducible representation $R(\Lambda_k)$ of highest weight $\Lambda_k$,[3] of the $K(G)$ invariant bilinear form $G(\mathcal{Q},\mathcal{Q})$ raised to the power minus $s$. For $M^{E_{d+1}}_{\Lambda_{d+1}}$ and $M^{E_{d+1}}_{\Lambda_1}$, the constrained lattice sum can be interpreted as a sum over 1/2-BPS states in string theory, that correspond respectively to particles and strings in $\mathbb{R}^{1,9-d}$ with BPS mass $M_{\text{BPS}}(\mathcal{Q}) = G(\mathcal{Q},\mathcal{Q})^{\frac{1}{2}}$ [26], see Table 1.

The coefficients (1.2) satisfy tensorial homogeneous differential equations on $\mathcal{M}_D$, reflecting the fact that they are only sensitive to 1/2- and 1/4-BPS instantons, respectively [12,13]. This implies that they are related to unipotent automorphic representations attached to the minimal and next-to-minimal nilpotent orbit, respectively. Supersymmetry Ward identities and U-duality determine uniquely the function $\mathcal{E}^{(d)}_{(0,0)}$ in (1.2) for $d \geq 3$ and $\mathcal{E}^{(d)}_{(1,0)}$ for $d \geq 5$, up to an overall

---

[1] Note that in formulae which are valid for all $d$ such as (1.2), we use the labelling associated to $E_{d+1}$ which differs from the standard labelling for $d \leq 4$ — for example the $E_5$ labelling is $\left[\begin{smallmatrix} 1 & 3 & 4 & \frac{2}{5} \end{smallmatrix}\right]$ whereas the $D_5$ labelling is $\left[\begin{smallmatrix} 1 & 2 & 3 & \frac{4}{5} \end{smallmatrix}\right]$.

[2] Here and elsewhere in the paper we denote by $\sum_\gamma f(x)|_\gamma$ the Poincaré sum over coset elements $\gamma$ acting on the seed function $f$ by $f(x)|_\gamma = f(\gamma \cdot x)$, where the action of $\gamma$ on $x$ is defined by the right action on the group element $g(x)$ as $g(x)\gamma$. A more precise definition of the $y_k$ will be given in Section 2.1.

[3] The constraint can be written in general as [25, (6.17)]

$$\mathcal{Q}_i \times \mathcal{Q}_j = \kappa_{\alpha\beta} T^\alpha \mathcal{Q}_i \otimes T^\beta \mathcal{Q}_j + (1 - (\Lambda_k, \Lambda_k))\mathcal{Q}_i \otimes \mathcal{Q}_j - \mathcal{Q}_j \otimes \mathcal{Q}_i = 0 \,,$$

with $\kappa_{\alpha\beta}$ the Killing Cartan form and $T^\alpha$ the generators of the algebra $\mathfrak{g}$. The constraint selects those charges $\mathcal{Q} \in R(\Lambda_k)$ whose symmetric square lies in $R(2\Lambda_k)$. In practice, the projection of this constraint on the largest irreducible submodule is usually sufficient to enforce $\mathcal{Q} \times \mathcal{Q} = 0$.

| $D$ | $d$ | $G_D = E_{d+1}$ | $M^{E_{d+1}}_{\Lambda_{d+1}}$ | $M^{E_{d+1}}_{\Lambda_1}$ | $M^{E_{d+1}}_{\Lambda_6}$ |
|---|---|---|---|---|---|
| $10B$ | $0$ | $SL(2)$ | $\emptyset$ | $\mathbf{2}$ | $\emptyset$ |
| $9$ | $1$ | $GL(2)$ | $\mathbf{2}^{(-3)} \oplus \mathbf{1}^{(4)}$ | $\mathbf{2}^{(1)}$ | $\emptyset$ |
| $8$ | $2$ | $SL(3) \times SL(2)$ | $(\overline{\mathbf{3}}, \mathbf{2})$ | $(\mathbf{3}, \mathbf{1})$ | $\emptyset$ |
| $7$ | $3$ | $SL(5)$ | $\overline{\mathbf{10}}$ | $\mathbf{5}$ | $\emptyset$ |
| $6$ | $4$ | $Spin(5,5)$ | $\mathbf{16}$ | $\mathbf{10}$ | $\mathbf{1}$ |
| $5$ | $5$ | $E_{6(6)}$ | $\overline{\mathbf{27}}$ | $\mathbf{27}$ | $\overline{\mathbf{27}}$ |
| $4$ | $6$ | $E_{7(7)}$ | $\mathbf{56}$ | $\mathbf{133}$ | $\mathbf{1539}$ |
| $3$ | $7$ | $E_{8(8)}$ | $\mathbf{248}$ | $\mathbf{3875}$ | $\mathbf{2450240}$ |

Table 1: U-duality group, particle multiplet ($\Lambda_{d+1}$) and string multiplet ($\Lambda_1$); the constraints on the particle and string multiplets are in $\Lambda_1$ and $\Lambda_6$, respectively.

coefficient. Using functional relations for Langlands–Eisenstein series, they can then be written alternatively as

$$\mathcal{E}^{(d)}_{(0,0)} \underset{d \geq 3}{=} 2\zeta(3) E^{E_{d+1}}_{\frac{3}{2}\Lambda_1} , \qquad \mathcal{E}^{(d)}_{(1,0)} \underset{d \geq 5}{=} \zeta(5) E^{E_{d+1}}_{\frac{5}{2}\Lambda_1} . \qquad (1.4)$$

Remarkably, for $d = 1$ in the small volume limit (or equivalently, for type IIB strings in $D = 10$) these couplings can also be computed in eleven-dimensional supergravity compactified on $T^2$ at one-loop and two-loop, respectively [27, 28]. For $d \geq 2$ or for $d = 1$ at finite volume, membrane and five-brane degrees of freedom become important, and can be incorporated using the framework of exceptional field theory in dimension $D = 10 - d$ [29]. In this formalism, all 1/2-BPS charges in the 'particle multiplet' lattice $M^{E_{d+1}}_{\Lambda_{d+1}}$ are allowed to propagate in the loops. At one-loop, this leads to a 'constrained lattice sum' which reproduces the Langlands–Eisenstein series $\mathcal{E}^{(d)}_{(0,0)}$ in (1.2) [1], while at two-loops it leads to a 'double constrained lattice sum' which again reproduces the Langlands–Eisenstein series $\mathcal{E}^{(d)}_{(1,0)}$ in (1.2) [1]. Note that the one-loop amplitude in exceptional field theory also produces a divergent contribution to the $\nabla^4\mathcal{R}^4$ coupling, but this is cancelled by a one-loop amplitude with 1/4-BPS states running in the loop [30].

At next-to-next-to-leading order, the coefficient $\mathcal{E}^{(d)}_{(0,1)}$ of the $\nabla^6\mathcal{R}^4$ coupling is less well understood. It satisfies a set of inhomogeneous differential equations, the simplest one being the Poisson equation [31, 22, 2]

$$\left(\Delta_{E_{d+1}} - \frac{6(4-d)(d+4)}{8-d}\right) \mathcal{E}^{(d)}_{(0,1)} = -\left(\mathcal{E}^{(d)}_{(0,0)}\right)^2 + 40\,\zeta(3)\,\delta_{d,4} + \frac{55}{3}\,\mathcal{E}^{(5)}_{(0,0)}\,\delta_{d,5} + \frac{85}{2\pi}\,\mathcal{E}^{(6)}_{(1,0)}\,\delta_{d,6} , \quad (1.5)$$

where $\Delta_{E_{d+1}}$ is the Laplace–Beltrami operator for the $E_{d+1}$-invariant metric on $\mathcal{M}_D$, and the right-hand side involves the *square* of the $\mathcal{R}^4$ coupling, plus anomalous terms when ultraviolet logarithmic divergences appear in supergravity. As a result, the weak coupling expansion includes perturbative contributions up to genus three, 1/8-BPS instantons as well as bound states of 1/2-BPS instantons and anti-instantons. The exact $\nabla^6\mathcal{R}^4$ coupling in ten-dimensional type IIB string theory was obtained from a two-loop computation in eleven-dimensional supergravity compactified on a torus $T^2$ in [31] and further analysed using the differential equation (1.5) in [32]. The same coupling in $D = 9$ and $D = 8$ was obtained using similar methods in [33, 22, 34], albeit in a rather implicit way.

An explicit proposal for the $\nabla^6 \mathcal{R}^4$ coupling in $D = 6$ was given in [2], by upgrading the genus-two string theory contribution [35, 36] to an invariant function under the U-duality group $Spin(5, 5, \mathbb{Z})$,[4]

$$\mathcal{E}^{(4)}_{(0,1)} = 8\pi \, \text{R.N.} \int_{\mathcal{F}_2} \frac{\mathrm{d}^6 \Omega}{|\Omega_2|^3} \, \Gamma_{5,5,2}(\Omega, \phi) \, \varphi_{\text{KZ}}(\Omega) + \frac{16\zeta(8)}{189} \widehat{E}^{D_5}_{4\Lambda_5} \, . \tag{1.6}$$

Here $\mathcal{F}_2$ is the standard fundamental domain of the action of the modular group $Sp(4, \mathbb{Z})$ on the Siegel upper-half plane and $\Gamma_{d,d,h}(\Omega, \phi)$ is the genus-$h$ Siegel–Narain theta series for the even self-dual lattice $II_{d,d}$ of signature $(d, d)$ given by

$$\Gamma_{d,d,h}(\Omega) \equiv \sum_{q_i \in II_{d,d}} |\Omega_2|^{\frac{d}{2}} e^{-\pi \Omega_2^{ij} G(q_i, q_j) - \pi i \Omega_1^{ij}(q_i, q_j)} \, , \tag{1.7}$$

where $i$ runs from 1 to $h$, $\Omega^{ij} = \Omega_1^{ij} + \mathrm{i}\Omega_2^{ij}$ is a symmetric $h \times h$ matrix with positive imaginary part. Furthermore, $G$ is the symmetric $SO(d, d)$ matrix parametrising the moduli space $SO(d, d)/(SO(d) \times SO(d))$, $\varphi_{\text{KZ}}(\Omega)$ is the Kawazumi–Zhang invariant [37, 38] for the genus-two curve with period matrix $\Omega$ and R.N. is a particular regularisation prescription for genus-two modular integrals introduced in [39, 40]. The ansatz (1.6) automatically satisfies the differential constraint (1.5) and reproduces the known perturbative contributions up to genus three [2]. Moreover, its decompactification predicts the full $SL(5, \mathbb{Z})$-invariant $\nabla^6 \mathcal{R}^4$ coupling in $D = 7$,

$$\mathcal{E}^{(3)}_{(0,1)} = \frac{4\pi}{3} \int_{S_+} \frac{\mathrm{d}^3 \Omega_2}{|\Omega_2|^{\frac{1}{2}}} \sum_{M_i \in M_{\Lambda_1}^{A_4}}' e^{-\pi \Omega_2^{ij} G(M_i, M_j)} \varphi_{\text{KZ}}^{\text{tr}}(\Omega_2) + \frac{5\pi\zeta(7)}{189} E^{SL(5)}_{\frac{7}{2}\Lambda_3} \tag{1.8}$$

where $G$ is the $5 \times 5$ unit-determinant positive definite symmetric matrix parametrising the moduli space $SL(5)/SO(5)$ in $D = 7$, the sum runs over pairs of non-zero vectors $(M_1, M_2)$ in the lattice $M_{\Lambda_1}^{A_4} \cong \mathbb{Z}^5$, and the integral runs over the 'positive Schwinger space'

$$S_+ = \left\{ \Omega_2 = \begin{pmatrix} L_1 + L_3 & L_3 \\ L_3 & L_2 + L_3 \end{pmatrix} \,\Bigg|\, L_1, L_2, L_3 \in \mathbb{R}_+ \right\} \, . \tag{1.9}$$

Finally, $\varphi_{\text{KZ}}^{\text{tr}}(\Omega_2)$ is the supergravity (a.k.a tropical [41]) limit of the Kawazumi–Zhang invariant, as computed in [36, §3.2]

$$\varphi_{\text{KZ}}^{\text{tr}}(\Omega_2) = \lim_{\lambda \to +\mathrm{i}\infty} \lambda^{-1} \varphi_{\text{KZ}}(\Omega_1 + \mathrm{i}\lambda \Omega_2) = \frac{\pi}{6} \left( L_1 + L_2 + L_3 - \frac{5L_1 L_2 L_3}{\det \Omega_2} \right) \tag{1.10}$$

The result (1.8) has a structure similar to the two-loop supergravity amplitude studied in [28, 31], but the summation variables $M_i$ transform as doublets of vectors of $SL(5)$, corresponding to the multiplet of string charges in $D = 7$, while the multiplet of particle charges in $D = 7$ as a $\overline{\mathbf{10}}$ of $SL(5)$.

---

[4]We define the regularised Eisenstein series $\widehat{E}^G_{s\Lambda_k}$ as the $\mathcal{O}(\epsilon^0)$ term in the Laurent expansion of $E^G_{(s+\epsilon)\Lambda_k}$ at $\epsilon = 0$, whenever $E^G_{(s+\epsilon)\Lambda_k}$ has a pole at that point.

Coming back to the exceptional field theory approach [1], the one-loop contribution to the $\nabla^6 \mathcal{R}^4$ coupling in exceptional field theory gives[5]

$$\mathcal{F}^{(d)}_{(0,1)} = \frac{8\pi^4}{567} \xi(d+4) \, E^{E_{d+1}}_{\frac{d+4}{2}\Lambda_{d+1}} \,. \tag{1.11}$$

while the two-loop contribution is

$$\mathcal{E}^{(d),\text{ExFT}}_{(0,1)} = \frac{4\pi}{3} \int_{S_+} \frac{\mathrm{d}^3\Omega_2}{|\Omega_2|^{\frac{6-d}{2}}} \sum_{\substack{\Gamma_1,\Gamma_2 \in M^{E_{d+1}}_{\Lambda_{d+1}} \\ \Gamma_i \times \Gamma_j = 0}}' e^{-\pi\Omega_2^{ij} G(\Gamma_i,\Gamma_j)} \varphi^{\text{tr}}_{\text{KZ}}(\Omega_2) \,, \tag{1.12}$$

where $\varphi^{\text{tr}}_{\text{KZ}}(\Omega_2)$ now arises from the Symanzik polynomial of the two-loop supergravity amplitude [42], as shown in [28]. $G$ is the symmetric bilinear form on the representation of highest weight $\Lambda_{d+1}$, depending on the moduli in $\mathcal{M}_D$. The sum runs over pairs $\Gamma_1, \Gamma_2$ of non-vanishing vectors in the particle multiplet lattice $M^{E_{d+1}}_{\Lambda_{d+1}}$, corresponding to the charges running in the two loops, subject to the 1/2-BPS conditions $\Gamma_i \times \Gamma_j = 0$ for $i, j = 1, 2$, where $\Gamma_i \times \Gamma_j$ denotes the projection of the tensor product on the representation of highest weight $\Lambda_1$, see footnote 3.

These two functions are associated to two distinct $\nabla^6 \mathcal{R}^4$-type supersymmetry invariants for $1 \leq d \leq 6$ (and $d = 0$ type IIA) [14]. In particular it was shown in [1] that (1.11) satisfies not only the differential equation (1.5) for all $d \leq 6$, but also the more constraining tensorial equation (2.10) for $d = 4, 5, 6$, using differential properties of $\varphi^{\text{tr}}_{\text{KZ}}$ and the lattice sum. This led to the proposal that the total non-perturbative $\nabla^6 \mathcal{R}^4$ coupling should be given by the sum of these two contributions [1]

$$\mathcal{E}^{(d)}_{(0,1)} = \mathcal{E}^{(d),\text{ExFT}}_{(0,1)} + \mathcal{F}^{(d)}_{(0,1)} \,. \tag{1.13}$$

One main aim of this work will be to check that this proposal does indeed produce the correct weak coupling and large radius expansions, and that it agrees with the proposal (1.6) and (1.8) in $D = 6$ and $D = 7$. Before addressing these expansions, a major task will be to provide a proper definition of the integral (1.12), which is otherwise divergent.

Specifically, in this work we shall

1. give a mathematically precise definition of the formal integral (1.12) via dimensional regularisation;

2. demonstrate that (1.13) (suitably renormalised, cf. point 6 below) coincides with (1.6) for $D = 6$, thanks to remarkable properties of double theta series associated to the particle and string multiplet (Section 2) and of the Kawazumi–Zhang invariant and its tropical limit (Section A);

3. generalise the string multiplet proposal (1.6) to other dimensions $D \geq 3$ and show that it formally[6] agrees with (1.12);

---

[5] For $d = 0$ and $d = 7$, there is a single $\nabla^6 \mathcal{R}^4$ invariant and $\mathcal{F}^{(d)}_{(0,1)}$ vanishes. For $d = 1, 2$, $\mathcal{F}^{(d)}_{(0,1)}$ are given in (F.4), (F.8), respectively. For $d = 3, 4$, $\mathcal{F}^{(d)}_{(0,1)}$ coincides with the second term in (1.8) and (1.6), respectively. The function (1.11) is divergent at the given value of the parameter [1] and we consider instead its regularised version defined in (1.30).

[6] To keep the length of this work within reasonable bounds, we refrain from describing the regularisation of the string multiplet formula in arbitrary dimensions.

4. extract the weak coupling expansion of (1.13) for $D \geq 4$ (Section 3.2) and show agreement with the perturbative corrections in string theory (Section 5.2);

5. extract the large radius limit of (1.13), reproducing the corresponding term in dimension $D + 1$ along with the expected threshold contributions (Sections 5.3, and 5.4);

6. obtain the complete Fourier expansions relative to the weak coupling and large radius limit for $D \geq 5$ (and part of it for $D \geq 3$), including instanton-anti-instanton contributions and the 1/8-BPS instanton measure for generic Fourier coefficients in $D = 4$ and $D = 3$ (Sections 3.3 and E.2);

7. analyse the two-loop amplitude with 1/4-BPS states running in one of the two loops, and show that it cancels the divergence of the two-loop amplitude in exceptional field theory, such that the total amplitude including both 1/2-BPS and 1/4-BPS states up to three loops is finite and gives the exact string theory coupling (Section 5.1)

The upshot of this analysis is that the appropriately renormalised form of (1.13) reproduces the expected perturbative amplitude in string theory, up to non-perturbative corrections that have yet not been computed from first principles but take the expected form of D-instanton corrections. Using similar methods, one could in principle also extract the constant terms with respect to the other maximal parabolic subgroups, e.g. the one relevant to the limit where the M-theory torus $T^{d+1}$ decompactifies keeping its shape fixed, and hence characterise the behavior at all cusps. Assuming that these constant terms also agree with predictions from M-theory, one may then apply the the conjecture that the relevant U-duality groups do not admit cuspidal automorphic representations attached to suitably small nilpotent orbits [43, 44] to conclude that (1.13), suitably renormalised, is indeed the full exact coupling in any dimension $D \geq 4$.

In the remainder of this introduction, we summarise our main results in view of the points above, leaving details of the derivation to the body of the paper.

**Double lattice sums and regularised integrals**

As stressed before, the integral (1.12) is divergent and requires regularisation. In analogy with dimensional regularisation in QFT, it is natural to replace $d \to d + 2\epsilon$ in the exponent of $|\Omega_2|$, and define the integral as the value at $\epsilon = 0$ after analytic continuation from the region $\mathrm{Re}(\epsilon) \gg 0$ where the integral converges. However, we expect the analytic continuation to have a pole at $\epsilon = 0$ when $d = 4, 5, 6$, which thus needs to be subtracted appropriately. In addition, we expect that the exact $\nabla^6 \mathcal{R}^4$ coupling also includes the three-loop contribution in exceptional field theory as well as contributions from 1/4-BPS states that play the role of counterterms in exceptional field theory [30]. As we show in Section 5.1, the one-loop contribution to $\nabla^6 \mathcal{R}^4$ in exceptional field theory is in fact cancelled by a one-loop diagram with 1/4-BPS states running in the loop, just as in the case of $\nabla^4 \mathcal{R}^4$, but the same contribution $\mathcal{F}_{(0,1)}^{(d)}$ reappears at three loops [30], as we shall review later.

For the purpose of discussing this regularisation, it will be useful to decompose the period matrix of the two-loop graph as in [28, 31, 45],

$$\Omega_2 = \begin{pmatrix} L_1 + L_3 & L_3 \\ L_3 & L_2 + L_3 \end{pmatrix} = \frac{1}{\tau_2 V} \begin{pmatrix} 1 & \tau_1 \\ \tau_1 & |\tau|^2 \end{pmatrix} , \tag{1.14}$$

such that $\tau = \tau_1 + i\tau_2$ runs over six copies of the standard fundamental domain $\mathcal{F} = \{\tau \in \mathcal{H}_1, |\tau| > 1, 0 < \tau_1 < \frac{1}{2}\}$ for the action of $PGL(2, \mathbb{Z})$ on the upper half plane $\mathcal{H}_1$, and set $\varphi_{\mathrm{KZ}}^{\mathrm{tr}}(\Omega_2) = \frac{\pi}{6}|\Omega_2|^{1/2}A(\tau)$ where $A(\tau)$ is the modular function defined in the fundamendal domain $\mathcal{F}$ by

$$A(\tau) = \frac{|\tau|^2 - \tau_1 + 1}{\tau_2} + \frac{5\tau_1(\tau_1 - 1)(|\tau|^2 - \tau_1)}{\tau_2^3} \ , \tag{1.15}$$

and elsewhere in $\mathcal{H}_1$ by enforcing $PGL(2, \mathbb{Z})$ invariance. This function belongs to the class of local modular functions, which appear at all orders in the derivative expansion of the two-loop supergravity amplitudes [45] (see [46] for the relation to genus-two string integrands). Rewriting the measure $\mathrm{d}^3\Omega_2/|\Omega_2|^3 = 2V^2\mathrm{d}V\mathrm{d}\tau_1\mathrm{d}\tau_2/\tau_2^2$, the integral over $V \in \mathbb{R}^+$ can be performed easily, leading to a modular integral over $\mathcal{F}$,

$$\mathcal{E}_{(0,1)}^{(d),\mathrm{ExFT}} = \frac{8\pi^2}{3}\frac{\Gamma(d-2)}{\pi^{d-2}}\int_{\mathcal{F}}\frac{\mathrm{d}\tau_1\mathrm{d}\tau_2}{\tau_2^2}A(\tau)\sum_{\substack{\Gamma_1,\Gamma_2\in M_{\Lambda_{d+1}}^{E_{d+1}}\\ \Gamma_i\times\Gamma_j=0}}'\left[\frac{\tau_2}{G(\Gamma_1 + \tau\Gamma_2, \Gamma_1 + \bar{\tau}\Gamma_2)}\right]^{d-2} \tag{1.16}$$

The divergence of the integral (1.12) at $V \to \infty$ is reflected in the non-convergence of the double lattice sum in (1.16). We shall regularise the latter by dimensional regularisation, i.e. by replacing $d \to d + 2\epsilon$ in the exponent of $|\Omega_2|$ appearing in the denominator of (1.12), or equivalently in the exponent of the summand in (1.16). It is indeed apparent in (1.16) that the sum will be absolutely convergent for $\mathrm{Re}\,(\epsilon)$ large enough. In order to regulate divergences due to collinear charges, i.e. pairs of charges such that $\Gamma_i \wedge \Gamma_j = 0$, we further introduce a cut-off[7] $\tau_2 < L$ on the domain $\mathcal{F}$ in the coordinates (1.14), and consider the integral

$$\mathcal{I}_d(\phi, \epsilon, L) = 8\pi\int_{\mathbb{R}^+\times\mathcal{F}(L)}\frac{\mathrm{d}^3\Omega_2}{|\Omega_2|^{\frac{6-d-2\epsilon}{2}}}\varphi_{\mathrm{KZ}}^{\mathrm{tr}}(\Omega_2)\,\theta_{\Lambda_{d+1}}^{E_{d+1}}(\phi, \Omega_2)\,, \tag{1.17}$$

where $\phi$ parametrises the moduli space $E_{d+1}/K_{d+1}$ and $\mathcal{F}(L) = \mathcal{F} \cap \{\tau_2 < L\}$. Here, $\theta_{\Lambda_k}^{E_{d+1}}$ for $k = 1, \ldots d+1$ denotes the 'double theta series' for the lattice $M_{\Lambda_k}^{E_{d+1}}$ transforming in the representation with highest weight $\Lambda_k$,

$$\theta_{\Lambda_k}^{E_{d+1}}(\phi, \Omega_2) = \sum_{\substack{\mathcal{Q}_i\in M_{\Lambda_k}^{E_{d+1}}\\ \mathcal{Q}_i\times\mathcal{Q}_j=0}}' e^{-\pi\Omega_2^{ij}G(\mathcal{Q}_i, \mathcal{Q}_j)}\,. \tag{1.18}$$

The integral (1.17) is absolutely convergent for $\mathrm{Re}\,(\epsilon)$ sufficiently large. We shall argue that it admits an analytic continuation to a meromorphic function of $\epsilon \in \mathbb{C}$, by relying on similar analytic properties of the Langlands–Eisenstein series which arise in its constant terms and of the functions appearing in its Fourier coefficients. The renormalised value will be defined as the value at $\epsilon = 0$, after subtracting a specific Eisenstein series canceling the poles that occur when $d = 4, 5, 6$ [47].

These poles can be interpreted physically as ultraviolet divergences of the two-loop amplitude in exceptional field theory; they should cancel in the full theory in which all states are allowed

---

[7]The cut-off $L$ plays the same rôle as the infrared cut-off $\mu \sim 1/\sqrt{L}$ introduced in [1, 30].

to propagate in the loops. Indeed, we shall show that the sum of the contributions of 1/2-BPS states (given by the above integral (1.16)) and 1/4-BPS states (which we compute separately in Section 5.1) gives a finite answer. It will also become clear that the 1/4-BPS states' contribution is needed to restore supersymmetry Ward identities for $d = 5$. The remaining $L$-dependent terms corresponds to infrared effects from the exchange of massless particles, which must cancel against non-local terms in the 1PI effective action. The apparent ambiguity in the regularisation drops out in the complete four-graviton amplitude. Since we shall not compute the full non-local amplitude, we shall not keep track of the cutoff-dependent terms in the effective coupling $\mathcal{E}^{(d)}_{(0,1)}$. It would be interesting to fix these finite terms by analysing the full genus-two string amplitude.

Performing the integral (1.17) over $V$ leads to the modular integral

$$\mathcal{I}_d(\phi, \epsilon, L) = \frac{8\pi^2}{3} \int_{\mathcal{F}(L)} \frac{\mathrm{d}\tau_1 \mathrm{d}\tau_2}{\tau_2^2} A(\tau) \, \Xi^{E_{d+1}}_{\Lambda_{d+1}}(\tau, \phi, d-2+2\epsilon) \qquad (1.19)$$

where the 'double Epstein series' $\Xi^{E_{d+1}}_{\Lambda_k}$ is defined for any $k = 1, \ldots, d+1$ and $\mathrm{Re}\,(r)$ sufficiently large[8] by

$$\Xi^{E_{d+1}}_{\Lambda_k}(\phi, \tau, r) = \frac{\Gamma(r)}{\pi^r} \sideset{}{'}\sum_{\substack{\mathcal{Q}_i \in M^{E_{d+1}}_{\Lambda_k} \\ \mathcal{Q}_i \times \mathcal{Q}_j = 0}} \left[ \frac{\tau_2}{G(\mathcal{Q}_1 + \tau\mathcal{Q}_2, \mathcal{Q}_1 + \bar{\tau}\mathcal{Q}_2)} \right]^r . \qquad (1.20)$$

Although we only analyse in detail the case $k = d+1$ in this paper, it should be possible to use similar methods to show that $\Xi^{E_{d+1}}_{\Lambda_k}(\phi, \tau, r)$ can generally be analytically continued to a meromorphic function of $r \in \mathbb{C}$ for any fundamental weight $\Lambda_k$. We will use these Epstein series outside the domain of convergence, with the understanding that they are defined by analytic continuation.

### Equivalence of particle and string multiplet formulae

The proof of the equivalence of the exceptional field theory computation (1.13) and the covariantised string theory answer (1.6) rests on two main claims. The first is a remarkable property of the Kawazumi–Zhang invariant $\varphi_{\mathrm{KZ}}$, namely that it coincides with the Poincaré series seeded by its tropical limit

$$\varphi_{\mathrm{KZ}}(\Omega) = \lim_{\epsilon \to 0} \left[ \sum_{\gamma \in (GL(2,\mathbb{Z}) \ltimes \mathbb{Z}^3) \backslash Sp(4,\mathbb{Z})} \left( |\Omega_2|^\epsilon \, \varphi^{\mathrm{tr}}_{\mathrm{KZ}}(\Omega_2) \right) \Big|_\gamma \right] , \qquad (1.21)$$

where the limit $\epsilon \to 0$ is to be taken after analytic continuation from the region $\mathrm{Re}\,\epsilon > \frac{5}{2}$ where the sum converges. Using the theta lift representation of $\varphi_{\mathrm{KZ}}$ established in [48], we trace the relation (1.21) to a similar property (Eq. (A.16) of Appendix A) relating genus-one Siegel–Narain theta series for lattices of signature $(3,2)$ and $(2,1)$. Inserting (1.21) inside the genus-two

---

[8]For the particle multiplet $k = d+1$, we shall argue that $\Xi^{E_{d+1}}_{\Lambda_{d+1}}(\tau, \phi, r)$ converges for $\mathrm{Re}\,(r) > 4, 6, 9, \frac{29}{2}$ for $d = 4, 5, 6, 7$, respectively; for the string multiplet $k = 1$, that $\Xi^{E_{d+1}}_{\Lambda_1}(\tau, \phi, r')$ converges for $\mathrm{Re}\,(r') > 4, 6, \frac{17}{2}, \frac{23}{2}$.

modular integral in (1.6) and unfolding the integration domain, one immediately arrives at

$$8\pi \, \text{R.N.} \int_{\mathcal{F}_2} \frac{\mathrm{d}^6\Omega}{|\Omega_2|^3} \, \Gamma_{5,5,2} \, \varphi_{\text{KZ}} = \frac{4\pi}{3} \text{R.N.} \int_{S_+} \frac{\mathrm{d}^3\Omega_2}{|\Omega_2|^{1/2}} \sideset{}{'}\sum_{\substack{\mathcal{Q}_1,\mathcal{Q}_2 \in M^{D_5}_{\Lambda_1} \\ (\mathcal{Q}_i,\mathcal{Q}_j)=0}} e^{-\pi\Omega_2^{ij} G(\mathcal{Q}_i,\mathcal{Q}_j)} \, \varphi_{\text{KZ}}^{\text{tr}}(\Omega_2) \,, \quad (1.22)$$

where $\mathcal{Q}_i$ runs over pairs of *vectors* in the even self-dual lattice $M^{D_5}_{\Lambda_1} \cong \mathit{II}_{5,5}$ of $SO(5,5)$, which are null and mutually orthogonal.

In order to match (1.22) with (1.12) for $D=6$, which involves a sum over pairs of *spinors* under $Spin(5,5)$, we invoke the special case $d=4$ of a second remarkable property, namely

$$\frac{\Gamma(d-2)}{\pi^{d-2}} \sideset{}{'}\sum_{\substack{\Gamma_1,\Gamma_2 \in M^{E_{d+1}}_{\Lambda_{d+1}} \\ \Gamma_i \times \Gamma_j=0}} \left[ \frac{\tau_2}{G(\Gamma_1 + \tau\Gamma_2, \Gamma_1 + \bar{\tau}\Gamma_2)} \right]^{d-2} \underset{d\geq 3}{=} \frac{\Gamma(3)}{\pi^3} \sideset{}{'}\sum_{\substack{\mathcal{Q}_1,\mathcal{Q}_2 \in M^{E_{d+1}}_{\Lambda_1} \\ \mathcal{Q}_i \times \mathcal{Q}_j=0}} \left[ \frac{\tau_2}{G(\mathcal{Q}_1 + \tau\mathcal{Q}_2, \mathcal{Q}_1 + \bar{\tau}\mathcal{Q}_2)} \right]^3$$

$$(1.23)$$

where $\Gamma_i$ runs over pairs of vectors in the particle multiplet (the spinor for $D=6$), while $\mathcal{Q}_i$ runs over pairs of vectors in the string multiplet (the vector for $D=6$). As in (1.12), $\Gamma_i \times \Gamma_j$ denotes the projection of the product on the representation of highest weight $\Lambda_1$, while $\mathcal{Q}_i \times \mathcal{Q}_j$ denotes the projection of the product on the representation of highest weight $\Lambda_6$, which is trivial for $d<4$ and understood as a singlet for $d=4$ (see the last column in Table 1). While both sides of (1.23) are in general divergent, the identity should be understood as a statement about the analytic continuation of the sums $\Xi^{E_{d+1}}_{\Lambda_{d+1}}(\phi,\tau,r)$ and $\Xi^{E_{d+1}}_{\Lambda_1}(\phi,\tau,r')$ defined in (1.20) as $(r,r') \to (d-2,3)$. We do not expect that a similar relation holds for generic values of $(r,r')$. When the analytic continuations happen to have a pole at the required value $(r,r') \to (d-2,3)$, we shall argue that the equality (1.23) still holds for appropriately renormalised expressions.

In order to justify this claim, we shall show in Section 2 that the integral of both sides of (1.23) against $SL(2,\mathbb{Z})$ Eisenstein series and cusp forms agree, thanks to Langlands' functional equation for Eisenstein series of $E_{d+1}$. This can be viewed as a spectral justification of the claim (1.23). Identities similar to (1.23) for double lattice sums associated to vector and spinor representations of orthogonal groups are also established using similar methods in Section 2.8.

Formally inserting (1.23) into (1.22) and restoring the integral over $V$, we obtain the first term in (1.13), hinting at the equivalence of the two proposals for $D=6$. This equivalence can be established more rigorously after regularising both (1.22) and (1.23). Conversely, we can insert (1.23) inside (1.16), and obtain an alternative representation of the two-loop amplitude (1.12) involving a sum over pairs of 1/2-BPS *string* charges,

$$\mathcal{E}^{(d),\text{ExFT}}_{(0,1)} \underset{d\geq 3}{=} \frac{8\pi^2}{3} \int_{\mathcal{F}} \frac{\mathrm{d}\tau_1 \mathrm{d}\tau_2}{\tau_2^2} \, A(\tau) \sideset{}{'}\sum_{\substack{\mathcal{Q}_1,\mathcal{Q}_2 \in M^{E_{d+1}}_{\Lambda_1} \\ \mathcal{Q}\times\mathcal{Q}=0}} \left[ \frac{\tau_2}{G(\mathcal{Q}_1 + \tau\mathcal{Q}_2, \mathcal{Q}_1 + \bar{\tau}\mathcal{Q}_2)} \right]^3$$

$$= \frac{4\pi}{3} \int_{S_+} \frac{\mathrm{d}^3\Omega_2}{|\Omega_2|^{1/2}} \, \varphi_{\text{KZ}}^{\text{tr}}(\Omega_2) \sideset{}{'}\sum_{\substack{\mathcal{Q}_1,\mathcal{Q}_2 \in M^{E_{d+1}}_{\Lambda_1} \\ \mathcal{Q}_i \times \mathcal{Q}_j=0}} e^{-\pi\Omega_2^{ij} G(\mathcal{Q}_i,\mathcal{Q}_j)} \,. \quad (1.24)$$

For $d = 4$ we recover (1.22), for $d = 3$ (1.8); for $d = 1$ (and $d = 0$ type IIB) the constraint $\mathcal{Q}_i \times \mathcal{Q}_j = 0$ is trivially satisfied and the r.h.s. of (1.24) reproduces the two-loop supergravity integral of [31]. Note however that the first equality (1.24) only holds for $d \geq 3$. For example, for $d = 1$ the constraint $\Gamma_i \times \Gamma_j = 0$ admits two independent solutions, and the sum over $\Gamma_i \in \mathbb{Z}^{2+1}$ splits into the sum over eleven-dimensional supergravity Kaluza–Klein momenta on $T^2$, which is equal to the right-hand-side of (1.24), along with an additional sum over M2 branes wrapping $T^2$ which has no analogue on the string multiplet side.

Finally, as a by-product of this analysis, we obtain two alternative representations of $\mathcal{E}_{(0,1)}^{(d),\mathrm{ExFT}}$ as Poincaré series for the parabolic subgroups $P_d$ and $P_3$ of $E_{d+1}$, whose seed involves a special function $\widetilde{A}_s$ on the Poincaré upper half-plane,[9]

$$\mathcal{E}_{(0,1)}^{(d),\mathrm{ExFT}} \underset{d \geq 2}{=} \frac{8\pi^2}{3} \sum_{\gamma \in P_{d+1}\backslash E_{d+1}} \left[ y_d^{2-d}\, \widetilde{A}_{\frac{d-2}{2}}(U) \right]\Big|_\gamma \underset{d \geq 3}{=} \frac{8\pi^2}{3} \sum_{\gamma \in P_3\backslash E_{d+1}} \left[ y_3^{-3}\, \widetilde{A}_{\frac{3}{2}}(U) \right]\Big|_\gamma . \qquad (1.25)$$

The function $\widetilde{A}_s$, defined in (2.43) below, satisfies the differential equation (2.44). For $s = 3/2$, it coincides (up to the overall factor $\frac{8\pi^2}{3}$ in (1.25)) with the exact $\nabla^6 \mathcal{R}^4$ coupling in ten-dimensional type IIB string theory considered in [31, 32]. Thus, the second equation in (1.25) can be summarised by saying that the exact $\nabla^6 \mathcal{R}^4$ coupling in $D$ dimensions is the sum of the covariantisation of the S-duality invariant coupling in $D = 10$ under U-duality and the homogeneous solution $\mathcal{F}_{(0,1)}^{(d)}$ in (1.11), which is separately U-duality invariant. Unfortunately, while conceptually pleasing, the identities (1.25) do not seem to be convenient for obtaining asymptotic expansions.

## Weak coupling expansion

In Section 3 we compute the asymptotic expansion of (1.16) at weak coupling by generalising techniques introduced in [1, 50], whereby the constraints $\mathcal{Q}_i \times \mathcal{Q}_j = 0$ are solved step by step for a suitable graded decomposition of $E_{d+1}$ which keeps T-duality manifest. Using this method, we find the expected perturbative contributions, up to instanton corrections:

$$\mathcal{E}_{(0,1)}^{(d),\mathrm{ExFT}} = g_D^{\frac{2d+8}{d-8}} \left( \frac{2\zeta(3)^2}{3g_D^2} + \frac{4\pi\zeta(3)}{3}\xi(d-2)\, E_{\frac{d-2}{2}\Lambda_1}^{D_d} + g_D^2\, \mathcal{E}_{(0,1)}^{(d,2)} + \frac{4\zeta(6)}{27}g_D^4\, \widehat{E}_{3\Lambda_{d-1}}^{D_d} + \mathcal{O}(e^{-1/g_D}) \right) \quad (1.26)$$

Here, the first term reproduces the tree-level contribution, the second term and fourth term reproduce part of the genus one (3.2) and genus three (3.4) contribution (the second part coming from the homogeneous solution (1.11) while the third term reproduces the full genus-two contribution (3.3), involving the Kawazumi–Zhang invariant $\varphi_{\mathrm{KZ}}$. This fact relies on the key equality (1.21) between $\varphi_{\mathrm{KZ}}$ and the Poincaré series seeded by its tropical limit $\varphi_{\mathrm{KZ}}^{\mathrm{tr}}$.

We are also able to extract the contributions to the constant term from bound states of instantons and anti-instantons for any $d \leq 6$. For $d = 0$, our approach provides a powerful computational method, alternative to the one used in [31], which reproduces the results found in [32] by integrating the differential equation.

---

[9]Another proposal for $\mathcal{E}_{(0,1)}^{(3)}$ was given in [49], based on a Poincaré sum over $P_1\backslash SL(5)$, which can be rewritten as a single lattice sum. In contrast, the double lattice sum in (1.8) can be rewritten as a Poincaré sum over $P_2\backslash SL(5)$, see (1.25) below.

Our method also allows us to analyse the non-zero Fourier modes of $\mathcal{E}_{(0,1)}^{(d),\text{ExFT}}$ that correspond to contributions to the scattering amplitude in the background of D-brane or NS-brane instantons. In Section 3.3 we express the D-instanton contribution in terms of nested orbit sums. The result is complete for $E_5 = D_5 = Spin(5,5)$, we argue that it is also complete for $E_6$ and we compute the generic Fourier coefficients (3.89) for $E_7$. The generic Fourier coefficients in $d = 6$ are particularly interesting because they are expected to be proportional to the helicity supertrace $\Omega_{14}$ counting 1/8-BPS D-brane bound states. We find agreement with [51–53] for the simplest D-brane configurations, but further analysis is required to understand the general case.

## Large radius/decompactification expansion

In Section 4 we study in the large radius limit using similar methods and obtain the expected expansion [22, 2]

$$\mathcal{E}_{(0,1)}^{(d),\text{ExFT}} = R^{\frac{12}{8-d}} \left[ \mathcal{E}_{(0,1)}^{(d-1),\text{ExFT}} + \frac{5}{\pi} \xi(d-6) R^{d-7} \mathcal{E}_{(1,0)}^{(d-1)} + \frac{2\pi}{3} \xi(d-2) R^{d-3} \mathcal{E}_{(0,0)}^{(d-1)} \right. \tag{1.27}$$
$$\left. + \frac{20\pi}{3} \xi(6)\xi(d+4) R^{d+3} + \frac{16\pi^2 [\xi(d-2)]^2}{(d+1)(6-d)} R^{2d-6} + \mathcal{O}(e^{-R}) \right].$$

where $R$ is the radius measured in $D$-dimensional Planck units. The first line in this formula represents the contribution from the decompactification of the $\nabla^6 \mathcal{R}^4$ term on $T^d$ to the one on $T^{d-1}$ along with threshold effects coming from the $\nabla^4 \mathcal{R}^4$ and $\mathcal{R}^4$ interactions. The second line, containing pure powers of the decompactifying radius $R$, represents one-loop and two-loop threshold effects in supergravity.

Our method also allows us to analyse the non-zero Fourier modes of $\mathcal{E}_{(0,1)}^{(d),\text{ExFT}}$, which can now be interpreted as instanton effects from Euclidean black holes wrapping the Euclidean time circle. The result is complete for $E_5 = Spin(5,5)$ and $E_6$, we argue that it is also complete for $E_7$ and we compute the generic abelian Fourier coefficients for $E_8$ in Appendix E. The generic abelian Fourier coefficients in $d = 7$ are also particularly interesting because they are expected to be proportional to the helicity supertrace counting 1/8-BPS black holes. We do find agreement with [51–53] for the simplest black hole charge configurations, but further analysis is required to understand the general case.

## 1/4-BPS contributions and renormalised function

The couplings derived from the two-loop exceptional field theory calculation alone diverge in dimension $d = 4, 5, 6$ whereas the full string theory amplitude is supposed to be finite. This discrepancy can be traced back to the fact that in exceptional field theory only 1/2-BPS states are allowed to propagate in the loops. However, the full theory also involves 1/4- and 1/8-BPS states as well as non-BPS states. For specific BPS protected couplings such as $\nabla^6 \mathcal{R}^4$, one may hope that only 1/2- and 1/4-BPS states contribute at two-loop. Indeed, the perturbative genus-two $\nabla^6 \mathcal{R}^4$ coupling in string theory [35, 36] exhibits precisely such contributions. The result of our analysis shows that the contributions from 1/2- and 1/4-BPS states up to three-loop indeed reproduces the exact low energy effective action up to $\nabla^6 \mathcal{R}^4$, leading to the conclusion that 1/8-BPS states do not contribute to this coupling.

A similar issue was already encountered in [30] in the context of the $\nabla^4 \mathcal{R}^4$ coupling where both 1/2- and 1/4-BPS states happen to contribute. The contribution of the latter was inferred in [30] by taking the perturbative one-loop string calculation, extracting the contribution of perturbative 1/4-BPS states, and covariantising this result under U-duality so as to obtain the contribution of the full non-perturbative spectrum of 1/4-BPS states. Following the same strategy for $\nabla^6 \mathcal{R}^4$, we find that the two-loop amplitude with 1/4-BPS charges running in the loops is given by a similar integral as in (1.12), where $\varphi_{\mathrm{KZ}}^{\mathrm{tr}}(\Omega_2)$ is replaced by $-E_{-3\Lambda_1}^{SL(2)}(\tau)/V$ in the variables (1.14). Combining this with the exceptional field theory result, we get

$$\mathcal{E}_{(0,1)}^{(d)2\text{-loop}} = \frac{4\pi}{3} \int_{S_+} \frac{\mathrm{d}^3\Omega_2}{|\Omega_2|^{\frac{6-d-2\epsilon}{2}}} \Big( \varphi_{\mathrm{KZ}}^{\mathrm{tr}} - \frac{\pi}{36} \frac{E_{-3\Lambda_1}^{SL(2)}(\tau)}{V} \Big) \theta_{\Lambda_{d+1}}^{E_{d+1}}(\phi, \Omega_2) . \tag{1.28}$$

We will give strong evidence that this expression has a finite limit as $\epsilon \to 0$ for all values of $d$. At three-loops, the structure of the perturbative genus-three superstring amplitude [54] suggests that only 1/2-BPS states run in the loop. The resulting three-loop contribution in exceptional field theory [30] produces the sum of two Eisenstein series (5.18), one of which formally cancels the 1/4-BPS contribution in (1.28), while the other reproduces the contribution (1.11)

$$\mathcal{E}_{(0,1)}^{(d)3\text{-loop}} = \frac{\pi^2}{27} \int_{S_+} \frac{\mathrm{d}^3\Omega_2}{|\Omega_2|^{\frac{6-d}{2}}} \frac{E_{-(3+2\epsilon)\Lambda_1}^{SL(2)}(\tau)}{V} \theta_{\Lambda_{d+1}}^{E_{d+1}}(\phi, \Omega_2) + \frac{8\pi^4}{567} \xi(d+4+2\epsilon) \, E_{\frac{d+4+2\epsilon}{2}\Lambda_{d+1}}^{E_{d+1}} . \tag{1.29}$$

The regularised Eisenstein series (1.11) is defined by

$$\widehat{\mathcal{F}}_{(0,1)}^{(d)} = \frac{8\pi^4}{567} \xi(d+4+2\epsilon) \, E_{\frac{d+4+2\epsilon}{2}\Lambda_{d+1}}^{E_{d+1}} \Big|_{\epsilon^0} = \frac{8\pi^4}{567} \xi(d+4) \, \widehat{E}_{\frac{d+4}{2}\Lambda_{d+1}}^{E_{d+1}} , \tag{1.30}$$

where $|_{\epsilon^0}$ denotes the zeroth order term in the Laurent series in $\epsilon$, which amounts to a minimal subtraction prescription. According to the discussion below (1.18), we shall not keep track of the finite terms proportional to the residue at the pole, which is similar to the difference between minimal substraction (MS) or modified minimal subtraction ($\overline{\mathrm{MS}}$) regularisation schemes in quantum field theory. We shall use the hat notation for similar zeroth order terms of any Eisenstein series throughout this work.

Although the two functions in (1.29) satisfy the same $E_{d+1}$ invariant differential equation (up to inhomogeneous terms), they do not satisfy the same differential equations [14] and correspond mathematically to distinct automorphic representations [47]. Physically this means that instantons with generic charges in a given limit may contribute to one function and not to the other. In particular $\widehat{\mathcal{F}}_{(0,1)}^{(6)}$ obtains contributions from generic Euclidean black holes instantons in the decompactification limit whereas the first term in (1.29) and the two functions in (1.28) do not, while these latter receive corrections from generic D-brane instantons in the weak-coupling limit whereas $\widehat{\mathcal{F}}_{(0,1)}^{(6)}$ does not. It is therefore more natural to combine the first component of (1.29) with the two-loop contribution (1.28) to define the renormalised coupling $\widehat{\mathcal{E}}_{(0,1)}^{(d),\mathrm{ExFT}} = \mathcal{E}_{(0,1),\epsilon}^{(d),\mathrm{ExFT}} \Big|_{\epsilon^0}$ as the finite part of

$$\mathcal{E}_{(0,1),\epsilon}^{(d),\mathrm{ExFT}} = \frac{4\pi}{3} \int_{S_+} \frac{\mathrm{d}^3\Omega_2}{|\Omega_2|^{\frac{6-d}{2}}} \Big( |\Omega_2|^\epsilon \varphi_{\mathrm{KZ}}^{\mathrm{tr}} - \frac{\pi}{36} \frac{E_{-3\Lambda_1}^{SL(2)}(\tau)}{V^{1+2\epsilon}} + \frac{\pi}{36} \frac{E_{-(3+2\epsilon)\Lambda_1}^{SL(2)}(\tau)}{V} \Big) \theta_{\Lambda_{d+1}}^{E_{d+1}}(\phi, \Omega_2) \tag{1.31}$$

as $\epsilon \to 0$. The two additional terms cancel each other for $d \leq 3$ and we shall see that the function (1.31) has the correct behaviour in $d = 4, 5, 6, 7$. In $d = 4, 5, 6$, these contributions are individually divergent, but the total result is well-defined and satisfies all expected weak coupling and decompactification limits. The sum of (1.31) and (1.30) defines the complete non-perturbative function

$$\mathcal{E}_{(0,1)}^{(d)} = \widehat{\mathcal{E}}_{(0,1)}^{(d),\mathrm{ExFT}} + \widehat{\mathcal{F}}_{(0,1)}^{(d)} , \tag{1.32}$$

which gives a precise definition to the formal formula (1.13).

## Outline

The remainder of this work is organised as follows. In Section 2 we establish the central identity (1.23) that relates the double lattice sums in the string and particle multiplets to show the equivalence (1.22) between the particle and string multiplet representations of the $\nabla^6 \mathcal{R}^4$ coupling $\mathcal{E}_{(0,1)}^{(d)}$. In Sections 3 and 4, we analyse the weak string coupling and single circle decompactification limits of $\mathcal{E}_{(0,1)}^{(d)}$, respectively. In particular, we compute the corresponding Fourier expansion of $\mathcal{I}_d(\phi, \epsilon, L)$ defined in (1.17), and find that it is a meromorphic function of $\epsilon$. In Section 5, we discuss in detail the renormalisation of the function $\mathcal{E}_{(0,1)}^{(d)}$ due to the contribution of 1/4-BPS states at two-loop order and show that their contribution cancels the divergences coming from the 1/2-BPS sector, leading to the well-defined total result (1.32). Several appendices contain additional technical details. In Appendix A, we present evidence for (1.21) that expresses the Kawazumi–Zhang invariant as a Poincaré series seeded by its tropical limit while Appendices B–D discuss certain integrals and auxiliary Fourier expansions that are used in the main body of the paper. Most of our calculations apply to $4 \leq D \leq 8$. Appendix E contains details for the special case of $D = 3$ with U-duality group $E_8$ that are also relevant to 1/8-BPS black holes and Appendix F summarises the cases $D \geq 8$.

## 2 From particle to string multiplet

In this section, we give very strong evidence for the identity (1.23) relating double Epstein sums in the particle and string multiplets of $E_{d+1}$. This is done by first including arbitrary parameters $(r, r')$ on either sides to obtain convergent expressions, and analytically continuing to the desired values $(r, r') \to (d - 2, 3)$ at the end. In Sections 2.1 and 2.2, we show that both sides satisfy the same differential equations invariant under $SL(2) \times E_{d+1}$. The representation-theoretic origin of the differential equations is discussed in Section 2.3. In the remaining subsections, we provide a spectral argument for (1.23) by computing the integral of both sides against Maaß cusp forms and Eisenstein series of $SL(2, \mathbb{Z})$, and showing that the two results are equal by virtue of Langlands' functional relation.

### 2.1 Laplace identities

We first establish that the lattice sum (1.20) in the particle multiplet representation satisfies

$$\left[ \Delta_{E_{d+1}} - \Delta_\tau - \frac{r[(d-10)r - 20 + 18d - d^2]}{d - 8} \right] \Xi_{\Lambda_{d+1}}^{E_{d+1}}(\phi, \tau, r) = 0 , \tag{2.1}$$

where $\Delta_{E_{d+1}}$ is the Laplace operator on $\mathcal{M} = E_{d+1}/K_{d+1}$, and $\Delta_\tau = \tau_2^2(\partial_{\tau_1}^2 + \partial_{\tau_2}^2)$ is the $SL(2)$ Laplace operator on the upper-half plane. To prove this, we proceed as in [1] and write the sum over doublets of charges $\Gamma_1, \Gamma_2$ such that $\Gamma_i \times \Gamma_j = 0$ as a Poincaré sum over $P_d \backslash E_{d+1}$, where $P_d$ is the maximal parabolic subgroup with Levi subgroup $GL(2) \times E_{d-1}$. Under this subgroup, the particle multiplet decomposes as [1, (4.28)] [10]

$$M_{\Lambda_{d+1}}^{E_{d+1}} \cong (\mathbf{2}, \mathbf{1})^{(10-d)} \oplus (\mathbf{1}, M_{\Lambda_{d-1}}^{E_{d-1}})^{(2)} \oplus (\mathbf{2}, M_{\Lambda_1}^{E_{d-1}})^{(d-6)} \oplus \dots \qquad (2.2)$$

corresponding to the various charges arises upon compactifying from dimension $D + 2 = 12 - d$ down to $D = 10 - d$ on a torus $T^2$: the Kaluza–Klein charges on $T^2$, particle charges in dimension $D + 2$, strings in dimension $D + 2$ wrapped on a circle in $T^2$, while the dots correspond to membranes wrapped on $T^2$ and Kaluza–Klein monopoles. The superscripts in (2.2) denote the scaling degree with respect to the action of $GL(1) \subset GL(2)$, normalised so as to take integer values. The representations of $SL(2) \times E_{d-1}$ are denoted as $(2j + 1, M_\lambda^{E_{d-1}})$ with $2j + 1$ the dimension of the $SL(2)$ representation and $\lambda$ the highest weight of the $E_{d-1}$ representation.

As explained in [1], one can always rotate a pair of vectors $\Gamma_1, \Gamma_2$ using $E_{d+1}$ into the top degree space $(\mathbf{2}, \mathbf{1})^{(10-d)}$ (i.e. the eigenspace in (2.2) of maximal $GL(1)$ eigenvalue), thereby allowing a rewriting of (1.20) as

$$\frac{\pi^r}{\Gamma(r)} \Xi_{\Lambda_{d+1}}^{E_{d+1}}(\phi, \tau, r) = \sum_{\gamma \in P_d \backslash E_{d+1}} \left( \sum_{\substack{M \in \mathbb{Z}^{2 \times 2} \\ \det M \neq 0}} \left[ \frac{\tau_2 U_2 \, y_d^{-1}}{|(1, \tau) M (1, U)|^2 + 2\tau_2 U_2 \det M} \right]^r \right) \Bigg|_\gamma$$
$$+ \sum_{\gamma \in P_{d+1} \backslash E_{d+1}} \left( \sideset{}{'}\sum_{(m_1, m_2) \in \mathbb{Z}^2} \left[ \frac{\tau_2}{|m_1 \tau + m_2|^2 y_{d+1}^2} \right]^r \right) \Bigg|_\gamma \qquad (2.3)$$

corresponding to non-collinear and collinear pairs of vectors, respectively, and where $y_k$ is the multiplicative character for the parabolic subgroup $P_k$, normalised [11] such that the Langlands–Eisenstein series is $E_{s\Lambda_k}^G = \sum_{\gamma \in P_k \backslash G} y_k^{-2s}|_\gamma$. Note that the second term can be viewed as the contribution of matrices $M$ with $\det M = 0$ in the first sum; moreover, it is recognised as $\zeta(2r) E_{r\Lambda_1}^{SL(2)}(\tau) E_{r\Lambda_{d+1}}^{E_{d+1}}$.

Next, we use the fact that upon acting on functions depending only on the $GL(2)/U(1)$ factor parametrised by $U = U_1 + iU_2 \in \mathcal{H}_1$ and $y_d$, the Laplacian on $E_{d+1}$ reduces to to [1, (4.65)]

$$\Delta_{E_{d+1}} = \frac{1}{(8-d)} y_d \partial_{y_d} \big( (10-d) y_d \partial_{y_d} + (20 + d(d-18)) \big) + \Delta_U . \qquad (2.4)$$

---

[10]For $d = 5, 6, 7$, this follows from embedding $GL(2) \times E_{d-1}$ in a dual pair inside $E_{d+1}$,

$$E_6 \supset SL(2) \times SL(6) : \quad \mathbf{27} \quad \cong \quad (\mathbf{2}, \mathbf{6}) \oplus (\mathbf{1}, \mathbf{15})$$
$$E_7 \supset SL(2) \times Spin(6,6) : \quad \mathbf{56} \quad \cong \quad (\mathbf{1}, \mathbf{32}) \oplus (\mathbf{2}, \mathbf{12})$$
$$E_8 \supset SL(3) \times E_6 : \quad \mathbf{248} \quad \cong \quad (\mathbf{8}, \mathbf{1}) \oplus (\mathbf{1}, \mathbf{78}) \oplus (\mathbf{3}, \mathbf{27}) \oplus (\overline{\mathbf{3}}, \overline{\mathbf{27}})$$

[11]The normalisation of the character is defined such that the action of the Cartan torus element on the lowest weight representation $\Lambda_k$ is normalised to $y_k$. In other words, we write the torus element as $\exp(-\sum_i \log(y_i) h_i)$, where the $h_i$ are the canonical Chevalley generators that need to be evaluated in the lowest weight representation. For example for $SL(2)$ this leads to the matrix $\mathrm{diag}(y_1, y_1^{-1})$.

Acting on the seed in the first term on right-hand side of (2.3) immediately leads to (2.1). The same is true using $y_{d+1}^2 = y_d U_2^{-1}$ for the sum in the second line of (2.3), which morally extends the sum on the first line to all non-zero matrices $M$, thus establishing (2.1).

Similarly, for any $4 \leq d \leq 7$, we claim that the lattice sum in the string multiplet satisfies

$$\left[\Delta_{E_{d+1}} - \Delta_\tau - \frac{dr'(2d-1-r')}{d-8}\right] \Xi_{\Lambda_1}^{E_{d+1}}(\phi, \tau, r') = 0. \tag{2.5}$$

To establish this, we proceed as before and write the constrained lattice sum as a Poincaré sum over $P_3 \backslash E_{d+1}$, where $P_3$ is the maximal parabolic subgroup with Levi factor $GL(2) \times SL(d)$. Under this subgroup, the string multiplet decomposes as[12]

$$M_{\Lambda_1}^{E_{d+1}} \cong (\mathbf{2},\mathbf{1})^{(1)} \oplus (\mathbf{1}, \wedge^2 V)^{(\frac{2d-8}{d})} \oplus (\mathbf{2}, \wedge^4 V)^{(\frac{3d-16}{d})} \oplus (\mathbf{1}, V \otimes \wedge^5 V)^{(\frac{4d-24}{d})} \oplus \dots, \tag{2.6}$$

corresponding to the various string charges appearing in the large volume limit of type IIB string theory compactified on $T^d$: $(p,q)$ strings, D3-branes, $(p,q)$ 5-branes, and Kaluza–Klein, with $V = \mathbb{Z}^d$. Using $E_{d+1}$, one can always rotate any pair of vectors $\mathcal{Q}_1, \mathcal{Q}_2$ into the top degree space $(\mathbf{2},\mathbf{1})^{(1)}$, obtaining

$$\frac{\pi^{r'}}{\Gamma(r')} \Xi_{\Lambda_1}^{E_{d+1}}(\phi, \tau, r') = \sum_{\gamma \in P_3 \backslash E_{d+1}} \left( \sum_{\substack{M \in \mathbb{Z}^{2\times 2} \\ \det M \neq 0}} \left[ \frac{\tau_2 U_2 y_3^{-1}}{|(1,\tau)M(1,u)|^2 + 2\tau_2 U_2 \det M} \right]^{r'} \right)\Bigg|_\gamma$$

$$+ \sum_{\gamma \in P_1 \backslash E_{d+1}} \left( {\sum_{(m_1,m_2) \in \mathbb{Z}^2}}' \left[ \frac{\tau_2}{|m_1 \tau + m_2|^2 y_1^2} \right]^{r'} \right)\Bigg|_\gamma. \tag{2.7}$$

We then use the fact that the Laplacian on $E_{d+1}$ acting on functions depending only on the $GL(2)/U(1)$ factor reduces to

$$\Delta_{E_{d+1}} = \frac{d}{8-d}(y_3 \partial_{y_3} + 2d - 1)y_3 \partial_{y_3} + \Delta_U. \tag{2.8}$$

Acting with this operator on the seed terms in (2.7) establishes (2.5). For $d = 5$, the two equations (2.7) and (2.3) are of course identical since the particle and string multiplets $\overline{\mathbf{27}}$ and $\mathbf{27}$ are related by conjugation.

For $r = d - 2$, $r' = 3$, the two eigenvalues in (2.1) and (2.5) agree and the two lattice sums satisfy the differential equation

$$\left[\Delta_{E_{d+1}} - \Delta_\tau + \frac{6d(d-2)}{8-d}\right] \Xi^{E_{d+1}} = 0, \tag{2.9}$$

where $\Xi^{E_{d+1}}$ stands for either the string multiplet double Epstein series $\Xi_{\Lambda_1}^{E_{d+1}}$ or the particle multiplet double Epstein series $\Xi_{\Lambda_{d+1}}^{E_{d+1}}$.

---

[12]For $d = 6, 7$, this follows by embedding $GL(2) \times SL(d)$ inside a dual pair,

$$E_7 \supset SL(3) \times SL(6) : \quad \mathbf{133} = (\mathbf{8},\mathbf{1}) \oplus (\mathbf{1},\mathbf{35}) \oplus (\mathbf{3},\overline{\mathbf{15}}) \oplus (\overline{\mathbf{3}},\mathbf{15})$$

$$E_8 \supset SL(2) \times E_7 : \quad \mathbf{3875} = (\mathbf{1},\mathbf{1539}) + (\mathbf{3},\mathbf{133}) + (\mathbf{2},\mathbf{56}) + (\mathbf{2},\mathbf{912}) + (\mathbf{1},\mathbf{1})$$

## 2.2 Tensorial differential equations

For the same values $(r, r') = (d-2, 3)$, it turns out that the two lattice sums satisfy a much stronger system of differential equations beyond the Laplace equation (2.9). This system of equations is given compactly for $d = 4, 5, 6$ as

$$\left(T^\alpha T^\beta T^\gamma \mathcal{D}_\alpha \mathcal{D}_\beta \mathcal{D}_\gamma - \frac{3}{2}(d + 4 - \tfrac{34}{9-d})T^\alpha \mathcal{D}_\alpha\right)\Xi^{E_{d+1}} = 0 , \qquad (2.10)$$

where $T^\alpha$ are the generators of $E_{d+1}$ written in the highest weight representation $R(\Lambda_{d+1})$ and $\mathcal{D}_\alpha = V_\alpha{}^M(\partial_M + \omega_M)$ the covariant derivative in tangent frame, where $V_\alpha{}^M$ denotes the inverse vielbein on the Riemannian symmetric space $E_{d+1}/K(E_{d+1})$ and $\omega_M$ is the $K(E_{d+1})$ connection defined by the $K(\mathfrak{e}_{d+1})$ component of the Maurer–Cartan form [12, 47]. To prove (2.10) for the particle multiplet sum, one uses the same Poincaré sum representation (2.3), and the restriction of the tensorial equation (2.10) on functions of $GL(2)$, which appeared in Eqs. (4.94) and (4.96) of [1]. For the string multiplet sum, (2.10) also holds for $d = 5$ since the particle and string multiplets are conjugate. For $d = 6$, additional work is required. Using the same techniques as in [1], one finds that for a function of the Levi subgroup $GL(2) \subset P_3 \subset E_7$,

$$T^\alpha \mathcal{D}_\alpha = \begin{pmatrix} y_3\partial_{y_3}\mathbb{1}_6 & 0 & 0 & 0 & 0 & 0 & 0 \\ 0 & \frac{1}{2}(y_3\partial_{y_3} + U_2\partial_{U_2})\mathbb{1}_6 & \frac{1}{2}U_2\partial_{U_1}\mathbb{1}_6 & 0 & 0 & 0 & 0 \\ 0 & \frac{1}{2}U_2\partial_{U_1}\mathbb{1}_6 & \frac{1}{2}(y_3\partial_{y_3} - U_2\partial_{U_2})\mathbb{1}_6 & 0 & 0 & 0 & 0 \\ 0 & 0 & 0 & 0_{20} & 0 & 0 & 0 \\ 0 & 0 & 0 & 0 & \frac{1}{2}(-y_3\partial_{y_3} + U_2\partial_{U_2})\mathbb{1}_6 & \frac{1}{2}U_2\partial_{U_1}\mathbb{1}_6 & 0 \\ 0 & 0 & 0 & 0 & \frac{1}{2}U_2\partial_{U_1}\mathbb{1}_6 & -\frac{1}{2}(y_3\partial_{y_3} + U_2\partial_{U_2})\mathbb{1}_6 & 0 \\ 0 & 0 & 0 & 0 & 0 & 0 & -y_3\partial_{y_3}\mathbb{1}_6 \end{pmatrix}.$$
$$(2.11)$$

This is a $(56 \times 56)$ matrix of first order differential operators since $R(\Lambda_{d+1})$ corresponds to the **56** of $E_7$, see Table 1. The differential operators $\mathcal{D}_\alpha$ normally act on the 70 coordinates of $E_7/SU(8)$ but here are reduced to the coordinates $U = U_1 + iU_2$ and $y_3$ of the $GL(2) \subset E_7$ part of the symmetric space. The matrix is blocked according to the branching of the representation $R(\Lambda_{d+1})$ under $GL(2) \times SL(5)$ that is

$$\mathbf{56} \cong (\mathbf{1}, \mathbf{6})^{(2)} \oplus (\mathbf{2}, \overline{\mathbf{6}})^{(1)} \oplus (\mathbf{1}, \mathbf{20})^{(0)} \oplus (\mathbf{2}, \mathbf{6})^{(-1)} \oplus (\mathbf{1}, \overline{\mathbf{6}})^{(-2)} , \qquad (2.12)$$

and we have written out the doublets of **6** separately in (2.11). The third power of (2.11) evaluates to

$$T^\alpha T^\beta T^\gamma \mathcal{D}_\alpha \mathcal{D}_\beta \mathcal{D}_\gamma = \begin{pmatrix} \mathcal{S}_1\mathbb{1}_6 & 0 & 0 & 0 & 0 & 0 & 0 \\ 0 & (\mathcal{S}_2 + \frac{1}{2}\mathcal{S}_3 U_2\partial_{U_2})\mathbb{1}_6 & \frac{1}{2}\mathcal{S}_3 U_2\partial_{U_1}\mathbb{1}_6 & 0 & 0 & 0 & 0 \\ 0 & \frac{1}{2}\mathcal{S}_3 U_2\partial_{U_1}\mathbb{1}_6 & (\mathcal{S}_2 - \frac{1}{2}\mathcal{S}_3 U_2\partial_{U_2})\mathbb{1}_6 & 0 & 0 & 0 & 0 \\ 0 & 0 & 0 & 0_{20} & 0 & 0 & 0 \\ 0 & 0 & 0 & 0 & (-\mathcal{S}_2 + \frac{1}{2}\mathcal{S}_3 U_2\partial_{U_2})\mathbb{1}_6 & \frac{1}{2}\mathcal{S}_3 U_2\partial_{U_1}\mathbb{1}_6 & 0 \\ 0 & 0 & 0 & 0 & \frac{1}{2}\mathcal{S}_3 U_2\partial_{U_1}\mathbb{1}_6 & -(\mathcal{S}_2 + \frac{1}{2}\mathcal{S}_3 U_2\partial_{U_2})\mathbb{1}_6 & 0 \\ 0 & 0 & 0 & 0 & 0 & 0 & -\mathcal{S}_1\mathbb{1}_6 \end{pmatrix}$$
$$(2.13)$$

with

$$\mathcal{S}_1 = (y_3\partial_{y_3})^3 - \frac{37}{4}(y_3\partial_{y_3})^2 + \frac{3}{4}\Delta_U + \frac{27}{4}y_3\partial_{y_3} ,$$

$$\mathcal{S}_2 = \frac{1}{8}(y_3\partial_{y_3})^3 + \frac{3}{8}\Delta_U y_3\partial_{y_3} - \frac{25}{8}(y_3\partial_{y_3})^2 - \frac{3}{4}\Delta_U + \frac{9}{4}y_3\partial_{y_3} ,$$

$$\mathcal{S}_3 = \frac{3}{4}(y_3\partial_{y_3})^2 + \frac{1}{4}\Delta_U - \frac{33}{4}y_3\partial_{y_3} \ . \tag{2.14}$$

Using these formulae, one can check that the seed function in (2.7) is annihilated by the operator in (2.10).

We have shown that the seed of the double Epstein series (1.20) for the string and particle multiplet satisfy the same homogeneous differential equations. By inserting these equations in (1.12) or (1.24), and using the differential equation satisfied by the (non-differentiable) modular function $A(\tau)$ (see [31] and (B.3) below), one may show that both proposals (1.12) and (1.24) satisfy the inhomogenous differential equations required by supersymmetry Ward identities [14], including the Poisson-type equation (1.5) [1]. To prove that the two double Epstein series (1.20) indeed satisfy the tensorial differential equations we need to take care of the poles that arise by analytic continuation from the domain of absolute convergence. We shall argue that they are indeed satisfied, but for $E_6$, in which case only the renormalised coupling does satisfy the equations.

## 2.3   Nilpotent orbits and BPS states

The structure of the tensorial differential equations (2.10) can be understood by using the language of nilpotent orbits of the group acting on its Lie algebra (see e.g. [55]). We shall be using Bala–Carter labels for complex nilpotent orbits. The Bala–Carter label, e.g. $A_1$, $A_2$ or $2A_1$ (where the last case designates two commuting $A_1 \cong SL(2)$ subgroups) indicates in what type of Levi subgroup a given nilpotent Lie algebra element is distinguished. For type $A_n$, a nilpotent element is distinguished if it is regular (a.k.a. principal), i.e. if it belongs to the largest possible nilpotent orbit of $A_n$.[13] If there are several non-conjugate Levi subgroups of the same type in $E_{d+1}$, the Bala–Carter label includes conventional primes to differentiate between the non-conjugate orbits, e.g. $(2A_1)'$ and $(2A_1)''$ in $E_5$.

Nilpotent orbits provide a useful classification of Fourier coefficients of automorphic forms. In physics terminology, Fourier coefficients describe effects from non-perturbative states coupling to axions, corresponding to coordinates along nilpotent generators in the symmetric moduli space $E_{d+1}/K(E_{d+1})$ in a given parabolic decomposition [5, 21, 23]. Therefore, the various types of non-perturbative effects can be labelled by nilpotent elements. The set of nilpotent orbits that support non-vanishing Fourier coefficients is often called the wavefront set of an automorphic form. The wavefront set of a generic Eisenstein series induced from a parabolic subgroup $P = LU \subset E_{d+1}$ can be easily determined from the Gelfand–Kirillov dimension, and is tabulated for various groups and parabolics e.g. in [56].

In the relation between nilpotent orbits and non-perturbative effects, 1/2-BPS states correspond to nilpotent elements of Bala–Carter label $A_1$, while 1/4-BPS states correspond to Bala–Carter label $2A_1$. This can be understood by noticing that certain 1/4-BPS states can be realised as an (orthogonal) intersection of two 1/2-BPS states. Similarly, certain 1/8-BPS states can be realised by an (orthogonal) intersection of three 1/2-BPS states, leading to the

---

[13]For other Levi types there can be a finite number of such distinguished nilpotent elements; those are written using conventional labels in parentheses following the Levi type, e.g., $D_4(a_1)$. Since these Levi types only appear for nilpotent orbits larger than the one encountered in the present paper, we refer the interested reader to [55].

Bala–Carter label $3A_1$. In addition, the labels $A_2$ and $4A_1$ are also associated with 1/8-BPS states and arise for a different class of non-perturbative effects [14, 47].

The order of nilpotency $p$ in $x^p = 0$ of a given nilpotent element $x$ of the Lie algebra depends on the finite-dimensional representation of $\mathfrak{e}_{d+1}$ in which it acts. Since the Lie algebra is represented by first order differential operators acting on functions on the symmetric space $E_{d+1}/K(E_{d+1})$, the nilpotency relations translate into differential equations of order $p$ satisfied by the automorphic form. For a given automorphic form, the strongest differential equation arises from the maximal orbit in its wavefront set. Physically, this equation corresponds to a supersymmetric Ward identity [12–14]. Often it suffices to consider these equations in only one of the fundamental representations, as the others will be consequences.

Equipped with this knowledge we now see that (2.10) is in fact the tensorial differential equation associated with the maximal orbit in the wavefront set of the Eisenstein series induced from the Heisenberg parabolic subgroup $P_H$, i.e. the maximal parabolic subgroup $P_{\Lambda_H}$ associated to the highest weight for the adjoint representation (respectively $\Lambda_2$, $\Lambda_2$, $\Lambda_1$, $\Lambda_8$ for $D_5$, $E_6$, $E_7$ and $E_8$). As will be shown in (2.37) and (2.40) below, integrating the lattice sums $\Xi_{\Lambda_1}^{E_{d+1}}$ at $r' = 3$ or respectively $\Xi_{\Lambda_{d+1}}^{E_{d+1}}$ at $r = d-2$ against an arbitrary $SL(2)$ Eisenstein series leads to an Eisenstein series $E_{s\Lambda_H}^{E_{d+1}}$ for the Heisenberg parabolic for a specific value of $s$. The wavefront set of any such 'adjoint' Eisenstein series is generically of Bala–Carter type $A_2$.

Since integrating the double lattice sums (1.23) against an $SL(2)$ Eisenstein series gives an Eisenstein series with a wavefront set associated to $A_2$ nilpotent orbit, we conclude that the Fourier coefficients of the double lattice sums themselves are also restricted to the same orbits, and thus are at most of Bala–Carter type $A_2$, as confirmed by equation (2.10). Automorphic representations of $E_{d+1}$ for $d \geq 4$ with this Bala–Carter type are uniquely represented by adjoint Eisenstein series $E_{s\Lambda_H}^{E_{d+1}}$, where $s$ is determined by the eigenvalue under the Laplacian. This already gives a strong indication that the two double lattice sums in (1.23) must be proportional to each other. We shall now present further evidence based on spectral considerations.

The claim that automorphic representations of $E_{d+1}$ for $d \geq 4$ with Bala–Carter type $A_2$ are uniquely represented by Eisenstein series relies on the conjecture that there is no cuspidal automorphic representation associated to such small nilpotent orbits. Recall that cuspidal forms are by definition exponentially suppressed at all cusps, and as such admit Fourier coefficients that are themselves exponentially suppressed at all cusps of the corresponding Levi subgroup. For the nilpotent orbit associated to the adjoint node $2\Lambda_H$, the generic Fourier coefficients in the Heisenberg parabolic $P_H$ saturate the Gelfand–Kirillov dimension and are functions of the Levi subgroup element $v \in L_H$ acting on the Fourier charge $v(Q)$. For exceptional groups, the following stabilisers $H_Q \subset L_H \subset E_{d+1}$ occur for generic charges $Q$

$$
\begin{aligned}
SO(1,1) \times SO(2,2)\,, \;\; SO(2) \times SO(1,3) \subset SL(2) \times SO(3,3) &\quad \subset Spin(5,5)\,, \\
SL(3) \times SL(3)\,, \;\; SL(3,\mathbb{C}) \subset SL(6) &\quad \subset E_{6(6)}\,, \\
SL(6)\,, \;\; SU(3,3) \subset Spin(6,6) &\quad \subset E_{7(7)}\,, \\
E_{6(6)}\,, \;\; E_{6(2)} \subset E_{7(7)} &\quad \subset E_{8(8)}\,. \quad (2.15)
\end{aligned}
$$

The stabilisers are all non-compact. It follows that the Fourier coefficients as a function of $v(Q)$ are constant along all the cusps of the stabiliser $H_Q$, and therefore cannot be cuspidal.

Applying this reasoning to orthogonal groups of type $SO(2n, 2n)$, one concludes that the first possible cuspidal representation can only appear for the nilpotent orbit of weight $2\Lambda_n$, for which the stabiliser of generic charges $SO(n) \times SO(n)$ is compact, in agreement with the conjecture in [43]. For exceptional groups one predicts in this way that cuspidal representations can only appear for higher dimensional nilpotent orbits, like the nilpotent orbit of weight $2\Lambda_2$ of type $A_2 + 3A_1$ for $E_7$ for example.

## 2.4 Integrating against cusp forms and against Eisenstein series

In order to prove the identity (1.23), we shall now integrate both sides against an arbitrary Maaß eigenform $f(\tau)$ that is annihilated by $\Delta_\tau - s(s-1)$ and an eigenmode of all Hecke operators $H_N : f(\tau) \mapsto \sum_{kp=N, 0 \le j < k} f(\frac{p\tau+j}{k})$. To avoid regularisation issues, we first consider the case where $f$ is a cusp form for $GL(2, \mathbb{Z})$, and then discuss the case of an Eisenstein series.

Starting with the string multiplet sum, we consider, for $\mathrm{Re}\,(r')$ large enough and $f$ a cusp form,

$$\mathcal{I}^{E_{d+1}}_{\Lambda_1}(f, r') \equiv \int_{\mathcal{F}} \frac{\mathrm{d}\tau_1 \mathrm{d}\tau_2}{\tau_2^2} f(\tau) \, \Xi^{E_{d+1}}_{\Lambda_1}(\tau, r') \,, \tag{2.16}$$

where $\mathcal{F}$ is the fundamental domain for $PGL(2, \mathbb{Z})$ defined below (1.14). Using (2.7), we rewrite $\Xi^{E_{d+1}}_{\Lambda_1}(\tau, r')$ as a sum over $\gamma \in P_3 \backslash E_{d+1}$ and over non-zero $2 \times 2$ matrices $M$. Restoring the integral over the volume factor, we get

$$\mathcal{I}^{E_{d+1}}_{\Lambda_1}(f, r') = \sum_{\gamma \in P_3 \backslash E_{d+1}} \left[ \int_0^\infty \frac{\mathrm{d}V}{V^{r'+1}} \int_{\mathcal{F}} \frac{\mathrm{d}\tau_1 \mathrm{d}\tau_2}{\tau_2^2} f(\tau) \sideset{}{'}\sum_{M \in \mathbb{Z}^{2\times 2}} e^{-\frac{\pi y_3}{V} \mathrm{Tr}[\mathcal{T}M\mathcal{U}M^{\mathsf{T}}]} \right] \Bigg|_\gamma \tag{2.17}$$

where

$$\mathcal{T} = \frac{1}{\tau_2} \begin{pmatrix} |\tau|^2 & -\tau_1 \\ -\tau_1 & 1 \end{pmatrix} \,, \quad \mathcal{U} = \frac{1}{U_2} \begin{pmatrix} 1 & U_1 \\ U_1 & |U|^2 \end{pmatrix} \,, \tag{2.18}$$

such that $(y_3, U)$ parametrise the $GL(2)$ factor in $P_3$. The integral over $\mathcal{F}$ can then be unfolded using the orbit method as in [57]. For cusp forms, the rank-one orbit does not contribute[14] and the sum over rank-two matrices can be restricted to summing over $M = \begin{pmatrix} k & j \\ 0 & p \end{pmatrix}$ with $0 \le j < k$, $k, p \ne 0$, provided the integral over $\tau$ is extended to the upper half plane. For fixed $N = kp$ with $k, p > 0$, the sum over $k, p, j$ is recognised as the action of the Hecke operator acting on modular functions in the $U$ variable, $H_N[f](U) = N^{-1/2} \sum_{k,p>0, kp=N} \sum_{j \bmod p} f(\frac{kU+j}{p})$. Thus, we get

$$\mathcal{I}^{E_{d+1}}_{\Lambda_1}(f, r) = 2 \sum_{\gamma \in P_3 \backslash E_{d+1}} \left\{ \sum_{N>0} H_N \left[ \int_0^\infty \frac{\mathrm{d}V}{V^{r'+1}} \int_{\mathcal{H}_1} \frac{\mathrm{d}\tau_1 \mathrm{d}\tau_2}{\tau_2^2} f(\tau) e^{-\frac{\pi N y_3 |\tau - U|^2}{V \tau_2 U_2} - \frac{2\pi y_3 N}{V}} \right] \right\} \Bigg|_\gamma \tag{2.19}$$

Now, we use the fact that $e^{-\pi t |\tau - U|^2/(\tau_2 U_2)}$ acts as a reproducing kernel on eigenmodes of $\Delta_\tau$ [58, 59]. More precisely, for any smooth solution of $[\Delta_\tau - s(s-1)]f(\tau) = 0$,

$$\int_{\mathcal{H}_1} \frac{\mathrm{d}\tau_1 \mathrm{d}\tau_2}{\tau_2^2} f(\tau) e^{-\frac{\pi t |\tau - U|^2}{\tau_2 U_2}} = \mathcal{N}(s, t) f(U) \,, \tag{2.20}$$

---

[14]For $f$ an Eisenstein series, the rank-one orbit gives cut-off dependent contributions which do not contribute to the renormalised integral for generic $s$.

where the factor $\mathcal{N}(s,t)$ is independent of $f$. This factor can be computed by choosing $f(\tau) = \tau_2^s$:

$$
\mathcal{N}(s,t) = \int_{\mathcal{H}_1} \frac{d\tau_1 d\tau_2}{\tau_2^2} \left(\frac{\tau_2}{U_2}\right)^s e^{-\frac{\pi t|\tau - U|^2}{\tau_2 U_2}} = \frac{U_2^{\frac{1}{2}-s}}{\sqrt{t}} \int_0^\infty \frac{d\tau_2}{\tau_2^{3/2}} e^{-\frac{\pi t}{\tau_2 U_2} - \frac{\pi t U_2}{\tau_2} + 2\pi t}
$$
$$
= \frac{2}{\sqrt{t}} K_{s-\frac{1}{2}}(2\pi t)\, e^{2\pi t}\,, \tag{2.21}
$$

where $K_\nu(x)$ is the modified Bessel function of the second kind. Setting $t = N y_3/V$, we thus get

$$
\mathcal{I}_{\Lambda_1}^{E_{d+1}}(f, r') = 4 \sum_{\gamma \in P_3 \backslash E_{d+1}} \left[ \sum_{N>0} \frac{1}{\sqrt{y_3 N}} \int_0^\infty \frac{dV}{V^{r'+\frac{1}{2}}} K_{s-\frac{1}{2}}\left(\frac{2\pi y_3 N}{V}\right) H_N^{(U)} f(U) \right]\bigg|_\gamma. \tag{2.22}
$$

The integral over $V$ can now be computed using

$$
\int_0^\infty \frac{dt}{t^{1-s'}} K_{s-\frac{1}{2}}(2\pi t) = \frac{\pi^{-s'}}{4} \Gamma\!\left(\frac{s'-s}{2} + \frac{1}{4}\right) \Gamma\!\left(\frac{s+s'}{2} - \frac{1}{4}\right), \tag{2.23}
$$

which is valid whenever $\mathrm{Re}\,(s') > |\mathrm{Re}\,(s - \frac{1}{2})|$. As for the action of the Hecke operator $H_N$, its action on the Fourier expansion

$$
f(\tau) = f_0\, \tau_2^s + f_0'\, \tau_2^{1-s} + \sum_{n>0} f_n\, \sqrt{2\pi\tau_2}\, K_{s-\frac{1}{2}}(2\pi|n|\tau_2)\, e^{2\pi n \tau_1} \tag{2.24}
$$

sends $f_n \mapsto \sqrt{N} \sum_{d|(n,N)} d^{-1} f_{nN/d^2}$. From looking at the first mode with $n = 1$, it follows that $H_N f(\tau) = \sqrt{N}\, f_N\, f(\tau)/f_1$ if $f(\tau)$ is a cuspidal Hecke eigenmode (i.e. $f_0 = f_0' = 0, f_1 \neq 0$). In this way, setting $s' = r' - \frac{1}{2}$ and assuming that $\mathrm{Re}\,(r')$ is large enough such that $1 - \mathrm{Re}\,(r) < \mathrm{Re}\,(s) < \mathrm{Re}\,(r)$, we arrive at

$$
\mathcal{I}_{\Lambda_1}^{E_{d+1}}(f, r') = \pi^{\frac{1}{2}-r'} \Gamma\!\left(\frac{r'-s}{2}\right) \Gamma\!\left(\frac{r'+s-1}{2}\right) \sum_{N>0} \frac{f_N}{f_1} N^{\frac{1}{2}-r'} \sum_{\gamma \in P_3 \backslash E_{d+1}} \left[ y_3^{-r'} f(U) \right]\bigg|_\gamma. \tag{2.25}
$$

Recalling the definition of the completed L-series associated to $f$ [60],

$$
L^\star(f, r) = \pi^{-r} \Gamma\!\left(\frac{r+s}{2} - \frac{1}{4}\right) \Gamma\!\left(\frac{r-s}{2} + \frac{1}{4}\right) \sum_{N>0} \frac{f_N}{f_1} N^{-r}\,, \tag{2.26}
$$

normalised such that $L^\star(f, 1-r)$ is equal to $L^\star(f, r)$ up to a phase, we get

$$
\mathcal{I}_{\Lambda_1}^{E_{d+1}}(f, r') = L^\star(f, r' - \tfrac{1}{2}) \sum_{\gamma \in P_3 \backslash E_{d+1}} \left[ y_3^{-r'} f(U) \right]\bigg|_\gamma. \tag{2.27}
$$

The right-hand side is recognised as an Eisenstein series induced from the cusp form $f(U)$ on the $GL(2)$ factor in the maximal parabolic subgroup $P_3$ with Levi subgroup $GL(2) \times SL(d)$.

If we now take for $f$ the non-holomorphic Eisenstein series $E_{s\Lambda_1}^{SL(2)}$, the same computation goes through, except that the rank-one orbit gets cut-off dependent coefficients from the constant terms in the Fourier expansion

$$
E_{s\Lambda_1}^{SL(2)}(\tau) = \tau_2^s + \frac{\xi(2s-1)}{\xi(2s)} \tau_2^{1-s} + \frac{2\tau_2^{1/2}}{\xi(2s)} \sum_{N \neq 0} |N|^{s-\frac{1}{2}} \sigma_{1-2s}(|N|)\, K_{s-\frac{1}{2}}(2\pi|N|\tau_2)\, e^{2\pi i N \tau_1}\,.
$$
$$\tag{2.28}$$

For $r'$ large enough, these terms vanish as the cut-off is removed, and the rank-two orbit picks up contributions from the non-zero Fourier coefficients in (2.28). Using Ramanujan's identity

$$\sum_{N=1}^{\infty} N^{s-s'-\frac{1}{2}} \sigma_{1-2s}(N) = \zeta(s+s'-\tfrac{1}{2})\,\zeta(s'-s+\tfrac{1}{2})\;, \tag{2.29}$$

one finds the L-series associated to $E_{s\Lambda_1}^{SL(2)}$,

$$L^{\star}(E_{s\Lambda_1}^{SL(2)}, r) = \xi(r+s-\tfrac{1}{2})\,\xi(r-s+\tfrac{1}{2}) \tag{2.30}$$

leading to a Langlands–Eisenstein series,

$$\begin{aligned}
\mathcal{I}_{\Lambda_1}^{E_{d+1}}(E_{s\Lambda_1}^{SL(2)}, r') &= \xi(r'-s)\,\xi(r'+s-1) \sum_{\gamma \in P_3 \backslash E_{d+1}} \left[ y_3^{-r'}\, E_{s\Lambda_1}^{SL(2)}(U) \right]\Big|_{\gamma} \\
&= \xi(r'-s)\,\xi(r'+s-1)\, E_{s\Lambda_1+\frac{r'-s}{2}\Lambda_3}^{E_{d+1}} \;.
\end{aligned} \tag{2.31}$$

We now turn to the particle multiplet sum. Using the same reasoning, the integral

$$\mathcal{I}_{\Lambda_{d+1}}^{E_{d+1}}(f, r) = \int_{\mathcal{F}} \frac{\mathrm{d}\tau_1 \mathrm{d}\tau_2}{\tau_2^2}\, f(\tau)\, \Xi_{\Lambda_{d+1}}^{E_{d+1}}(\tau, r) \tag{2.32}$$

for $f$ a normalised Hecke eigenform evaluates to

$$\mathcal{I}_{\Lambda_{d+1}}^{E_{d+1}}(f, r) = L^{\star}(f, r-\tfrac{1}{2}) \sum_{\gamma \in P_d \backslash E_{d+1}} \left[ y_d^{-r}\, f(U) \right]\Big|_{\gamma} \tag{2.33}$$

or, for $f = E_{s\Lambda_1}^{SL(2)}$,

$$\mathcal{I}_{\Lambda_{d+1}}^{E_{d+1}}(E_{s\Lambda_1}^{SL(2)}, r) = \xi(r-s)\,\xi(r+s-1)\, E_{s\Lambda_{d+1}+\frac{r-s}{2}\Lambda_d}^{E_{d+1}} \;. \tag{2.34}$$

Note that we use the Bourbaki labelling of $E_{d+1}$, with the slight abuse of notation that $\Lambda_d$ corresponds to a sum of fundamental weights for $d \leq 3$ as used in [1] to allow for general formulae. In particular, for $d = 3$, the weights $\Lambda_{d+1}$ and $\Lambda_d$ in $E_4$ correspond to $\Lambda_3$ and $\Lambda_2 + \Lambda_4$ in $A_4$.

## 2.5 Relating the particle, string multiplet and adjoint Eisenstein series

In order to relate the Eisenstein series (2.31) and (2.34), we use the general functional relation for Langlands–Eisenstein series with infinitesimal weight parameter $2\lambda - \rho$,

$$E_{\lambda}^{G} = M(w, 2\lambda - \rho)\, E_{w(\lambda-\frac{\rho}{2})+\frac{\rho}{2}}^{G} \tag{2.35}$$

for any element $w$ of the Weyl group [61] (see [24] for an exposition targeted at physicists). Here, $\rho$ is the Weyl vector and the prefactor $M(w, \lambda)$, known as the intertwiner (between different

principal series representations), is given by a product over positive roots that are reflected into negative roots under $w$:

$$M(w, \lambda) = \prod_{\substack{\alpha > 0 \\ w\alpha < 0}} \frac{\xi(\langle \alpha, \lambda \rangle)}{\xi(1 + \langle \alpha, \lambda \rangle)} \,. \tag{2.36}$$

Using suitable Weyl elements[15] we find for $d \geq 3$ that (2.31) and (2.34) coincide for $r = d - 2$, $r' = 3$,

$$\xi(s - d + 3)\,\xi(4 - d - s)\, E^{E_{d+1}}_{s\Lambda_{d+1} + \frac{d-2-s}{2}\Lambda_d} = \xi(3 - s)\,\xi(2 + s)\, E^{E_{d+1}}_{s\Lambda_1 + \frac{3-s}{2}\Lambda_3} \tag{2.37}$$

hence confirming the relation (1.23). In the following, we shall also need a dimensionally regularised version of the Eisenstein series in (2.37) with $d_\epsilon = d + 2\epsilon$, which satisfy the modified identity

$$\xi(s - d_\epsilon + 3)\,\xi(4 - d_\epsilon - s)\, E^{E_{d+1}}_{s\Lambda_{d+1} + \frac{d_\epsilon - 2 - s}{2}\Lambda_d} = \xi(3 - s - 2\epsilon)\,\xi(2 + s - 2\epsilon)\, E^{E_{d+1}}_{s\Lambda_1 + 2\epsilon\Lambda_2 + \frac{3-s-2\epsilon}{2}\Lambda_3} \,. \tag{2.38}$$

When $f$ is an $SL(2)$ cusp form, the expressions (2.27) and (2.33) describe more general Eisenstein series induced from cusp forms on the parabolic subgroups $P_3$ and $P_{d-1}$, respectively. Langlands has also provided a functional relation for this case [61], see also [62, 63], and the intertwiner now depends on the cusp form $f$ as well as on $\lambda$. For the case of $SL(2)$ it evaluates to the corresponding quotient of completed $L$-functions [63], implying the equality

$$\mathcal{I}^{E_{d+1}}_{\Lambda_{d+1}}(f, d - 2) = \mathcal{I}^{E_{d+1}}_{\Lambda_1}(f, 3) \tag{2.39}$$

for all cusp forms. This completes the proof of (1.23).

It is also interesting to note that the same functional equation (2.35) also allows to rewrite either side of (2.37) as an Eisenstein series $E^{E_8}_{\frac{s+14}{2}\Lambda_8}$, $E^{E_7}_{\frac{s+8}{2}\Lambda_1}$, $E^{E_6}_{\frac{s+5}{2}\Lambda_2}$, $E^{D_5}_{\frac{s+3}{2}\Lambda_2}$ for the adjoint representation, *i.e.* induced from the Heisenberg parabolic, in agreement with the discussion of the last subsection:

$$\mathcal{I}^{E_{d+1}}_{\Lambda_{d+1}}\big(E^{SL(2)}_{s\Lambda_1}, d - 2\big) = \xi(s - d + 3)\,\xi(s + d - 3) E^{E_{d+1}}_{s\Lambda_{d+1} + \frac{d-2-s}{2}\Lambda_d}$$

$$= \frac{\xi(s - 4 + 2s_{d+1})\xi(s - d - 1 - \delta_{d,7} + 2s_{d+1})\xi(s + d - 3 + \delta_{d,7})}{\xi(s)} E^{E_{d+1}}_{(\frac{s-4}{2} + s_{d+1})\Lambda_H} \tag{2.40}$$

where $s_{d+1} = \frac{7}{2}, \frac{9}{2}, 6, 9$ for $d = 4, 5, 6, 7$, respectively.[16]

---

[15]For the values $d = 3, 4, 5, 6, 7$ we use $w_{A_4} = w_2 w_1 w_3 w_2$, $w_{D_5} = w_3 w_2 w_1 w_5 w_3 w_2$, $w_{E_6} = w_5 w_4 w_3 w_1 w_6 w_5 w_4 w_3$, $w_{E_7} = w_6 w_5 w_4 w_3 w_1 w_7 w_6 w_5 w_4 w_3$ and $w_{E_8} = w_7 w_6 w_5 w_4 w_3 w_1 w_8 w_7 w_6 w_5 w_4 w_3$, respectively. Recall that $s\Lambda_{d+1} + \frac{d-2-s}{2}\Lambda_d$ for $E_4$ is $s\Lambda_3 + \frac{d-2-s}{2}(\Lambda_2 + \Lambda_4)$ in the $A_4$ basis.

[16]Here, we have used the following Weyl elements for the cases $d = 4, 5, 6, 7$: $w_{D_5} = w_2 w_1 w_3 w_2 w_4 w_3$, $w_{E_6} = w_2 w_4 w_3 w_1 w_5 w_4 w_2 w_3 w_4 w_5$, $w_{E_7} = w_1 w_3 w_4 w_2 w_5 w_4 w_3 w_1 w_6 w_5 w_4 w_2 w_3 w_4 w_5 w_6$ and finally $w_{E_8} = w_8 w_7 w_6 w_5 w_4 w_2 w_3 w_1 w_4 w_3 w_5 w_4 w_2 w_6 w_5 w_4 w_3 w_1 w_7 w_6 w_5 w_4 w_2 w_3 w_4 w_5 w_6 w_7$.

## 2.6 Poincaré series representations

For the case $f(\tau) = A(\tau)$ of (1.15), the identity (2.20) is no longer valid, due to the non-differentiability of $A(\tau)$ on the locus $\tau_1 = 0$ and its images under $GL(2, \mathbb{Z})$. Moreover, $A(\tau)$ is not an eigenmode of Hecke operators. Nevertheless, the manipulation in (2.17) and its analogue for $\mathcal{I}_{\Lambda_{d+1}}^{E_{d+1}}(A, r)$ are still valid, and lead to the Poincaré series representations

$$\mathcal{I}_{\Lambda_{d+1}}^{E_{d+1}}(A, r) = \sum_{\gamma \in P_d \backslash E_{d+1}} \left[ y_d^{-r} \, \widetilde{A}_{\frac{r}{2}}(U) \right] \Big|_\gamma \tag{2.41}$$

and

$$\mathcal{I}_{\Lambda_1}^{E_{d+1}}(A, r') = \sum_{\gamma \in P_3 \backslash E_{d+1}} \left[ y_3^{-r'} \, \widetilde{A}_{\frac{r'}{2}}(U) \right] \Big|_\gamma , \tag{2.42}$$

where $\widetilde{A}_s(U)$ is the $SL(2, \mathbb{Z})$-invariant function

$$\widetilde{A}_s(U) \equiv \int_0^\infty \frac{dV}{V^{1+2s}} \int_{\mathcal{F}} \frac{d\tau_1 d\tau_2}{\tau_2^2} A(\tau) \sum_{M \in \mathbb{Z}^{2 \times 2}}' e^{-\frac{\pi}{V} \text{Tr}[\mathcal{T} M \mathcal{U} M^\intercal]} . \tag{2.43}$$

This satisfies the differential equation

$$\Delta \widetilde{A}_s(U) = 12 \widetilde{A}_s(U) - 6 \left( \xi(2s) E_{s\Lambda_1}^{SL(2)}(U) \right)^2 . \tag{2.44}$$

For the case of interest for the $\nabla^6 \mathcal{R}^4$ coupling, using the identity between the particle and string multiplet sums, we get

$$\mathcal{E}_{(0,1)}^{(d),\text{ExFT}} \underset{d \geq 2}{=} \frac{8\pi^2}{3} \sum_{\gamma \in P_d \backslash E_{d+1}} \left[ y_d^{2-d} \, \widetilde{A}_{\frac{d-2}{2}}(U) \right] \Big|_\gamma \underset{d \geq 3}{=} \frac{8\pi^2}{3} \sum_{\gamma \in P_3 \backslash E_{d+1}} \left[ y_3^{-3} \, \widetilde{A}_{\frac{3}{2}}(U) \right] \Big|_\gamma \tag{2.45}$$

This identity is formal however, since the value of $r$ typically corresponds to a pole. The function $\frac{8\pi^2}{3} \widetilde{A}_{\frac{3}{2}}(U)$ corresponds to the exact $\nabla^6 \mathcal{R}^4$ coupling in ten-dimensional type IIB string theory in the form given in [32].

## 2.7 Convergence

To determine the domain of convergence of the double Epstein series $\Xi_{\Lambda_{d+1}}^{E_{d+1}}(\phi, \tau, r)$ for the particle multiplet, let us insert an additional regulating power of $y_d^{-\tilde{\epsilon}}$ in the sum, and assume that $\text{Re}(r)$ and $\text{Re}(\tilde{\epsilon})$ are both large enough such that the second term in (2.3) can be combined with the first by allowing all matrices with $\text{rk} \, M \geq 1$. We can then perform a Poisson resummation on the second row of $M$ and obtain the Fourier expansion with respect to $\tau_1$,

$$\pi^{-r} \Gamma(r) \sum_{\gamma \in P_d \backslash E_{d+1}} \left( y_d^{-\tilde{\epsilon}} \sum_{M \in \mathbb{Z}^{2 \times 2}}' \left[ \frac{\tau_2 U_2 \, y_d^{-1}}{|(1, \tau) M (1, U)|^2 + 2\tau_2 U_2 \det M} \right]^r \right) \Big|_\gamma \tag{2.46}$$

$$= \xi(2r) \tau_2^r E_{r\Lambda_{d+1} + \frac{\tilde{\epsilon}}{2} \Lambda_d}^{E_{d+1}} + \xi(2r - 2) \tau_2^{2-r} E_{\frac{1+\tilde{\epsilon}}{2} \Lambda_d + (r-1) \Lambda_{d+1}}^{E_{d+1}}$$

$$+4\tau_2 \sum_{\gamma\in P_d\backslash E_{d+1}} \left(y_d^{-\tilde\epsilon-r} \sideset{}{'}\sum_{m,n} \frac{\sigma_{r-1}(\gcd(m,n))}{\gcd(m,n)^{2r-2}} K_{r-1}(2\pi\tau_2 \tfrac{|mU+n|^2}{U_2})\right)\bigg|_\gamma$$

$$+2\tau_2 \sum_{\gamma\in P_d\backslash E_{d+1}} \left(y_d^{-\tilde\epsilon-r} \sideset{}{'}\sum_{m_1,m_2} \sideset{}{'}\sum_{n_1,n_2} \frac{|m_1+Um_2|^{r-1}}{|n_1+Un_2|^{r-1}} K_{r-1}\left(2\pi\tau_2 \tfrac{|m_1+m_2U||n_1+n_2U|}{U_2}\right) e^{2\pi i\tau_1(m_1n_2-m_2n_1)}\right)\bigg|_\gamma$$

By Godement's criterion [64, 24], the first term $E^{E_{d+1}}_{r\Lambda_{d+1}+\frac{\tilde\epsilon}{2}\Lambda_d}$ in the limit $\tilde\epsilon\to 0$ converges for $\mathrm{Re}\,(r) > 4, 6, 9, \frac{29}{2}$ when $d = 4, 5, 6, 7$. The second term $E^{E_{d+1}}_{\frac{1+\tilde\epsilon}{2}\Lambda_d+(r-1)\Lambda_{d+1}}$ never converges when $\tilde\epsilon\to 0$, but its analytic continuation at $\tilde\epsilon = 0$ can be shown to vanish. Thus, we conclude that the double Epstein series $\Xi^{E_{d+1}}_{\Lambda_{d+1}}(\phi,\tau,r)$ has no pole for $\mathrm{Re}\,(r) > 4, 6, 9, \frac{29}{2}$, respectively, which indicates that it is absolutely convergent in the same range. Similarly, we find that the double Epstein series $\Xi^{E_{d+1}}_{\Lambda_1}(\phi,\tau,r')$ for the string multiplet converges absolutely for $\mathrm{Re}\,(r') > 4, 6, \frac{17}{2}, \frac{23}{2}$.

## 2.8 From vector to spinor double lattice sums

In this section, we generalise the observation on the equivalence of different lattice sums from $E_{d+1}$ to $Spin(d,d)$, as this will be used in our later analysis. Using the same techniques, one can establish the relation between the double lattice sums in vector and spinor representations of $Spin(d,d)$ with $d \geq 3$,

$$\frac{\Gamma(d-2)}{\pi^{d-2}} \sideset{}{'}\sum_{\substack{Q_i\in II_{d,d}\\ Q_i\times Q_j=0}} \left[\frac{\tau_2}{g(Q_1+\tau Q_2, Q_1+\bar\tau Q_2)}\right]^{d-2} = \frac{\Gamma(2)}{\pi^2} \sideset{}{'}\sum_{\substack{Q_i\in S_\pm\\ Q_i\gamma_{d-4}Q_j=0}} \left[\frac{\tau_2}{g(Q_1+\tau Q_2, Q_1+\bar\tau Q_2)}\right]^2 \quad (2.47)$$

where $II_{d,d}$ is the even self-dual lattice in the vector representation of $Spin(d,d)$, and $S_\pm$ are the lattices in the Weyl spinor prepresentation of $Spin(d,d)$, for either chirality. More precisely, using the same notation as in (1.20),

$$\lim_{r\to d-2} \Xi^{D_d}_{\Lambda_1}(\tau,r) = \lim_{r'\to 2} \Xi^{D_d}_{\Lambda_d}(\tau,r') = \lim_{r'\to 2} \Xi^{D_d}_{\Lambda_{d-1}}(\tau,r') . \quad (2.48)$$

For $d = 5$, this reduces to the identity (1.23) for $G = SO(5,5)$. For $d = 4$, it expresses invariance under triality of $SO(4,4)$. As a consistency check, note that the differential equations satisfied by the two Epstein series

$$[\Delta_{D_d} - \Delta_\tau - r(r+3-2d)]\,\Xi^{D_d}_{\Lambda_1}(\tau,r) = 0 \quad (2.49)$$

$$[\Delta_{D_d} - \Delta_\tau - \tfrac{1}{2}(d-2)r'(r'-d-1)]\,\Xi^{D_d}_{\Lambda_d}(\tau,r') = 0 , \quad (2.50)$$

agree for $(r,r') = (d-2,2)$. Integrating both sides against the Eisenstein series $E^{SL(2)}_{s\Lambda_1}(\tau)$, one gets

$$\xi(d-2-s)\,\xi(d+s-3)\,E^{D_d}_{s\Lambda_1+\frac{d-2-s}{2}\Lambda_2} = \xi(2-s)\,\xi(s+1)\,E^{D_d}_{s\Lambda_d+\frac{2-s}{2}\Lambda_{d-2}} \quad (2.51)$$

where the equality follows from Langlands' functional relation (2.35). It is worth noting that these series are related by functional equations to the adjoint Eisenstein series $E^{D_d}_{\frac{s+d-2}{2}\Lambda_2}$. A

similar functional identity should hold for Eisenstein series induced from $SL(2)$ cusp forms $f$ on parabolic subgroups $P_2$ and $P_{d-2}$, namely

$$\mathcal{I}^{D_d}_{\Lambda_1}(f, d-2) = \mathcal{I}^{D_d}_{\Lambda_d}(f, 2) = \mathcal{I}^{D_d}_{\Lambda_{d-1}}(f, 2) \, . \tag{2.52}$$

Assuming the relation (2.47) as well as the Poincaré series representation (1.21), we obtain several equivalent ways of expressing the modular integral of the product of $\varphi_{\mathrm{KZ}}$ with the Siegel–Narain lattice sum,[17]

$$
\begin{aligned}
\int_{\mathcal{F}_2} \frac{\mathrm{d}^6\Omega}{|\Omega_2|^3} \varphi_{\mathrm{KZ}} \, \Gamma_{d,d,2} &= \int_{\mathcal{G}} \frac{\mathrm{d}^3\Omega_2}{|\Omega_2|^{3-\frac{d}{2}}} \varphi_{\mathrm{KZ}}^{\mathrm{tr}}(\Omega_2) \, \theta^{D_d}_{\Lambda_1} \\
&= \int_{\mathcal{G}} \frac{\mathrm{d}^3\Omega_2}{|\Omega_2|} \varphi_{\mathrm{KZ}}^{\mathrm{tr}}(\Omega_2) \, \theta^{D_d}_{\Lambda_d} = \int_{\mathcal{G}} \frac{\mathrm{d}^3\Omega_2}{|\Omega_2|} \varphi_{\mathrm{KZ}}^{\mathrm{tr}}(\Omega_2) \, \theta^{D_d}_{\Lambda_{d-1}}
\end{aligned}
\tag{2.53}
$$

where $\theta^{D_d}_{\Lambda_k}$ is defined as in (1.18) and $\mathcal{G} = \mathbb{R}^+ \times \mathcal{F}$. In Appendix C, we study the asymptotics of the various integrals and find further support for the relations (2.53), hence for the Poincaré series representation (1.21).

# 3 Weak coupling limit

In this section, we study the weak coupling limit of the integral (1.17). We first discuss the expected form of the expansion, known from general physical considerations, before turning to a detailed analysis of the constrained lattice sum $\theta^{E_{d+1}}_{\Lambda_{d+1}}$ of (1.18) entering in (1.17).

## 3.1 Expectation

The weak coupling limit (1.1) of the exact non-perturbative $\nabla^6 \mathcal{R}^4$ coupling (which is invariant under the U-duality group $E_{d+1}(\mathbb{Z})$) in generic dimension $D = 10 - d$ takes the form

$$\mathcal{E}^{(d)}_{(0,1)} = g_D^{\frac{2d+8}{d-8}} \left[ \frac{\frac{2}{3}\zeta(3)^2}{g_D^2} + \mathcal{E}^{(d,1)}_{(0,1)} + g_D^2 \, \mathcal{E}^{(d,2)}_{(0,1)} + g_D^4 \mathcal{E}^{(d,3)}_{(0,1)} + \mathcal{O}(e^{-1/g_D}) \right] \tag{3.1}$$

where $2\zeta(3)^2/3g_D^2$ is the tree-level contribution while the genus one, genus two and genus three contributions are given by [48, §2.1.1][18]

$$\mathcal{E}^{(d,1)}_{(0,1)} = \frac{4\pi\zeta(3)}{3} \xi(d-2) E^{D_d}_{\frac{d-2}{2}\Lambda_1} + \frac{8\pi^4}{567} \xi(d+4) E^{D_d}_{\frac{d+4}{2}\Lambda_1} \tag{3.2}$$

$$\mathcal{E}^{(d,2)}_{(0,1)} = 8\pi \int_{\mathcal{F}_2} \frac{\mathrm{d}^6\Omega}{|\Omega_2|^3} \varphi_{\mathrm{KZ}}(\Omega) \, \Gamma_{d,d,2} \tag{3.3}$$

$$\mathcal{E}^{(d,3)}_{(0,1)} = \frac{4\zeta(6)}{27} \left( \widehat{E}^{D_d}_{3\Lambda_{d-1}} + \widehat{E}^{D_d}_{3\Lambda_d} \right) \, . \tag{3.4}$$

---

[17]In each of these equations, we assume that a factor $|\Omega_2|^\epsilon$ is inserted in the integral, divergences are subtracted and the limit $\epsilon \to 0$ is taken after analytic continuation.

[18]The Eisenstein series in (3.2) and (3.4) originate from genus-one and genus-three modular integrals, respectively. The genus-two integration measure $\mathrm{d}^6\Omega/|\Omega_2|^3$ in this paper differs by a factor $1/8$ from $\mathrm{d}\mu_2$ in [48].

The exponentially suppressed terms in (3.1) originate from 1/8-BPS instantons, as well as pairs of 1/2-BPS and anti-1/2-BPS instantons, as required by the quadratic source term in the Laplace equation (1.5). In special dimensions where the local $\nabla^6 \mathcal{R}^4$ coupling mixes with the non-local part of the one-particle irreducible effective action, there are also non-analytic terms proportional to $\log g_D$ [8, 48] which we will discuss in more detail in Section 5.2.

In the weak coupling limit, the behaviour of the homogeneous solution (1.11) can be determined using standard constant term formulae [65, 24] to be

$$\mathcal{F}_{(0,1)}^{(d)} = g_D^{-\frac{24}{8-d}+2} \left[ \frac{8\pi^4}{567} \xi(d+4)\, E_{\frac{d+4}{2}\Lambda_1}^{D_d} + g_D^2\, \mathcal{F}_{(0,1)}^{(d,2)} + \frac{4\zeta(6)}{27} g_D^4\, \widehat{E}_{3\Lambda_d}^{D_d} + \mathcal{O}(e^{-1/g_D}) \right]. \tag{3.5}$$

The $\mathcal{O}(g_D^2)$ contribution arises for $d = 5, 6$ only, and corresponds to a two-loop threshold term proportional to $\log g_D$. Such a term is known to arise from the non-analytic part of the string amplitude, after Weyl rescaling to Einstein frame [8]. Substituting this behaviour into (1.13), it follows from the above equation, (3.1) and (1.2) that the two-loop exceptional field theory amplitude must behave as (for $D > 3$)

$$\mathcal{E}_{(0,1)}^{(d),\text{ExFT}} = g_D^{-\frac{24}{8-d}+2} \left[ \frac{2\zeta(3)^2}{3g_D^2} + \frac{4\pi\zeta(3)}{3} \xi(d-2)\, E_{\frac{d-2}{2}\Lambda_1}^{D_d} + g_D^2\, \mathcal{E}_{(0,1)}^{(d,2)} + \frac{4\zeta(6)}{27} g_D^4\, \widehat{E}_{3\Lambda_{d-1}}^{D_d} + \mathcal{O}(e^{-\frac{1}{g_D}}) \right] \tag{3.6}$$

up to logarithmic corrections discussed in Section 5.2. The three-loop amplitude is invariant under the outer automorphism of $D_d$ which exchanges the two spinor nodes due to the fact the four-graviton amplitudes in type IIA and type IIB are the same up to order $\nabla^8 \mathcal{R}^4$ [66]. The constituent functions (3.5) and (3.6) are not invariant individually under this exchange since they involve the two distinct spinor series associated to the fundamental weights $\Lambda_d$ and $\Lambda_{d-1}$, respectively.

## 3.2 Weak coupling limit of the particle multiplet lattice sum

We are interested in the weak coupling limit of the integral (1.17), which, after subtraction of the divergent power law $L$-dependent terms we denote by R.N., reads

$$\mathcal{I}_d(\phi, \epsilon) = 8\pi\, \text{R.N.} \int_{\mathcal{G}} \frac{d^3\Omega_2}{|\Omega_2|^{\frac{6-d-2\epsilon}{2}}} \varphi_{\text{KZ}}^{\text{tr}}(\Omega_2)\, \theta_{\Lambda_{d+1}}^{E_{d+1}}(\phi, \Omega_2), \tag{3.7}$$

where $\mathcal{G}$ is the fundamental domain $\mathbb{R}^+ \times \mathcal{F}$ for the action of $PGL(2,\mathbb{Z})$ on $\Omega_2$ (of which the positive Schwinger domain $S_+$ is a six-fold cover). The integral and the sum are absolutely convergent for $\text{Re}(\epsilon)$ large enough. By analyzing the Fourier expansion in this region, we shall find evidence that $\mathcal{I}_d(\phi, \epsilon)$ has a meromorphic continuation to $\epsilon \in \mathbb{C}$, with a pole at $\epsilon = 0$ for $d = 4, 5, 6$. As we explain in Section 5.1, these poles are cancelled by contributions from 1/4-BPS states running in the loops. Since we are interested in the limit $\epsilon \to 0$, we shall retain the $\epsilon$ dependence only when there is a potential pole for some value of $d$.

The theta series $\theta_{\Lambda_{d+1}}^{E_{d+1}}$ involves a sum over pairs of vectors $\Gamma_i$ in the particle multiplet, subject to the constraints $\Gamma_i \times \Gamma_j = 0$ valued in the string multiplet. Under $E_{d+1} \supset GL(1) \times Spin(d,d)$, the particle multiplet representation branches as

$$M_{\Lambda_{d+1}}^{E_{d+1}} \to II_{d,d}^{\left(\frac{2}{8-d}\right)} \oplus \overline{S}_+^{\left(\frac{d-6}{8-d}\right)} \oplus \left[ \wedge^{d-5} II_{d,d} + \wedge^{d-7} II_{d,d} \right]^{\left(\frac{2d-14}{8-d}\right)} \oplus \ldots \tag{3.8}$$

where the superscript denotes the charge under the $GL(1)$ factor, $II_{d,d}$ the even-self-dual lattice in the vector representation, $\wedge^k II_{d,d}$ the lattice in the $k$-th exterior power of the vector representation (which is trivial for $k > 2d$), and $\overline{S}_+$ the Weyl spinor representation lattice.[19] The branching (3.8) is complete for $d \le 6$; for $E_8$ there are additional terms indicated by the ellipses. For $d \le 6$ we denote the components of the charge $\Gamma_i$ of (3.8) by $q_i \in II_{d,d}$, $\chi_i \in \overline{S}_+$ and $N_i \in \wedge^{d-5} II_{d,d}$.

On the other hand, the string multiplet, appearing in the constraint $\Gamma_i \times \Gamma_j = 0$ of the lattice sum, decomposes under $GL(1) \times Spin(d,d)$ as

$$M^{E_{d+1}}_{\Lambda_1} \to \mathbb{Z}^{(\frac{4}{8-d})} \oplus \overline{S}^{(\frac{d-4}{8-d})}_- \oplus \left[\wedge^{d-4} II_{d,d} + \wedge^{d-6} II_{d,d}\right]^{(\frac{2d-12}{8-d})} \oplus \left[\wedge^{d-7} II_{d,d} \otimes \overline{S}_-\right]^{(\frac{3d-20}{8-d})} \oplus \dots , \quad (3.9)$$

where the dots denote additional components that arise only for $d \ge 6$ and play no role in our analysis. Thus, the particle multiplet components $(q_i, \chi_i, N_i)$ along the decomposition (3.8) must satisfy

$$(q_i, q_j) = 0 , \quad q^a_{(i} \gamma_a \chi_{j)} = 0 , \quad q_{(i} \wedge N_{j)} + \chi_{(i} \gamma_{d-4} \chi_{j)} = 0 , \quad q_i \cdot N_j = 0 \quad (3.10)$$

where the last constraint arises only in $d \ge 6$. Here, we have denoted by $\gamma_a$ the gamma matrices of $Spin(d,d)$ and $\gamma_{d-4}$ denotes the antisymmetric product of $d-4$ such gamma matrices. In terms of these components, the quadratic form $G(\Gamma, \Gamma)$ occurring in the double lattice sum $\theta^{E_{d+1}}_{\Lambda_{d+1}}$ of (1.18) can be expressed as

$$G(\Gamma, \Gamma) = g_D^{\frac{4}{8-d}} |v_1(q + a\gamma\chi + (\tfrac{1}{2} a\gamma_{d-4} a + b)N)|^2 + g_D^{2\frac{d-6}{8-d}} |v_2(\chi + aN)|^2 + g_D^{4\frac{d-7}{8-d}} |v_3(N)|^2 , \quad (3.11)$$

where $a \in S_+, b \in \mathbb{R}$ denote the Ramond–Ramond and Neveu–Schwarz axions, respectively, parametrising the unipotent part of the parabolic subgroup $P_{d+1}$ with Levi subgroup $GL(1) \times Spin(d,d)$ (note that $b$ is only present for $d = 6$). The norms $|v_1(q)|^2, |v_2(\chi)|^2, |v_3(N)|^2$ denote the $Spin(d,d)$ invariant quadratic forms in the respective representations, and depend on the $SO(d,d)/(SO(d) \times SO(d))$ moduli parametrising the metric and $B$-field on the torus. To avoid cluttering, we denote all these norms by $|v(\cdot)|^2$. The $\gamma^a$ matrices are integral valued in the canonical null basis associated to the even self-dual lattice $II_{d,d}$ with the normalisation $\{\gamma_a, \gamma_b\} = \eta_{ab}$.

As in [1, 50], we shall split the theta series $\theta^{E_{d+1}}_{\Lambda_{d+1}}$ into contributions where the components $(q_i, \chi_i, N_i)$ along the graded decomposition (3.8) are gradually populated, such that the constraints can be solved explicitly. We shall refer to the gradually populated subsets of charges that arise in this way as 'layers'. We first focus on constant terms, which are independent of the axions $a, b$ and then consider non-trivial Fourier coefficients. A similar analysis for $Spin(d,d)$ lattice sums is presented in Appendix C.

## 1) The first layer

The contribution of the layer with $\chi_i = N_i = 0$ but $q_i \ne 0$ gives

$$\theta^{(1)}_{\Lambda_{d+1}}(\phi, \Omega_2) = \sum_{\substack{q_i \in II_{d,d} \\ (q_i, q_j)=0}}' e^{-\pi \Omega_2^{ij} g_D^{\frac{4}{8-d}} g(q_i, q_j)} . \quad (3.12)$$

---

[19]$\overline{S}_+ \cong S_+$ when $d$ is even and $\overline{S}_+ \cong S_-$ when it is odd. For the corresponding parabolic subgroups, we likewise denote $\overline{P}_d \cong P_d$ for $d$ even, and $\overline{P}_d \cong P_{d-1}$ for $d$ odd.

Integrating against $\varphi_{\mathrm{KZ}}^{\mathrm{tr}}$ in order to obtain the contribution to (3.7) and using the Poincaré series representation (1.21), the domain $\mathcal{G}$ can be folded into the fundamental domain $\mathcal{F}_2$ for $Sp(4,\mathbb{Z})$,

$$
\begin{aligned}
\mathcal{I}_d^{(1)} &:= 8\pi \text{ R.N.} \int_{\mathcal{G}} \frac{\mathrm{d}^3\Omega_2}{|\Omega_2|^{\frac{6-d-2\epsilon}{2}}} \varphi_{\mathrm{KZ}}^{\mathrm{tr}}(\Omega_2)\, \theta_{\Lambda_{d+1}}^{(1)}(\phi,\Omega_2) \\
&= 8\pi g_D^{-\frac{24+8\epsilon}{8-d}+4} \int_{\mathcal{F}_2} \frac{\mathrm{d}^6\Omega}{|\Omega_2|^3} \varphi_{\mathrm{KZ}}^{\epsilon}(\Omega)\, \Gamma_{d,d,2}(\Omega)
\end{aligned}
\tag{3.13}
$$

where

$$
\varphi_{\mathrm{KZ}}^{\epsilon} = |\Omega_2|^{\epsilon}\, \varphi_{\mathrm{KZ}}^{\mathrm{tr}}(\Omega_2)
\tag{3.14}
$$

denotes the Poincaré series seed in (1.21) before taking the limit $\epsilon \to 0$. The expression (3.13) is recognised as the perturbative two-loop contribution (3.3). Note that the Narain partition function $\Gamma_{d,d,2}$ includes the zero vector $q_i = 0$ which is absent in $\theta_{\Lambda_{d+1}}^{(1)}(\phi,\Omega_2)$, but the contribution of this vector is removed by the renormalisation prescription mentioned above and discussed in more detail in Section 5.

## 2) The second layer

The second contribution corresponds to $q_i$ arbitrary, $\chi_i \neq 0$ but linearly dependent $\chi_i \wedge \chi_j = 0$, while $N_i = 0$. For $d \geq 5$, the constraints $\chi_i \gamma_{d-4} \chi_j = 0$ are solved by $\chi_i = n_i \hat{\chi}$ where $\hat{\chi} \in \overline{S}_+$ is a primitive pure spinor *i.e.* $\hat{\chi}\gamma_{d-4}\hat{\chi} = 0$ and such that no integer divides $\hat{\chi}$ (for $d \leq 4$ there are no constraints to solve). The primitive pure spinor $\hat{\chi}$ can always be rotated to a standard form by $Spin(d,d,\mathbb{Z})$ with stabiliser $\overline{P}_d \subset Spin(d,d,\mathbb{Z})$. Therefore, the sum over $\chi_i$ can be written as a Poincaré sum over $\overline{P}_d\backslash Spin(d,d,\mathbb{Z})$ together with a sum over $n_i \in \mathbb{Z}$. Under this parabolic decomposition, $II_{d,d} = \overline{\mathbb{Z}}^d \oplus \mathbb{Z}^d$, and the constraints $(q_i,q_j) = 0$ from (3.10) imply that $q_i \in \mathbb{Z}^d$ so that their $Spin(d,d)$ invariant norm vanishes automatically. Since the sum over $q_i \in \mathbb{Z}^d$ is unconstrained, one can perform a Poisson resummation to obtain

$$
\begin{aligned}
\theta_{\Lambda_{d+1}}^{(2)}(\phi,\Omega_2) &= \sum_{n_i\in\mathbb{Z}}' \sum_{q_i^a\in\mathbb{Z}^d} \sum_{\gamma\in P_d\backslash D_d} \left( e^{-\pi\Omega_2^{ij}g_D^{\frac{4}{8-d}}\left(y^{\frac{4}{d}}u_{ab}(q_i^a+a^an_i)(q_j^b+a^bn_j)+g_D^{-2}y^2 n_i n_j\right)} \right)\bigg|_{\gamma} \\
&= \frac{g_D^{-\frac{4d}{8-d}}}{|\Omega_2|^{\frac{d}{2}}} \sum_{n_i\in\mathbb{Z}}' \sum_{q_a^i\in\mathbb{Z}^d} \sum_{\gamma\in P_d\backslash D_d} \left( y^{-4} e^{-\pi\Omega_2^{ij}g_D^{2\frac{d-6}{8-d}}y^2 n_i n_j - \pi\Omega_{2ij}^{-1}g_D^{-\frac{4}{8-d}}y^{-\frac{4}{d}}u^{ab}q_a^i q_b^j + 2\pi i n_i q_a^i a^a} \right)\bigg|_{\gamma}.
\end{aligned}
\tag{3.15}
$$

Here, $y$ and $u_{ab}$ parametrise the Levi subgroup $GL(d) \subset SO(d,d)$ while the axions $a \in \wedge^2\mathbb{Z}^d$ parametrise the unipotent subgroup within $SO(d,d)$, and $\gamma$ is understood to act on them through the non-linear $SO(d,d)$ action.[20] The scalar $y$ is defined such that $y^{-2s}$ is the canonical character defining the Eisenstein series $E_{s\Lambda_d}^{D_d} = \sum_{\gamma\in P_d\backslash D_d} y^{-2s}$.

The term (3.15) contributes both to constant terms and to Fourier coefficients of (3.7). Constant terms may come from a) from $q_a^i = 0$ or b) from $n_i q_a^i = 0$ and $q_a^i \neq 0$. The contribution

---

[20]The axion $a$ is not to be confused with the summation index $a = 1,\ldots,d$.

from a) $q_a^i = 0$ diverges at $\epsilon = 0$, but it can be obtained by analytic continuation in $\epsilon$ as above to give

$$\mathcal{I}_d^{(2a)} = 8\pi g_D^{-\frac{24}{8-d}} \sum_{\gamma \in P_d \backslash D_d} \left( y^{-2\epsilon} \int_{\mathcal{G}} \frac{\mathrm{d}^3 \Omega_2}{|\Omega_2|^{3-\epsilon}} \varphi_{\mathrm{KZ}}^{\mathrm{tr}}(\Omega_2) \sideset{}{'}\sum_{n_i \in \mathbb{Z}} e^{-\pi \Omega_2^{ij} n_i n_j} \right)\Bigg|_\gamma . \tag{3.16}$$

After integrating over the volume factor $V$ using the parametrisation (1.14), the sum over $n_i$ produces an Eisenstein series $E_{(2\epsilon-2)\Lambda_1}^{SL(2)}(\tau)$. The remaining integral over $\tau$ can be computed using the following formula, that we establish in Appendix B,

$$\mathrm{R.N.} \int_{\mathcal{F}} \frac{\mathrm{d}\tau_1 \mathrm{d}\tau_2}{\tau_2^2} A(\tau) \, E_{s\Lambda_1}^{SL(2)}(\tau) = \frac{3 \, [\xi(s)]^2}{[12 - s(s-1)]\xi(2s)} . \tag{3.17}$$

Using this formula we get

$$\mathcal{I}_d^{(2a)} = g_D^{-\frac{24}{8-d}} \frac{16\pi^2 \xi(-2+2\epsilon)^2}{(1+2\epsilon)(6-2\epsilon)} \sum_{\gamma \in P_d \backslash D_d} y^{-2\epsilon}\Bigg|_\gamma \underset{\epsilon \to 0}{\to} \frac{2\zeta(3)^2}{3} g_D^{-\frac{24}{8-d}} , \tag{3.18}$$

which is recognised as the perturbative tree-level contribution in (3.1).

The contribution from $n_i q_a^i = 0$, $q_a^i \neq 0$ is computed by unfolding the fundamental domain of $PGL(2,\mathbb{Z})$ to the strip, so as to set $(n_1, n_2) = (n, 0)$, $(q^1, q^2) = (0, q)$, leading to

$$\frac{8\pi^2}{3} g_D^{-\frac{4d}{8-d}} \int_0^\infty \frac{\mathrm{d}V}{V^{-1+2\epsilon}} \int_0^L \frac{\mathrm{d}\tau_2}{\tau_2^2} \left( \int_{-\frac{1}{2}}^{\frac{1}{2}} \mathrm{d}\tau_1 \, A(\tau) \right) \tag{3.19}$$

$$\times \sum_{\gamma \in P_d \backslash D_d} \left( y^{-4} \sum_{n \geq 1} \sideset{}{'}\sum_{q_a \in \mathbb{Z}^d} e^{-\frac{\pi}{V\tau_2} g_D^{2\frac{d-6}{8-d}} y^2 n^2 - \frac{\pi V}{\tau_2} g_D^{-\frac{4}{8-d}} y^{-\frac{4}{d}} u^{ab} q_a q_b} \right)\Bigg|_\gamma .$$

Note that the boundary of the unfolded domain $\cup_{\gamma \in P_1 \backslash SL(2)} \gamma \mathcal{F}(L)$ includes boundaries at each image of the cusp, but since there are no divergences at these points one can safely extend the unfolded regularised domain to the bounded strip with $\tau_2 < L$.

Naïvely assuming that the expression (1.15) for $A(\tau)$ holds for all $\tau_2 > 0$, the integral on the first line would evaluate to

$$\int_{-\frac{1}{2}}^{\frac{1}{2}} \mathrm{d}\tau_1 A(\tau) \approx \int_{-\frac{1}{2}}^{\frac{1}{2}} \mathrm{d}\tau_1 \left( \frac{1}{\tau_2} + \frac{(|\tau|^2 - \tau_1)(\tau_2^2 + 5(\tau_1^2 - \tau_1))}{\tau_2^3} \right) = \tau_2 + \frac{1}{6\tau_2^3} . \tag{3.20}$$

We will see that this gives the correct powerlike terms in $g_D$ but misses exponentially suppressed corrections to be discussed below and the full (non-naïve) result will be presented in (3.27). To compute the integral over $\tau_2$ it is convenient to modify the regulator. Note that the integral of the second term $\frac{1}{6\tau_2^3}$ in (3.20) is finite, while the first term $\tau_2$ gives an incomplete Gamma function; in the limit $L \to \infty$, the result coincides with the result of the integral over $\tau_2 \in \mathbb{R}^+$ with an insertion of a factor $\tau_2^{-2\tilde{\epsilon}}$ in the integral with the identification $\tilde{\epsilon} = \frac{1}{2\log L} \to 0$. Using this regulator instead of $L$ to simplify the computation, inserting this result in (3.19), and changing variables to $\rho_2 = 1/(\tau_2 V)$, $t = \tau_2/V$, we get

$$\mathcal{I}_d^{(2b)} = \frac{4\pi^2}{3} g_D^{-\frac{4d}{8-d}} \int_0^\infty \frac{\mathrm{d}t}{t^{3-\epsilon+\tilde{\epsilon}}} \int_0^\infty \frac{\mathrm{d}\rho_2}{\rho_2^{2-\epsilon-\tilde{\epsilon}}} \left( t + \frac{\rho_2^2}{6t} \right) \tag{3.21}$$

$$\times \sum_{\gamma \in P_d \backslash D_d} \left( y^{-4} \sum_{n \geq 1} {\sum_{q_a \in \mathbb{Z}^d}}' e^{-\pi \rho_2 g_D^{2\frac{d-6}{8-d}} y^2 n^2 - \frac{\pi}{t} g_D^{-\frac{4}{8-d}} y^{-\frac{4}{d}} u^{ab} q_a q_b} \right) \Big|_\gamma$$

$$= \frac{4\pi^2}{3} g_D^{-\frac{24}{8-d} - 2\epsilon \frac{d-4}{8-d} + 2\tilde{\epsilon}} \sum_{\gamma \in P_d \backslash D_d} \left[ y^{-4\epsilon} {\sum_{q_a \in \mathbb{Z}^d}}' \left( \frac{\xi(2 + 2\epsilon + 2\tilde{\epsilon}) \pi^{-3+\epsilon-\tilde{\epsilon}} \Gamma(3 - \epsilon + \tilde{\epsilon}) g_D^6}{6(y^2 \frac{d-2}{d} u^{ab} q_a q_b)^{3-\epsilon+\tilde{\epsilon}}} \right. \right.$$

$$\left. \left. + \frac{\xi(-2 + 2\epsilon + 2\tilde{\epsilon}) \pi^{-1+\epsilon-\tilde{\epsilon}} \Gamma(1 - \epsilon + \tilde{\epsilon}) g_D^2}{(y^2 \frac{d-2}{d} u^{ab} q_a q_b)^{1-\epsilon+\tilde{\epsilon}}} \right) \right] \Big|_\gamma$$

$$= \frac{8\pi^2}{3} g_D^{-\frac{24}{8-d} - 2\epsilon \frac{d-4}{8-d} + 2\tilde{\epsilon}} \left( \frac{\xi(2 + 2\epsilon + 2\tilde{\epsilon}) \xi(6 - 2\epsilon + 2\tilde{\epsilon})}{6} g_D^6 E^{D_d}_{(3-\epsilon+\tilde{\epsilon})\Lambda_{d-1}+2\epsilon\Lambda_d} \right.$$

$$\left. + \xi(-2 + 2\epsilon + 2\tilde{\epsilon}) \xi(2 - 2\epsilon + 2\tilde{\epsilon}) g_D^2 E^{D_d}_{(1-\epsilon+\tilde{\epsilon})\Lambda_{d-1}+2\epsilon\Lambda_d} \right)$$

$$\overset{\tilde{\epsilon} \to 0}{\to} g_D^{-\frac{24}{8-d} - 2\epsilon \frac{d-4}{8-d}} \left( \frac{4\zeta(3)}{3} g_D^2 \zeta(2) E^{D_d}_{\Lambda_{d-1}} + \frac{4\pi^2 \xi(2+2\epsilon)\xi(6-2\epsilon)}{9} g_D^6 E^{D_d}_{(3-\epsilon)\Lambda_{d-1}+2\epsilon\Lambda_d} + \mathcal{O}(\epsilon) \right)$$

where we used

$$\sideset{}{'}\sum_{q_a \in \mathbb{Z}^d} \sum_{\gamma \in P_d \backslash D_d} \left( \frac{y^{-2t}}{(y^2 \frac{d-2}{d} u^{ab} q_a q_b)^s} \right) \Big|_\gamma = 2\zeta(2s) E^{D_d}_{s\Lambda_{d-1}+t\Lambda_d} . \tag{3.22}$$

Using the fact that a vector $q_a$ parametrises the highest weight component of a conjugate Weyl spinor of opposite chirality under the parabolic decomposition associated to $P_d$, one has

$$\sideset{}{'}\sum_{q_a \in \mathbb{Z}^d} \sum_{\gamma \in P_d \backslash D_d} \left[ f(y^2 \frac{d-2}{d} u^{ab} q_a q_b) \right] \Big|_\gamma = \sideset{}{'}\sum_{N \in S_-} f(g(N, N)) , \tag{3.23}$$

for any function $f(x)$ suitably decaying at infinity. Decomposing the more general sum with a factor of $y^{-2t}$ one gets the sum over the non-maximal parabolic coset $P_{d-1,d}$ of a product of the two multiplicative characters that gives (3.22). Thus we get, in generic dimension

$$\mathcal{I}_d^{(2b)} = g_D^{-\frac{24}{8-d}} \left( \frac{4}{27} g_D^6 \zeta(6) E^{D_d}_{3\Lambda_{d-1}} + \frac{4\zeta(3)}{3} g_D^2 \zeta(2) E^{D_d}_{\Lambda_{d-1}} \right) . \tag{3.24}$$

The two constant terms on the last line of (3.24) reproduce the expected one-loop and three-loop contributions in (3.6). We shall explain in Section 5.2 how the renormalised coupling (1.28) gives indeed the correct constant terms for all $d$.

**Additional contributions to the second layer**

However, (3.24) is only part of the constant term generated by (3.19), since the naïve formula (3.20) only holds for $\tau_2 > \frac{1}{2}$, where the representation is (1.15) is valid. To compute the integral over the full half-line $\tau_2 \in \mathbb{R}^+$, it is convenient to extend the Laplace equation in (B.3) to the full upper half-plane by $GL(2, \mathbb{Z})$ invariance,

$$(\Delta - 12) A(\tau) = -12 \sum_{\gamma \in PGL(2,\mathbb{Z})/(\mathbb{Z}_2 \times \mathbb{Z}_2)} \frac{\tau_2}{|c\tau + d|^2} \delta\left(\frac{(ad+bc)\tau_1 + bd + ac|\tau|^2}{|c\tau+d|^2}\right) \tag{3.25}$$

where $\gamma = \begin{pmatrix} a & b \\ c & d \end{pmatrix}$, $ad - bc = \pm 1$ and the stabiliser subgroup $\mathbb{Z}_2 \times \mathbb{Z}_2$ is generated by $\begin{pmatrix} 1 & 0 \\ 0 & -1 \end{pmatrix}$ and $\begin{pmatrix} 0 & 1 \\ 1 & 0 \end{pmatrix}$. The locus $(ad + bc)\tau_1 + bd + ac|\tau|^2 = 0$ is a geodesic circle of radius $\frac{1}{|2ac|}$. going from $-\frac{b}{a}$ to $-\frac{d}{c}$ on the boundary at $\tau_2 = 0$. For fixed coprime $(a, c)$, the pair $(b, d)$ is determined up to shifts by $(a, c)$, which translate the circle by integers. There is only one circle among these translates that intersects the region $[-\frac{1}{2}, \frac{1}{2}] \times i\mathbb{R}$ and the possible values of $\tau_2$ are restricted to $\tau_2 \leq \frac{1}{|2ac|}$ due to the radius and both possible signs of $c$ are identical in this respect. Therefore the integral of $A(\tau)$ along the segment $[-\frac{1}{2}, \frac{1}{2}]$ satisfies the Laplace equation

$$\left( \tau_2^2 \frac{\partial^2}{\partial \tau_2^2} - 12 \right) \int_{-\frac{1}{2}}^{\frac{1}{2}} d\tau_1 A(\tau) = -12\tau_2 - 24\tau_2 \sum_{\substack{a,c \geq 1 \\ \gcd(a,c)=1}} \frac{H(1 - (2ac\tau_2)^2)}{\sqrt{1 - (2ac\tau_2)^2}} \tag{3.26}$$

where $H(x)$ is the Heaviside function, equal to 1 if $x > 0$ or 0 otherwise. The first term on the r.h.s. is the contribution of $(a, c) = (1, 0)$. The unique solution to (3.26) with the correct behaviour at $\tau_2 > 1$ is[21]

$$\int_{-\frac{1}{2}}^{\frac{1}{2}} d\tau_1 A(\tau) = \tau_2 + \frac{1}{6\tau_2^3} - \frac{1}{7} \sum_{\substack{a,c \geq 1 \\ \gcd(a,c)=1 \\ 2ac < 1/\tau_2}} \frac{1 + \frac{3}{2}(2ac\tau_2)^2 + (2ac\tau_2)^4}{ac(ac\tau_2)^3}(1 - (2ac\tau_2)^2)^{\frac{3}{2}} . \tag{3.27}$$

The first two terms reproduce the naïve answer (3.20), but the last term, upon insertion into (3.19), produces an additional contribution

$$\mathcal{I}_d^{(2c)} = -\frac{2 \cdot 8^2 \pi^2}{21} \int_0^\infty \frac{dV}{V^{-1+2\epsilon}} \int_0^1 \frac{dt}{t^5}(1 + \tfrac{3}{2}t^2 + t^4)(1 - t^2)^{\frac{3}{2}} g_D^{-\frac{4d}{8-d}} \tag{3.28}$$

$$\times \sum_{\gamma \in P_d \backslash D_d} \left( y^{-4} \sum_{\substack{a,c \geq 1 \\ \gcd(a,c)=1}} \sum_{n \geq 1} \sideset{}{'}\sum_{q_a \in \mathbb{Z}^d} e^{-\frac{\pi}{Vt}g_D^2 \frac{d-6}{8-d} y^2(2acn^2) - \frac{\pi V}{t}g_D^{-\frac{4}{8-d}} y^{-\frac{4}{d}} u^{ab}(2acq_a q_b)} \right) \Bigg|_\gamma .$$

Using (3.23) and observing that $an$ and $cn$ are independent divisors of $N^a = acnq^a$, we get

$$\mathcal{I}_d^{(2c)} = -\frac{256\pi^2}{21} g_D^{-\frac{24}{8-d}+2} \int_0^1 \frac{dt}{t^5}(1 + \tfrac{3}{2}t^2 + t^4)(1 - t^2)^{\frac{3}{2}} \sideset{}{'}\sum_{\substack{N \in S_- \\ N \times N = 0}} \sigma_2(N)^2 \frac{K_2(\frac{4\pi}{g_D t}|v(N)|)}{|v(N)|^2}$$

$$= -\frac{16\pi^2}{21} g_D^{-\frac{24}{8-d}+2} \sideset{}{'}\sum_{\substack{N \in S_- \\ N \times N = 0}} \frac{\sigma_2(N)^2}{|v(N)|^2} B_2(\tfrac{2\pi}{g_D}|v(N)|) , \tag{3.29}$$

where we introduced the special function

$$B_s(z) = 16 \int_0^1 \frac{dt}{t^5}(1 + \tfrac{3}{2}t^2 + t^4)(1 - t^2)^{\frac{3}{2}} K_s(\tfrac{2z}{t}) , \tag{3.30}$$

---

[21]The homogeneous solution $(\frac{1}{(2n\tau_2)^3} - (2n\tau_2)^4)H(1 - 2n\tau_2)$ would have a $\delta$ source non-vanishing and is thus ruled out.

which evaluates to

$$\begin{aligned}
B_s(z) &= 16 \int_1^\infty \frac{\mathrm{d}u}{\sqrt{u^2-1}}\Big(1+\frac{1}{2}(u^2+u^{-2})-u^4-u^{-4}\Big)K_s(2uz) \qquad (3.31)\\
&= \Big(\frac{s^2-4}{z^2}-2-\frac{8z^2}{s^2-9}+\frac{64z^4}{(s^2-9)(s^2-1)}\Big)(K_{\frac{s}{2}-1}(z))^2 \\
&\quad +\Big(\frac{s(s^2-4)}{z^3}-\frac{2(s+2)}{z}-\frac{8z}{s+3}+\frac{64z^3}{(s^2-9)(s+1)}\Big)K_{\frac{s}{2}-1}(z)K_{\frac{s}{2}}(z) \\
&\quad +\Big(-\frac{s(s+2)}{z^2}+\frac{2(s+9)}{s+3}+\frac{8z^2}{(s+3)(s+1)}-\frac{64z^4}{(s^2-9)(s^2-1)}\Big)(K_{\frac{s}{2}}(z))^2 \,.
\end{aligned}$$

For $s=2$, this reduces to

$$zB_2(2z) = \frac{1}{60}\sum_{i=0,1} r_{ij}(\tfrac{z}{2})K_i(z)K_j(z) \qquad (3.32)$$

where $r_{ij}$ are the functions defined in [32, (2.45)]. As a result, for $d=0$ (3.29) reproduces the formula [32, (2.44)], *i.e.* $-\frac{16\pi^2}{21}\sum'_{n\in\mathbb{Z}}\frac{\sigma_2(n)^2}{|n|^3}\frac{|n|}{g_D}B_2(\frac{2\pi}{g_D}|n|)$. In the limit $g_D \to 0$, using the standard asymptotics of the modified Bessel function, we find that (3.29) reduces to

$$\mathcal{I}_d^{(2c)} \sim -g_D^{-\frac{24}{8-d}+5} \sideset{}{'}\sum_{\substack{N\in S_- \\ N\times N=0}} \sigma_2(N)^2 \frac{e^{-\frac{4\pi}{g_D}|v(N)|}}{|v(N)|^5}\,, \qquad (3.33)$$

which can be interpreted as contribution from bound states of instantons and anti-instantons with vanishing total charge. Indeed, these effects are required by the differential equation (1.5), given that $\mathcal{E}_{(0,0)}$ contains instanton corrections of the form (see e.g. [5, (66)] for $d=0$, [23, (4.84)] for $d=4$)

$$4\pi g_D^{-\frac{12}{8-d}+\frac{3}{2}} \sideset{}{'}\sum_N \sigma_2(N) \frac{e^{-\frac{2\pi}{g_D}|v(N)|+2\pi\mathrm{i}Na}}{|v(N)|^{\frac{3}{2}}}\,. \qquad (3.34)$$

**Consistency with the Poisson equation**

In order to check that the contributions (3.29) do satisfy the inhomogeneous Laplace equation (1.5) sourced by the instanton terms in $\mathcal{E}_{(0,0)}^{(d)}$, we use (2.1) to compute

$$\Big(\Delta_{E_{d+1}} - \frac{6(4-d)(d+4)}{8-d}\Big)\mathcal{E}_{(0,1)}^{(d),\mathrm{ExFT}} = \frac{8\pi^2}{3}\frac{\Gamma(d-2)}{\pi^{d-2}}\int_\mathcal{F}\frac{\mathrm{d}\tau_1\mathrm{d}\tau_2}{\tau_2^2}A(\tau)\left[\Delta_\tau - 12\right]\Xi_{\Lambda_{d+1}}^{E_{d+1}}\,. \qquad (3.35)$$

Restoring the integral over $V$, integrating by parts over $\tau$, and focusing on the contribution to the term (3.19) of type 2b), we get

$$\begin{aligned}
\frac{16\pi^2}{3}\,g_D^{-\frac{4d}{8-d}}\int_0^\infty V\mathrm{d}V\int_0^\infty \frac{\mathrm{d}\tau_2}{\tau_2^2}\Bigg[(\tau_2^2\partial_{\tau_2}^2-12)\int_{-1/2}^{1/2}A\,\mathrm{d}\tau_1\Bigg] & \\
\times \sum_{\gamma\in P_d\backslash D_d}\Bigg(y^{-4}\sideset{}{'}\sum_{n\geq1}\sum_{q_a\in\mathbb{Z}^d} e^{-\frac{\pi}{V\tau_2}g_D^{2\frac{d-6}{8-d}}y^2n^2-\frac{\pi V}{\tau_2}g_D^{-\frac{4}{8-d}}y^{-\frac{4}{d}}u^{ab}q_aq_b}\Bigg)\Bigg|_\gamma\,. & \qquad (3.36)
\end{aligned}$$

We now substitute the source term on the r.h.s. of (3.26) into the square bracket, obtaining

$$4\pi^2 \, g_D^{-\frac{4d}{8-d}} \sum_{\substack{a,c \geq 1 \\ \gcd(a,c)=1}} \int_0^{\frac{1}{2ac}} \frac{\mathrm{d}\tau_2}{\tau_2 \sqrt{1-(4ac\tau_2)^2}} \int_0^\infty V \mathrm{d}V$$

$$\times \sum_{\gamma \in P_d \backslash D_d} \left( y^{-4} \sum_{n \geq 1} \sum_{q_a \in \mathbb{Z}^d}' e^{-\frac{\pi}{V\tau_2} g_D^{2\frac{d-6}{8-d}} y^2 n^2 - \frac{\pi V}{\tau_2} g_D^{-\frac{4}{8-d}} y^{-\frac{4}{d}} u^{ab} q_a q_b} \right) \Bigg|_\gamma . \quad (3.37)$$

The integral over $V$ is of Bessel type, giving

$$\frac{2^7 \pi^2}{g_D^{\frac{24}{8-d}-2}} \sum_{\substack{a,c \geq 1 \\ \gcd(a,c)=1}} \sum_{n \geq 1} \int_0^{\frac{1}{2ac}} \frac{\mathrm{d}\tau_2}{\tau_2 \sqrt{1-(4ac\tau_2)^2}} \sum_{\gamma \in P_d \backslash D_d} \left[ \frac{y^{\frac{2(2-d)}{d}} n^2}{u^{ab} q_a q_b} K_2 \left( \frac{2\pi}{\tau_2} y^{\frac{d-2}{d}} n \sqrt{u^{ab} q_a q_b} \right) \right] \Bigg|_\gamma . \quad (3.38)$$

The integral over $\tau_2$ can be computed by changing variables to $u = 1/(2ac\tau_2)$ and using

$$\int_1^\infty \frac{\mathrm{d}u}{\sqrt{u^2-1}} K_s(2uz) = \frac{1}{2} \left[ K_{s/2}(z) \right]^2 . \quad (3.39)$$

Setting $acnq^a = N^a$, the sum over $a, c, n$ amounts to a sum over pairs of divisors $(an, cn)$ of $N^a$. As a result, we get

$$2^6 \pi^2 \, g_D^{-\frac{24}{8-d}+2} \sum_{\substack{N \in S_- \\ N \times N = 0}}' \left[ \frac{\sigma_2(N)}{|v(N)|} K_1 \left( \frac{2\pi}{g_s} \sqrt{u^{ab} N_a N_b} \right) \right]^2 \quad (3.40)$$

which we recognise as the square of the D-instanton contributions in $\mathcal{E}_{(0,0)}^{(d)}$ consistent with (3.29).

**3) The third layer**

The third contribution to (3.7) is obtained when $\chi_1$ and $\chi_2$ are non-zero and linearly independent, while the $N_i$ still vanish. The $\chi_i$s can then be rotated into the degree-one doublet of the $SL(2)$ factor in the Levi subgroup associated to the graded decomposition[22]

$$\left( \mathfrak{gl}_1 \oplus \mathfrak{sl}_2 \oplus \mathfrak{sl}_2' \oplus \mathfrak{sl}_{d-2} \right)^{(0)} \oplus \left( \mathbf{2} \otimes \mathbf{2}' \otimes \mathbb{Z}^{d-2} \right)^{(\frac{2}{d-2})} \oplus \left( \wedge^2 \mathbb{Z}^{d-2} \right)^{(\frac{4}{d-2})} \subset \mathfrak{so}_{d,d} ,$$

$$\chi_i \in \overline{S}_+ = \cdots \oplus \left( \mathbf{2} \otimes \wedge^2 \mathbb{Z}^{d-2} \right)^{(\frac{d-6}{d-2})} \oplus \left( \mathbf{2}' \otimes \mathbb{Z}^{d-2} \right)^{(\frac{d-4}{d-2})} \oplus \mathbf{2}^{(1)} , \quad (3.41)$$

$$q_i \in V = \left( \mathbb{Z}^{d-2} \right)^{(-\frac{2}{d-2})} \oplus \left( \mathbf{2} \otimes \mathbf{2}' \right)^{(0)} \oplus \left( \mathbb{Z}^{d-2} \right)^{(\frac{2}{d-2})} .$$

We denote the variables parametrising the Levi subgroup $GL(1) \times SL(2) \times SL(2)' \times SL(d-2)$ by $(y, v_{\hat{i}\hat{j}}, \rho^{\alpha\beta}, u_{ab})$, and the coordinates on the unipotent part $\mathbf{2} \otimes \mathbf{2}' \otimes \mathbb{Z}^{d-2}$ by $c_{\hat{k}}^{\alpha\beta}$. The coordinates

---

[22]Such a doublet of spinors defines a $(d-2)$-form $\chi_1 \gamma_{d-2} \chi_2$ which is in the $Spin(d,d)$ orbit of a highest weight representative, which can be rotated into a standard form using $Spin(d,d)$ to a specific representative.

of $\chi_i$ are $(0, \ldots, 0, n_i{}^{\hat{j}})$, while the constraint $q_{(i}^a \gamma_a \chi_{j)} = 0$ in (3.10) implies that $q_i = (0, \hat{n}_i{}^{\hat{j}} p_\alpha, q_i^a)$ where $\hat{n}_i{}^{\hat{j}} := n_i{}^{\hat{j}}/\gcd\left(n_i{}^{\hat{j}}\right)$. Using these variables one can write the Poincaré sum

$$
\theta_{\Lambda_{d+1}}^{(3)}(\phi, \Omega_2) = \sum_{\gamma \in P_{d-2} \backslash D_d} \left( \sum_{\substack{n_i{}^{\hat{j}} \in \mathbb{Z}^2 \\ \det n \neq 0}} \sum_{\substack{q_i^a \in \mathbb{Z}^{d-2} \\ p_\alpha \in \mathbb{Z}^2}} e^{-\pi \Omega_2^{ij} g_D^{\frac{4}{8-d}} \left( (g_D^{-2} y n^2 + \rho^{\alpha\beta}(p_\alpha + a_\alpha n)(p_\beta + a_\beta n)) v_{\hat{k}\hat{l}} \hat{n}_i{}^{\hat{k}} \hat{n}_j{}^{\hat{l}} \right)} \right.
$$
$$
\left. \times\, e^{-\pi \Omega_2^{ij} g_D^{\frac{4}{8-d}} y^{\frac{2}{d-2}} u_{ab}(q_i^a + a_{\hat{k}}^a n_i{}^{\hat{k}} + c_{\hat{k}}^{a\alpha} \hat{n}_i{}^{\hat{k}}(p_\alpha + a_\alpha n))(q_j^b + a_{\hat{l}}^b n_j{}^{\hat{l}} + c_{\hat{l}}^{b\beta} \hat{n}_j{}^{\hat{l}}(p_\beta + a_\beta n))} \right)\Bigg|_\gamma
$$
$$
= \frac{g_D^{-4\frac{d-1}{8-d}}}{|\Omega_2|^{\frac{d-2}{2}}} \sum_{\gamma \in P_{d-2} \backslash D_d} \left( \sum_{\substack{n_i{}^{\hat{j}} \in \mathbb{Z}^2 \\ \det n \neq 0}} \sum_{\substack{q_a^i \in \mathbb{Z}^{d-2} \\ p^\alpha \in \mathbb{Z}^2}} \frac{y^{-2}}{\Omega_2^{ij} v_{\hat{k}\hat{l}} \hat{n}_i{}^{\hat{k}} \hat{n}_j{}^{\hat{l}}} e^{-\pi \Omega_2^{ij} g_D^{2\frac{d-6}{8-d}} y v_{\hat{k}\hat{l}} n_i{}^{\hat{k}} n_j{}^{\hat{l}}} \right.
$$
$$
\left. \times\, e^{-\pi g_D^{-\frac{4}{8-d}} \left( y^{-\frac{2}{d-2}} \Omega_{2ij}^{-1} u^{ab} q_a^i q_b^j + \frac{1}{\Omega_2^{ij} v_{\hat{k}\hat{l}} \hat{n}_i{}^{\hat{k}} \hat{n}_j{}^{\hat{l}}} \rho_{\alpha\beta}(p^\alpha - c_{\hat{k}}^{a\alpha} \hat{n}_i{}^{\hat{k}} q_a^i)(p^\beta - c_{\hat{l}}^{b\beta} \hat{n}_j{}^{\hat{l}} q_b^j) \right) + 2\pi i (q_a^i n_i{}^{\hat{j}} a_{\hat{j}}^a + n p^\alpha a_\alpha)} \right)\Bigg|_\gamma
$$
(3.42)

where we have used Poisson resummation on the unconstrained variables $p_\alpha$ and $q_i^a$. The constant term comes from $q_a^i n_i{}^{\hat{j}} = 0$ and $n p^\alpha = 0$, implying $p^\alpha = q_a^q = 0$. Replacing $d \to d + 2\epsilon$ for the analytic continuation, one obtains the constant term

$$
\mathcal{I}_d^{(3a)} = 8\pi g_D^{-4\frac{d-1}{8-d}} \int_{\mathcal{G}} \frac{\mathrm{d}^3 \Omega_2}{|\Omega_2|^{2-\epsilon}} \varphi_{\text{KZ}}^{\text{tr}}(\Omega_2) \sum_{\gamma \in P_{d-2} \backslash D_d} \left( \sum_{\substack{n_i{}^{\hat{j}} \in \mathbb{Z}^2 \\ \det n \neq 0}} \frac{y^{-2}}{\Omega_2^{ij} v_{\hat{k}\hat{l}} \hat{n}_i{}^{\hat{k}} \hat{n}_j{}^{\hat{l}}} e^{-\pi \Omega_2^{ij} g_D^{2\frac{d-6}{8-d}} y v_{\hat{k}\hat{l}} n_i{}^{\hat{k}} n_j{}^{\hat{l}}} \right)\Bigg|_\gamma
$$
$$
= 8\pi g_D^{-\frac{24+8\epsilon}{8-d}+2+4\epsilon} \frac{\xi(4\epsilon - 2)}{\xi(4\epsilon)} \sum_{\gamma \in P_{d-2} \backslash D_d} \left( y^{-1} \sum_{\substack{n_i{}^{\hat{j}} \in \mathbb{Z}^2 \\ \det(n_i{}^{\hat{j}}) \neq 0}} \int_{\mathcal{G}} \frac{\mathrm{d}^3 \Omega_2}{|\Omega_2|^{2-\epsilon}} \varphi_{\text{KZ}}^{\text{tr}}(\Omega_2) e^{-\pi \Omega_2^{ij} y v_{\hat{k}\hat{l}} n_i{}^{\hat{k}} n_j{}^{\hat{l}}} \right)\Bigg|_\gamma
$$
(3.43)

where we have done the integral over $V = |\Omega_2|^{-\frac{1}{2}}$ and the sum over $\gcd(n)$ and then rewritten the result as a new simpler integral over $V$ and sum over the matrices $n_i{}^{\hat{j}}$ without explicit $\gcd(n)$. In Appendix C.3, we argue that in the limit $\epsilon \to 0$, this gives a finite Eisenstein series

$$
\mathcal{I}_d^{(3a)} = \frac{4\pi^2}{9} \frac{\xi(4\epsilon - 2)}{\xi(4\epsilon)} \xi(5 - 2\epsilon) \xi(3 + 2\epsilon) \frac{\xi(8)}{\xi(7)} g_D^{-\frac{24+8\epsilon}{8-d}+2+4\epsilon} E_{(-\frac{3}{2}+\epsilon)\Lambda_{d-2}+4\Lambda_d}^{D_d}
$$
$$
\underset{\epsilon \to 0}{=} -80\xi(2)\xi(6)\xi(8) g_D^{-\frac{24}{8-d}+2} E_{-\frac{3}{2}\Lambda_{d-4}+4\Lambda_d}^{D_d}.
$$
(3.44)

As we shall see in Section 5.2, this undesired term cancels against the counterterm in (1.28) and does not appear in the renormalised coupling.

## 4) The fourth layer

Up to now, we have considered only contributions with $N_i = 0$, which exhaust all layers when $d \leq 4$. The fourth layer includes $N_i \neq 0$, but linearly dependent ($N_i \wedge N_j = 0$), which is

automatic for $d = 5$, where $N_i \in \mathbb{Z}$. We shall argue that the contribution from this layers drops out in the renormalised coupling (1.28).

For $d = 5$ one has

$$
\mathcal{I}_5^{(4)}(\phi, \epsilon) = 8\pi \, \mathrm{R.N.} \int_{\mathcal{G}} \frac{\mathrm{d}^3 \Omega_2}{|\Omega_2|^{\frac{1-2\epsilon}{2}}} \varphi_{\mathrm{KZ}}^{\mathrm{tr}}(\Omega_2) \sum_{N_i \in \mathbb{Z}}'
$$

$$
\sum_{\substack{\chi_i \in S_- \\ q_i \in II_{5,5} \\ \chi_i \gamma \chi_j = N_{(i} q_{j)} \\ (q_i, q_j) = 0}} e^{-\pi \Omega_2^{ij} \left( g_5^{\frac{4}{3}} g(q_i + a\gamma\chi_i + \frac{1}{2}(a\gamma a)N_i, q_j + a\gamma\chi_j + \frac{1}{2}(a\gamma a)N_j) + g_5^{-\frac{2}{3}} v(\chi_i + aN_i) \cdot v(\chi_j + aN_j) + g_5^{-\frac{8}{3}} N_i N_j \right)} \quad (3.45)
$$

while the same term for $d = 6$ can be written as a Poincaré sum

$$
\mathcal{I}_6^{(4)}(\phi, \epsilon) = 8\pi \, \mathrm{R.N.} \int_{\mathcal{G}} \frac{\mathrm{d}^3 \Omega_2}{|\Omega_2|^{-\epsilon}} \varphi_{\mathrm{KZ}}^{\mathrm{tr}}(\Omega_2) \sum_{\gamma \in P_1 \backslash SO(6,6)} \sum_{N_i \in \mathbb{Z}}'
$$

$$
\sum_{\substack{\chi_i \in S_- \\ q_i \in II_{5,5} \\ \chi_i \gamma \chi_j = N_{(i} q_{j)} \\ (q_i, q_j) = 0}} e^{-\pi \Omega_2^{ij} \left( g_4^2 g(q_i + a\gamma\chi_i + \frac{1}{2}(a\gamma a)N_i, q_j + a\gamma\chi_j + \frac{1}{2}(a\gamma a)N_j) + yv(\chi_i + aN_i) \cdot v(\chi_j + aN_j) + g_4^{-2} y^2 N_i N_j \right)}
$$

$$
\times \sum_{m_i \in \mathbb{Z}} e^{-\pi \Omega_2^{ij} g_4^2 y^2 (m_i + \bar{a}(\chi_i + a n_i) + b n_i)(m_j + \bar{a}(\chi_j + a n_j) + b n_j)} \Bigg|_\gamma
$$

$$
= \frac{8\pi}{g_4^2} \sum_{\gamma \in P_1 \backslash SO(6,6)} y^{-2} \left( \mathrm{R.N.} \int_{\mathcal{G}} \frac{\mathrm{d}^3 \Omega_2}{|\Omega_2|^{\frac{1-2\epsilon}{2}}} \varphi_{\mathrm{KZ}}^{\mathrm{tr}}(\Omega_2) \sum_{N_i \in \mathbb{Z}}' \right.
$$

$$
\sum_{\substack{\chi_i \in S_- \\ q_i \in II_{5,5} \\ \chi_i \gamma \chi_j = N_{(i} q_{j)} \\ (q_i, q_j) = 0}} e^{-\pi \Omega_2^{ij} \left( g_4^2 g(q_i + a\gamma\chi_i + \frac{1}{2}(a\gamma a)N_i, q_j + a\gamma\chi_j + \frac{1}{2}(a\gamma a)N_j) + yv(\chi_i + aN_i) \cdot v(\chi_j + aN_j) + g_4^{-2} y^2 N_i N_j \right)}
$$

$$
\times \sum_{\tilde{m}^i \in \mathbb{Z}} e^{-\pi \Omega_{2\,ij}^{-1} g_4^{-2} y^{-2} \tilde{m}_i \tilde{m}_j + 2\pi i \tilde{m}^i (\bar{a}(\chi_i + a n_i) + b n_i)} \left. \right) \Bigg|_\gamma . \quad (3.46)
$$

The abelian Fourier coefficient is obtained by setting $\tilde{m}^i = 0$, leading to

$$
\mathcal{I}_6^{(4a)}(\phi, \epsilon) = g_4^{-4-\frac{4\epsilon}{3}} \sum_{\gamma \in P_1 \backslash SO(6,6)} y^{-4-\frac{4\epsilon}{3}} \mathcal{I}_5^{(4)} (g_5 = y^{-\frac{1}{2}} g_4) \Big|_\gamma . \quad (3.47)
$$

Therefore the contributions to the constant terms and abelian Fourier coefficients in $d = 6$ are determined from the ones in $d = 5$ through a Poincaré sum.

In Appendix D.1, we study a similar integral $\mathcal{I}_{\Lambda_{d+1}}^{E_{d+1}}(E_{s\Lambda_1}^{SL(2)}, d + 2\epsilon - 2)$ where $A(\tau)$ is replaced by an Eisenstein series $E_{s\Lambda_1}^{SL(2)}$. There we find for generic $s$ that the constant terms from the orbit with $N_i \neq 0$, $N_i \wedge N_j = 0$ disappear as $\epsilon = 0$, due to an overall factor of $\frac{1}{\xi(4\epsilon)}$. Therefore we

expect this factor of $\frac{1}{\xi(4\epsilon)}$ to appear in the computation irrespective of the function (e.g. $A(\tau)$ or $E^{SL(2)}_{s\Lambda_1}$) on $SL(2)/SO(2)$ one considers. However, for the specific value $s = -3$ corresponding to the counterterm in (1.28), one finds that the coefficient diverges in $\xi(1 + 2\epsilon)$ and there is a finite contribution in the limit. Consistency requires that this finite contribution disappears in the renormalised coupling (1.28), see Section 5.2.

## 5) The fifth layer

For $d = 6$ one must also consider the cases with $N_i$ non-collinear. One can write the sum as a Poincaré sum over $P_2 \subset SO(6,6)$ such that

$$I\!I_{6,6} \cong (\mathbb{Z}^2)^{(-2)} \oplus (S_-)^{(0)} \oplus (\mathbb{Z}^2)^{(2)} , \qquad \bar{S}_+ \cong (I\!I_{4,4})^{(-2)} \oplus (\mathbb{Z}^2 \otimes S_+)^{(0)} \oplus (I\!I_{4,4})^{(2)} , \qquad (3.48)$$

where the embedding $SO(4,4) \subset SO(6,6)$ differs from the standard one by triality. The solution to the constraints (3.10) decomposes in this basis as

$$q_i = (0, 0, n_i{}^{\hat{\jmath}}) , \qquad \chi_i = (0, n_i{}^{\hat{\jmath}} \tfrac{p_\alpha}{k}, q_{ia}) , \qquad N_i = (n_i{}^{\hat{\jmath}} \tfrac{(p,p)}{2k^2}, \gamma^{a\,\alpha\dot{\alpha}} \tfrac{p_\alpha}{k} q_{ia}, m_i{}^{\hat{\jmath}}) , \qquad (3.49)$$

where $k$ can be chosen as an integer coprime to $p_\alpha$ that divides $n_i{}^{\hat{\jmath}}$, $\gamma^{a\,\alpha\dot{\alpha}} p_\alpha q_{ia}$ and $\tfrac{n_i{}^{\hat{\jmath}}}{k} \tfrac{(p,p)}{2}$. The integer $k$ can be decomposed as $k = k_1 k_2$ such that $k_1 k_2^2 | n_i{}^{\hat{\jmath}}$ and $k_1 | \tfrac{(p,p)}{2}$. For any $p_\alpha$, one can find a pair of primitive null vectors $u_\alpha$ and $v_\alpha$ such that $(u,v) = 1$ and

$$p_\alpha = \gcd(p) u_\alpha + \frac{(p,p)}{2\gcd(p)} v_\alpha \qquad (3.50)$$

and the condition $k | \gamma^{a\,\alpha\dot{\alpha}} p_\alpha q_{ia}$ reduces to the property that the component of $q_{ia}$ in the null space of $u_\alpha$ is divisible by $k_2$ and the one in the null space of $v_\alpha$ is divisible by $k_1 k_2$. So $q_{ia} \in k_2 I\!I_{4,4}[k_1] \cong (k_1 k_2 \mathbb{Z})^4 \oplus (k_2 \mathbb{Z})^4$ in the appropriate decomposition. The bilinear form reads

$$G(\Gamma_i, \Gamma_j) = \tilde{y}_1 \upsilon_{\hat{\imath}\hat{\jmath}} (m_i{}^{\hat{\imath}} + \tilde{a}^{\hat{\imath}a} q_{ia} + \tfrac{1}{2} \tilde{a}^{\hat{\imath}a} \tilde{a}_{\hat{k}a} n_i{}^{\hat{k}} + \tilde{b} n_i{}^{\hat{\jmath}})(m_j{}^{\hat{\jmath}} + \tilde{a}^{\hat{\jmath}b} q_{jb} + \tfrac{1}{2} \tilde{a}^{\hat{\jmath}b} \tilde{a}_{\hat{l}b} n_j{}^{\hat{l}} + \tilde{b} n_j{}^{\hat{l}})$$
$$+ \sqrt{\tilde{y}_1 \tilde{y}_2} \tilde{u}^{ab} (q_{ia} + \tilde{a}_{\hat{\imath}a} n_i{}^{\hat{\imath}})(q_{jb} + \tilde{a}_{\hat{\jmath}b} n_j{}^{\hat{\jmath}}) + \tilde{y}_2 \upsilon_{\hat{\imath}\hat{\jmath}} n_i{}^{\hat{\jmath}} n_j{}^{\hat{\jmath}} \quad (3.51)$$

where

$$\tilde{y}_1 = g_4^2 y , \qquad \tilde{y}_2 = g_4^{-2} y + u^{\alpha\beta}(\tfrac{p_\alpha}{k} + a_\alpha)(\tfrac{p_\beta}{k} + a_\beta) + \frac{g_4^2}{4y}[(\tfrac{p_\alpha}{k} + a_\alpha)(\tfrac{p^\alpha}{k} + a^\alpha)]^2 ,$$

$$\tilde{u}^{ab} = \frac{u^{ab} + g_4^2 y^{-1} u^{\dot{\alpha}\dot{\beta}} \gamma^a{}_{\alpha\dot{\alpha}} \gamma^b{}_{\beta\dot{\beta}} (\tfrac{p^\alpha}{k} + a^\alpha)(\tfrac{p^\beta}{k} + a^\beta)}{\sqrt{1 + g_4^2 y^{-1} u^{\gamma\delta}(\tfrac{p_\gamma}{k} + a_\gamma)(\tfrac{p_\delta}{k} + a_\delta) + \frac{g_4^4}{4y^2}[(\tfrac{p_\gamma}{k} + a_\gamma)(\tfrac{p^\gamma}{k} + a^\gamma)]^2}} ,$$

$$\tilde{a}^{\hat{\imath}a} = \varepsilon^{\hat{\imath}\hat{\jmath}} \eta^{ab} \tilde{a}_{\hat{\jmath}b} = a^{\hat{\imath}a} + c^{\hat{\imath}\dot{\alpha}} \gamma^a{}_{\alpha\dot{\alpha}}(\tfrac{p^\alpha}{k} + a^\alpha) ,$$

$$\tilde{b} = b + \bar{a}^\alpha(\tfrac{p_\alpha}{k} + a_\alpha) + \tfrac{1}{2} c\,(\tfrac{p_\alpha}{k} + a_\alpha)(\tfrac{p^\alpha}{k} + a^\alpha) , \qquad (3.52)$$

with $(a_\alpha, a_{\hat{\imath}a}, \bar{a}_\alpha) \in S_-$ parametrising the unipotent in $P_1 \subset E_7$ in

$$S_- \cong (S_+)^{(-2)} \oplus (\mathbb{Z}^2 \otimes I\!I_{4,4})^{(0)} \oplus (S_+)^{(2)} , \qquad (3.53)$$

and $(c_{i\dot\alpha}, c)$ the unipotent $\mathbb{Z}^2 \otimes S_- \oplus \mathbb{Z}$ of $P_2 \subset SO(6,6)$ and $u^{ab}$ and $u^{\dot\alpha\dot\beta}$ the Levi subgroup $Spin(4,4)$ in the vector and spinor representation and $v_{\hat i\hat j}$ the $SL(2)$ Levi subgroup of $P_2$. For fixed $p_\alpha$ and $k$, the sum over $n_i{}^{\hat j}$, $q_{ia}$ and $m_i{}^{\hat j}$ reproduces a genus two Siegel–Narain theta series over the lattice $k_2 II_{4,4}[k_1] \oplus II_{2,2}[k_1 k_2^2]$, with $n_i{}^{\hat j}$ non-degenerate. The computation at this level would involve the consideration of the Poincaré sum of $|\Omega_2|^\epsilon \varphi_{\mathrm{KZ}}^{\mathrm{tr}}(\Omega_2)$ over all congruent subgroups $\Gamma_0(k_1 k_2^2)$ of $Sp(4,\mathbb{Z})$ ($\gamma = \left(\begin{smallmatrix} A & B \\ C & D \end{smallmatrix}\right)$ with $C$ a multiple of $k_1 k_2^2$), which seems out of reach.

Rather than pursuing this approach, we shall argue that the sum over $p$ and $k$ in this expression can be seen as a Poincaré series over $P_1 \backslash SO(5,5)$ acting on the overall unconstrained lattice sum in $II_{6,6}$ with $n_i{}^{\hat j}$ non-degenerate. The reason is that one obtains exactly the same sum in the $T^2$ decompactification limit of the same coupling, i.e. in the parabolic $P_6 \subset E_7$

$$M_{\Lambda_7}^{E_7} = (\mathbb{Z}^2)^{(-2)} \oplus S_-^{(-1)} \oplus (\mathbb{Z}^2 \otimes II_{5,5})^{(0)} \oplus S_+^{(1)} \oplus (\mathbb{Z}^2)^{(2)} \; . \tag{3.54}$$

The decomposition of this series can be computed explicitly when all strictly negative degree charges are zero while the degree 0 ones are non-degenerate, in which case they match exactly the set of charges we have defined above, i.e.

$$\left(n_i{}^{\hat j}, n_i{}^{\hat j}\frac{p_\alpha}{k}, \frac{(p,p)}{2k^2}n_i{}^{\hat j}\right) \in (\mathbb{Z}^2 \otimes II_{5,5})^{(0)} \; , \qquad \left(q_{ia}, \gamma^a{}_{\alpha\dot\alpha}\frac{p^\alpha}{k}q_{ia}\right) \in S_+^{(1)} \; , \quad m_i{}^{\hat j} \in (\mathbb{Z}^2)^{(2)} \; . \tag{3.55}$$

This does not parametrise the whole set of charges in the large $T^2$ volume, but only those for which $n_i{}^{\hat j}$ is non-degenerate in $\mathbb{Z}^2 \otimes II_{5,5}$, which we call the principal layer in the decomposition of the $SO(5,5)$ Poincaré sum.[23]

With this interpretation, the sum over $p$ and $k$ of each Narain theta series over $k_2 II_{4,4}[k_1] \oplus II_{2,2}[k_1 k_2^2]$, with $n_i{}^{\hat j}$ is the Poincaré sum acting on the Narain theta series over $II_{6,6}$. So one can apply the orbit method for the single $Sp(4,\mathbb{Z})$ invariant theta series, and then carry out the sum over $p$ and $k$ on the resulting expression. This leads to

$$\int_{\mathcal{G}} \frac{\mathrm{d}^3\Omega_2}{|\Omega_2|^3} \int_{\mathbb{Z}^3\backslash\mathbb{R}^3} \mathrm{d}^3\Omega_1 \; |\Omega_2|^\epsilon \varphi_{\mathrm{KZ}}^{\mathrm{tr}}(\Omega_2) |\Omega_2|^3 \sum_{\substack{n_i{}^{\hat j}\in\mathbb{Z}^2 \\ \det n\neq 0}} \sum_{\substack{q_{ia}\in II_{4,4} \\ m_i{}^{\hat j}\in\mathbb{Z}^2}} e^{-\pi\Omega_2^{ij}G(\Gamma_i,\Gamma_j)+\pi\mathrm{i}\Omega_1^{ij}(2\varepsilon_{\hat i\hat j}m_i{}^{\hat i}n_j{}^{\hat j}-(q_i,q_j))}$$

$$= \int_{\mathcal{G}} \frac{\mathrm{d}^3\Omega_2}{|\Omega_2|^3} \int_{\mathbb{Z}^3\backslash\mathbb{R}^3} \mathrm{d}^3\Omega_1 \; |\Omega_2|^\epsilon \varphi_{\mathrm{KZ}}^{\mathrm{tr}}(\Omega_2) \frac{|\Omega_2|^2}{\tilde y_1^{3+\epsilon}\tilde y_2^{1+\epsilon}} \sum_{\substack{n_i{}^{\hat j}\in\mathbb{Z}^2 \\ \det n\neq 0}} \sum_{m^{i\hat j}\in\mathbb{Z}^2} e^{-\pi\sqrt{\frac{\tilde y_2}{\tilde y_1}}\Omega_{2\hat i\hat j}^{-1}v_{\hat i\hat j}(m^{i\hat i}+\Omega^{ik}n_k{}^{\hat i})(m^{j\hat j}+\bar\Omega^{jl}n_l{}^{\hat j})}$$

$$\times \sum_{q_{ia}\in II_{4,4}} e^{\pi\mathrm{i}\Omega^{ij}p_L(q_i+\tilde a n_i)\cdot p_L(q_i+\tilde a n_i)-\pi\mathrm{i}\bar\Omega^{ij}p_R(q_i+\tilde a n_i)\cdot p_R(q_i+\tilde a n_i)+2\pi\mathrm{i}m^i(q_i\tilde a+\frac{1}{2}\tilde a\tilde a n_i+\tilde b n_i)}$$

$$= \int_{\mathcal{G}} \frac{\mathrm{d}^3\Omega_2}{|\Omega_2|^3} \int_{\mathbb{Z}^3\backslash\mathbb{R}^3} \mathrm{d}^3\Omega_1 \sum_{\substack{\gamma\in P_2\backslash Sp(4,\mathbb{Z}) \\ \det C(\gamma)\neq 0}} \left(|\Omega_2|^\epsilon \varphi_{\mathrm{KZ}}^{\mathrm{tr}}(\Omega_2)\right)\big|_\gamma \frac{|\Omega_2|^2}{\tilde y_1^{3+\epsilon}\tilde y_2^{1+\epsilon}} \sum_{\substack{m^{i\hat j}\in\mathbb{Z}^2 \\ \det m\neq 0}} e^{-\pi\sqrt{\frac{\tilde y_2}{\tilde y_1}}\Omega_{2\hat i\hat j}^{-1}v_{\hat i\hat j}m^{i\hat i}m^{j\hat j}}$$

$$\times \sum_{q_{ia}\in II_{4,4}} e^{-\pi\Omega_2^{ij}\tilde u^{ab}q_{ia}q_{jb}-\pi\mathrm{i}\Omega_1^{ij}\eta^{ab}q_{ia}q_{jb}+2\pi\mathrm{i}m^{i\hat j}q_{ia}\tilde a_{\hat j}{}^a} + \dots \; , \tag{3.56}$$

---

[23]The Poincaré series turns the vector $(0,0,m^{\hat i})$ into an arbitrary null vector $(n^{\hat i}, q_\alpha, m^{\hat i})$ with the same gcd. The trivial element gives $(0,0,m^{\hat i})$, elements in the first layer are vectors of type $(0, p_\alpha, m^{\hat i})$ while elements in the principal layer are $(n^{\hat i}, p_\alpha, m^{\hat i})$ with $n^{\hat i} \neq 0$.

where the ellipsis denotes non-abelian Fourier coefficients. In words, we first enforce the constraint by introducing the integral over $\Omega_1$, then rescale $\Omega_2$ to identify the sum as a Narain theta series over $II_{6,6}$ and use Poisson summation over $m_i^{\hat{j}}$, and in the last step convert the 'partial' $P_2 \backslash Sp(4, \mathbb{Z})$ Poincaré sum over linearly independent $(m^{i\hat{j}}, n_i^{\hat{j}})$ but with trivial symplectic product into a 'partial' Poincaré sum of $|\Omega_2|^\epsilon \varphi_{\mathrm{KZ}}^{\mathrm{tr}}(\Omega_2)$. Indeed, the sum over $(m^{i\hat{j}}, n_i^{\hat{j}})$ with $n_i^{\hat{j}}$ non-degenerate can be promoted to an $Sp(4, \mathbb{Z})$ invariant sum over doublet of symplectic vectors that are linearly independent. The $Sp(4, \mathbb{Z})$ orbit of doublets of symplectic vectors with a non-trivial symplectic product contribute to the non-abelian Fourier coefficient and can be computed similarly. The $Sp(4, \mathbb{Z})$ orbit of doublets of symplectic vectors with a vanishing symplectic product can be written as a Poincaré sum over $\gamma \in P_2 \backslash Sp(4, \mathbb{Z})$ of the representatives with $n_i^{\hat{j}}$ and $m^{i\hat{j}}$ non-degenerate, but only when the $2 \times 2$ matrix $C$ in the lower-left block of $\gamma$ is non-degenerate is the resulting $n_i^{\hat{j}} = C_{ij} m^{j\hat{j}}$ non-degenerate.

Now we shall argue that the missing terms in the Poincaré sum over $P_1 \backslash Sp(4, \mathbb{Z})$ only contribute to degenerate Fourier coefficients, such that the following refinement of (1.21) holds[24]

$$\lim_{\epsilon \to \infty} \sum_{\substack{\gamma \in P_2 \backslash Sp(4, \mathbb{Z}) \\ \det C(\gamma) \neq 0}} \left( |\Omega_2|^\epsilon \varphi_{\mathrm{KZ}}^{\mathrm{tr}}(\Omega_2) \right)\big|_\gamma = \varphi_{\mathrm{KZ}}(\Omega) - \varphi_{\mathrm{KZ}}^{\mathrm{tr}}(\Omega_2) - \sum_{\substack{M \in \mathcal{S}_+ \\ \det M = 0}}' F_M(\Omega_2) e^{2\pi i \mathrm{tr}[M\Omega_1]}, \quad (3.57)$$

where $\mathcal{S}_+$ is the set of symmetric matrices with positive integral diagonal components $M_{ii} \geq 0$ and half-integral off-diagonal $M_{12}$ that is moreover $> 0$ if $M_{11} = M_{22} = 0$. The function $F_M(\Omega_2)$ removes part of the Fourier coefficients of the KZ invariant in (A.11) supported on rank-one matrices. For $\epsilon \neq 0$, one expects, by analogy with the Siegel modular form $\sum_{\gamma \in P_2 \backslash Sp(4, \mathbb{Z})} \frac{E_{-3\Lambda_1}^{SL(2)}(\tau)}{V^{1+2\epsilon}}\big|_\gamma$, that the constant term at $\epsilon \neq 0$ takes the form

$$\sum_{\substack{\gamma \in P_2 \backslash Sp(4, \mathbb{Z}) \\ \det C(\gamma) \neq 0}} \left( |\Omega_2|^\epsilon \varphi_{\mathrm{KZ}}^{\mathrm{tr}}(\Omega_2) \right)\big|_\gamma \sim \frac{5\zeta(3)}{4\pi^2} V^2 E_{2\epsilon\Lambda_1}^{SL(2)}(\tau) + \frac{\pi}{36} \frac{\xi(4\epsilon - 1)}{\xi(4\epsilon)} \frac{\xi(-4+2\epsilon)\xi(3+2\epsilon)}{\xi(-3+2\epsilon)\xi(4+2\epsilon)} \frac{E_{-3\Lambda_1}^{SL(2)}}{V^{2-2\epsilon}}. \quad (3.58)$$

The second term vanishes in the limit $\epsilon \to 0$. Inserting the first term in the previous integral, one obtains

$$
\begin{aligned}
\mathcal{I}_6^{(5a)}(\phi, \epsilon) &= 8\pi \int_{\mathcal{G}} \frac{\mathrm{d}^3 \Omega_2}{|\Omega_2|^3} \frac{5\zeta(3)}{4\pi^2} \frac{E_{2\epsilon\Lambda_1}^{SL(2)}(\tau)}{|\Omega_2|} \sum_{\gamma \in P_2 \backslash SO(6,6)} \sum_{\substack{p_\alpha \in S_+ \\ k \geq 1 \\ \gcd(k,p)=1}} \frac{|\Omega_2|^2}{\tilde{y}_1^{3+\epsilon} \tilde{y}_2^{1+\epsilon}} \sum_{\substack{m^{i\hat{j}} \in \mathbb{Z}^2 \\ \det m \neq 0}} e^{-\pi \sqrt{\frac{\tilde{y}_2}{\tilde{y}_1}} \Omega_{2ij}^{-1} v_{\hat{i}\hat{j}} m^{i\hat{i}} m^{j\hat{j}}}\Bigg|_\gamma \\
&= \frac{20\zeta(3)}{\pi} \xi(2\epsilon)^2 \sum_{\gamma \in P_2 \backslash SO(6,6)} \sum_{\substack{p_\alpha \in S_+ \\ k \geq 1 \\ \gcd(k,p)=1}} \frac{g_4^{-2} y^{-4-2\epsilon} E_{2\epsilon\Lambda_1}^{SL(2)}(v)}{\left(1 + g_4^2 y^{-1} u^{\gamma\delta} (\frac{p_\gamma}{k} + a_\gamma)(\frac{p_\delta}{k} + a_\delta) + \frac{g_4^4}{4y^2}[(\frac{p_\gamma}{k} + a_\gamma)(\frac{p^\gamma}{k} + a^\gamma)]^2\right)^{\frac{3}{2}+\epsilon}}\Bigg|_\gamma
\end{aligned}
$$

---

[24] For the Siegel–Eisenstein series (A.25) with $s_1 = s_2 = s$, one checks that the sum over rank-one matrices $C$ gives the constant term $\frac{\xi(2s-1)}{\xi(2s)} \frac{E_{(2s-1)\Lambda_1}^{SL(2)}}{V}$ and contributes to the degenerate Fourier coefficients $e^{2\pi i \mathrm{tr}[M\Omega_1]}$ with $M$ rank-one. We shall argue in Appendix D that for $s_1$ and $s_2$ generic, the principal layer of the Poincaré sum over rank-two matrices $C$ gives all the constant terms of the two-parameter Siegel–Eisenstein series.

$$
= \frac{20\zeta(3)}{\pi}\xi(2\epsilon)^2\frac{\xi(2\epsilon-1)\xi(2\epsilon-4)}{\xi(2\epsilon)\xi(2\epsilon+3)}g_4^{-10}\sum_{\gamma\in P_2\backslash SO(6,6)}E_{2\epsilon\Lambda_1}^{SL(2)}(v)\Big(y^{-2\epsilon}+\dots\Big)\Big|_\gamma
$$

$$
= \frac{40\xi(2\epsilon)\xi(3)\xi(2-2\epsilon)\xi(5-2\epsilon)}{\xi(3+2\epsilon)}g_4^{-10}E_{2\epsilon\Lambda_1}^{D_6}+\dots\;,\tag{3.59}
$$

where the ellipses are Fourier coefficients. Here, we rewrote the sum over $(p,k)$ as a sum over unconstrained $(p,k)$ and $n=\frac{(p,p)}{2k}$ not zero, up to an overall factor of $\frac{1}{\zeta(3+2\epsilon)}$, and then performed a Poisson summation over $n$ and set $k$ to zero through the introduction of the theta lift of $E_{\epsilon\Lambda_1}^{SL(2)}$,

$$
\sum_{\substack{p_\alpha\in S_+\\k\geq 1\\\gcd(k,p)=1}}\frac{1}{\left(\frac{y}{g_4^2}+u^{\gamma\delta}(\frac{p_\gamma}{k}+a_\gamma)(\frac{p_\delta}{k}+a_\delta)+\frac{g_4^2}{4y}[(\frac{p_\gamma}{k}+a_\gamma)(\frac{p^\gamma}{k}+a^\gamma)]^2\right)^{\frac{3}{2}+\epsilon}}
$$

$$
= \frac{1}{\xi(3+2\epsilon)}\int_0^\infty\frac{d\rho_2}{\rho_2^2}\int_{-\frac12}^{\frac12}d\rho_1\rho_2^{\frac52+\epsilon}\sum_{\substack{n\in\mathbb{Z}\\p_\alpha\in S_+\\k\geq 1}}e^{-\pi\rho_2\left(\frac{y}{g_4^2}k^2+u(p+ak,p+ak)+\frac{g_4^2}{y}(n+\bar p a+\frac12\bar a ak)^2\right)+i\pi\rho_1(2nk-\bar p p)}
$$

$$
= \frac{g_4^{-1}y^{\frac12}}{\xi(3+2\epsilon)}\int_0^\infty d\rho_2\int_{-\frac12}^{\frac12}d\rho_1(E_{\epsilon\Lambda_1}^{SL(2)}(\rho)-\rho_2^\epsilon)\sum_{\substack{\tilde n\geq 1\\p_\alpha\in S_+}}e^{-\pi\rho_2 u(p,p)-\frac{\pi}{\rho_2}\frac{y}{g_4^2}\tilde n^2+2\pi i\tilde n\bar p a-i\pi\rho_1\bar p p}
$$

$$
= \frac{\xi(2\epsilon-1)\xi(2\epsilon-4)}{\xi(3+2\epsilon)\xi(2\epsilon)}g_4^{-5+2\epsilon}y^{\frac52-\epsilon}+\dots\;.\tag{3.60}
$$

Although several steps in the computation just outlined remain to be clarified, in Appendix D.1 we apply the same reasoning to a similar modular integral with $A(\tau)$ replaced by an Eisenstein series $E_{s\Lambda_1}^{SL(2)}$, and find that it reproduces the correct constant terms (namely the last three terms in (D.2)) predicted by Langlands' formula. This agreement is a strong indication that this reasoning is indeed correct.

In Section 5.2 we shall see that the sum of the contributions from the five layers to the perturbative part of the renormalised coupling (1.31) reproduce the expected terms in (3.1), including logarithmic terms in the string coupling constant, while the divergent one-loop contribution in $\mathcal{I}_6^{(5a)}(\phi,\epsilon)$ disappears in the renormalised function (1.28).

## 3.3 Fourier coefficients

Beyond the constant terms, our method also gives access to non-zero Fourier coefficients, which we now turn to.

The first source of Fourier coefficients comes from what was called the second term above, more specifically $n_iq_a^i\neq 0$ in (3.15). The corresponding terms simplify to

$$
\mathcal{I}_d^{(2d)} = 8\pi g_{\scriptscriptstyle D}^{-\frac{24}{8-d}}\int_{\mathcal{G}}\frac{d^3\Omega_2}{|\Omega_2|^3}\varphi_{\rm KZ}^{\rm tr}(\Omega_2)\sum_{\substack{q^i\in S_+\\q^i\times q^j=0}}\sum_{\substack{n_i\in\mathbb{Z}\\n_iq^i\neq 0}}e^{-\pi\Omega_2^{ij}n_in_j-\pi\Omega_{2ij}^{-1}g_{\scriptscriptstyle D}^{-2}g(q_i,q_j)+2\pi i n_i(q^i,a)}\;.\tag{3.61}
$$

To analyse this expression, it is convenient to unfold the integral domain $\mathcal{G}$ to the set of positive matrices $\mathbb{R}^+\times\mathcal{H}_1/\mathbb{Z}$ by fixing $n_i=(n,0)$ for $n>0$. Setting $N=nq^1$, one can solve the

constraint for $q^2$ in the $P_d \subset SO(d,d)$ parabolic decomposition associated to $N$ such that

$$
\begin{aligned}
\mathcal{I}_d^{(2d)} &= \frac{8\pi^2}{3} g_D^{-\frac{24}{8-d}} \int_0^\infty V \mathrm{d}V \int_0^\infty \frac{\mathrm{d}\tau_2}{\tau_2^2} \int_{-\frac{1}{2}}^{\frac{1}{2}} \mathrm{d}\tau_1 \, A(\tau) \sum_{\gamma \in P_d \backslash SO(d,d)} {\sum_{N \in \mathbb{N}}}' \sum_{n|N} \\
&\quad \times \sum_{\substack{j \in \mathbb{Z} \\ q \in \mathbb{Z}^{\frac{d(d-1)}{2}} \\ q \wedge q = 0}} e^{-\pi \left( \frac{n^2}{V\tau_2} + V\tau_2 \frac{y^2 N^2}{g_D^2 n^2} + \frac{V}{\tau_2 g_D^2}(y^2(j+(\varsigma,q) - \frac{\tau_1}{n}N)^2 + y^2 \frac{d-4}{d}|v(q)|^2)\right) + 2\pi i Na} \Big|_\gamma \\[2mm]
&= \frac{8\pi^2}{3} g_D^{-\frac{24}{8-d}+1} \int_0^\infty V^{\frac{1}{2}} \mathrm{d}V \int_0^\infty \frac{\mathrm{d}\tau_2}{\tau_2^{\frac{3}{2}}} \sum_{\gamma \in P_d \backslash SO(d,d)} {\sum_{N \in \mathbb{N}}}' \sum_{n|N} \sum_{\tilde{j} \in \mathbb{Z}} \int_{-\frac{1}{2}}^{\frac{1}{2}} \mathrm{d}\tau_1 \, A(\tau) e^{-2\pi i \tau_1 \frac{N\tilde{j}}{n}} \\
&\quad \times y^{-1} \sum_{\substack{q \in \mathbb{Z}^{\frac{d(d-1)}{2}} \\ q \wedge q = 0}} e^{-\pi \left( \frac{n^2}{V\tau_2} + V\tau_2 \frac{y^2 N^2}{g_D^2 n^2} + \frac{V}{\tau_2 g_D^2} y^2 \frac{d-4}{d}|v(q)|^2 + \frac{\tau_2 g_D^2}{V y^2} \tilde{j}^2 \right) + 2\pi i(\tilde{j}(q,\varsigma) + Na)} \Big|_\gamma \qquad (3.62)
\end{aligned}
$$

For $\tilde{j} \neq 0$ and $q \neq 0$, this term involves the integral of the Fourier coefficient of $A(\tau)$ with a saddle point at

$$
\tau_2 = n \sqrt{\frac{y^{\frac{d-4}{d}}|v(q)|}{g_D \tilde{j} N}} \, , \qquad V = \frac{n}{y} \sqrt{\frac{g_D^3 \tilde{j}}{y^{\frac{d-4}{d}}|v(q)|N}} \, , \qquad (3.63)
$$

which is exponentially suppressed in $e^{-2\pi \frac{y}{g_D}N - 2\pi y^{-\frac{4}{d}}|v(\tilde{j}q)|}$. One can compute explicitly the contribution from the leading part (3.20) of the constant term of $A(\tau)$, and similarly for its Fourier coefficients. Using the same method as in (3.26), (3.27) one solves the differential equation for the Fourier coefficients[25]

$$
\left( \tau_2^2 \frac{\partial^2}{\partial \tau_2^2} - (2\pi \tilde{j} \tau_2)^2 - 12 \right) \int_{-\frac{1}{2}}^{\frac{1}{2}} \mathrm{d}\tau_1 A(\tau) e^{-2\pi i \tilde{j} \tau_1} \qquad (3.64)
$$

$$
= -12\tau_2 - 24\tau_2 \sum_{\substack{a,c \geq 1 \\ \gcd(a,c)=1}} \frac{H(1 - (2ac\tau_2)^2)}{\sqrt{1 - (2ac\tau_2)^2}} \cos\left(2\pi \tilde{j}\left(\frac{1}{2ac} + \frac{b}{a}\right)\right) \cos\left(\frac{\pi \tilde{j}}{ac}\sqrt{1 - (2ac\tau_2)^2}\right)
$$

where $b$ is the solution modulo $a$ to $ad - bc = 1$. One finds the unique continuous solution that reproduces $A(\tau)$ for $\tau_2 > \frac{1}{2}$

$$
\begin{aligned}
\int_{-\frac{1}{2}}^{\frac{1}{2}} \mathrm{d}\tau_1 A(\tau) e^{-2\pi i \tilde{j} \tau_1} &= \frac{3}{(\tilde{j}\pi)^2 \tau_2} - \frac{15}{2(\tilde{j}\pi)^4 \tau_2^3} \\
&\quad + \sum_{\substack{a,c \geq 1 \\ \gcd(a,c)=1 \\ 2ac < 1/\tau_2}} \cos\left(2\pi \tilde{j}\left(\frac{1}{2ac} + \frac{b}{a}\right)\right) \left( 3\left( \frac{75(ac)^2}{(\pi\tilde{j})^6 \tau_2^3} + \frac{2}{(\pi\tilde{j})^2 \tau_2} - \frac{5}{(\pi\tilde{j})^4 \tau_2^3}(1 - (2ac\tau_2)^2) \right) \right)
\end{aligned}
$$

---

[25] For each positive coprime $a$ and $c$ there are two solutions $\tau_1 = -\frac{1}{2ac} - \frac{b}{a} \pm \frac{\sqrt{1-(2ac\tau_2)^2}}{2ac}$ where $b$ is the same modulo $a$, leading to the same source term as in (3.26) multiplied by $e^{2\pi i \tilde{j}(\frac{1}{2ac} + \frac{b}{a})} \cos(\frac{\pi\tilde{j}}{ac}\sqrt{1-(2ac\tau_2)^2})$. Because the function $A(\tau)$ is even in $\tau_1$, its Fourier coefficients are real. For each coprime $a, c$, there is the permuted pair $c, a$, with $b$ and $-d$ permuted, and the contribution carries the complex conjugate phase $e^{2\pi i \tilde{j}(\frac{1}{2ac} - \frac{d}{c})} = e^{2\pi i \tilde{j}(-\frac{1}{2ac} - \frac{b}{a})}$.

$$\times \left( \sqrt{1-(2ac\tau_2)^2}\cos\left(\tfrac{\pi\tilde{\jmath}}{ac}\sqrt{1-(2ac\tau_2)^2}\right) - \tfrac{ac}{\pi\tilde{\jmath}}\sin\left(\tfrac{\pi\tilde{\jmath}}{ac}\sqrt{1-(2ac\tau_2)^2}\right)\right)$$
$$+ \frac{75ac}{(\pi\tilde{\jmath})^5\tau_2^3}(1-(2ac\tau_2)^2)\sin\left(\tfrac{\pi\tilde{\jmath}}{ac}\sqrt{1-(2ac\tau_2)^2}\right)\Bigg). \quad (3.65)$$

The saddle point (3.63) is at large $\tau_2$ at small coupling $g_D$, therefore the contributions from $A(\tau)$ at $\tau_2 < \frac{1}{2}$ will be further exponentially suppressed and at leading order one can neglect them. The integral gives then

$$\mathcal{I}_d^{(2d)} \sim \frac{16\pi^2}{3} g_D^{-\frac{24}{8-d}+3} \sum_{\substack{N\in S_+\\N\times N=0}}' \sigma_2(N) \Bigg( \frac{y_N^{\frac{4}{d}-3}}{\gcd(N)}\xi(2)E_{\Lambda_1}^{SL_d}(v_N)K_1\big(2\pi\tfrac{\sqrt{g(N,N)}}{g_D}\big)$$
$$+ g_D^2 \frac{y_N^{\frac{20}{d}-5}}{6\gcd(N)}\xi(5)E_{\frac{5}{2}\Lambda_2}^{SL_d}(v_N)K_1\big(2\pi\tfrac{\sqrt{g(N,N)}}{g_D}\big)$$
$$+ \frac{3g_D}{\pi^2\gcd(N)^2 y_N^4} \sum_{\substack{Q\in\mathbb{Z}^{\frac{d(d-1)}{2}}\\Q\wedge Q=0}}' \sigma_1(Q)e^{2\pi i(Q,\varsigma_N)} \frac{K_{\frac{3}{2}}(2\pi y_N^{-\frac{4}{d}}|v_N(Q)|)}{(y_N^{-\frac{4}{d}}|v_N(Q)|)^{\frac{3}{2}}} K_0\big(2\pi\tfrac{\sqrt{g(N,N)}}{g_D}\big)$$
$$- \frac{15g_D^2}{2\pi^4\gcd(N)^3 y_N^5} \sum_{\substack{Q\in\mathbb{Z}^{\frac{d(d-1)}{2}}\\Q\wedge Q=0}}' \sigma_1(Q)e^{2\pi i(Q,\varsigma_N)} \frac{K_{\frac{5}{2}}(2\pi y_N^{-\frac{4}{d}}|v_N(Q)|)}{(y_N^{-\frac{4}{d}}|v_N(Q)|)^{\frac{5}{2}}} K_1\big(2\pi\tfrac{\sqrt{g(N,N)}}{g_D}\big)\Bigg) e^{2\pi i(N,a)} \quad (3.66)$$

where we kept the variable $y_N = \frac{\sqrt{g(N,N)}}{\gcd(N)}$ for simplicity, and the sum over $Q\in\mathbb{Z}^{\frac{d(d-1)}{2}}$ is a sum over characters of the unipotent stabilisers of the charge $N$. The leading term in $g_D$ factorises as an Eisenstein series over the Levi stabiliser of $N$, while the full Fourier coefficient depends non-trivially on the whole parabolic stabiliser.

The neglected terms in (3.65) give rise to integrals over the truncated domain $\tau_2 \in [0,\frac{1}{2ac}]$ for any coprime $a$ and $c$, which are therefore further exponentially suppressed. As we shall see, these corrections can be ascribed to instanton anti-instanton corrections, similarly to (3.29) for the constant term. To see this, it is convenient to do the inverse Poisson summation over $\tilde{\jmath}$. Note that the function $f_{a,c,\tilde{\jmath}}(\tau_2)$ appearing in the sum over coprime $a,c$

$$\int_{-\frac{1}{2}}^{\frac{1}{2}} d\tau_1 A(\tau)e^{-2\pi i\tilde{\jmath}\tau_1} = \frac{3}{(\tilde{\jmath}\pi)^2\tau_2} - \frac{15}{2(\tilde{\jmath}\pi)^4\tau_2^3} + \sum_{\substack{a,c\geq 1\\\gcd(a,c)=1\\2ac<1/\tau_2}} f_{a,c,\tilde{\jmath}}(\tau_2) \quad (3.67)$$

is regular at $\tilde{\jmath}=0$ and gives

$$f_{a,c,0}(\tau_2) = -\frac{1+\frac{3}{2}(2ac\tau_2)^2+(2ac\tau_2)^4}{7ac(ac\tau_2)^3}(1-(2ac\tau_2)^2)^{\frac{3}{2}} \quad (3.68)$$

as in (3.27). The Poisson formula involves the inverse Fourier transform

$$\tilde{f}_{a,c,j}(\tau_2) = \int_{\mathbb{R}} d\tilde{\jmath}\, e^{2\pi ij\tilde{\jmath}} f_{a,c,\frac{N}{n}\tilde{\jmath}}(\tau_2)\, e^{-\pi\frac{\tau_2 g_D^2}{Vy^2}\tilde{\jmath}^2} \quad (3.69)$$

which evaluates to

$$\tilde{f}_{a,c,j}(\tau_2) = \left(P^{(1)}_{a,c,j}(\tau_2) + P^{(2)}_{a,c,j}(\tau_2)\sqrt{1-(2ac\tau_2)^2}\right)e^{-\pi\frac{Vy^2}{\tau_2 g_D^2}\frac{[2acn(j+(q,\varsigma))+N(1+2bc+\sqrt{1-(2ac\tau_2)^2})]^2}{(2acn)^2}}$$

$$- \left(P^{(1)}_{a,c,j}(\tau_2) - P^{(2)}_{a,c,j}(\tau_2)\sqrt{1-(2ac\tau_2)^2}\right)e^{-\pi\frac{Vy^2}{\tau_2 g_D^2}\frac{[2acn(j+(q,\varsigma))+N(1+2bc-\sqrt{1-(2ac\tau_2)^2})]^2}{(2acn)^2}}$$

$$+ P^{(0)}_{a,c,j}(\tau_2)\left(\text{erf}\left(\sqrt{\pi\tfrac{V}{\tau_2}}\tfrac{y}{g_D}\tfrac{2acn(j+(q,\varsigma))+N(1+2bc+\sqrt{1-(2ac\tau_2)^2})}{2acn}\right) - \text{erf}\left(\sqrt{\pi\tfrac{V}{\tau_2}}\tfrac{y}{g_D}\tfrac{2acn(j+(q,\varsigma))+N(1+2bc-\sqrt{1-(2ac\tau_2)^2})}{2acn}\right)\right) \quad (3.70)$$

Here $P^{(k)}_{a,c,j}$ are polynomials in the various parameters which we omit since they are not particularly illuminating. In the saddle point approximation, one computes that these corrections are exponentially suppressed with the action[26]

$$S_{I\bar{I}} = \frac{2\pi}{g_D}\sqrt{y^2(N+N_1+(\varsigma,Q))^2 + y^{2-\frac{8}{d}}|v(Q)|^2} + \frac{2\pi}{g_D}\sqrt{y^2(N_1+(\varsigma,Q))^2 + y^{2-\frac{8}{d}}|v(Q)|^2} \quad (3.71)$$

which corresponds to the sum of the actions of an instanton of charges $(N+N_1, Q)$ and anti-instanton of charge $(-N_1, -Q)$ with

$$N_1 = bcN + acnj , \qquad Q = acnq . \quad (3.72)$$

It is convenient to change variables to $N_1$, $Q$ and

$$\tau_2 = \frac{t}{2ac} , \qquad V = acn^2\nu , \quad (3.73)$$

and define

$$F\left(t, \tfrac{y^2}{g_D^2}\nu, N, N_1+(\varsigma,Q)\right) = \frac{\sqrt{2}}{n}\tilde{f}_{a,c,j}(\tau_2) , \quad (3.74)$$

where the dependence on the arguments is made explicit in $F$. Then the complete function $\mathcal{I}^{(2d)}_d$ reduces to the sum of (3.66) and

$$\frac{8\pi^2}{3}g_D^{-\frac{24}{8-d}+1}\int_0^\infty \nu^{\frac{1}{2}}\mathrm{d}\nu\int_0^1\frac{\mathrm{d}t}{t^{\frac{3}{2}}}\sum_{\gamma\in P_d\backslash SO(d,d)}\sideset{}{'}\sum_{N\in\mathbb{N}}e^{2\pi i Na_\gamma}\sum_{\substack{q\in\mathbb{Z}^{\frac{d(d-1)}{2}}\\q\wedge q=0}}\sum_{j\in\mathbb{Z}}\sum_{n|N}\sum_{\substack{a,c\geq 1\\ \gcd(a,c)=1}}a^2c^2n^4$$

$$\times y^{-1}e^{-\pi\left(\frac{2}{\nu t}+\nu t\frac{y^2 N^2}{2g_D^2}+\frac{2\nu}{tg_D^2}y^{2\frac{d-4}{d}}|v(acnq)|^2\right)}F\left(t, \tfrac{y^2}{g_D^2}\nu, N, bcN+acj+(\varsigma,acq)\right)\Bigg|_\gamma$$

$$= \frac{8\pi^2}{3}g_D^{-\frac{24}{8-d}+1}\int_0^\infty \nu^{\frac{1}{2}}\mathrm{d}\nu\int_0^1\frac{\mathrm{d}t}{t^{\frac{3}{2}}}\sum_{\gamma\in P_d\backslash SO(d,d)}\sideset{}{'}\sum_{N\in\mathbb{N}}e^{2\pi i Na_\gamma}\sideset{}{'}\sum_{\substack{N_1\in\mathbb{Z}\\Q\in\mathbb{Z}^{\frac{d(d-1)}{2}}\\Q\wedge Q=0}}\sigma_2(N_1,Q)\sigma_2(N+N_1,Q)$$

---

[26]To do this computation it is convenient to introduce $z_1 = \sqrt{y^2(N+N_1+(\varsigma,Q))^2 + y^{2-\frac{8}{d}}|v(Q)|^2}$ and $z_2 = \sqrt{y^2(N_1+(\varsigma,Q))^2 + y^{2-\frac{8}{d}}|v(Q)|^2}$. The saddle point for $e^{-\pi\frac{Vy^2}{\tau_2 g_D^2}\frac{[2acn(j+(q,\varsigma))+N(1+2bc+\sqrt{1-(2ac\tau_2)^2})]^2}{(2acn)^2}}$ lies within the integration domain when $z_2 > z_1$ and the action takes the minimum value $\frac{2\pi}{g_D}(z_1+z_2)$, whereas the saddle point for $e^{-\pi\frac{Vy^2}{\tau_2 g_D^2}\frac{[2acn(j+(q,\varsigma))+N(1+2bc-\sqrt{1-(2ac\tau_2)^2})]^2}{(2acn)^2}}$ lies within the integration domain when $z_1 > z_2$ and attains the same minimum value. The error functions involve the same exponential in their asymptotic expansion at large $x$ using $\text{erf}(x) = \text{sign}(x) - \frac{1}{\pi x}e^{-x^2}\sum_{k=0}^\infty \Gamma(\tfrac{1}{2}+k)(-\tfrac{1}{x^2})^k$.

$$\times \, y^{-1} e^{-\pi\left(\frac{2}{\nu t}+\nu t\frac{y^2 N^2}{2g_D^2}+\frac{2\nu}{tg_D^2}y^{2\frac{d-4}{d}}|v(Q)|^2\right)} F\big(t,\tfrac{y^2}{g_D^2}\nu, N, N_1 + (\varsigma, Q)\big)\bigg|_\gamma \qquad (3.75)$$

where we used the property that $cn$ divides $(N_1, Q)$ and $an$ divides $(N+N_1, Q)$ using $1+bc = ad$. Note that the case $(N_1, Q) = 0$ is excluded from the second sum: In this case $a$ and $j$ are fixed such that $b\frac{N}{n} + aj = 0$ and the sum over $c$ (after replacing $a^2 c^2 n^4$ by $(acn^2)^{2-2\epsilon}$ in dimensional regularisation) leads to a factor of $\zeta(2\epsilon - 2)$ which vanishes at $\epsilon = 0$. The factors $\sigma_2(N_1, Q)$ and $\sigma_2(N + N_1, Q)$ are the measure factors of the 1/2-BPS instanton and anti-instantons, see [5, 67].

We conclude that the dominant contribution (3.66) to $\mathcal{I}_d^{(2d)}$ is of the expected form to correspond to 1/2-BPS Euclidean D-brane instantons, with the spinor $N$ identified as the D-brane charge satisfying the 1/2-BPS constraint $N \times N = 0$. The overall factor of $\sigma_2(N)/N^2$ is recognised as the partition function of the world-volume theory of N Euclidean D$p$-branes on the torus [67]. For a D$(d-1)$ Euclidean brane instanton $y_N^{-\frac{4}{d}} v_N$ defines the string frame metric and $\varsigma_N$ the $B$ field components along the torus, so that the sum over $Q$ in (3.66) can be interpreted as contributions from world-sheet instantons over the Euclidean brane background. The subleading correction (3.75) can instead be interpreted as the instanton anti-instanton corrections, which also carry the measure factor of the two constitutive instantons.

The second source of Fourier coefficients comes from the third layer of charges, more specifically from (3.42) with $q_a^i n_{i\hat{j}} \neq 0$ or $np^\alpha \neq 0$,

$$\mathcal{I}_d^{(3b)} = 8\pi g_D^{-\frac{24}{8-d}+2} \sum_{\gamma \in P_{d-2}\backslash D_d} \left( \sideset{}{'}\sum_{n_{i\hat{j}} \in \mathbb{Z}^2} \int_{\mathcal{G}} \frac{\mathrm{d}^3\Omega_2}{|\Omega_2|^2} \varphi_{\mathrm{KZ}}^{\mathrm{tr}}(\Omega_2) \frac{e^{-\pi\Omega_2^{ij} v_{\hat{k}\hat{l}} n_i{}^{\hat{k}} n_j{}^{\hat{l}}}}{y \Omega_2^{ij} v_{\hat{k}\hat{l}} \hat{n}_i{}^{\hat{k}} \hat{n}_j{}^{\hat{l}}} \right. \qquad (3.76)$$

$$\left. \times \sum_{\substack{q_a^i \in \mathbb{Z}^{d-2} \\ p^\alpha \in \mathbb{Z}^2}} e^{-\frac{\pi}{g_D^2}\left(y^{\frac{d-4}{d-2}}\Omega_{2ij}^{-1} u^{ab} q_a^i q_b^j + \frac{y}{\Omega_2^{ij} v_{\hat{k}\hat{l}} \hat{n}_i{}^{\hat{k}} \hat{n}_j{}^{\hat{l}}} \rho_{\alpha\beta}(p^\alpha - c_{\hat{k}}^{a\alpha}\hat{n}_i{}^{\hat{k}} q_a^i)(p^\beta - c_{\hat{l}}^{b\beta}\hat{n}_j{}^{\hat{l}} q_b^j)\right)+2\pi \mathrm{i}(q_a^i n_{i\hat{j}} a_a^a + np^\alpha a_\alpha)} \right)$$

The integral over $\mathcal{G}$ can be unfolded to $\mathbb{R}^+ \times \mathcal{H}_1$ at the expense of restricting the sum over $n_i{}^{\hat{j}}$ to $\mathbb{Z}^{2\times 2}/GL(2,\mathbb{Z})$. The integral over $\Omega_2$ is once again dominated by a saddle point as $g_D \to \infty$. The modulus of the exponential is of the form $e^{-S(\Omega)}$ where $S$ is the 'action'

$$S(\Omega_2) = \pi\Big(\mathrm{Tr}\,\Omega_2 Y + \mathrm{Tr}\,\Omega_2^{-1} X + \frac{M}{\mathrm{Tr}\,\Omega_2 Y}\Big) \qquad (3.77)$$

where $X, Y$ are symmetric positive matrices and $M > 0$. The extremum with respect to $\Omega_2$ is given by

$$\Omega_2^\star = \frac{\sqrt{M + \mathrm{Tr}\,XY + 2\sqrt{|XY|}}}{\mathrm{Tr}\,XY + 2\sqrt{|XY|}}\big(X + \sqrt{|XY|}Y^{-1}\big)\,, \qquad (3.78)$$

and satisfies

$$S(\Omega_2^\star) = 2\pi\sqrt{M + \mathrm{Tr}\,XY + 2\sqrt{|XY|}}\,. \qquad (3.79)$$

Provided the integrand $\varphi_{\mathrm{KZ}}^{\mathrm{tr}}(\Omega_2)$ is continuous around $\Omega_2^\star$, the integral in the saddle point approximation reduces to

$$\int_{\mathcal{H}(\mathbb{R})} \frac{\mathrm{d}^3\Omega_2}{|\Omega_2|^2} \frac{\varphi_{\mathrm{KZ}}^{\mathrm{tr}}(\Omega_2)}{\mathrm{Tr}\,\Omega_2 Y} e^{-S(\Omega_2)} \sim \frac{\varphi_{\mathrm{KZ}}^{\mathrm{tr}}(X + \sqrt{|XY|}Y^{-1}) e^{-2\pi\sqrt{M+\mathrm{Tr}\,XY+2\sqrt{|XY|}}}}{\sqrt{|X|}(\mathrm{Tr}\,XY + 2\sqrt{|XY|})(M + \mathrm{Tr}\,XY + 2\sqrt{|XY|})^{\frac{1}{4}}}\,. \qquad (3.80)$$

For the integral (3.76) one obtains, setting $P^\alpha = np^\alpha$ and $Q_a^{\hat{i}} = n_j^{\hat{i}} q_a^j$,

$$S(\Omega_2^\star) = \frac{2\pi}{g_D}\sqrt{y\rho_{\alpha\beta}(P^\alpha - c_{\hat{i}}^{a\alpha}Q_a^{\hat{i}})(P^\beta - c_{\hat{j}}^{b\beta}Q_b^{\hat{j}}) + y^{\frac{d-4}{d-2}}\left(v_{\hat{i}\hat{j}}u^{ab}Q_a^{\hat{i}}Q_b^{\hat{j}} + 2\sqrt{(u^{ab}u^{cd} - u^{ac}u^{bd})Q_a^{\hat{1}}Q_b^{\hat{1}}Q_c^{\hat{2}}Q_d^{\hat{2}}}\right)} \quad (3.81)$$

which is identified as the classical action for a 1/4-BPS D-brane of charge $N \in S_+$ with $\bar{N}\gamma_{d-4}N \neq 0$, (and $\gamma_2 N \cdot (\bar{N}\gamma_{d-4}N) = 0$ in $\wedge^{d-6}II_{d,d}$ for $d \geq 6$)

$$S(\Omega_2^\star) = \frac{2\pi}{g_D}\sqrt{|v(N)|^2 + \sqrt{2|\overline{v(N)}\gamma_{d-4}v(N)|^2}} \; . \quad (3.82)$$

We shall now express the Fourier coefficients (3.76) in a covariant fashion by resolving the sum over $P_{d-2}\backslash D_d$. For each non-zero spinor $N \in S_+$, one has a sum over the doublets of spinor $\chi_i \in S_-$ satisfying instead $\bar{\chi}_i\gamma_{d-4}\chi_j = 0$ such that $(\bar{\chi}_i\gamma_{d-2}\chi_j) \cdot \gamma_2 N = 0$ in $\wedge^{d-4}II_{d,d}$ and such that

$$\frac{1}{\gcd(\bar{\chi}_i\gamma_{d-2}\chi_j)}\bar{\chi}_i\gamma_{d-3}N \in \wedge^{d-3}II_{d,d} \;, \qquad \frac{1}{\gcd(\chi_i)}N \in S_+ \; . \quad (3.83)$$

Introducing the same notation $g(\cdot,\cdot)$ for the metric on any module of $Spin(d,d)$ parametrised by the coset $SO(d,d)/(SO(d) \times SO(d))$, and unfolding the integration domain against the sum over $\chi_i$, one can write (3.76) as

$$
\begin{aligned}
\mathcal{I}_d^{(3b)} &= 16\pi g_D^{-\frac{24}{8-d}+2} \sum_{\substack{N \in S_+ \\ \gamma_2 N \cdot (\bar{N}\gamma_{d-4}N)=0}}' e^{2\pi i \bar{N}a} \sum_{\substack{\chi_i \in S_- \otimes \mathbb{Z}^2/GL(2,\mathbb{Z}) \\ \bar{\chi}_i\gamma_{d-4}\chi_j=0,\, \chi_i\gamma_{d-2}\chi_j\neq 0 \\ \bar{\chi}_i\gamma_{d-2}\chi_j \cdot \gamma_2 N=0 \\ \frac{1}{\gcd(\bar{\chi}_i\gamma_{d-2}\chi_j)}\bar{\chi}_i\gamma_{d-3}N\in\wedge^{d-3}II_{d,d} \\ \frac{1}{\gcd(\chi_i)}N\in S_+}} \int_{\mathcal{H}(\mathbb{R})} \frac{d^3\Omega_2}{|\Omega_2|^2}\varphi_{KZ}^{tr}(\Omega_2)\frac{\gcd(\chi_i)^2}{\Omega_2^{ij}g(\chi_i,\chi_j)} \\
&\quad \times e^{-\pi\Omega_2^{ij}g(\chi_i,\chi_j)-\frac{\pi}{g_D^2}\left(\frac{\Omega_{2ij}^{-1}\varepsilon^{ik}\varepsilon^{jl}g(\bar{\chi}_k\gamma_{d-3}N,\bar{\chi}_l\gamma_{d-3}N)}{g(\bar{\chi}_1\gamma_{d-2}\chi_2,\bar{\chi}_1\gamma_{d-2}\chi_2)} + \frac{g(N,N)}{\Omega_2^{ij}g(\chi_i,\chi_j)} - \frac{\varepsilon^{ik}\varepsilon^{jl}g(\chi_i,\chi_j)g(\bar{\chi}_k\gamma_{d-3}N,\bar{\chi}_l\gamma_{d-3}N)}{\Omega_2^{ij}g(\chi_i,\chi_j)g(\bar{\chi}_1\gamma_{d-2}\chi_2,\bar{\chi}_1\gamma_{d-2}\chi_2)}\right)} \\[2mm]
&\sim 8\pi g_D^{-\frac{24}{8-d}+4+\frac{1}{4}} \sum_{\substack{N \in S_+ \\ \Gamma_2 N \cdot (\bar{N}\Gamma_{d-4}N)=0}}' \frac{e^{2\pi i \bar{N}a - \frac{2\pi}{g_D}\sqrt{g(N,N)+2\sqrt{g(\bar{N}\gamma_{d-4}N,\bar{N}\gamma_{d-4}N)}}}}{(g(N,N) + 2\sqrt{g(\bar{N}\gamma_{d-4}N,\bar{N}\gamma_{d-4}N)})^{\frac{1}{4}}} \\
&\quad \times \sum_{\substack{\chi_i \in S_- \\ \bar{\chi}_i\gamma_{d-4}\chi_j=0,\, \chi_i\gamma_{d-2}\chi_j\neq 0 \\ \bar{\chi}_i\gamma_{d-2}\chi_j \cdot \gamma_2 N=0 \\ \frac{1}{\gcd(\bar{\chi}_i\gamma_{d-2}\chi_j)}\bar{\chi}_i\gamma_{d-3}N\in\wedge^{d-3}II_{d,d} \\ \frac{1}{\gcd(\chi_i)}N\in S_+}} \frac{\gcd(\chi_i)^2}{\sqrt{|g(\chi_i,\chi_j)|}}\varphi_{KZ}^{tr}\big[\big(\tfrac{g(\bar{\chi}_i\gamma_{d-3}N,\bar{\chi}_j\gamma_{d-3}N)}{g(\bar{\chi}_1\gamma_{d-2}\chi_2,\bar{\chi}_1\gamma_{d-2}\chi_2)} + \tfrac{\sqrt{2g(\bar{N}\gamma_{d-4}N,\bar{N}\gamma_{d-4}N)}g(\chi_i,\chi_j)}{2|g(\chi_k,\chi_l)|}\big)^{-1}\big] \; .
\end{aligned}
\quad (3.84)
$$

where we used the same saddle point approximation as above in the second step. To find this formula one can use the fact that the sum over non-zero $\chi_i$ decomposes into the Poincaré sum over $P_{d-2}\backslash D_d$ and $\chi_i = (0, 0, n_i^{\hat{j}})$. Then $\bar{\chi}_i\gamma_{d-2}\chi_j = \varepsilon_{ij}(\ldots, 0, \det n)$, and the constraint $\bar{\chi}_i\gamma_{d-2}\chi_j \cdot \gamma_2 N = 0$ imposes that $N = (0, Q_a^{\hat{i}}, P^\alpha)$. Then the only non-zero component of $\bar{\chi}_i\gamma_{d-3}N$ is $\varepsilon_{\hat{i}\hat{j}}n_i^{\hat{i}}Q_a^{\hat{j}}$, so that the one of $\frac{1}{\gcd(\bar{\chi}_i\gamma_{d-2}\chi_j)}\bar{\chi}_i\gamma_{d-3}N$ is $-\varepsilon_{ij}n_{\hat{i}}^{-1\,j}Q_a^{\hat{i}}$, which in order to be an integers requires that $Q_a^{\hat{i}} = n_j^{\hat{i}}q_a^j$ for an integral $q_a^i$. Finally the condition that $\frac{1}{\gcd(\chi_i)}N \in S_+$ is automatically satisfied for $Q_a^{\hat{i}}$ and requires that $P^\alpha$ be divisible by $\gcd(n_i^{\hat{j}})$.

In $D = 4$ there are additional contributions from the last orbit to the Fourier coefficients. For generic abelian Fourier coefficients, one can insert the Fourier expansion (A.11) in (3.56) without having to worry about the discrepancy (3.57). The integral over $\Omega_1$ sets the matrix $M$ in (A.11) equal to $\frac{1}{2}\eta_{ab}q_i^a q_j^b$. As the Fourier coefficient is defined by the charge $Q_{\hat{i}}^a = m_{\hat{i}}^{\ j} q_j^a$, we recast the sum over $q_i^a$ and $m_{\hat{i}}^{\ j}$ into a sum over integral charges $Q_{\hat{i}} \in II_{4,4}$ and matrices $A_{\hat{i}}^{\ j}$ dividing them in $II_{4,4}$. One obtains in this way

$$\mathcal{I}_6^{(5c)} = \sum_{\gamma \in P_2 \backslash SO(6,6)} \left( \frac{1}{g_4^4 y^4} \sum_{\substack{Q \in \mathbb{Z}^2 \otimes II_{4,4} \\ \Delta(Q) \geq 1}} \sum_{\substack{A \in \mathbb{Z}^{2\times 2}/GL(2,\mathbb{Z}) \\ A^{-1}Q \in \mathbb{Z}^2 \otimes II_{4,4}}} \sum_{d | A^{-1} Q \cdot Q^{\mathsf{T}} A^{-\mathsf{T}}} |A|^{-1} d^{-3} \tilde{c}\left(\tfrac{\Delta(Q)}{|A|^2 d^2}\right) \qquad (3.85)$$

$$\times \sum_{\substack{k \geq 1 \\ p \in S_+ \\ \gcd(k,p)=1 \\ \frac{(p,p)}{2k} \in \mathbb{Z} \\ \frac{A^{-1}Qp}{k} \in S_-}} \int_{\mathcal{H}_+} \frac{\mathrm{d}^3\Omega_2}{|\Omega_2|} \left( \frac{4\pi\Delta(Q)}{L(\frac{p}{k}+a)} + \frac{5}{\pi|\Omega_2|}\left( \frac{1}{L(\frac{p}{k}+a)^3} + \frac{\pi}{L(\frac{p}{k}+a)^2} \mathrm{tr}[\Omega_2 vQ \cdot Q^{\mathsf{T}} v^{\mathsf{T}}] \right) \right)$$

$$\times e^{-\pi \mathrm{tr}\left[v^{\mathsf{T}}\Omega_2 v\left(L(p+a)Q \cdot Q^{\mathsf{T}} + QuQ^{\mathsf{T}} + \frac{g_4^2}{y}(\frac{p}{k}+a)^{\mathsf{T}} Q u Q(\frac{p}{k}+a)\right)\right] - \frac{\pi}{g_4^2}\mathrm{tr}[\Omega_2^{-1}] + 2\pi i(Q, a + c\gamma^a(\frac{p}{k}+a))} \right) \Bigg|_\gamma$$

where

$$\Delta(Q) = \det[\eta^{ab} Q_{\hat{i}a} Q_{\hat{j}b}] , \qquad L(p) = \sqrt{1 + \frac{g_4^2}{y} u^{\alpha\beta} p_\alpha p_\beta + \frac{g_4^4}{4y^2}(p,p)^2} . \qquad (3.86)$$

Note that $\Delta(Q)$ is the usual quartic invariant of electromagnetic charge vectors in an $\mathcal{N} = 4$ truncation of $\mathcal{N} = 8$ supergravity [68].

To exhibit the Fourier expansion, we still need to decompose the sum over $(k,p)$ into $p$ mod $k$ and the integral part $p'$, and Poisson resum over $p'$. We define the function

$$f_Q(\tfrac{g_4 x}{\sqrt{y}}) = \int_{\mathcal{H}_+} \frac{\mathrm{d}^3\Omega_2}{|\Omega_2|} \left( \frac{4\pi\Delta(Q)}{L(x)} + \frac{5}{\pi|\Omega_2|}\left( \frac{1}{L(x)^3} + \frac{2\pi}{L(x)^2}\mathrm{tr}[\Omega_2 vQ \cdot Q^{\mathsf{T}} v^{\mathsf{T}}] \right) \right)$$

$$e^{-\pi \mathrm{tr}\left[v^{\mathsf{T}}\Omega_2 v\left(L(x)Q \cdot Q^{\mathsf{T}} + QuQ^{\mathsf{T}} + \frac{g_4^2}{y} x^{\mathsf{T}} Q u Q x\right)\right] - \frac{\pi}{g_4^2}\mathrm{tr}[\Omega_2^{-1}]} , \quad (3.87)$$

which can be evaluated in terms of matrix variate Bessel functions [80, 50], and its Fourier transform

$$\tilde{f}_Q(\chi) = \int \mathrm{d}^8 x \, f_Q(x) e^{2\pi i(\chi,x)} , \qquad (3.88)$$

where we have rescaled variables such that $\tilde{f}_Q(\chi)$ does not depend on $y$. While we do not have an explicit formula for $\tilde{f}_Q(\chi)$, we note that it is a well-defined, absolutely convergent integral. Moreover, we expect that it should have the characteristic exponential suppression for 1/8-BPS D-brane instantons. The generic Fourier coefficients can be written as[27]

---

[27]The condition $d|Q \cdot Q^{\mathsf{T}}$ is a shorthand notation for $d|(Q_1,Q_1)/2, (Q_2,Q_2)/2, (Q_1,Q_2)$.

$$\mathcal{I}_6^{(5c)} = \sum_{\gamma \in P_2 \backslash SO(6,6)} \left( \frac{1}{g_4^{12}} \sum_{\substack{Q \in \mathbb{Z}^2 \otimes II_{4,4} \\ \Delta(Q) \geq 1}} \sum_{\chi \in S_+} \sum_{\substack{A \in \mathbb{Z}^{2 \times 2}/GL(2,\mathbb{Z}) \\ A^{-1}Q \in \mathbb{Z}^2 \otimes II_{4,4}}} \sum_{d | A^{-1}Q \cdot Q^\intercal A^{-\intercal}} |A|^{-1} d^{-3} \tilde{c}\left(\frac{\Delta(Q)}{|A|^2 d^2}\right) \right.$$
$$\left. \times \sum_{\substack{k \geq 1 \\ p \in S_+ \bmod kS_+ \\ \frac{(p,p)}{2k} \in \mathbb{Z} \\ \frac{A^{-1}Qp}{k} \in S_-}} e^{2\pi i(\frac{p}{k},\chi)} \tilde{f}_Q\left(\frac{\sqrt{y}}{g_4}(\chi + Qc)\right) e^{2\pi i(Q,a) + 2\pi i(\chi,a)} \right)\Bigg|_\gamma . \quad (3.89)$$

where the coefficients $\tilde{c}(n)$ are defined in (A.12). As expected for a generic Fourier coefficient saturating the Gelfand–Kirillov dimension of the automorphic representation, these Fourier coefficients decompose into a 'measure factor'

$$\mu_{P_2}(Q,\chi) = \sum_{\substack{A \in \mathbb{Z}^{2 \times 2}/GL(2,\mathbb{Z}) \\ A^{-1}Q \in \mathbb{Z}^2 \otimes II_{4,4}}} \sum_{d | A^{-1}Q \cdot Q^\intercal A^{-\intercal}} |A|^{-1} d^{-3} \tilde{c}\left(\frac{\Delta(Q)}{|A|^2 d^2}\right) \sum_{\substack{k \geq 1 \\ p \in S_+ \bmod kS_+ \\ \frac{(p,p)}{2k} \in \mathbb{Z} \\ \frac{A^{-1}Qp}{k} \in S_-}} e^{2\pi i(\frac{p}{k},\chi)} \quad (3.90)$$

and an analytic function $\frac{1}{g_4^{12}} \tilde{f}_Q\left(\frac{\sqrt{y}}{g_4}(\chi + Qc)\right)$ of $g_4$ and the Levi factor $v$ acting on the charge $(0, Q, \chi)$ only, but not on the number-theoretic properties of the charge. Note that the dependence in $y$ and the axions $c$ is such that the function is covariant under $P_2$ as a $U(1) \times SO(4) \times SO(4) \subset P_2$ invariant function of $v(0, Q, \chi)$.

A significant complication is that the true measure factor $\mu(Q, \chi)$ differs from (3.90), due to the fact that the charges in the form $(0, Q, \chi)$ do not define a unique representative of the $P_2 \subset Spin(6,6)$ orbits. We shall not attempt to compute the measure $\mu$ for general $(Q, \chi)$, although we expect that it will take a similar form with different powers of the determinant $|A|$ of the dividing matrix $A$ and of the integer $d$. In the special case however where $Q$ is a projective charge, in the language of [69], then the problem simplifies drastically. One example of such a projective charge is a configuration of one D5 and three D1 Euclidean branes wrapping three orthogonal $T^2 \subset T^6$, two once and one $N$ times, possibly along with one unit of $D(-1)$ brane, i.e. 1 D5 +1 D1+1 D1+$N$ D1 (+1 D(-1)). Then each representative has $\gcd(Q) = 1$, $\gcd(Q_{\hat{i}} \cdot Q_{\hat{j}}) = 1$ and $\gcd(Q_{\hat{1}} \wedge Q_{\hat{2}}) = 1$ such that $A = 1$, $d = 1$ and $k = 1$. In this case there is no sum over $p$ and the measure reduces to $\tilde{c}(\Delta(Q)) = \tilde{c}(\Delta(0, Q, \chi))$, where $\Delta(0, Q, \chi) = \Delta(Q)$ is also the quartic $Spin(6,6)$ invariant of the total charge $(0, Q, \chi)$. Then the measure is the same for all possible representatives in the Poincaré sum $P_2 \backslash Spin(6,6)$, so the $Spin(6,6,\mathbb{Z})$ invariant measure will be preserved and equal to $\tilde{c}(\Delta(0, Q, \chi))$. For such a projective charge configuration, the partition function associated to the stack of Euclidean D-branes on $T^6$ is indeed expected to coincide with the helicity supertrace $\tilde{c}(\Delta(0, Q, \chi))$ of the corresponding 1/8-BPS D-particles determined in [51–53], which counts four-dimensional BPS black holes.

### The full Fourier expansion for $D_5$

For $d = 4$, we have exhausted all the contributions in the theta series $\theta_{\Lambda_{d+1}}^{E_{d+1}}(\phi, \Omega_2)$, and have thus obtained the complete expansion of the integral (3.7) that we record here for reference

$$\mathcal{E}_{(0,1)}^{(4),\mathrm{ExFT}} = \frac{2\zeta(3)^2}{3}g_6^{-6} + \frac{2\pi^2}{9}\zeta(3)g_6^{-4}E_{\Lambda_1}^{D_4} + 8\pi g_6^{-2}\mathrm{R.N.}\int_{\mathcal{F}_2}\frac{\mathrm{d}^6\Omega}{|\Omega_2|^3}\varphi_{\mathrm{KZ}}(\Omega)\Gamma_{4,4,2}(\Omega,t)$$

$$+ \frac{4}{27}\zeta(6)\,E_{3\Lambda_3}^{D_4} + 8\pi g_6^{-6}\int_{\mathcal{G}}\frac{\mathrm{d}^3\Omega_2}{|\Omega_2|^3}\varphi_{\mathrm{KZ}}^{\mathrm{tr}}(\Omega_2)\underset{\substack{q^i\in S_+\\ \bar{q}^i q^j=0}}{\sum}{}'\underset{\substack{n_i\in\mathbb{Z}\\ n_i q^i\neq 0}}{\sum}e^{-\pi\Omega_2^{ij}n_i n_j - \pi\Omega_{2ij}^{-1}g_6^{-2}g(q^i,q^j)+2\pi i n_i(q^i,a)}$$

$$+ 8\pi g_6^{-4}\underset{N\in S_+}{\sum}{}'\,e^{2\pi i \bar{N}a}\underset{\substack{\chi_i\in S_-\\ \bar{\chi}_i\chi_j=0,\,\chi_i\gamma_2\chi_j\neq 0\\ \bar{\chi}_i\gamma_2\chi_j\cdot\gamma_2 N=0\\ \frac{1}{\gcd(\bar{\chi}_i\gamma_2\chi_j)}\bar{\chi}_i\gamma_1 N\in II_{4,4}\\ \frac{1}{\gcd(\chi_i)}N\in S_+}}{\sum}\int_{\mathcal{G}}\frac{\mathrm{d}^3\Omega_2}{|\Omega_2|^2}\varphi_{\mathrm{KZ}}^{\mathrm{tr}}(\Omega_2)\frac{\gcd(\chi_i)^2}{\Omega_2^{ij}g(\chi_i,\chi_j)}$$

$$\times\,e^{-\pi\Omega_2^{ij}g(\chi_i,\chi_j)-\frac{\pi}{g_6^2}\left(\frac{\Omega_{2ij}^{-1}\varepsilon^{ik}\varepsilon^{jl}g(\bar{\chi}_k\gamma_1 N,\bar{\chi}_l\gamma_1 N)}{g(\bar{\chi}_1\gamma_2\chi_2,\bar{\chi}_1\gamma_2\chi_2)} + \frac{g(N,N)}{\Omega_2^{ij}g(\chi_i,\chi_j)} - \frac{\varepsilon^{ik}\varepsilon^{jl}g(\chi_i,\chi_j)g(\bar{\chi}_k\gamma_1 N,\bar{\chi}_l\gamma_1 N)}{\Omega_2^{ij}g(\chi_i,\chi_j)g(\bar{\chi}_1\gamma_2\chi_2,\bar{\chi}_1\gamma_2\chi_2)}\right)}$$

$$-\frac{16\pi^2}{21}g_6^{-4}\underset{\substack{q^i\in II_{4,4}\\ (q^i,q^j)=0}}{\sum}{}'\frac{\sigma_2(q)^2}{|v(q)|^2}B_2\left(\frac{2\pi}{g_6}|v(q)|\right)\quad(3.91)$$

As shown in Section C.2, the theta lift formula $\int\frac{\mathrm{d}^6\Omega}{|\Omega_2|^3}\varphi_{\mathrm{KZ}}(\Omega)\Gamma_{5,5,2}(\Omega,t)$ gives indeed the same constant terms. The integral in the second line can be simplified as in (3.66) and (3.75).

# 4   Decompactification limit

In this section, we study the integral (1.17) in the limit where one circle inside $T^d$ becomes very large. We first discuss the expected form of the expansion, known from general physical considerations, before turning to a detailed analysis of the constrained lattice sum (1.18). For $d = 5$, it is worth noting that the decompactification limit is equivalent to the weak coupling limit under exchanging $g_5$ with $R^{-1}$, due to the symmetry of the Dynkin diagram of $E_6$ and (1.23) with $d - 2 = 3$, which will allow us to cross-check our computations.

## 4.1   Expectation

The decompactification limit of the non-perturbative $\nabla^6\mathcal{R}^4$ coupling takes the formal generic form [22] [2, (2.28)]

$$\mathcal{E}_{(0,1)}^{(d)} \sim R^{\frac{12}{8-d}}\left(\mathcal{E}_{(0,1)}^{(d-1)} + \frac{5}{\pi}\xi(d-6)R^{d-7}\mathcal{E}_{(1,0)}^{(d-1)} + 40\xi(2)\xi(6)\xi(d+4)R^{d+3}\right.$$
$$\left. + \frac{2\pi}{3}\xi(d-2)R^{d-3}\mathcal{E}_{(0,0)}^{(d-1)} + \frac{16\pi^2\xi(d-2)^2}{(d+1)(6-d)}R^{2d-6}\right)\quad(4.1)$$

where $R = r_d/\ell_{D+1}$ is the radius of the circle in Planck units, up to logarithmic terms that depend on the specific dimension and can be found in the Appendix B of [2]. These terms are determined by matching the decompactification limit of the perturbative string theory answer together with the requirement that the result must be expressed in terms of the functions

multiplying the lower-derivative terms in $\mathcal{R}$, $\mathcal{R}^4$ and $\nabla^4 \mathcal{R}^4$ in the effective action. On the other hand, the decompactification limit of the homogeneous solution $\mathcal{F}_{(0,1)}^{(d)}$ in (1.11) (coming from the one-loop amplitude in exceptional field theory) gives

$$\mathcal{F}_{(0,1)}^{(d)} \sim R^{\frac{12}{8-d}} \left( \mathcal{F}_{(0,1)}^{(d-1)} + \frac{5}{\pi} \xi(d-6)\zeta(5)\, R^{d-7} E_{\frac{5}{2}\Lambda_1}^{E_d} + 40\xi(2)\xi(6)\xi(d+4)\, R^{d+3} \right) \quad (4.2)$$

where $\zeta(5)E_{\frac{5}{2}\Lambda_1}^{E_d} = \mathcal{E}_{(1,0)}^{(d-1)}$ for $d = 6$ and for $d = 0$ in type IIB, whereas

$$\mathcal{E}_{(1,0)}^{(d-1)} = \zeta(5)\, E_{\frac{5}{2}\Lambda_1}^{E_d} + \frac{4\pi^3}{45}\xi(d+1)\, E_{\frac{d+1}{2}\Lambda_d}^{E_d} \quad (4.3)$$

for $1 \le d \le 5$ and $d = 0$ type IIA.[28]

It follows from (1.13) that for $d \le 5$, the two-loop exceptional field theory amplitude must behave as

$$\mathcal{E}_{(0,1)}^{(d),\mathrm{ExFT}} \sim R^{\frac{12}{8-d}} \left( \mathcal{E}_{(0,1)}^{(d-1),\mathrm{ExFT}} + \frac{4\pi^2}{9}\xi(d-6)\xi(d+1)\, R^{d-7} E_{\frac{d+1}{2}\Lambda_d}^{E_d} + \frac{2\pi}{3}\xi(d-2)\, R^{d-3} \mathcal{E}_{(0,0)}^{(d-1)} \right.$$
$$\left. + \frac{16\pi^2 \xi(d-2)^2}{(d+1)(6-d)}\, R^{2d-6} \right) \quad (4.4)$$

up to logarithmic corrections that will be discussed in detail in Section 5.3. This formula applies up to non-analytic terms to $d = 6$ if one omits the term $E_{\frac{d+1}{2}\Lambda_d}^{E_d}$ which is divergent.

## 4.2 Decompactification limit of the particle multiplet lattice sum

We are interested in the decompactification limit of the integral (3.7). Under $E_{d+1} \supset GL(1) \times E_d$, the particle multiplet decomposes as

$$M_{\Lambda_{d+1}}^{E_{d+1}} \to \mathbb{Z}^{(9-d)} \oplus \left[ M_{\Lambda_d}^{E_d} \right]^{(1)} \oplus \left[ M_{\Lambda_1}^{E_d} \right]^{(d-7)} \oplus \left[ M_{\Lambda_7}^{E_d} \right]^{(2d-15)}, \quad (4.5)$$

where we define $M_{\Lambda_{d+1}}^{E_d} = \mathbb{Z}$ and $M_{\Lambda_k}^{E_d} = \{0\}$ for $k > d+1$. We denote the charges $\Gamma_i$ accordingly by $(m_i, Q_i, P_i, n_i)$. The constraints $\Gamma_i \times \Gamma_j = 0$ are valued in the string multiplet, which decomposes as

$$M_{\Lambda_1}^{E_{d+1}} \to \left[ M_{\Lambda_1}^{E_d} \right]^{(2)} \oplus \left[ M_{\Lambda_2}^{E_d} \oplus M_{\Lambda_7}^{E_d} \right]^{(d-6)} \oplus \left[ M_{\Lambda_6}^{E_d} \right]^{(2d-14)}. \quad (4.6)$$

Thus, the components $(m_i, Q_i, P_i, n_i)$ are subject to the constraints

$$Q_i \times Q_j = m_{(i} P_{j)}, \quad P_{(i} \times Q_{j)}\big|_{\Lambda_2} = 0, \quad P_{(i} \cdot Q_{j)} = -3m_{(i}\, n_{j)}, \quad P_i \times P_j = -n_{(i} Q_{j)} \quad (4.7)$$

where the third only arises for $d = 6$, and the last constraint simplifies to $(P_i, P_j) = 0$ for $d = 5$ and disappears for $d \le 4$. In terms of these components, the quadratic form $G(\Gamma, \Gamma)$ can be expressed as

$$G(\Gamma, \Gamma) = R^{-2\frac{9-d}{8-d}}(m + \langle a, Q \rangle + \langle a \times a, P \rangle - \det a\, n)^2$$
$$+ R^{-\frac{2}{8-d}} g(Q + 2a \times P - a \times an, Q + 2a \times P - a \times an)$$
$$+ R^{2\frac{7-d}{8-d}} g(P - an, P - an) + R^{2\frac{15-2d}{8-d}} n^2 \quad (4.8)$$

---

[28] For $d \le 3$ one must understand $E_{s\Lambda_d}^{E_d}$ as the sum over the 1/2-BPS particle charges, so one gets $E_{s\Lambda_3}^{E_3} = E_{s\Lambda_1}^{SL(2)} E_{s\Lambda_2}^{SL(3)}$, $E_{s\Lambda_2}^{E_2} = \nu^{\frac{6s}{7}} E_{s\Lambda_1}^{SL(2)} + \nu^{-\frac{8}{7}s}$, $E_{s\Lambda_1}^{E_1} = g_A^{\frac{3}{2}s}$ for type IIA, and zero for type IIB.

where $a$ denotes the axions parametrising the unipotent part of the parabolic subgroup $P_{d+1}$. As in Section 3.2, we shall split the theta series $\theta^{E_{d+1}}_{\Lambda_{d+1}}$ into contributions where the components $(m_i, Q_i, P_i, n_i)$ along the graded decomposition (4.5) are gradually populated, such that the constraints can be solved explicitly.

**1) The first layer**

The first layer corresponds to all charges being zero except $m_i$, in which case one has the contribution

$$\theta^{(1)}_{\Lambda_{d+1}}(\phi, \Omega_2) = \sum_{m_i \in \mathbb{Z}}{}' e^{-\pi \Omega_2^{ij} R^{-2\frac{9-d}{8-d}} m_i m_j} \quad . \tag{4.9}$$

This term corresponds to the Kaluza–Klein states running in the loop. It is infrared divergent and requires regularisation. Integrating against $\varphi^{\mathrm{tr}}_{\mathrm{KZ}}$, we get

$$
\begin{aligned}
\mathcal{I}^{(1)}_d \;&=\; 8\pi \int_{\mathcal{G}} \frac{\mathrm{d}^3 \Omega_2}{|\Omega_2|^{\frac{6-d}{2}}} \varphi^{\mathrm{tr}}_{\mathrm{KZ}}(\Omega_2) \theta^{(1)}_{\Lambda_{d+1}}(\phi, \Omega_2) \\[2mm]
&=\; \frac{8\pi^2}{3} \int_0^\infty \frac{\mathrm{d}V}{V^{3-d}} \int_{\mathcal{F}} \frac{\mathrm{d}\tau_1 \mathrm{d}\tau_2}{\tau_2^2} A(\tau) \sum_{(m,n) \in \mathbb{Z}^2}{}' e^{-\pi V R^{-2\frac{9-d}{8-d}} \frac{|m+n\tau|^2}{\tau_2}} \\[2mm]
&=\; \frac{16\pi^2}{3} \xi(2d-4)\, R^{\frac{12}{8-d}+2(d-3)} \int_{\mathcal{F}} \frac{\mathrm{d}\tau_1 \mathrm{d}\tau_2}{\tau_2^2} A(\tau)\, E^{SL(2)}_{(d-2)\Lambda_1}(\tau) \\[2mm]
&=\; \frac{16\pi^2\, \xi(d-2)^2}{(6-d)(d+1)} R^{\frac{12}{8-d}+2(d-3)} \tag{4.10}
\end{aligned}
$$

where we used (3.17). More precisely, using the regularisation (1.17) one obtains after taking the limit $L \to \infty$

$$\mathcal{I}^{(1)}_{d,\epsilon} = \frac{16\pi^2\, \xi(d+2\epsilon-2)^2}{(6-d-2\epsilon)(d+2\epsilon+1)} R^{\frac{12+4\epsilon}{8-d}+2(d-3)+4\epsilon} \quad . \tag{4.11}$$

It has a double pole in $d=2$ and $d=3$ associated to the double pole in the eight-dimensional supergravity amplitude, and a simple pole in $d=6$. It is associated to both the log divergence proportional to the $\mathcal{E}_{(1,0)}$ coupling in $d=6$ and the log divergence proportional to the $\mathcal{E}_{(0,0)}$ coupling in $d=5$ and is not proportional to a sum of them. We shall discuss the logarithmic contribution for $d=6$ in more detail in Section 5.3.

**2) The second layer**

The second term comes from $m_i \in \mathbb{Z}$, $Q_i \neq 0$, $P_i = n_i = 0$:

$$
\begin{aligned}
\theta^{(2)}_{\Lambda_{d+1}}(\phi, \Omega_2) \;&=\; \sum_{\substack{Q_i \in M^{E_d}_{\Lambda_d} \\ Q_i \times Q_j = 0}}{}' \sum_{m_i \in \mathbb{Z}} e^{-\pi \Omega_2^{ij}\left(R^{-2\frac{9-d}{8-d}}(m_i + \langle a, Q_i\rangle)(m_j + \langle a, Q_j\rangle) + R^{-\frac{2}{8-d}} g(Q_i, Q_j)\right)} \\[2mm]
&=\; \sum_{\substack{Q_i \in M^{E_d}_{\Lambda_d} \\ Q_i \times Q_j = 0}}{}' \sum_{m^i \in \mathbb{Z}} \frac{R^{2\frac{9-d}{8-d}}}{|\Omega_2|^{\frac{1}{2}}} e^{-\pi \Omega_{2\,ij}^{-1} R^{2\frac{9-d}{8-d}} m^i m^j - \pi \Omega_2^{ij} R^{-\frac{2}{8-d}} g(Q_i, Q_j) + 2\pi \mathrm{i} \langle m^i Q_i, a\rangle} \quad .
\end{aligned}
\tag{4.12}
$$

The term $\mathcal{I}_d^{(2a)}$ with $m^i = 0$ is recognised as the theta series $\theta_{\Lambda_d}^{E_d}(\phi, \Omega_2)$, up to factors of $|\Omega_2|$ and $R$, whereas the term $\mathcal{I}_d^{(2b)}$ coming from $m_i \neq 0$ but $m_i Q^i = 0$ can be treated as $\mathcal{I}_d^{(2b)}$ and $\mathcal{I}_d^{(2c)}$ in Section 3.2. In the same way we unfold the fundamental domain of $PGL(2,\mathbb{Z})$ to the strip, so as to set $(m^1, m^2) = (0, m)$ and $(Q_1, Q_2) = (Q, 0)$, leading to, upon using (3.27),

$$
\begin{aligned}
\mathcal{I}_d^{(2b)} &= \frac{8\pi^2 R^{2\frac{9-d}{8-d}}}{3} \int_0^\infty \frac{\mathrm{d}V}{V^{d-2+2\epsilon}} \int_0^L \frac{\mathrm{d}\tau_2}{\tau_2^2} \left( \int_{-\frac{1}{2}}^{\frac{1}{2}} \mathrm{d}\tau_1 \, A(\tau) \right) \sideset{}{'}\sum_{\substack{Q \in M_{\Lambda_d}^{E_d} \\ Q \times Q = 0}} \sum_{m \geq 1} e^{-\frac{\pi V}{\tau_2} R^{2\frac{9-d}{8-d}} m^2 - \frac{\pi R^{-\frac{2}{8-d}}}{V \tau_2} |Z(Q)|^2} \\
&= \frac{8\pi^2 R^{\frac{12+4\epsilon}{8-d}}}{3} \int_0^\infty \frac{\mathrm{d}V}{V^{d-2+2\epsilon}} \int_0^L \frac{\mathrm{d}\tau_2}{\tau_2^2} \sideset{}{'}\sum_{\substack{Q \in M_{\Lambda_d}^{E_d} \\ Q \times Q = 0}} \sum_{m \geq 1} e^{-\frac{\pi V}{\tau_2} R^2 m^2 - \frac{\pi}{V \tau_2} |Z(Q)|^2} \\
&\qquad \times \left( \tau_2 + \frac{1}{6\tau_2^3} - \frac{1}{7} \sum_{\substack{a,c \geq 1 \\ \gcd(a,c)=1 \\ 2ac < 1/\tau_2}} \frac{1 + \frac{3}{2}(2ac\tau_2)^2 + (2ac\tau_2)^4}{ac(ac\tau_2)^3} (1 - (2ac\tau_2)^2)^{\frac{3}{2}} \right) \\
&= \frac{8\pi^2 R^{\frac{12+4\epsilon}{8-d}}}{3} \left( \xi(d_\epsilon - 2)\xi(d_\epsilon - 3) R^{d_\epsilon - 3} \, E_{\frac{d_\epsilon-3}{2}\Lambda_d}^{E_d} + \frac{\xi(d_\epsilon - 6)\xi(d_\epsilon + 1)}{6} R^{d_\epsilon - 7} \, E_{\frac{d_\epsilon+1}{2}\Lambda_d}^{E_d} \right. \\
&\qquad \left. - \frac{2\pi^2}{7} R^{d_\epsilon - 3} \sideset{}{'}\sum_{\substack{Q \in M_{\Lambda_d}^{E_d} \\ Q \times Q = 0}} \frac{(\sigma_{d_\epsilon-3}(Q))^2}{|Z(Q)|^{d_\epsilon - 3}} B_{d_\epsilon - 3}(2\pi R |Z(Q)|) \right) \tag{4.13}
\end{aligned}
$$

where $d_\epsilon = d + 2\epsilon$ and $B_{d-3}(x)$ is the function defined in (3.30). In total, one obtains

$$
\begin{aligned}
\mathcal{I}_d^{(2)} &= R^{\frac{12+4\epsilon}{8-d}} \left( 8\pi \int \frac{\mathrm{d}^3 \Omega_2}{|\Omega_2|^{\frac{7-d_\epsilon}{2}}} \varphi_{\mathrm{KZ}}^{\mathrm{tr}}(\Omega_2) \sideset{}{'}\sum_{\substack{Q_i \in M_{\Lambda_d}^{E_d} \\ Q_i \times Q_j = 0}} e^{-\pi \Omega_2^{ij} g(Q_i, Q_j)} \right. \tag{4.14} \\
&\qquad + \frac{8\pi^2}{3} \xi(d_\epsilon - 2)\xi(d_\epsilon - 3) R^{d_\epsilon - 3} \, E_{\frac{d_\epsilon-3}{2}\Lambda_d}^{E_d} + \frac{4\pi^2}{9} \xi(d_\epsilon - 6)\xi(d_\epsilon + 1) R^{d_\epsilon - 7} \, E_{\frac{d_\epsilon+1}{2}\Lambda_d}^{E_d} \\
&\qquad - \frac{16\pi^2}{21} R^{d-3} \sideset{}{'}\sum_{\substack{Q \in M_{\Lambda_d}^{E_d} \\ Q \times Q = 0}} \frac{(\sigma_{d-3}(Q))^2}{|Z(Q)|^{d-3}} B_{d-3}(2\pi R |Z(Q)|) \\
&\qquad \left. + 8\pi R^{2(d-3)} \int \frac{\mathrm{d}^3 \Omega_2}{|\Omega_2|^{\frac{7-d}{2}}} \varphi_{\mathrm{KZ}}^{\mathrm{tr}}(\Omega_2) \sideset{}{'}\sum_{\substack{Q_i \in M_{\Lambda_d}^{E_d} \\ Q_i \times Q_j = 0}} \sum_{\substack{m^i \in \mathbb{Z} \\ m^i Q_i \neq 0}} e^{-\pi \Omega_{2\,ij}^{-1} m^i m^j - \pi \Omega_2^{ij} R^2 g(Q_i, Q_j) + 2\pi i \langle m^i Q_i, a \rangle} \right) .
\end{aligned}
$$

The first term, corresponding to $m^i = 0$, reproduces the function $R^{\frac{12}{8-d}} \mathcal{I}_{d-1}$ in dimension $D + 1$. The two terms in the second line formally give respectively (part of) the threshold functions $\mathcal{E}_{(0,0)}^{(d-1)}$ and $\mathcal{E}_{(1,0)}^{(d-1)}$ in $D + 1$ dimensions, while the third line corresponds to non-perturbative corrections to the constant term. We shall discuss the renormalised expression at $\epsilon = 0$ shortly. Note that unlike in the weak coupling limit, the regularisation does not give rise to non-maximal parabolic Eisenstein series. This is because there is no additional Poincaré series in this computation, so

one cannot deviate from the $P_d$ maximal parabolic Poincaré series associated to the sum over $Q_i$. The last term corresponds to non-trivial Fourier coefficients associated to 1/2-BPS instantons, which can be analysed similarly as $\mathcal{I}_d^{(2d)}$ in Sections 3.3 and C.1.

Following the steps as in Section C.1 and in particular (C.19), one computes that

$$
8\pi\, R^{2(d-3)} \int \frac{\mathrm{d}^3\Omega_2}{|\Omega_2|^{\frac{7-d}{2}}} \varphi_{\mathrm{KZ}}^{\mathrm{tr}}(\Omega_2) \underset{\substack{Q_i \in M_{\Lambda_d}^{E_d} \\ Q_i \times Q_j = 0}}{\sum{}'} \underset{\substack{m^i \in \mathbb{Z} \\ m^i Q_i \neq 0}}{\sum{}'} e^{-\pi\Omega_{2\,ij}^{-1} m^i m^j - \pi\Omega_2^{ij} R^2 g(Q_i, Q_j) + 2\pi\mathrm{i}\langle m^i Q_i, a\rangle}
$$

$$
= \frac{16\pi^2}{3} R^{\frac{d-3}{2}} \underset{\substack{Q \in M_{\Lambda_d}^{E_d} \\ Q \times Q = 0}}{\sum{}'} \Bigg( \sigma_{d-3}(Q) \Bigg( \frac{y_Q^{\frac{9-3d}{2} + \frac{(9-d)(d-4)}{10-d}}}{\gcd(Q)^{\frac{d-3}{2}}} \xi(d-4) E_{\frac{d-4}{2}\Lambda_{d-1}}^{E_{d-1}}(v_Q) K_{\frac{d-3}{2}}\big(2\pi R|Z(Q)|\big)
$$

$$
+ \frac{3 y_Q^{\frac{7-3d}{2}}}{\pi^2 R \gcd(Q)^{\frac{d-1}{2}}} \underset{\substack{q \in M_{\Lambda_{d-1}}^{E_{d-1}} \\ q \times q = 0}}{\sum{}'} \sigma_{d-4}(q) e^{2\pi\mathrm{i}(q, \varsigma_Q)} \frac{K_{\frac{d-2}{2}}\big(2\pi y_Q^{-2\frac{9-d}{10-d}}|v_Q(q)|\big)}{\big(y_Q^{-2\frac{9-d}{10-d}}|v_Q(q)|\big)^{\frac{d-2}{2}}} K_{\frac{d-5}{2}}\big(2\pi R|Z(Q)|\big)
$$

$$
- \frac{15 y_Q^{\frac{5-3d}{2}}}{2\pi^4 R^2 \gcd(Q)^{\frac{d+1}{2}}} \underset{\substack{q \in M_{\Lambda_{d-1}}^{E_{d-1}} \\ q \times q = 0}}{\sum{}'} \sigma_{d-4}(q)\, e^{2\pi\mathrm{i}(q, \varsigma_Q)} \frac{K_{\frac{d}{2}}\big(2\pi y_Q^{-2\frac{9-d}{10-d}}|v_Q(q)|\big)}{\big(y_Q^{-2\frac{9-d}{10-d}}|v_Q(q)|\big)^{\frac{d}{2}}} K_{\frac{d-7}{2}}\big(2\pi R|Z(Q)|\big) \Bigg)
$$

$$
+ \frac{y_Q^{\frac{5-3d}{2} + \frac{(9-d)d}{10-d}}}{6R^2} \frac{\sigma_{d-7}(Q)}{\gcd(Q)^{\frac{d-7}{2}}} \xi(d)\, E_{\frac{d}{2}\Lambda_{d-1}}^{E_{d-1}}(v_Q) K_{\frac{d-7}{2}}\big(2\pi R|Z(Q)|\big) \Bigg) e^{2\pi\mathrm{i}(Q,a)}
$$

$$
+ \frac{8\pi^2}{3} R^{2d-7} \int_0^\infty \frac{\mathrm{d}\nu}{\nu^{\frac{9}{2}-d}} \int_0^1 \frac{\mathrm{d}t}{t^{\frac{3}{2}}} \sum_{\gamma \in P_d \backslash E_d} \underset{N \in \mathbb{N}}{\sum{}'} \underset{\substack{N_1 \in \mathbb{Z} \\ q \in M_{\Lambda_{d-1}}^{E_{d-1}} \\ q \times q = 0}}{\sum{}'} \sigma_{d-3}(N_1, q)\, \sigma_{d-3}(N + N_1, q)
$$

$$
\times\, y^{-1} e^{-\pi\big(\frac{2}{\nu t} + \nu t \frac{R^2 y^2 N^2}{2} + \frac{2\nu}{t} R^2 y^{\frac{2}{10-d}}|v(q)|^2\big)} F\big(t, R^2 y^2 \nu, N, N_1 + (\varsigma, q)\big) e^{2\pi\mathrm{i}Q a_\gamma}\bigg|_\gamma \quad (4.15)
$$

where $y_Q = \frac{|Z(Q)|}{\gcd(Q)}$, the sum over $q \in M_{\Lambda_{d-1}}^{E_{d-1}}$ runs over over characters of the unipotent stabilisers of the charge $Q$, and $F$ is the function defined in (3.74). The leading term in $R$ factorises as an Eisenstein series over the Levi stabiliser of $Q$ in the minimal representation, while the full Fourier coefficient depends non-trivially on the whole parabolic stabiliser. One recognises $\sigma_{d-3}(Q)$ as the measure for 1/2-BPS charges $Q \in M_{\Lambda_d}^{E_d}$, and similarly $\sigma_{d-4}(q)$ as the measure for 1/2-BPS charges $q \in M_{\Lambda_{d-1}}^{E_{d-1}}$, just like for Fourier coefficients of $\mathcal{E}_{(0,0)}^{(d-1)}$ and $\mathcal{E}_{(0,0)}^{(d-2)}$ in the decompactification limit, see [23, 1]. To interpret these Fourier coefficients it is relevant to combine them with the 1/2-BPS Fourier coefficients of the homogeneous solution [50]

$$
\int_{R(\Lambda_d)/M_{\Lambda_d}^{E_d}} da\; e^{-2\pi\mathrm{i}(Q,a)} \mathcal{F}_{(0,1)}^{(d)} = 80\xi(2) R^{\frac{12+4\epsilon}{8-d} + \frac{d+3}{2}} \Bigg( \xi(6)\sigma_{d+3}(Q) \frac{K_{\frac{d+3}{2}}\big(2\pi R|Z(Q)|\big)}{|Z(Q)|^{\frac{d+3}{2}}}
$$

$$
+ \frac{\sigma_{d-7}(Q)}{R^5 \gcd(Q)^{\frac{d-7}{2}}} \xi(5) E_{\frac{5}{2}\Lambda_1}^{E_{d-1}}(v_Q) K_{\frac{d-7}{2}}\big(2\pi R|Z(Q)|\big) \Bigg) \quad (4.16)
$$

where $Q \times Q = 0$. Altogether, the abelian Fourier coefficients of $\mathcal{E}^{(d)}_{(0,1)}$ involving an Eisenstein series of the Levi stabiliser of the charge $Q$ combine in the full coupling $\mathcal{E}^{(d)}_{(0,1)}$ as

$$
R^{\frac{12}{8-d}+\frac{d+3}{2}}\left( \frac{16\pi^4}{567}\sigma_{d+3}(Q)\frac{K_{\frac{d+3}{2}}(2\pi R|Z(Q)|)}{|Z(Q)|^{\frac{d+3}{2}}} \right.
$$
$$
+ \frac{4\pi\sigma_{d-3}(Q)}{3(Ry_Q^{\frac{2}{10-d}})^3}\mathcal{E}^{(d-2)}_{(0,0)}(v_Q)\frac{K_{\frac{d-3}{2}}(2\pi R|Z(Q)|)}{|Z(Q)|^{\frac{d-3}{2}}}
$$
$$
\left. + \frac{10\sigma_{d-7}(Q)}{\pi(Ry_Q^{\frac{2}{10-d}})^5}\mathcal{E}^{(d-2)}_{(1,0)}(v_Q)\frac{K_{\frac{d-7}{2}}(2\pi R|Z(Q)|)}{|Z(Q)|^{\frac{d-7}{2}}} \right) \quad (4.17)
$$

and we assembled the effective couplings in $D + 2$ dimensions that appear with the expected power of the torus volume $Ry_Q^{\frac{2}{10-d}}$ associated to the charge $Q$.

The last term in (4.15) involving the function $F$ can be ascribed to instanton anti-instanton corrections of charges $Q = (N, 0, 0)$ and $Q_1 = (N_1, q, 0)$, and is further exponentially suppressed in $e^{-2\pi R(|Z(Q_1)|+|Z(Q-Q_1)|)}$. The measure factor also reproduces the 1/2-BPS measure appearing in the Fourier expansion of $\mathcal{E}^{(d)}_{(0,0)}$, consistently with the property that these terms are the solution to the Laplace equation (1.5) with a quadratic source term in $\mathcal{E}^{(d)}_{(0,0)}$.

### 3) The third layer

The next layer corresponds to $m_i \in \mathbb{Z}$, $Q_i \in M^{E_d}_{\Lambda_d}$ and $P_i \neq 0$, with $P_i \wedge P_j = 0$, so that $P_i = n_iP$ for two relative prime integers $n_i$ and some $P \in M^{E_d}_{\Lambda_1}$. In this case, the last constraint in (4.7) implies that $P$ is in the orbit of the highest weight representative in the parabolic decomposition (3.9) with respect to $P_1 \subset E_d$. Within this decomposition, the constraints (4.7) for the charge $Q_i$ in the decomposition (3.8) imply that $Q_i = (q_i, 0, \dots)$ for a doublet of vectors $q_i \in II_{d-1,d-1}$, subject to the conditions $(q_i, q_j) = 2m_{(i}n_{j)}$. Writing this third contribution as a Poincaré series over $P_1 \backslash E_d$, the seed is recognised as a Siegel–Narain genus-two theta series for the lattice $II_{d,d}$ as follows,

$$
\theta^{(3)}_{E_{d+1}}(\phi, \Omega_2)
$$
$$
= \sum_{\gamma \in P_1 \backslash E_d}\left( \sum_{\substack{q_i \in II_{d-1,d-1} \\ n_i,m_i \in \mathbb{Z} \ n_i \neq 0 \\ (q_i,q_j)=2m_{(i}n_{j)}}} e^{-\pi\Omega_2^{ij}yR^{\frac{-2}{8-d}}\left((yR^2)^{-1}(m_i+(a,q_i)+\frac{(a,a)}{2}n_i)(m_j+(a,q_j)+\frac{(a,a)}{2}n_j)+g(q_i+an_i,q_j+an_j)+yR^2n_in_j\right)} \right)\Bigg|_\gamma
$$
$$
= \sum_{\gamma \in P_1 \backslash E_d}\left( |yR^{-\frac{2}{8-d}}\Omega_2|^{-\frac{d}{2}}\int_{[0,1]^3}\mathrm{d}^3\Omega_1 \ \Gamma_{d,d,2}(Ry^{\frac{1}{2}}, \Omega_1 + iyR^{-\frac{2}{8-d}}\Omega_2) \right.
$$
$$
\left. - \sum_{\substack{q_i \in II_{d-1,d-1} \\ (q_i,q_j)=0}}\sum_{m_i \in \mathbb{Z}} e^{-\pi\Omega_2^{ij}yR^{-\frac{2}{8-d}}\left((yR^2)^{-1}(m_i+(a,q_i))(m_j+(a,q_j))+g(q_i,q_i)\right)} \right)\Bigg|_\gamma \quad (4.18)
$$

where $\Gamma_{d,d,2}(Ry^{\frac{1}{2}}, \Omega)$ is the genus-two partition function for the lattice $II_{d,d} = II_{d-1,d-1} \oplus II_{1,1}$, with radius $Ry^{\frac{1}{2}}$ on $II_{1,1}$. To compute the integral of the first line against $\varphi_{\mathrm{KZ}}^{\mathrm{tr}}$, we first rescale $\Omega_2 \mapsto R^{\frac{2}{8-d}}\Omega_2/y$, so that the argument of the Siegel–Narain theta series becomes $\Omega = \Omega_1 + i\Omega_2$; we then use its invariance under $Sp(4,\mathbb{Z})$ to fold the integration domain to the fundamental domain $\mathcal{F}_2$, at the cost of replacing $\varphi_{\mathrm{KZ}}^{\mathrm{tr}}$ by the sum of its images under $Sp(4,\mathbb{Z})$; the latter sum produces the Kawazumi–Zhang invariant $\varphi_{\mathrm{KZ}}$ by virtue of (1.21). As for the second line in (4.18), we perform a Poisson summation over $m_i$, obtaining finally

$$\mathcal{I}_d^{(3)} = 8\pi\, R^{2\frac{d-2}{8-d}} \sum_{\gamma \in P_1 \backslash E_d} \left( y^{2-d} \int_{\mathcal{F}_2} \frac{\mathrm{d}^6\Omega}{|\Omega_2|^3} \varphi_{\mathrm{KZ}}^{\epsilon}(\Omega)\, \Gamma_{d,d,2}(\Omega, Ry^{\frac{1}{2}}) \right)\Bigg|_{\gamma} \tag{4.19}$$

$$-8\pi\, R^{\frac{12}{8-d}} \sum_{\gamma \in P_1 \backslash E_d} \left( y^{3-d} \int_{\mathcal{G}} \frac{\mathrm{d}^3\Omega_2}{|\Omega_2|^{\frac{7-d}{2}-\epsilon}} \varphi_{\mathrm{KZ}}^{\mathrm{tr}}(\Omega_2) \sum_{\substack{n^i \in \mathbb{Z} \\ q_i \in II_{d-1,d-1} \\ (q_i,q_j)=0}} e^{-\pi\Omega_{2ij}^{-1}R^2 y n^i n^j - \pi\Omega_2^{ij}g(q_i,q_j)+2\pi i(n^i q_i,a)} \right)\Bigg|_{\gamma}$$

The next step is to integrate over $\Omega_1$ in the first line. In principle one should do the computation using the unfolding method and the Fourier–Jacobi expansion of $\varphi_{\mathrm{KZ}}^{\epsilon}(\Omega)$ at $\epsilon \neq 0$. Instead, we shall do the computation at $\epsilon = 0$, and argue a posteriori that we do not miss any term for $d \geq 4$. At $\epsilon = 0$, we can use the equivalence (2.53) between the constrained lattice sum over the vectors in $II_{d,d}$ and the constrained lattice sum over spinors in $S_+$. As for the second line, it is useful to change variable from $\Omega_2 \to \Omega_2^{-1}$, using the fact that $\mathrm{d}^3\Omega_2/|\Omega_2|^s \varphi_{\mathrm{KZ}}^{\mathrm{tr}}(\Omega_2) \to \mathrm{d}^3\Omega_2/|\Omega_2|^{4-s}\varphi_{\mathrm{KZ}}^{\mathrm{tr}}(\Omega_2)$ under this operation. After these steps, one obtains

$$\mathcal{I}_d^{(3)} = 8\pi\, R^{2\frac{d-2}{8-d}} \sum_{\gamma \in P_1 \backslash E_d} \Bigg[ y^{2-d}\Bigg( \int_{\mathcal{G}} \frac{\mathrm{d}^3\Omega_2}{|\Omega_2|} \varphi_{\mathrm{KZ}}^{\mathrm{tr}}(\Omega_2)\, \Xi_{\Lambda_d}^{D_d}(\Omega_2, Ry^{\frac{1}{2}})$$

$$- R^2 y \int_{\mathcal{G}} \frac{\mathrm{d}^3\Omega_2}{|\Omega_2|^{\frac{d+1}{2}}} \varphi_{\mathrm{KZ}}^{\mathrm{tr}}(\Omega_2) \sum_{\substack{n_i \in \mathbb{Z} \\ q^i \in II_{d-1,d-1} \\ (q^i,q^j)=0}} e^{-\pi\Omega_2^{ij}R^2 y n_i n_j - \pi\Omega_{2ij}^{-1}g(q^i,q^j)+2\pi i(n_i q^i,a)} \Bigg) \Bigg]\Bigg|_{\gamma} \tag{4.20}$$

The integral on the first line can then be computed by inserting (C.24) with $R$ replaced by $Ry^{1/2}$. Most terms in (C.24) coincide with the terms appearing in the second line of (4.20) and cancel out, leaving only

$$\mathcal{I}_d^{(3)} = 8\pi R^{\frac{12}{8-d}+d-3} \sum_{\substack{\gamma \in P_1 \backslash E_d \\ \delta \in P_{d-3} \backslash D_{d-1}}} \left( y^{\frac{3-d}{2}} \sum_{\substack{n_i^{\hat{j}} \in \mathbb{Z}^2 \\ \det n \neq 0}} \int \frac{\mathrm{d}^3\Omega_2}{|\Omega_2|^2} \varphi_{\mathrm{KZ}}^{\mathrm{tr}}(\Omega_2) \frac{e^{-\pi\Omega_2^{ij}v_{\hat{k}\hat{l}}n_i^{\hat{k}}n_j^{\hat{l}}}}{y'(\Omega_2^{ij}v_{\hat{k}\hat{l}}\hat{n}_i^{\hat{k}}\hat{n}_j^{\hat{l}})^{\frac{d-3}{2}}} \right. \tag{4.21}$$

$$\left. \times \sum_{\substack{q_a^i \in \mathbb{Z}^2 \\ p^{\alpha} \in \mathbb{Z}^{d-3}}}' e^{-\pi R^2 y\left(\Omega_{2ij}^{-1}u^{ab}q_a^i q_b^j + \frac{y'^{\frac{2}{d-3}}}{\Omega_2^{ij}v_{\hat{k}\hat{l}}\hat{n}_i^{\hat{k}}\hat{n}_j^{\hat{l}}}\rho_{\alpha\beta}(p^{\alpha}-c_{\hat{k}}^{a\alpha}\hat{n}_i^{\hat{k}}q_a^i)(p^{\beta}-c_{\hat{l}}^{b\beta}\hat{n}_j^{\hat{l}}q_b^j)\right)+2\pi i(q_a^i n_i^{\hat{j}}a_{\hat{j}}^a + np^{\alpha}a_{\alpha})} \right)\Bigg|_{\delta\gamma},$$

where $y$ and $y'$ are the coordinates on the $GL(1)$ factors of the Levi subgroups of $P_1$ and $P_{d-3}$, respectively, i.e. the associated multiplicative parabolic characters. At this point we change

variables $y = y_4^{\frac{2}{3}} v$, $y' = y_4^{\frac{9-d}{2}} v^{\frac{3-d}{2}}$ such that $v v_{\hat{k}\hat{l}} n_i{}^{\hat{k}} n_j{}^{\hat{l}}$ is identified as $v_{\hat{k}\hat{l}} n_i{}^{\hat{k}} n_j{}^{\hat{l}}$ over $SL(3)$ and $y_4$ is the multiplicative character for $P_4$

$$\mathfrak{e}_d \supset (\mathfrak{gl}_1 \oplus \mathfrak{sl}_2 \oplus \mathfrak{sl}_3 \oplus \mathfrak{sl}_{d-3})^{(0)} \oplus (\mathbf{2},\mathbf{3},\mathbf{d}-\mathbf{3})^{(\frac{9-d}{3d-9})} \oplus (\overline{\mathbf{3}}, \binom{d-3}{2})^{(\frac{18-2d}{3d-9})} \oplus (\mathbf{2}, \binom{d-3}{3})^{(\frac{9-d}{d-3})} \quad (4.22)$$

under which

$$M_{\Lambda_d}^{E_d} \to (\mathbb{Z}^{d-3})^{(\frac{2}{d-3})} \oplus (\mathbb{Z}^2 \otimes \mathbb{Z}^3)^{(\frac{1}{3})} \oplus (\overline{\mathbb{Z}^3} \otimes \mathbb{Z}^{d-3})^{(\frac{2d-12}{3d-9})} \oplus (\mathbb{Z}^2 \otimes \wedge^2 \mathbb{Z}^{d-3})^{(\frac{d-7}{d-3})} \oplus (\mathbb{Z}^3 \otimes \wedge^3 \mathbb{Z}^{d-3})^{(\frac{4d-30}{3d-9})} . \quad (4.23)$$

In this way, denoting $y_4$ by $y$ again for simplicity, we obtain

$$\mathcal{I}_d^{(3)} = 8\pi R^{d-3} \sum_{\gamma \in P_4 \backslash E_d} \left( \sum_{\substack{n_i{}^{\hat{j}} \in \mathbb{Z}^3 \\ \mathrm{rk}(n)=2}} \int \frac{\mathrm{d}^3 \Omega_2}{|\Omega_2|^2} \varphi_{\mathrm{KZ}}^{\mathrm{tr}}(\Omega_2) \frac{e^{-\pi \Omega_2^{ij} y v_{\hat{k}\hat{l}} n_i{}^{\hat{k}} n_j{}^{\hat{l}}}}{y^2 (\Omega_2^{ij} y v_{\hat{k}\hat{l}} \hat{n}_i{}^{\hat{k}} \hat{n}_j{}^{\hat{l}})^{\frac{d-3}{2}}} \right. \quad (4.24)$$

$$\times \left. \sideset{}{'}\sum_{\substack{q_a^i \in \mathbb{Z}^2 \\ p^\alpha \in \mathbb{Z}^{d-3}}} e^{-\pi R^2 \left( \Omega_{2ij}^{-1} y^{-\frac{1}{3}} u^{ab} q_a^i q_b^j + \frac{y^{\frac{4}{d-3}}}{\Omega_2^{ij} y v_{\hat{k}\hat{l}} \hat{n}_i{}^{\hat{k}} \hat{n}_j{}^{\hat{l}}} \rho_{\alpha\beta}(p^\alpha - c_{\hat{k}}^{a\alpha} \hat{n}_i{}^{\hat{k}} q_a^i)(p^\beta - c_{\hat{l}}^{b\beta} \hat{n}_j{}^{\hat{l}} q_b^j) \right) + 2\pi i (q_a^i n_i{}^{\hat{j}} a_{\hat{j}}^a + n p^\alpha a_\alpha)} \right)\Bigg|_\gamma .$$

where $c_{\hat{k}}^{a\alpha}$ are the axions in the component $(\mathbf{2},\mathbf{3},\mathbf{d}-\mathbf{3})^{(\frac{9-d}{3d-9})}$ of the unipotent of $P_4$ and $Q = (n p^\alpha, q_a^i n_i{}^{\hat{j}}, 0, 0, 0)$ is the Fourier charge in (4.23). Using the saddle point approximation as in (3.77), (3.79) one obtains that these terms are exponentially suppressed in

$$2\pi R \sqrt{y^{\frac{4}{d-3}} \rho_{\alpha\beta}(n p^\alpha - c_{\hat{k}}^{a\alpha} n_i{}^{\hat{k}} q_a^i)(n p^\beta - c_{\hat{l}}^{b\beta} n_j{}^{\hat{l}} q_b^j) + y^{\frac{2}{3}} \left( u^{ab} q_a^i q_b^j v_{\hat{k}\hat{l}} n_i{}^{\hat{k}} n_j{}^{\hat{l}} + 2|\det q| \sqrt{\det(n v n^{\mathsf{T}})} \right)}$$

$$= 2\pi R \sqrt{|Z(Q)|^2 + 2\sqrt{\Delta(Q \times Q)}} , \quad (4.25)$$

the BPS mass of a 1/4-BPS charge $Q$. The charge $Q$ is 1/2-BPS if $\det q = 0$.

### 4) The fourth layer

Next we consider $m_i \in \mathbb{Z}$, $Q_i \in M_{\Lambda_d}^{E_d}$ and $P_i \in M_{\Lambda_1}^{E_d}$, with $P_1 \wedge P_2 \neq 0$. In this case $P_1 \wedge P_2 \in M_{\Lambda_3}^{E_d}$ is non-zero, and it is in the minimal orbit such that one can decompose the sum over $P_i$ as a Poincaré sum over $P_3 \backslash E_d$ and a sum over non-degenerate 2 by 2 matrices $n_i{}^{\hat{j}}$ in this $GL(2) \times SL(d-1)$ decomposition

$$M_{\Lambda_d}^{E_d} = (\mathbb{Z}^{d-1})^{(\frac{4}{d-1})} \oplus (\mathbb{Z}^{2 \times (d-1)})^{(\frac{d-5}{d-1})} \oplus (\wedge^3 \mathbb{Z}^{d-1})^{(\frac{2d-14}{d-1})} \oplus (\mathbb{Z}^2 \otimes \wedge^5 \mathbb{Z}^{d-1})^{(\frac{3d-23}{d-1})} ,$$

$$M_{\Lambda_1}^{E_d} = (\mathbb{Z}^2)^{(1)} \oplus (\wedge^2 \mathbb{Z}^{d-1})^{(\frac{2d-10}{d-1})} \oplus (\mathbb{Z}^2 \otimes \wedge^4 \mathbb{Z}^{d-1})^{(\frac{3d-19}{d-1})} \oplus (\mathbb{Z}^{d-1} \otimes \wedge^5 \mathbb{Z}^{d-1})^{(\frac{4d-28}{d-1})} ,$$

$$M_{\Lambda_2}^{E_d} = (\mathbb{Z}^{d-1})^{(\frac{2d-6}{d-1})} \oplus (\mathbb{Z}^2 \otimes \wedge^3 \mathbb{Z}^{d-1})^{(\frac{3d-15}{d-1})} \oplus \dots , \quad (4.26)$$

The general solution to the constraints $P_{(i} \times Q_{j)}\big|_{\Lambda_2} = 0$ is then

$$P_i = (n_i{}^{\hat{j}}, 0, 0, 0) , \qquad Q_i = (q_i^a, n_i{}^{\hat{j}} \tfrac{p_a}{k}, 0, 0) , \quad (4.27)$$

with $q_i$ and $p$ in $\mathbb{Z}^{d-1}$ and $k$ relative prime to $p$ that divides $n_i{}^{\hat{j}}$, while the additional constraint $Q_i \times Q_j = m_{(i} P_{j)}$ implies that

$$m_i = \tfrac{p_a}{k} q_i^a , \quad (4.28)$$

so that $k$ divides $p_a q_i^a$. One can then check that all the other constraints in (4.7) are satisfied. The bilinear form then reduces to

$$R^{\frac{2}{8-d}} G(\Gamma_i, \Gamma_j) = \left(R^2 y + y^{\frac{d-5}{d-1}} u^{ab} (\tfrac{p_a}{k} + a_a)(\tfrac{p_b}{k} + a_b)\right) v_{\hat{k}\hat{l}} n_i{}^{\hat{k}} n_j{}^{\hat{l}}$$
$$+ \left(y^{\frac{4}{d-1}} u_{ab} + R^{-2}(\tfrac{p_a}{k} + a_a)(\tfrac{p_b}{k} + a_b)\right) \left(q_i^a + \left(a_{\hat{k}}^a + c_{\hat{k}}^{ac}(\tfrac{p_c}{k} + a_c)\right) n_i{}^{\hat{k}}\right) \left(q_j^b + \left(a_{\hat{l}}^b + c_{\hat{l}}^{bd}(\tfrac{p_d}{k} + a_d)\right) n_j{}^{\hat{l}}\right) .$$

$$(4.29)$$

For fixed $k$ and $p_a$, the sum over $q_i^a$ is in $k\mathbb{Z} \oplus \mathbb{Z}^{d-2}$, and one can do a Poisson summation to the dual lattice $\frac{1}{k}\mathbb{Z} \oplus \mathbb{Z}^{d-2}$

$$\sum_{\substack{q_i \in \mathbb{Z}^{d-1} \\ p_a q_i^a \in k\mathbb{Z}}} e^{-\pi \Omega_2^{ij} \left(y^{\frac{4}{d-1}} u_{ab} + R^{-2}(\frac{p_a}{k}+a_a)(\frac{p_b}{k}+a_b)\right)(q_i^a + \tilde{a}_{\hat{k}}^a n_i{}^{\hat{k}})(q_j^b + \tilde{a}_{\hat{l}}^b n_j{}^{\hat{l}})}$$

$$(4.30)$$

$$= \frac{1}{y^4 |\Omega_2|^{\frac{d-1}{2}}} \sum_{\substack{q^i \in \mathbb{Z}^{d-1} \\ p \wedge q_i \in k\mathbb{Z}^{\frac{(d-1)(d-2)}{2}}}} \frac{e^{-\pi \Omega_{2ij}^{-1} y^{-\frac{4}{d-1}} \left(u^{ab} k^{-2} - \frac{u^{ac} u^{bd}(\frac{p_c}{k}+a_c)(\frac{p_d}{k}+a_d)}{y^{\frac{4}{d-1}} R^2 k^2 + u(p+ak)(p+ak)}\right) q_a^i q_b^j}}{k^2 + \frac{y^{-\frac{4}{d-1}}}{R^2} u^{ab}(p_a + a_a k)(p_b + a_b k)} e^{2\pi i \frac{n_i{}^{\hat{j}} q_a^i}{k} \tilde{a}_{\hat{j}}^a} .$$

Writing instead the sum over $n_i{}^{\hat{j}}/k$ which we write $n_i{}^{\hat{j}}$ for brevity and changing variable

$$\Omega_2 \to R^{\frac{2}{8-d}} \left(R^2 k^2 + y^{-\frac{4}{d-1}} u^{ab}(p_a + a_a k)(p_b + a_b k)\right)^{-1} \Omega_2 \qquad (4.31)$$

one obtains

$$\mathcal{I}_d^{(4)} = 8\pi R^{\frac{12+4\epsilon}{8-d} - 4\epsilon} \sum_{\gamma \in P_3 \backslash E_d} \int_{\mathcal{G}} \frac{\mathrm{d}^3 \Omega_2}{|\Omega_2|^{\frac{5}{2}-\epsilon}} \varphi_{\mathrm{KZ}}^{\mathrm{tr}}(\Omega_2) \sum_{\substack{n_i{}^{\hat{j}} \in \mathbb{Z}^2 \\ \det n \neq 0}} e^{-\pi \Omega_2^{ij} y v_{\hat{k}\hat{l}} n_i{}^{\hat{k}} n_j{}^{\hat{l}}} \sum_{q_i \in \mathbb{Z}^{d-1}} \sum_{\substack{(k,p) \\ k | p \wedge q_i}}$$

$$e^{2\pi i \frac{n_i{}^{\hat{j}} q_a^i}{k}(a_{\hat{j}}^a + c_{\hat{j}}^{ab}(\frac{p_b}{k}+a_b))} \frac{e^{-\pi \Omega_{2ij}^{-1} \left(y^{-\frac{4}{d-1}} R^2 u^{ab} q_a^i q_b^j + y^{-\frac{8}{d-1}} (u_{ac} u_{bd} - u_{ad} u_{bc})(\frac{p_c}{k}+a_c)(\frac{p_d}{k}+a_d) q_a^i q_b^j\right)}}{y^4 (k^2 + y^{-\frac{4}{d-1}} R^{-2} u(p+ak, p+ak))^{2\epsilon}} \Bigg|_{\gamma} \qquad (4.32)$$

The contribution from $q_a^i$ can be computed by Poisson resumming over $p_a \in \mathbb{Z}^{d-1}$,

$$\mathcal{I}_d^{(4a)} = \frac{8\pi}{\xi(4\epsilon)} R^{\frac{12+4\epsilon}{8-d}} \sum_{\gamma \in P_3 \backslash E_d} y^{-2} \int_{\mathcal{G}} \frac{\mathrm{d}^3 \Omega_2}{|\Omega_2|^{\frac{5}{2}-\epsilon}} \varphi_{\mathrm{KZ}}^{\mathrm{tr}}(\Omega_2) \sum_{\substack{n_i{}^{\hat{j}} \in \mathbb{Z}^2 \\ \det n \neq 0}} e^{-\pi \Omega_2^{ij} y v_{\hat{k}\hat{l}} n_i{}^{\hat{k}} n_j{}^{\hat{l}}}$$

$$\times \left(\xi(4\epsilon - d + 1) R^{d-1-4\epsilon} + 2R^{\frac{d-1}{2} - 2\epsilon} \sum_{p \in \mathbb{Z}^{d-1}}' \sigma_{d-1-4\epsilon}(p) \frac{K_{\frac{d-1}{2} - 2\epsilon}\left(2\pi R \sqrt{y^{\frac{4}{d-1}} u_{ab} p^a p^b}\right)}{\left(y^{\frac{4}{d-1}} u_{ab} p^a p^b\right)^{\frac{d-1}{4} - \epsilon}} e^{2\pi i p^a a_a}\right)\Bigg|_{\gamma}$$

$$
= \frac{8\pi}{\xi(4\epsilon)} R^{\frac{12+4\epsilon}{8-d}} \Bigg( \xi(4\epsilon - d + 1) R^{d-1-4\epsilon} \sum_{\substack{\mathcal{Q}_i \in M_{\Lambda_1}^{E_d} \\ \mathcal{Q}_i \times \mathcal{Q}_j = 0 \\ \mathcal{Q}_1 \wedge \mathcal{Q}_2 \neq 0}} \left| \frac{\gcd(\mathcal{Q}_1 \wedge \mathcal{Q}_2)}{v(\mathcal{Q}_1 \wedge \mathcal{Q}_2)} \right|^2 \int_{\mathcal{G}} \frac{\mathrm{d}^3 \Omega_2}{|\Omega_2|^{\frac{5}{2}-\epsilon}} \varphi_{\mathrm{KZ}}^{\mathrm{tr}}(\Omega_2) e^{-\pi \Omega_2^{ij} G(\mathcal{Q}_i, \mathcal{Q}_j)}
$$

$$
+ 2 R^{\frac{d-1}{2} - 2\epsilon} \sideset{}{'}\sum_{\substack{Q \in M_{\Lambda_d}^{E_d} \\ Q \times Q = 0}} \sum_{\substack{\mathcal{Q}_i \in M_{\Lambda_1}^{E_d} \\ \mathcal{Q}_i \times Q = 0 \\ \mathcal{Q}_i \times \mathcal{Q}_j = 0 \\ \mathcal{Q}_1 \wedge \mathcal{Q}_2 \neq 0}} \left| \frac{\gcd(\mathcal{Q}_1 \wedge \mathcal{Q}_2)}{v(\mathcal{Q}_1 \wedge \mathcal{Q}_2)} \right|^2 \int_{\mathcal{G}} \frac{\mathrm{d}^3 \Omega_2}{|\Omega_2|^{\frac{5}{2}-\epsilon}} \varphi_{\mathrm{KZ}}^{\mathrm{tr}}(\Omega_2) e^{-\pi \Omega_2^{ij} G(\mathcal{Q}_i, \mathcal{Q}_j)}
$$

$$
\times \sigma_{d-1-4\epsilon}(Q) \frac{K_{\frac{d-1}{2}-2\epsilon}(2\pi R |Z(Q)|)}{|Z(Q)|^{\frac{d-1}{2}-2\epsilon}} e^{2\pi \mathrm{i}(Q,a)} \Bigg) \tag{4.33}
$$

where the sum over $M_{\Lambda_1}^{E_d}$ with $\mathcal{Q}_i \times Q = 0$ in $M_{\Lambda_2}^{E_d}$, defines a function on the Levi stabiliser $E_{d-1}$ of $Q$ as a constrained double lattice sum over $M_{\Lambda_1}^{E_{d-1}}$.

We shall now argue that this contribution disappears in the renormalised integral (1.28). One can use the same argument as in Appendix C.3 to compute that the source term for the Laplace equation satisfied by this function vanishes as $\epsilon \to 0$, with

$$
\left( \Delta_{E_d} - 12 + 2 \frac{(1+2\epsilon)(d-1)(d-2-\epsilon)}{9-d} \right) \sum_{\substack{\mathcal{Q}_i \in M_{\Lambda_1}^{E_d} \\ \mathcal{Q}_i \times \mathcal{Q}_j = 0 \\ \mathcal{Q}_1 \wedge \mathcal{Q}_2 \neq 0}} \left| \frac{\gcd(\mathcal{Q}_1 \wedge \mathcal{Q}_2)}{v(\mathcal{Q}_1 \wedge \mathcal{Q}_2)} \right|^2 \int_{\mathcal{G}} \frac{\mathrm{d}^3 \Omega_2}{|\Omega_2|^{\frac{5}{2}-\epsilon}} \varphi_{\mathrm{KZ}}^{\mathrm{tr}}(\Omega_2) e^{-\pi \Omega_2^{ij} G(\mathcal{Q}_i, \mathcal{Q}_j)}
$$

$$
= -2\pi \xi(2\epsilon-1) \sum_{\gamma \in P_1 \backslash E_d} g^{-\frac{8+16\epsilon}{9-d}+2\epsilon} \left( \xi(2\epsilon-2) E_{\epsilon \Lambda_{d-2}}^{D_{d-1}} + 2 g^{1-\epsilon} \sideset{}{'}\sum_{\substack{Q \in S_+ \\ Q \times Q = 0}} \frac{\sigma_{2\epsilon-2}(Q)}{\gcd(Q)^2} \frac{K_{1-\epsilon}(\frac{2\pi}{g}|v(Q)|)}{|v(Q)|^{1+\epsilon}} e^{2\pi \mathrm{i}(Q,a)} \right)
$$

$$
= 2\pi \xi(2\epsilon-1) \xi(d-3+2\epsilon) \left( E_{(\epsilon-\frac{1}{2})\Lambda_1 + \frac{d-3+2\epsilon}{2}\Lambda_d}^{E_d} - E_{(\epsilon-\frac{1}{2})\Lambda_1}^{E_d} E_{\frac{d-3+2\epsilon}{2}\Lambda_d}^{E_d} + \mathcal{O}(\epsilon) \right) = \mathcal{O}(\epsilon) \tag{4.34}
$$

which vanishes at $\epsilon \to 0$. Therefore the potentially dangerous constant term in $\mathcal{I}_d^{(4a)}$ must be proportional to an Eisenstein series. The coefficient follows by computing the first non-trivial orbit in the string perturbation limit

$$
8\pi \sum_{\substack{\mathcal{Q}_i \in M_{\Lambda_1}^{E_d} \\ \mathcal{Q}_i \times \mathcal{Q}_j = 0 \\ \mathcal{Q}_1 \wedge \mathcal{Q}_2 \neq 0}} \left| \frac{\gcd(\mathcal{Q}_1 \wedge \mathcal{Q}_2)}{v(\mathcal{Q}_1 \wedge \mathcal{Q}_2)} \right|^2 \int_{\mathcal{G}} \frac{\mathrm{d}^3 \Omega_2}{|\Omega_2|^{\frac{5}{2}-\epsilon}} \varphi_{\mathrm{KZ}}^{\mathrm{tr}}(\Omega_2) e^{-\pi \Omega_2^{ij} G(\mathcal{Q}_i, \mathcal{Q}_j)} + \mathcal{O}(\epsilon) \tag{4.35}
$$

$$
= \frac{4\pi^2}{9} \xi(6-2\epsilon) \xi(2+2\epsilon) E_{-3\Lambda_1 + (2+\epsilon)\Lambda_3}^{E_d}
$$

$$
= \frac{2\pi^2}{9} \sum_{\substack{\mathcal{Q}_i \in M_{\Lambda_1}^{E_d} \\ \mathcal{Q}_i \times \mathcal{Q}_j = 0 \\ \mathcal{Q}_1 \wedge \mathcal{Q}_2 \neq 0}} \left| \frac{\gcd(\mathcal{Q}_1 \wedge \mathcal{Q}_2)}{v(\mathcal{Q}_1 \wedge \mathcal{Q}_2)} \right|^2 \int_{\mathcal{G}} \frac{\mathrm{d}^3 \Omega_2}{|\Omega_2|^{\frac{5}{2}-\epsilon}} \frac{E_{-3\Lambda_1}^{SL(2)}(\tau)}{V} e^{-\pi \Omega_2^{ij} G(\mathcal{Q}_i, \mathcal{Q}_j)},
$$

In the last equality, that can be computed in the same way as in Section 2.4, we recognise the same constant term as in the counterterm in (1.28). Using a functional equation, the Eisenstein series in the middle line of (4.35) is seen to diverge as $\xi(1+2\epsilon) E_{-\frac{3}{2}\Lambda_1 + \Lambda_5}^{E_d}$ for $d = 4, 5$, while it

has a finite limit otherwise. The same computation applies to the Fourier coefficients in $\mathcal{I}_d^{(4a)}$, since the function of the Levi stabiliser $E_{d-1}$ is the same. We conclude that $\mathcal{I}_d^{(4a)} \underset{\epsilon \to 0}{\to} 0$ for $d \neq 4, 5$, and expect that the same holds for the whole contribution $\mathcal{I}_d^{(4)}$. For $d = 4, 5$, $\mathcal{I}_d^{(4a)}$ has a finite limit, but cancels in the renormalised coupling (1.28).

**5) The fifth layer**

We now briefly discuss the last layer for which $n_i \neq 0$. This layer only occurs for $d = 6$. The analysis from Appendix D.1 shows that for the similar integral $\mathcal{I}_{\Lambda_7}^{E_7}\left(E_{s\Lambda_1}^{SL(2)}, 4 + 2\epsilon\right)$ where $A(\tau)$ is replaced by an arbitrary Eisenstein series, the contribution from this layer contains a general factor $\frac{\xi(4\epsilon-5)\xi(4\epsilon-9)}{\xi(4\epsilon)\xi(4\epsilon-4)}$, so we expect the same for the case of interest. However, at the specific value $s = -3$, the factor $\frac{\xi(4\epsilon-5)\xi(4\epsilon-9)}{\xi(4\epsilon)\xi(4\epsilon-4)}$ multiplies a divergent function containing $\xi(2\epsilon)$ that compensates for the $1/\xi(4\epsilon)$ and gives the finite contribution $\frac{8\pi}{27}\xi(2)\xi(10)R^{15}$ in the limit. As for the other cases we expect that this finite contribution cancels out in the renormalised function (1.28), as it must for consistency. Note that the fifth layer also contributes to generic Fourier coefficients with charges $Q$ with a non-trivial $E_6$ cubic invariant $I_3(Q) \neq 0$. However, the supersymmetric Ward identity for the renormalised coupling requires that such Fourier coefficients must vanish [14]. It is therefore consistent that this fifth layer should not contribute to the renormalised coupling after canceling the counterterm.

The contributions from the five layers produce exactly the expected constant terms in the decompactification limit shown in (4.1). For this it is crucial to take into account the renormalisation and contribution from 1/4-BPS states. This will be discussed in more detail in Section 5.3.

# 5 Regularisation and divergences

In the previous sections, we have computed the perturbative and decompactification limits of the $\nabla^6 \mathcal{R}^4$ coupling based on (1.12) that represents mutually 1/2-BPS states running inside a two-loop diagram of exceptional field theory. In the calculation we have encountered various divergent contributions, see for instance (3.44). In the present section, we analyse these singular terms in more detail and provide a renormalisation prescription that also includes 1/4-BPS states running in the loops and will be shown to cancel the pole in the two-loop 1/2-BPS contribution in space-time dimension $D = 4, 5, 6$, so that the sum of the two is finite in the limit $\epsilon \to 0$ in all dimensions. We will show that this regularisation gives the expected logarithmic term in the string coupling constant in perturbation theory.

## 5.1 Contributions from 1/4-BPS states

We follow the same reasoning as in [30], where the analogous one-loop contribution to $\nabla^4 \mathcal{R}^4$ with 1/4-BPS states running in the loop was obtained. The idea there was to interpret the perturbative genus-one string integrand in the limit $\tau_2 \to \infty$ as a sum over perturbative string states running in the loop, and extending the sum to the full non-perturbative spectrum of 1/4-BPS states. In this way, the 1/4-BPS state contribution to the $\nabla^4 \mathcal{R}^4$ coupling at one-loop

was given in [30] as

$$\mathcal{E}^{\text{1-loop } \frac{1}{4}\text{-BPS}}_{(1,0)} = 4\pi \sum_{\substack{\Gamma \in M^{E_{d+1}}_{\Lambda_{d+1}} \\ \Gamma \times \Gamma \neq 0 \ \Delta'(\Gamma)=0}} \frac{\sigma_3(\Gamma \times \Gamma)}{|\mathcal{V}(\Gamma \times \Gamma)|^2} \int_0^\infty \frac{\mathrm{d}L}{L^{\frac{6-d-2\epsilon}{2}}} \left( L + \frac{1}{2\pi|\mathcal{V}(\Gamma \times \Gamma)|} \right) e^{-\pi L \mathcal{M}(\Gamma)^2} , \quad (5.1)$$

where $|\mathcal{V}(\Gamma \times \Gamma)|^2$ is the $E_{d+1}$-invariant quadratic norm on $R(\Lambda_1)$ and $\mathcal{M}(\Gamma)$ is the mass of a state satisfying the 1/4-BPS constraints $\Gamma \times \Gamma \neq 0$ and $\Delta'(\Gamma) = 0$ with $\Delta$ the quartic invariant on $R(\Lambda_{d+1})$ and $\Delta'$ its gradient. The explicit form of the 1/4-BPS mass is

$$\mathcal{M}(\Gamma) = \sqrt{|Z(\Gamma)|^2 + 2|\mathcal{V}(\Gamma \times \Gamma)|} . \tag{5.2}$$

The contribution for each charge $\Gamma$ to (5.1) is weighted by $\sigma_3(\Gamma \times \Gamma)$, which we recognise as the twelfth helicity supertrace $\Omega_{12}(\Gamma) = \frac{1}{12!}\text{Tr}(-1)^{2J_3}(2J_3)^{12}$ counting 1/4-BPS multiplets of charge $\Gamma$, as computed in [70, 50].

Turning to $\nabla^6 \mathcal{R}^4$, a similar contribution must appear as a one-loop sub-diagram in the two-loop integrand by factorisation. We propose that the $\nabla^6 \mathcal{R}^4$ coupling receives a two-loop contribution of the form

$$\mathcal{E}^{\text{2-loop } \frac{1}{4}\text{-BPS}}_{(0,1)} = 20 \sum_{\substack{\Gamma_i \in M^{E_{d+1}}_{\Lambda_{d+1}} \\ \Gamma_1 \times \Gamma_i = 0 \\ \Gamma_2 \times \Gamma_2 \neq 0 \ \Delta'(\Gamma_2)=0}} \frac{\sigma_3(\Gamma_2 \times \Gamma_2)}{|\mathcal{V}(\Gamma_2 \times \Gamma_2)|^2} \int_{\mathbb{R}^3_+} \frac{\mathrm{d}L_1 \, \mathrm{d}L_2 \, \mathrm{d}L_3}{\left(\sum_{i>j} L_i L_j\right)^{\frac{8-d-2\epsilon}{2}}} \left( L_2 + L_3 + \frac{1}{2\pi|\mathcal{V}(\Gamma_2 \times \Gamma_2)|} \right)$$

$$\times e^{-\pi(L_1\mathcal{M}(\Gamma_1)^2 + L_2\mathcal{M}(\Gamma_2)^2 + L_3\mathcal{M}(\Gamma_1+\Gamma_2)^2)} . \tag{5.3}$$

Here the two edges with Schwinger parameters $L_2$ and $L_3$ carry 1/4-BPS multiplets of charge $\Gamma_2$ and $\Gamma_1 + \Gamma_2$, whereas the edge of length $L_1$ carries a 1/2-BPS multiplet of charge $\Gamma_1$. We assume that the two-loop contribution with a 1/2-BPS charge $\Gamma_1$ and a 1/4-BPS charge $\Gamma_2$, but with $\Gamma_1 \times \Gamma_2 \neq 0$, vanishes, as well as contributions where none of the charges $\Gamma_1, \Gamma_2, \Gamma_1 + \Gamma_2$ is 1/2-BPS.

Let us first analyse the contribution (5.3) from the point of view of perturbative string theory, by writing it as a Poincaré sum over $P_1 \backslash E_{d+1}$ of a charge sum in $II_{d,d}$. This is possible because one can always rotate the vector $\Gamma_2 \times \Gamma_2$ to a highest weight representative using $\gamma \in P_1 \backslash E_{d+1}$. In the corresponding graded decomposition, the charge $\Gamma_2$ is $q_2 \in II_{d,d}$ with $(q_2, q_2) \neq 0$ and the constraint $\Gamma_1 \times \Gamma_2 = 0$ implies according to (3.10)

$$(q_2, q_1) = 0 , \quad q_2^a \gamma_a \chi_1 = 0 , \quad q_2 \wedge N_1 = 0 , \quad q_2 \cdot N_1 = 0 , \tag{5.4}$$

from which one concludes that $\chi_1 = N_1 = 0$ and moreover from $\Gamma_1 \times \Gamma_1 = 0$ that $(q_1, q_1) = 0$. It will be useful to consider the change of variables on the Schwinger parameters

$$L_1 = g_D^{-\frac{4}{8-d}}(t + \rho_2 u_2(1 - u_2)) , \quad L_2 = g_D^{-\frac{4}{8-d}}\rho_2(1 - u_2) , \quad L_3 = g_D^{-\frac{4}{8-d}}\rho_2 u_2 , \tag{5.5}$$

where $\rho_2$ and $t$ are positive reals and $u_2 \in [0,1]$. One obtains from (5.3)

$$\mathcal{E}_{(0,1)}^{\text{2-loop } \frac{1}{4}\text{-BPS}} = 20 \sum_{\gamma \in P_1 \backslash E_{d+1}} \left[ g_D^{-\frac{24}{8-d}+4} \sum_{\substack{q_i \in II_{d,d} \\ (q_1,q_i)=0 \\ q_2^2 \neq 0}} \frac{\sigma_3(\frac{q_2^2}{2})}{(\frac{q_2^2}{2})^2} \int_{\mathbb{R}_+^2} \frac{d\rho_2\, dt}{\rho_2^2 t^3} \int_0^1 du_2 \frac{1}{t}\left(1 + \frac{1}{2\pi\rho_2|\frac{q_2^2}{2}|}\right) \right.$$

$$\left. \left. \times (\rho_2 t)^{\frac{d}{2}+\epsilon} e^{-\pi t g(q_1,q_1) - \pi\rho_2 g(q_2+u_2 q_1, q_2+u_2 q_1) - \pi\rho_2 q_2^2} \right] \right|_\gamma \tag{5.6}$$

$$= 40 \sum_{\gamma \in P_1 \backslash E_{d+1}} \left[ g_D^{-\frac{24}{8-d}+4} \int_{P_{1,2}\backslash\mathcal{H}_2(\mathbb{C})} \frac{d^6\Omega}{|\Omega_2|^{3-\epsilon}} \frac{1}{t} \sum_{n \in \mathbb{Z}}' \left( \rho_2^{\frac{1}{2}} \frac{\sigma_3(|n|)}{|n|^{\frac{3}{2}}} K_{\frac{3}{2}}(2\pi|n|\rho_2) e^{2\pi i n \rho_1} \right) \Gamma_{d,d,2}(\Omega) \right] \right|_\gamma ,$$

where

$$\Omega = \begin{pmatrix} \rho & \rho u_2 + u_1 \\ \rho u_2 + u_1 & \sigma_1 + \mathrm{i}t + \rho u_2^2 \end{pmatrix} , \tag{5.7}$$

and the integration domain $P_{1,2}\backslash\mathcal{H}_2(\mathbb{C})$ ranges over $[-\frac{1}{2}, \frac{1}{2}]$ for $\rho_1$, $u_1$, $u_2$ and $\sigma_1$ and over $\mathbb{R}_+$ for $\rho_2$ and $t$. In the last line we used the identity $K_{3/2}(2\pi|n|\rho_2) = \frac{e^{-2\pi|n|\rho_2}}{2\sqrt{|n|\rho_2}}\left(1 + \frac{1}{2\pi|n|\rho_2}\right)$.

Before evaluating (5.6) further, we note the consistency of the known expression with the genus-two contribution to the $\nabla^6\mathcal{R}^4$ coupling, given by [35, 36]

$$8\pi g_D^{-\frac{24}{8-d}+4} \int_{\mathcal{F}_2} \frac{d^6\Omega}{|\Omega_2|^3} \varphi_{\text{KZ}}(\Omega)\, \Gamma_{d,d,2}(\Omega) . \tag{5.8}$$

From (A.13) we recall that

$$\int_0^1 du_2 \int_0^1 du_1 \int_0^1 d\sigma_1 \varphi_{\text{KZ}} = \frac{\pi}{6} t + \frac{\pi}{36} \frac{E_{2\Lambda_1}^{SL(2)}(\rho)}{t} , \tag{5.9}$$

where the $SL(2)$ series has the well-known Fourier expansion

$$8\pi \cdot \frac{\pi}{36} E_{2\Lambda_1}^{SL(2)}(\rho) = \frac{2\pi^2}{9}\rho_2^2 + \frac{10\zeta(3)}{\pi}\rho_2^{-1} + 40 \sum_{n \in \mathbb{Z}}' \rho_2^{\frac{1}{2}} \frac{\sigma_3(|n|)}{|n|^{\frac{3}{2}}} K_{\frac{3}{2}}(2\pi|n|\rho_2) e^{2\pi i n \rho_1} . \tag{5.10}$$

exhibiting the same sum of Bessel functions as in (5.6). This shows that our non-perturbative proposal (5.8) does include the known perturbative contribution from perturbative string theory.

Returning to (5.6), we next fold the integral from $P_{1,2}\backslash\mathcal{H}_2(\mathbb{C})$ to the standard Siegel modular domain $\mathcal{F}_2 = Sp(4, \mathbb{Z})\backslash\mathcal{H}_2(\mathbb{C})$. The resulting Poincaré sum of the non-zero Fourier mode does not converge, but can be evaluated formally as in [30] to give

$$\sum_{\gamma \in P_{1,2}\backslash Sp(4,\mathbb{Z})} \frac{\sqrt{|n|\rho_2} K_{\frac{3}{2}}(2\pi|n|\rho_2) e^{2\pi i n \rho_1}}{t}\bigg|_\gamma \tag{5.11}$$

$$= \sum_{\gamma \in P_1 \backslash Sp(4,\mathbb{Z})} \sum_{\delta \in P_1 \backslash SL(2)} \frac{\sqrt{|n|\rho_2} K_{\frac{3}{2}}(2\pi|n|\rho_2) e^{2\pi i n \rho_1}}{t}\bigg|_\delta \bigg|_\gamma$$

$$= \frac{2\pi^2}{3\zeta(3)} \frac{\sigma_3(|n|)}{|n|} \sum_{\gamma \in P_1 \backslash Sp(4,\mathbb{Z})} \left. \frac{E_{2\Lambda_1}^{SL(2)}(\rho)}{t} \right|_\gamma = \frac{2\pi^2}{3\zeta(3)} \frac{\sigma_3(|n|)}{|n|} \sum_{\gamma \in P_2 \backslash Sp(4,\mathbb{Z})} \left. \frac{E_{-3\Lambda_1}^{SL(2)}(\tau)}{V} \right|_\gamma ,$$

where in the last step we use the analytic continuation of the Poincaré sums

$$\sum_{\gamma \in P_1 \backslash Sp(4,\mathbb{Z})} \left. \frac{E_{(2+\epsilon-\delta)\Lambda_1}^{SL(2)}(\rho)}{t^{1-\epsilon-\delta}} \right|_\gamma = E_{(2\delta-3)\Lambda_1 + (2+\epsilon-\delta)\Lambda_2}^{Sp(4)} = \sum_{\gamma \in P_2 \backslash Sp(4,\mathbb{Z})} \left. \frac{E_{(2\delta-3)\Lambda_1}^{SL(2)}(\tau)}{V^{1+2\epsilon}} \right|_\gamma , \qquad (5.12)$$

from the convergent range $\epsilon + 1 > \delta > 4$ to their value at $\epsilon = \delta = 0$. The Eisenstein series in the last term of (5.11) is recognised as a Siegel–Eisenstein series of $Sp(4)$ satisfying the functional identity

$$E_{-3\Lambda_1 + 2\Lambda_2}^{Sp(4)} = \frac{\xi(8)\xi(5)}{\xi(7)\xi(4)} E_{\frac{5}{2}\Lambda_2}^{Sp(4)} . \qquad (5.13)$$

Using these equalities between (divergent) Poincaré sums and ignoring the fact that the regularising factor $|\Omega_2|^\epsilon$ spoils modular invariance, one obtains from (5.6) that

$$\begin{aligned}
\mathcal{E}_{(0,1)}^{\text{2-loop } \frac{1}{4}\text{-BPS}} &= \frac{160\pi^2}{3\zeta(3)} \sum_{n=1}^\infty \frac{\sigma_3(n)^2}{n^3} \sum_{\gamma \in P_1 \backslash E_{d+1}} \left( g_D^{-\frac{24}{8-d}+4} \int_{\mathcal{F}_2} \frac{\mathrm{d}^6\Omega}{|\Omega_2|^{3-\epsilon}} \frac{\xi(8)\xi(5)}{\xi(7)\xi(4)} E_{\frac{5}{2}\Lambda_2}^{Sp(4)}(\Omega)\Gamma_{d,d,2}(\Omega) \right)\Big|_\gamma \\
&= \frac{160\pi^2}{3\zeta(3)} \sum_{n=1}^\infty \frac{\sigma_3(n)^2}{n^3} \int_{\mathcal{G}} \frac{\mathrm{d}^3\Omega_2}{|\Omega_2|^{\frac{6-d-2\epsilon}{2}}} \frac{E_{-3\Lambda_1}^{SL(2)}(\tau)}{V} \theta_{\Lambda_{d+1}}^{E_{d+1}}(\phi, \Omega_2) .
\end{aligned} \qquad (5.14)$$

Making use, as in [30], of the formal Ramanujan identity

$$\sum_{n=1}^\infty \frac{\sigma_3(n)^2}{n^3} = -\frac{\zeta(3)}{240} \qquad (5.15)$$

one concludes that

$$\mathcal{E}_{(0,1)}^{\text{2-loop } \frac{1}{4}\text{-BPS}} = -\frac{2\pi^2}{9} \int_{\mathcal{G}} \frac{\mathrm{d}^3\Omega_2}{|\Omega_2|^{\frac{6-d-2\epsilon}{2}}} \frac{E_{-3\Lambda_1}^{SL(2)}(\tau)}{V} \theta_{\Lambda_{d+1}}^{E_{d+1}}(\phi, \Omega_2) . \qquad (5.16)$$

Even though we have used the formal identities (5.11) and (5.15) in the derivation, we stress that the original expression (5.3) is regular and well-defined for large enough $\epsilon$.

As stated in (1.28) in the introduction, the complete two-loop amplitude with both 1/2- and 1/4-BPS states running in the loops is therefore given by the sum

$$\mathcal{E}_{(0,1)}^{(d)2\text{-loop}} = 8\pi \int_{\mathcal{G}} \frac{\mathrm{d}^3\Omega_2}{|\Omega_2|^{\frac{6-d-2\epsilon}{2}}} \left( \varphi_{\text{KZ}}^{\text{tr}} - \frac{\pi}{36} \frac{E_{-3\Lambda_1}^{SL(2)}(\tau)}{V} \right) \theta_{\Lambda_{d+1}}^{E_{d+1}}(\phi, \Omega_2) , \qquad (5.17)$$

which should be finite as $\epsilon \to 0$ in all dimensions $d \geq 4$. We show that it is indeed finite in the weak coupling and decompactification limits for $d = 4$, and for all the terms that we can compute for $d = 5$ and $d = 6$. With hindsight, the reason for this finiteness is that the divergences at $\epsilon = 0$ in $d = 4, 5, 6$ come from the constant terms proportional to $\tau_2^{-3}$ in $\varphi_{\text{KZ}}^{\text{tr}}$ and $E_{-3\Lambda_1}^{SL(2)}(\tau)/V$, that drop out in the difference (5.17).

The three-loop exceptional field theory contribution was computed and analysed in [30, (2.19)]. The analytic contribution has poles in $d = 4, 5, 6$, but those poles cancel against non-analytic contributions, leading in these dimensions to

$$\mathcal{E}_{(0,1)}^{(d)3\text{-loop}} = 40\xi(2)\,\xi(6)\,\xi(d+4)\,\widehat{E}_{\frac{d+4}{2}\Lambda_{d+1}}^{E_{d+1}} + 40\frac{\xi(8)}{\xi(7)}\xi(d+1)\xi(2s_{d+1}-d+3)\xi(2s_{d+1})\,\widehat{E}_{s_{d+1}\Lambda_H}^{E_{d+1}} \quad (5.18)$$

where $s_5 = \frac{7}{2}$, $s_6 = \frac{9}{2}$ and $s_7 = 6$ as in (2.40). For $d = 3$, the three-loop contribution decomposes similarly as[29]

$$\mathcal{E}_{(0,1)}^{(3)3\text{-loop}} = 40\,\xi(2)\,\xi(6)\,\xi(7)\,E_{\frac{7}{2}\Lambda_3}^{SL(5)} + 40\,\xi(2)\,\xi(3)\,\xi(4)\,E_{-\frac{1}{2}\Lambda_1-\Lambda_4}^{SL(5)} \quad (5.19)$$

which is finite and does not require regularisation. Both terms in (5.18) or (5.19) are homogeneous solutions of the Laplace equation (1.5), but they satisfy different tensorial equations [14] and thus belong to two distinct automorphic representations. In particular, the second function solves (2.10) whereas the first one does not. The first function in (5.18) is recognised as $\widehat{\mathcal{F}}_{(0,1)}^{(d)}$ in (1.30), while the second can be written using (2.40) as the finite part of

$$\mathcal{E}_{(0,1)}^{(d)\text{Adj}} = \frac{2\pi^2}{9}\int_{\mathcal{G}}\frac{\mathrm{d}^3\Omega_2}{|\Omega_2|^{\frac{6-d}{2}}}\frac{E_{-(3+2\epsilon)\Lambda_1}^{SL(2)}(\tau)}{V}\theta_{\Lambda_{d+1}}^{E_{d+1}}(\phi,\Omega_2) \quad (5.20)$$

$$= 40\frac{\xi(4)\xi(8+4\epsilon)}{\xi(4+2\epsilon)\xi(7+4\epsilon)}\xi(d+1+2\epsilon)\xi(2s_{d+1}+2\epsilon-d+3)\xi(2s_{d+1}+2\epsilon)\,\widehat{E}_{(s_{d+1}+\epsilon)\Lambda_H}^{E_{d+1}}\,,$$

at $\epsilon \to 0$ in agreement with the last term in (1.31).[30] One finds therefore that the sum of the second term in the three-loop contribution with the full two-loop contribution (5.16) reproduces (1.31).

Thus one gets the exact coupling (1.32) stated in the introduction, with a now precise prescription for defining the divergent integral (1.12). It will be useful in the analysis below to rewrite (1.31) as the finite function

$$\widehat{\mathcal{E}}_{(0,1)}^{(d),\text{ExFT}} = \frac{8\pi^2}{3}\mathcal{I}_{\Lambda_{d+1}}^{E_{d+1}}(A(\tau) - \tfrac{1}{6}E_{-3\Lambda_1}^{SL(2)}, d-2) + \frac{40\xi(8)\xi(d+1)\xi(2s_{d+1}+3-d)\xi(2s_{d+1})}{\xi(7)}\widehat{E}_{s_{d+1}\Lambda_H}^{E_{d+1}}\,, \quad (5.21)$$

where $\mathcal{I}_{\Lambda_{d+1}}^{E_{d+1}}(f,s)$ was defined in general in (2.32).

In summary, we have explained how the total $\nabla^6\mathcal{R}^4$ coupling arises from the sum of the one-, two- and three-loop four-graviton amplitudes including massive 1/2-BPS and 1/4-BPS states running in the loops. Since for $d < 7$ there are two distinct supersymmetry invariants completing $\nabla^6\mathcal{R}^4$ [14], it is natural to decompose this coupling into two functions as in (1.32).

---

[29]The weight $\Lambda_{d-1}$ in [30, (2.19)] is in general the highest weight of the third order antisymmetric product of $\Lambda_{d+1}$, which for $d = 3$ gives two solutions: $\Lambda_1 + 2\Lambda_4$ and $2\Lambda_2$ of $SL(5)$, corresponding respectively to type IIA and type IIB. The three-loop contribution is therefore $40\xi(-1)\xi(-2)\xi(-3)(E_{-\frac{1}{2}(\Lambda_1+2\Lambda_4)}^{SL(5)} + E_{-\frac{1}{2}\Lambda_2}^{SL(5)})$.

[30]The same formula holds for $d = 3$ with

$$\frac{2\pi^2}{9}\int_{\mathcal{G}}\frac{\mathrm{d}^3\Omega_2}{|\Omega_2|^{\frac{3}{2}}}\frac{E_{-(3+2\epsilon)\Lambda_1}^{SL(2)}(\tau)}{V}\theta_{\Lambda_3}^{A_4}(\phi,\Omega_2) = 40\xi(2+2\epsilon)\xi(3+2\epsilon)\xi(4)E_{-(\frac{1}{2}+\epsilon)\Lambda_1-(1+\epsilon)\Lambda_4}^{SL(5)}\,.$$

The computation of the 1/4-BPS states contributions at one- and two-loops are rather formal as in (5.11), and we do not understand the analytic continuation that would lead to a justification of the formal infinite sums we have been doing. The derivation of these contributions can therefore be considered as heuristic. Nonetheless, the final definition (5.21) can be justified independently as the unique regularisation of the two-loop integral that is consistent with supersymmetry Ward identities and the string theory perturbative expansion. Indeed, as we show in detail in Appendix D in (D.30) and below, the term $\mathcal{I}_{\Lambda_6}^{E_6}(E_{-3\Lambda_1}^{SL(2)}, 3 + 2\epsilon)$ in (5.21) yields an Eisenstein series that belongs to an automorphic representation associated to a bigger nilpotent orbit than the one required by supersymmetry according to equation (2.10). It is therefore apparent that only the finite combination involving $A(\tau) - \frac{1}{6}E_{-3\Lambda_1}^{SL(2)}$ in (5.17) solves (2.10) with the appropriate source term as written in [1], and is therefore consistent with supersymmetry. The last term in (5.21), involving an adjoint Eisenstein series, is the appropriate homogeneous solution to the homogeneous tensorial equation (2.10). As we shall argue in Section 5.2, its presence in the full coupling is required for consistency with string perturbation theory.

The renormalisation prescription (5.21) also makes the equality of the particle and string multiplet sums stated in (1.23) meaningful. While the identity (1.23) is divergent at the values of interest for the functional relation, the renormalised integral $\widehat{\mathcal{E}}_{(0,1)}^{(d),\mathrm{ExFT}}$ of (5.21) makes sense on either side and the equality holds for these renormalised couplings.

## 5.2 Divergences and threshold terms in the weak-coupling limit

We shall now analyse the cancelation of divergences and the contributions to logarithmic terms from (5.21) for each of the constant terms derived in Section 3. For brevity we shall refer to the last term in (5.21) as the 'adjoint Eisenstein series'

$$\mathcal{E}_{(0,1)}^{(\mathrm{d})\mathrm{Adj}} = \frac{40\xi(8)\xi(d+1)\xi(2s_{d+1}+3-d)\xi(2s_{d+1})}{\xi(7)}\widehat{E}_{s_{d+1}\Lambda_H}^{E_{d+1}}. \tag{5.22}$$

The second term $\mathcal{I}_{\Lambda_{d+1}}^{E_{d+1}}(E_{-3\Lambda_1}^{SL(2)}, d + 2\epsilon - 2)$ in (5.21) coming from the 1/4-BPS state sum (5.16) will be referred to as the 'counterterm', the idea being that exceptional field theory contains only loops of 1/2-BPS states, while additional contributions from 1/4-BPS states are described by suitable counterterms. To analyse the perturbative terms we must deal with the poles at $\epsilon = 0$ using the renormalisation prescription for the integral (5.17). There are divergent contributions from what we called the second layer in (3.21), the third layer in (3.44), the fourth layer in (3.47) and lastly, from the fifth layer in (3.58).

We have already argued that the contributions from the third and the fourth layers, which are both proportional to $\frac{1}{\xi(4\epsilon)}$, should cancel when using the renormalisation prescription (5.17). We expect the same to happen for the fifth layer in $d = 6$ proportional to $\frac{1}{\xi(4\epsilon)}$. This is indeed the case if the last term in the conjectured expansion (3.58) is correct. We are not able to check this property at this stage, and leave it as a conjecture.

Let us now turn to the terms that contribute at three-loop order in string perturbation theory. The divergent contribution from the second perturbative layer $\mathcal{I}_d^{(2b)}$ in (3.21) is given by

$$g_D^{-\frac{24}{8-d}-2\epsilon\frac{d-4}{8-d}}\frac{4\pi^2\xi(2+2\epsilon)\xi(6-2\epsilon)}{9}g_D^6 E_{(3-\epsilon)\Lambda_{d-1}+2\epsilon\Lambda_d}^{D_d}. \tag{5.23}$$

while the correct contribution appearing at three loops in string theory, computed using the same regularisation as in [30], is instead the zeroth order term at $\epsilon \to 0$ of

$$g_D^{-\frac{24}{8-d}+2\epsilon} \frac{4\pi^2 \xi(2+2\epsilon)\xi(6+2\epsilon)}{9} g_D^6 E^{D_d}_{(3+\epsilon)\Lambda_{d-1}} \,. \tag{5.24}$$

By examining the constant terms of the two functions one sees that these two results differ. The discrepancy is, however, resolved using the renormalised integral (5.21) as follows. Using the Langlands constant term formula (D.2) in Appendix D.1 one finds that the contribution from the counterterm in (5.17) cancels against (5.23), such that (5.17) is indeed finite, while the contribution from the adjoint Eisenstein series in (5.21) gives precisely the perturbative term (5.24) as was already observed in [47].

In addition, the counterterm (5.16) and adjoint series (5.20) both include a spurious correction in $g_D^{-3}$ in string frame, that cancels out in the total coupling. In $d = 6$ there is an additional spurious contribution from the adjoint series in $g_D^{-12}$ in string frame, that cancels the same one from the counterterm in (3.58), while the divergent one-loop term in (3.59) is canceled in the renormalised coupling (3.58).

In summary, the full $\nabla^6 \mathcal{R}^4$ coupling (5.21) reproduces all the expected perturbative corrections detailed in Section 3.1: The tree-level term appears in (3.18). The one-loop correction comes from (3.24), with the additional logarithmic term for $d = 6$ that comes from the adjoint series (5.20) that we shall discuss below. The two-loop term comes from (3.13). The three-loop correction in (3.24) is canceled by the counterterm and replaced by the function (5.24) from the adjoint series (5.20).

We close the perturbative analysis of the $\nabla^6 \mathcal{R}^4$ coupling by a discussion of the logarithmic terms. To analyse them we shall need the precise weak coupling expansion of $\widehat{\mathcal{F}}^{(d)}_{(0,1)}$ which corresponds to the first term in (5.18) and of $\widehat{\mathcal{E}}^{(d),\mathrm{ExFT}}_{(0,1)}$. The weak coupling expansion of $\widehat{\mathcal{F}}^{(d)}_{(0,1)}$ is given for $1 \leq d \leq 6$ by

$$\widehat{\mathcal{F}}^{(1)}_{(0,1)} \sim \frac{4\zeta(2)\zeta(5)}{63} g_9^{-10/7}(r^5 + r^{-5}) + \frac{4}{27}\zeta(6)\, r^3\, g_9^{18/7}\,,$$

$$\widehat{\mathcal{F}}^{(2)}_{(0,1)} \sim \left[ \frac{8\pi\zeta(6)}{567 g_8^2} E^{SL(2)}_{3\Lambda_1}(U) + \frac{4\zeta(6)}{27} g_8^2 \right] E^{SL(2)}_{3\Lambda_1}(T)\,,$$

$$\widehat{\mathcal{F}}^{(3)}_{(0,1)} \sim \frac{5\pi\zeta(7)}{189} g_7^{-14/5} E^{D_3}_{\frac{7}{2}\Lambda_1} + \frac{4\zeta(6)}{27} g_7^{6/5} E^{D_3}_{3\Lambda_3}\,,$$

$$\widehat{\mathcal{F}}^{(4)}_{(0,1)} \sim \frac{16\zeta(8)}{189} \frac{1}{g_6^4} E^{D_4}_{4\Lambda_3} + \frac{5}{3}\zeta(3) \log g_6 + \frac{4\zeta(6)}{27} \widehat{E}^{D_4}_{3\Lambda_4}\,, \tag{5.25}$$

$$\widehat{\mathcal{F}}^{(5)}_{(0,1)} \sim \frac{5\zeta(9)}{54 g_5^6} E^{D_5}_{\frac{9}{2}\Lambda_1} + \left[ \frac{40}{9 g_5^4}\zeta(3) + \frac{10}{9 g_5^2}\zeta(3) E^{D_5}_{\frac{3}{2}\Lambda_1} \right] \log g_5 + \frac{4\zeta(6)}{27 g_5^2} E^{D_5}_{3\Lambda_5}\,,$$

$$\widehat{\mathcal{F}}^{(6)}_{(0,1)} \sim \frac{64\zeta(10)}{189\pi g_4^{10}} \widehat{E}^{D_6}_{5\Lambda_1} + \left[ -\frac{5\zeta(5)}{\pi g_4^{10}} + \frac{8\zeta(8)}{3\pi^2 g_4^8} E^{D_6}_{4\Lambda_1} \right] \log g_4 - \frac{2\zeta(6)}{15 g_4^8} \partial_\epsilon E^{D_6}_{(4+\epsilon)\Lambda_1}\Big|_{\epsilon=0} + \frac{4\zeta(6)}{27 g_4^6} \widehat{E}^{D_6}_{3\Lambda_6}\,,$$

where we have shown the complete result of the constant term calculation. We first discuss the logarithmic terms for $d = 4, 5, 6$ and then the derivative of the Eisenstein series that appears for $d = 6$. And finally we will discuss the special case $d = 2$.

The integral in (5.17) is finite layer-by-layer for $d = 4, 5, 6$ and the cancellation of the pole between the two-loop integral and the counterterm also hold for the logarithmic terms. Therefore

the logarithmic terms in $\widehat{\mathcal{E}}_{(0,1)}^{(d),\text{ExFT}}$ must come exclusively from the three-loop contribution (5.18). As was shown in [47], the adjoint series corresponding to the second term in (5.18) produces the logarithmic terms

$$\widehat{\mathcal{E}}_{(0,1)}^{(4),\text{ExFT}} \underset{\log g_6}{\sim} \mathcal{E}_{(0,1)}^{(4)\text{Adj}} \underset{\log g_6}{\sim} \frac{10}{3}\zeta(3)\log g_6 + \dots,$$

$$\widehat{\mathcal{E}}_{(0,1)}^{(5),\text{ExFT}} \underset{\log g_5}{\sim} \mathcal{E}_{(0,1)}^{(5)\text{Adj}} \underset{\log g_5}{\sim} \frac{10}{3g_5^2}\zeta(3) E_{\frac{3}{2}\Lambda_1}^{D_5}\log g_5 + \dots, \tag{5.26}$$

$$\widehat{\mathcal{E}}_{(0,1)}^{(6),\text{ExFT}} \underset{\log g_4}{\sim} \mathcal{E}_{(0,1)}^{(6)\text{Adj}} \underset{\log g_4}{\sim} \left[\frac{10\zeta(5)}{\pi g_4^{10}} + \frac{2\pi^3}{27g_4^6} E_{2\Lambda_6}^{D_6}\right]\log g_4 + \dots.$$

Combining the contributions from (5.25) and (5.26) then produces the following total logarithmic terms for $\mathcal{E}_{(0,1)}^{(d)}$

$$\mathcal{E}_{(0,1)}^{(4)} \underset{\log g_6}{\sim} 5\zeta(3)\log g_6, \tag{5.27}$$

$$\mathcal{E}_{(0,1)}^{(5)} \underset{\log g_5}{\sim} \left[\frac{40}{9g_5^4}\zeta(3) + \frac{40}{9g_5^2}\zeta(3)E_{\frac{3}{2}\Lambda_1}^{D_5}\right]\log g_5 \sim \frac{20}{9}\mathcal{E}_{(0,0)}^{(5)}\log g_5, \tag{5.28}$$

$$\mathcal{E}_{(0,1)}^{(6)} \underset{\log g_4}{\sim} \left[\frac{5\zeta(5)}{\pi g_4^{10}} + \frac{2\pi^3}{27g_4^6} E_{2\Lambda_6}^{D_6} + \frac{8\zeta(8)}{3\pi^2 g_4^8} E_{4\Lambda_1}^{D_6}\right]\log g_4 \sim \frac{5}{\pi}\mathcal{E}_{(1,0)}^{(6)}\log g_4. \tag{5.29}$$

where $\mathcal{E}_{(0,0)}^{(d)}$ and $\mathcal{E}_{(1,0)}^{(d)}$ are the coefficients of the effective $\mathcal{R}^4$ and $\nabla^4 \mathcal{R}^4$ couplings given by (1.2). This indeed produces the expected non-analytic terms in the weak coupling expansion of the $\nabla^6\mathcal{R}^4$ couplings [2]. Note that the coefficient of the $\log g_D$ correction had to recombine into a U-duality invariant function, since it is related to the scale of a logarithm in Mandelstam variables [8], determined by form factor divergences in supergravity.

In $d = 6$ one must be more careful with the two-loop contributions since they potentially include an additional logarithmic term and a derivative of an Eisenstein series. Adding the two-loop contribution (3.13), the similar contribution from the counterterm and the two-loop contribution from the adjoint series leads to

$$8\pi g_4^{-8-4\epsilon} \int_{\mathcal{F}_2} \frac{\mathrm{d}^6\Omega}{|\Omega_2|^3} \left(\varphi_{\text{KZ}}^\epsilon(\Omega) - \frac{\pi}{36} E_{-3\Lambda_1+(2+\epsilon)\Lambda_2}^{Sp(4)}(\Omega)\right)\Gamma_{6,6,2}(\Omega)$$

$$+ \frac{2\zeta(6)}{15g_4^8}\partial_\epsilon E_{(4-2\epsilon)\Lambda_1+\epsilon\Lambda_2}^{D_6}\Big|_{\epsilon^0}, \tag{5.30}$$

and where $|_{\epsilon^0}$ denotes the constant term in the Laurent expansion around $\epsilon = 0$. In Appendix C we provide evidence that this genus-two integral is finite at $\epsilon \to 0$, such that (5.17) is indeed finite as claimed. If so, it cannot contribute to $\log g_4$ terms, and there is therefore no $\log g_4$ at two-loop order. The finite two-loop contributions from the Eisenstein series then add up to

$$-\frac{2\pi^2}{9g_4^8} \int_{\mathcal{F}_2} \frac{\mathrm{d}^6\Omega}{|\Omega_2|^3} E_{-3\Lambda_1+(2+\epsilon)\Lambda_2}^{Sp(4)}(\Omega)\,\Gamma_{6,6,2}(\Omega) + \frac{2\zeta(6)}{15g_4^8}\partial_\epsilon E_{(4-2\epsilon)\Lambda_1+\epsilon\Lambda_2}^{D_6}\Big|_{\epsilon^0}$$

$$= -\frac{2\zeta(6)}{15g_4^8}\partial_\epsilon E_{4\Lambda_1+\epsilon\Lambda_2}^{D_6}\Big|_{\epsilon=0} + \frac{2\zeta(6)}{15g_4^8}\partial_\epsilon E_{(4-2\epsilon)\Lambda_1+\epsilon\Lambda_2}^{D_6}\Big|_{\epsilon^0}$$

$$= -\frac{4\zeta(6)}{15g_4^8}\partial_\epsilon E_{(4+\epsilon)\Lambda_1}^{D_6}\Big|_{\epsilon^0}. \tag{5.31}$$

In order to reproduce the genus-two string theory amplitude (5.8), it should be that

$$\text{R.N.} \int_{\mathcal{F}_2} \frac{\mathrm{d}^6\Omega}{|\Omega_2|^3} \varphi_{\text{KZ}}(\Omega) \, \Gamma_{6,6,2}(\Omega) = \int_{\mathcal{F}_2} \frac{\mathrm{d}^6\Omega}{|\Omega_2|^3} \varphi_{\text{KZ}}^\epsilon(\Omega) \, \Gamma_{6,6,2}(\Omega) \Big|_{\epsilon=0} - \frac{\zeta(6)}{20\pi} \partial_\epsilon E_{(4+\epsilon)\Lambda_1}^{D_6} \Big|_{\epsilon^0}, \quad (5.32)$$

where we factored out $8\pi g_4^{-8}$ from (5.8). This identity may hold up to terms proportional to $E_{4\Lambda_1}^{D_6}$ which can be absorbed by adjusting the splitting between the analytic and non-analytic parts of the full amplitude. This ambiguity appears in the renormalisation of the pole

$$\int_{\mathcal{F}_2} \frac{\mathrm{d}^6\Omega}{|\Omega_2|^3} \varphi_{\text{KZ}}^\epsilon(\Omega) \, \Gamma_{6,6,2}(\Omega) = \frac{\pi}{18} \xi(1+2\epsilon)\xi(8+2\epsilon) E_{(4+\epsilon)\Lambda_1}^{D_6} + \mathcal{O}(\epsilon^0) \quad (5.33)$$

at $\epsilon \to 0$. A more detailed analysis would be needed to establish (5.32), which we again leave as a conjecture.

Let us end this discussion with the case $d = 2$, for which there is a logarithmic supergravity divergence with a double pole in dimensional regularisation at two loops. The sources of logarithmic divergences in $\mathcal{I}_2(\phi, \epsilon)$ come from the two-loop integral $\mathcal{I}_2^{(1)}$ (3.13) that behaves as (cf. Appendix F)

$$\mathcal{I}_2^{(1)} = 8\pi g_8^{-\frac{4}{3}\epsilon} \int_{\mathcal{F}_2} \frac{\mathrm{d}^6\Omega}{|\Omega_2|^3} \varphi_{\text{KZ}}^\epsilon(\Omega) \, \Gamma_{2,2,2}(\Omega) \quad (5.34)$$

$$= g_8^{-\frac{4}{3}\epsilon} \left( \frac{16\pi^2 \xi(2\epsilon)^2}{(4-2\epsilon)(3+2\epsilon)} E_{2\epsilon\Lambda_1}^{SL(2)}(U) E_{2\epsilon\Lambda_1}^{SL(2)}(T) + \mathcal{O}(\epsilon^0) \right),$$

and the contribution

$$\frac{8\pi^2}{3} g_8^{-2+\frac{2}{3}\epsilon} \xi(-2+2\epsilon)\xi(2-2\epsilon) E_{(1-\epsilon)\Lambda_1+2\epsilon\Lambda_2}^{D_2} = \frac{8\pi^2}{3} g_8^{-2+\frac{2}{3}\epsilon} \xi(3-2\epsilon)\xi(2-2\epsilon) E_{(1-\epsilon)\Lambda_1}^{SL(2)}(U) \quad (5.35)$$

from (3.21). To cancel the pole in $\frac{1}{\epsilon}$, we must include the divergent component of the supergravity amplitude, with massless legs, as well as the divergent component of the one-loop $\mathcal{R}^4$ form-factor associated to the two-loop exceptional field theory amplitude with only massless states running in one of the loops. Implementing an infrared cut-off $\mu$ as in [1,30], one obtains [31]

$$\mathcal{I}_{2,\epsilon} + \frac{\pi}{3} \pi^{-\epsilon} \Gamma(\epsilon) \, \mu^{-2\epsilon} \, \mathcal{E}_{(0,0),\epsilon}^{(2)} + \frac{\pi^2}{3} \left( \pi^{-2\epsilon} \Gamma(\epsilon)^2 + \tfrac{1}{6}\pi^{-\epsilon}\Gamma(\epsilon) + \mathcal{O}(\epsilon^0) \right) \mu^{-4\epsilon} \quad (5.36)$$

$$\underset{\epsilon\to 0}{\sim} \frac{4\pi^2}{27} \log(g_8)^2 + \frac{2\pi}{9} \log(g_8) \left( \frac{2\zeta(2)}{g_8^2} + 4\zeta(4)(\widehat{E}_{\Lambda_1}^{SL(2)}(U) + \widehat{E}_{\Lambda_1}^{SL(2)}(T)) + \frac{\pi}{3} \right) + \dots$$

$$+ \frac{4\pi^2}{3} \log(2\pi\mu)^2 - \frac{2\pi}{3} \log(2\pi\mu) \left( \frac{2\zeta(2)}{g_8^2} + 4\zeta(4)(\widehat{E}_{\Lambda_1}^{SL(2)}(U) + \widehat{E}_{\Lambda_1}^{SL(2)}(T)) + \frac{4\pi}{3} \log(g_8) + \frac{\pi}{3} \right)$$

where the dots stand for analytic terms in the string coupling constant coming from $\mathcal{I}_{2,\epsilon}$, and

$$\mathcal{E}_{(0,0),\epsilon}^{(2)} = 4\pi\xi(2\epsilon) E_{\epsilon\Lambda_1}^{SL(2)} E_{\epsilon\Lambda_2}^{SL(3)} = 4\pi\xi(2\epsilon) + \mathcal{E}_{(0,0)}^{(2)} + \mathcal{O}(\epsilon)$$

$$\sim 4\pi \left( \xi(2\epsilon) g_8^{-\frac{2}{3}} E_{\epsilon\Lambda_1}^{SL(2)}(U) E_{\epsilon\Lambda_1}^{SL(2)}(T) + \xi(3-2\epsilon) g_8^{-2+\frac{4}{3}\epsilon} E_{\epsilon\Lambda_1}^{SL(2)}(U) \right) \quad (5.37)$$

---

[31] The term $\tfrac{1}{6}\pi^{-\epsilon}\Gamma(\epsilon)L^{2\epsilon}$ has not been derived but must be there for cancelling the first order pole.

is the dimensionally regularised one-loop exceptional field theory $\mathcal{R}^4$ coupling [1]. The result is finite at $\epsilon \to 0$ and reproduces the logarithmic terms in the string coupling constant computed in [2, (2.19)], up to the additive, scheme-dependent constant $\frac{\pi}{3}$.

We conclude that the renormalised coupling (1.32) reproduces correctly all the required terms in the weak coupling expansion, including the terms that are logarithmic in the string coupling constant.

## 5.3   Divergences and threshold terms in the large radius limit

In the decompactification limit, similar divergent terms arise in the calculation presented in Section 4 and have to be considered along with the renormalisation and three-loop contribution shown in (5.21). More specifically, there are divergences in the first layer in (4.11), in the second layer in (4.13), in the fourth layer in (4.33) and in the fifth layer. Most of them were already discussed in detail after their derivation and we now focus in more detail on the second layer.

In the derivation of the second layer we used identities that are only valid at $\epsilon = 0$. However, the calculation in Appendix D.1 shows that the derivation gives the correct result at $\epsilon \neq 0$ if the local modular function $A(\tau)$ is replaced by an ordinary non-holomorphic Eisenstein series. Indeed we find a consistent result for $d = 4, 5, 6$ using the regularised expression (4.13). Taking this contribution from the second layer, subtracting the 1/4-BPS counterterm (5.17) and adding the three-loop contribution to give (1.31), one gets

$$
\mathcal{I}_{d,\epsilon}^{(2b)}\big[A(\tau) - \tfrac{\pi}{6}E_{-3\Lambda_1}^{SL(2)}(\tau)\big] + \mathcal{I}_{d,0}^{(2b)}\big[\tfrac{\pi}{6}E_{-(3+2\epsilon)\Lambda_1}^{SL(2)}\big] \tag{5.38}
$$

$$
\sim \frac{4\pi^2}{9} R^{\frac{12+4\epsilon}{8-d}}\left(6\xi(d_\epsilon - 2)\xi(d_\epsilon - 3)R^{d_\epsilon - 3}\, E_{\frac{d_\epsilon-3}{2}\Lambda_d}^{E_d} + \xi(d_\epsilon - 6)\xi(d_\epsilon + 1)R^{d_\epsilon - 7}\, E_{\frac{d_\epsilon+1}{2}\Lambda_d}^{E_d}\right.
$$

$$
\left. - \frac{\xi(8)}{\xi(7)}\xi(d_\epsilon - 6)\xi(d_\epsilon + 1)R^{d_\epsilon}\, E_{\frac{d_\epsilon-6}{2}\Lambda_d}^{E_d} - \xi(d_\epsilon - 6)\xi(d_\epsilon + 1)R^{d_\epsilon - 7}\, E_{\frac{d_\epsilon+1}{2}\Lambda_d}^{E_d}\right)
$$

$$
+ \frac{4\pi^2}{9}\xi(d - 6 - 2\epsilon)\xi(d + 1 + 2\epsilon)R^{\frac{12}{8-d}}\left(\frac{\xi(8)}{\xi(7)}R^{d+2\epsilon}\, E_{\frac{d-6-2\epsilon}{2}\Lambda_d}^{E_d} + R^{d-7-2\epsilon}\, E_{\frac{d+1+2\epsilon}{2}\Lambda_d}^{E_d}\right)
$$

$$
\sim R^{\frac{12}{8-d}}\left(\frac{8\pi^2}{3}\xi(d - 2)\xi(d + 2\epsilon - 3)R^{d-3}\, E_{\frac{d+2\epsilon-3}{2}\Lambda_d}^{E_d} + \delta_{d,6}\frac{8\zeta(10)}{\pi^2}R^{12}\log(R)\right.
$$

$$
\left. + \frac{4\pi^2\xi(d - 6 - 2\epsilon)\xi(d + 1 + 2\epsilon)}{9}R^{d-7-2\epsilon}\, E_{\frac{d+1+2\epsilon}{2}\Lambda_d}^{E_d}\right).
$$

The first term above is regular at $\epsilon = 0$ for $d \geq 4$. For $d = 6$, the extra term above cancels against the contribution from the first term (4.11). The last term gives

$$
\frac{4\pi^2\xi(3 + 2\epsilon)\xi(5 + 2\epsilon)}{9}R^{-3-2\epsilon}\, E_{(\frac{5}{2}+\epsilon)\Lambda_3}^{A_4} \sim \frac{10\zeta(3)}{3R^3}\left(\tfrac{1}{2\epsilon} - \log(R) + \frac{6}{\pi}\xi(4)\xi(6)\widehat{E}_{\frac{5}{2}\Lambda_3}^{A_4}\right)
$$

$$
\frac{4\pi^2\xi(2 + 2\epsilon)\xi(6 + 2\epsilon)}{9}R^{-2-2\epsilon}\, E_{(3+\epsilon)\Lambda_5}^{D_5} \sim \frac{5}{3R^2}\left(\left(\tfrac{1}{2\epsilon} - \log(R)\right)4\pi\xi(2)E_{\Lambda_4}^{D_5} + 4\pi\xi(4)\xi(6)\widehat{E}_{3\Lambda_5}^{D_5}\right)
$$

$$
\frac{4\pi^2\xi(1 + 2\epsilon)\xi(7 + 2\epsilon)}{9}R^{-1-2\epsilon}\, E_{(\frac{7}{2}+\epsilon)\Lambda_6}^{E_6} \sim 40\xi(2)R^{-2\epsilon}\xi(1 + 2\epsilon)\xi(5 - 2\epsilon)E_{(\frac{5}{2}-\epsilon)\Lambda_1}^{E_6} \tag{5.39}
$$

for $d = 4, 5, 6$, respectively, reproducing the expected result displayed in Section 4.1.

We close this subsection by considering the logarithmically divergent contributions in the decompactification limit for $d \geq 4$. The logarithmic terms in the radius $R$ arising from $\widehat{\mathcal{F}}_{(0,1)}^{(d)}$ as given by the first term in (5.18) are for $d = 4, 5, 6$

$$
\widehat{\mathcal{F}}_{(0,1)}^{(4)} \underset{\log R}{\sim} -\frac{5}{2}\zeta(3)\log R \, ,
$$
$$
\widehat{\mathcal{F}}_{(0,1)}^{(5)} \underset{\log R}{\sim} \frac{10}{9}\zeta(3)R^4 \log R - \frac{20}{9}\zeta(3)R^2 \, E_{\frac{3}{2}\Lambda_1}^{D_5} \log R \, ,
$$
$$
\widehat{\mathcal{F}}_{(0,1)}^{(6)} \underset{\log R}{\sim} -\frac{4}{\pi^2}\zeta(8)R^{12}\log R + \frac{5}{3}\zeta(3)R^6 E_{\frac{3}{2}\Lambda_1}^{E_6} \log R - \frac{5}{2\pi}\zeta(5)R^5 E_{\frac{5}{2}\Lambda_1}^{E_6} \log R \, . \tag{5.40}
$$

Turning to the decompactification limit of $\widehat{\mathcal{E}}_{(0,1)}^{(d),\mathrm{ExFT}}$ we note that there was a logarithmic contribution for $d = 6$ coming from the first constant term in (4.11) as well as the counterterm and there are additional contributions coming from the counterterm as well as from the three-loop amplitude given by the adjoint series given in (5.20). Therefore we have the following logarithmic terms

$$
\widehat{\mathcal{E}}_{(0,1)}^{(4),\mathrm{ExFT}} \underset{\log R}{\sim} \mathcal{E}_{(0,1)}^{(4)\mathrm{Adj}} \underset{\log R}{\sim} -\frac{10}{3}\zeta(3)\log R + \ldots
$$
$$
\widehat{\mathcal{E}}_{(0,1)}^{(5),\mathrm{ExFT}} \underset{\log R}{\sim} \mathcal{E}_{(0,1)}^{(5)\mathrm{Adj}} \underset{\log R}{\sim} -\frac{10}{3}\zeta(3)R^2 \, E_{\frac{3}{2}\Lambda_1}^{D_5} \log R \tag{5.41}
$$
$$
\widehat{\mathcal{E}}_{(0,1)}^{(6),\mathrm{ExFT}} \underset{\log R}{\sim} -\frac{8}{\pi^2}\zeta(8)R^{12}\log + \frac{16}{3\pi^2}\zeta(8)R^{12}\log R + \mathcal{E}_{(0,1)}^{(6)\mathrm{Adj}} \underset{\log R}{\sim} -\frac{5}{\pi}\zeta(5)R^5 \, E_{\frac{5}{2}\Lambda_1}^{E_6} \log R \, .
$$

Combining the two contributions then gives the following logarithmic terms

$$
\mathcal{E}_{(0,1)}^{(4)} \underset{\log R}{\sim} -\frac{35}{6}\zeta(3)\log R \, ,
$$
$$
\mathcal{E}_{(0,1)}^{(5)} \underset{\log R}{\sim} -\frac{25}{9}R^2 \mathcal{E}_{(0,0)}^{(4)} \log R + \frac{10}{9}\zeta(3)R^4 \log R \, ,
$$
$$
\mathcal{E}_{(0,1)}^{(6)} \underset{\log R}{\sim} -\frac{15}{2\pi}R^5 \mathcal{E}_{(1,0)}^{(5)} \log R + \frac{5}{6}R^6 \mathcal{E}_{(0,0)}^{(5)} \log R - \frac{4}{\pi^2}\zeta(8)R^{12}\log R \tag{5.42}
$$

where $\mathcal{E}_{(0,0)}^{(d-1)}$ and $\mathcal{E}_{(1,0)}^{(d-1)}$ are the coefficients of the $\mathcal{R}^4$ and $\nabla^4 \mathcal{R}^4$ couplings in dimension $D + 1$, given by (1.2) (after using Langlands functional equations). This agrees with the coefficients of the logarithms found in [2, (B.62)].

For $d = 6$, we must also consider the derivative of the Eisenstein series. This derivative arises from the term proportional to $R^5$ in $\widehat{\mathcal{F}}_{(0,1)}^{(6)}$ that takes the form

$$
\widehat{\mathcal{F}}_{(0,1)}^{(6)} \underset{R^5}{\sim} 40\xi(2)\xi(6)\left( R^{5-\epsilon}\frac{\xi(1+2\epsilon)\xi(5+2\epsilon)}{\xi(6+2\epsilon)}E_{(\frac{5}{2}+\epsilon)\Lambda_1}^{E_6} - R^5 \frac{\xi(5)}{2\epsilon\xi(6)}E_{\frac{5}{2}\Lambda_1}^{E_6}\right)\Big|_{\epsilon\to 0} \, . \tag{5.43}
$$

There is a similar term in $\widehat{\mathcal{E}}_{(0,1)}^{(6),\mathrm{ExFT}}$ that reads

$$
\widehat{\mathcal{E}}_{(0,1)}^{(6),\mathrm{ExFT}} \underset{R^5}{\sim} 40\xi(2)\left( R^{5+2\epsilon}\xi(1-2\epsilon)\xi(5+2\epsilon)E_{(\frac{5}{2}+\epsilon)\Lambda_1}^{E_6} + R^5 \frac{\xi(5)}{2\epsilon}E_{\frac{5}{2}\Lambda_1}^{E_6}\right)\Big|_{\epsilon\to 0} \tag{5.44}
$$

such that the unphysical derivative of the Eisenstein series $(\partial_s E_{s\Lambda_1}^{E_6})|_{s=\frac{5}{2}}$ drops out in the total coupling $\mathcal{E}_{(0,1)}^{(6)}$.

For $d = 2$ the combination (1.28) is not finite because there is a logarithmic divergence in supergravity. One obtains

$$
\begin{aligned}
\mathcal{I}_{2,\epsilon} &\sim R^{\frac{12+4\epsilon}{6}}\left(\frac{16\pi^2\xi(2\epsilon)^2}{(4-2\epsilon)(3+2\epsilon)}R^{-2+4\epsilon} + \frac{2\pi}{3}R^{2\epsilon-1}\xi(2\epsilon)\mathcal{E}_{(0,0),\epsilon}^{(1)} + \mathcal{O}(\epsilon^0)\right) \\
&\sim \frac{1}{12}\left(\mathcal{E}_{(0,0),\epsilon}^{(2)} + \tfrac{\pi}{6}\right)^2 + \mathcal{O}(\epsilon^0)
\end{aligned}
\tag{5.45}
$$

where

$$
\begin{aligned}
\mathcal{E}_{(0,0),\epsilon}^{(2)} &= 4\pi\xi(2\epsilon)E_{\epsilon\Lambda_1}^{SL(2)}E_{\epsilon\Lambda_2}^{SL(3)} = 4\pi\xi(2\epsilon) + \mathcal{E}_{(0,0)}^{(2)} + \mathcal{O}(\epsilon) \ , \\
\mathcal{E}_{(0,0),\epsilon}^{(1)} &= 4\pi\xi(2\epsilon-1)\left(\nu^{-\frac{3}{7}(1-2\epsilon)}E_{(\epsilon-\frac{1}{2})\Lambda_1}^{SL(2)} + \nu^{\frac{4}{7}(1-2\epsilon)}\right) = \mathcal{E}_{(0,0)}^{(1)} + \mathcal{O}(\epsilon) \ .
\end{aligned}
\tag{5.46}
$$

Taking into account the divergence coming from the supergravity amplitude and the $\mathcal{R}^4$ form-factor as in (5.37), one obtains instead

$$
\begin{aligned}
&\mathcal{I}_{2,\epsilon} + \frac{\pi}{3}\pi^{-\epsilon}\Gamma(\epsilon)\mu^{-2\epsilon}\,\mathcal{E}_{(0,0),\epsilon}^{(2)} + \frac{\pi^2}{3}\left(\pi^{-2\epsilon}\Gamma(\epsilon)^2 + \tfrac{1}{6}\pi^{-\epsilon}\Gamma(\epsilon) + \mathcal{O}(\epsilon^0)\right)\mu^{-4\epsilon} \\
&\sim \frac{49\pi^2}{27}\log(R)^2 - \frac{7\pi}{9}\log(R)\left(R\,\mathcal{E}_{(0,0)}^{(1)} + \tfrac{\pi}{3}\right) \\
&\quad + \frac{4\pi^2}{3}\log(2\pi\mu)^2 - \frac{2\pi}{3}\log(2\pi\mu)\left(R\,\mathcal{E}_{(0,0)}^{(1)} - \tfrac{14\pi}{3}\log(R) + \tfrac{\pi}{3}\right)
\end{aligned}
\tag{5.47}
$$

which is finite as $\epsilon \to 0$ and reproduces the expected logarithmic terms from [2, (B.49)].

For $d = 3$ one obtains the constant terms

$$
\begin{aligned}
\mathcal{I}_{3,\epsilon} &\sim R^{\frac{12+4\epsilon}{5}}\left(\frac{16\pi^2\xi(1+2\epsilon)^2}{(3-2\epsilon)(4+2\epsilon)}R^{4\epsilon} + \mathcal{I}_{2,\epsilon} + \frac{8\pi^2}{3}R^{2\epsilon}\xi(1+2\epsilon)\mathcal{E}_{(0,0),\epsilon}^{(2)} + \mathcal{O}(\epsilon^0)\right) \\
&\sim R^{\frac{12}{5}}\left[\frac{4\pi^2}{3}\log(R)^2 + \frac{2\pi}{3}\log(R)\,\mathcal{E}_{(0,0)}^{(2)} + \dots\right]
\end{aligned}
\tag{5.48}
$$

which reproduces the expected logarithmic terms from [2, (B.38)].

In summary, our renormalisation prescription leading ultimately to the renormalised coupling (1.31), reproduces correctly the expected expansion of the $\nabla^6\mathcal{R}^4$ coupling in the weak coupling and decompactification limits, including logarithmic terms in the string coupling and the radius $R$. This lends very strong support to the claim that (1.31) is the correct full coupling.

## 5.4 Generalisation to $E_8$

In three dimensions there exists a unique $\nabla^6\mathcal{R}^4$ type supersymmetry invariant [14]. Thus, the second term in (1.13) should be omitted, leading to

$$
\mathcal{E}_{(0,1)}^{(7)} = \widehat{\mathcal{E}}_{(0,1)}^{(7),\text{ExFT}} = 8\pi\int_{\mathcal{G}}\frac{d^3\Omega_2}{|\Omega_2|^{\frac{-1}{2}}}\left(|\Omega_2|^\epsilon\varphi_{\text{KZ}}^{\text{tr}} - \frac{\pi}{36}\frac{E_{-3\Lambda_1}^{SL(2)}(\tau)}{V^{1+2\epsilon}} + \frac{\pi}{36}\frac{E_{-(3+2\epsilon)\Lambda_1}^{SL(2)}(\tau)}{V}\right)\theta_{\Lambda_8}^{E_8}(\phi,\Omega_2)\Big|_{\epsilon^0} \ ,
\tag{5.49}
$$

consistently with the sum of the exceptional field theory amplitude contributions up to three loops [30], and the 1/4-BPS states contribution discussed in this section.

The analysis of Section 4.2 can be applied to the lattice $M^{E_8}_{\Lambda_8}$ in the adjoint representation of $E_8$. As we explain in Appendix E, the computation is very similar for the first five layers, but there are two additional layers of charges. We are able to compute the constant term and the generic Fourier coefficients for the sixth layer of charges. Using the Langland constant term formula for the Eisenstein series $\mathcal{I}^{E_8}_{\Lambda_8}(E^{SL(2)}_{s\Lambda_1}, 5 + 2\epsilon)$, we argue that the last layer does not contribute, so that the constant terms that we are able to compute do exhaust the non-vanishing contributions. Despite the fact that the three contributions in (5.49) are individually finite in the limit $\epsilon \to 0$, $\mathcal{I}^{E_8}_{\Lambda_8}(E^{SL(2)}_{-(3+\delta)\Lambda_1}, 5 + 2\epsilon)$ is not analytic at $(\epsilon, \delta) = (0,0)$ in $\mathbb{C}^2$, because its limit includes a factor of $\frac{\delta + 2\epsilon}{\delta - 2\epsilon}$. Therefore the renormalisation prescription (1.31) gives a finite contribution that must be taken into account to reproduce the correct coupling. One obtains eventually

$$\mathcal{E}^{(7)}_{(0,1)} \sim R^{12}\left(\mathcal{E}^{(6)}_{(0,1)} + \frac{5}{\pi}\log R\, \mathcal{E}^{(6)}_{(1,0)} + \frac{\zeta(5)}{2\pi}R^4 \mathcal{E}^{(6)}_{(0,0)} - \frac{9\zeta(5)^2}{8\pi^2}R^8 + \frac{5\zeta(11)}{12\pi}R^{10}\right), \qquad (5.50)$$

which reproduces the expected result from [2, (B.70)]. The first three terms come from the second layer of charges with

$$\mathcal{I}^{(2)}_{7,\epsilon}\left[A(\tau) - \tfrac{\pi}{6}E^{SL(2)}_{-3\Lambda_1}(\tau)\right] + \mathcal{I}^{(2)}_{7,0}\left[\tfrac{\pi}{6}E^{SL(2)}_{-(3+2\epsilon)\Lambda_1}\right] \qquad (5.51)$$

$$\sim R^{12}\left(R^{4\epsilon}\mathcal{I}^{E_7}_{\Lambda_7}\left(A(\tau) - \tfrac{\pi}{6}E^{SL(2)}_{-3\Lambda_1}(\tau), 5 + 2\epsilon\right) + \mathcal{I}^{E_7}_{\Lambda_7}\left(\tfrac{\pi}{6}E^{SL(2)}_{-(3+2\epsilon)\Lambda_1}(\tau), 5\right)\right)$$

$$+ \frac{4\pi^2}{9}R^{12+4\epsilon}\left(6\xi(5+2\epsilon)\xi(4+2\epsilon)R^{4+2\epsilon}\,E^{E_7}_{(2+\epsilon)\Lambda_7} + \xi(1+2\epsilon)\xi(8+2\epsilon)R^{2\epsilon}\,E^{E_7}_{(4+2\epsilon)\Lambda_7}\right.$$

$$\left. - \frac{\xi(8)}{\xi(7)}\xi(1+2\epsilon)\xi(8+2\epsilon)R^{7+2\epsilon}\,E^{E_7}_{(\frac{1}{2}+\epsilon)\Lambda_7} - \xi(1+2\epsilon)\xi(8+2\epsilon)R^{2\epsilon}\,E^{E_7}_{(4+2\epsilon)\Lambda_7}\right)$$

$$+ \frac{4\pi^2}{9}\xi(1-2\epsilon)\xi(8+2\epsilon)R^{12}\left(\frac{\xi(8)}{\xi(7)}R^{7+2\epsilon}\,E^{E_7}_{(\frac{1}{2}-\epsilon)\Lambda_7} + R^{-2\epsilon}\,E^{E_7}_{(4+2\epsilon)\Lambda_7}\right)$$

$$\sim R^{12}\left(R^{4\epsilon}\mathcal{I}^{E_7}_{\Lambda_7}\left(A(\tau) - \tfrac{\pi}{6}E^{SL(2)}_{-3\Lambda_1}(\tau), 5 + 2\epsilon\right) + \mathcal{I}^{E_7}_{\Lambda_7}\left(\tfrac{\pi}{6}E^{SL(2)}_{-(3+2\epsilon)\Lambda_1}(\tau), 5\right)\right.$$

$$\left. + \frac{4\pi^2\xi(1-2\epsilon)\xi(8+2\epsilon)}{9}R^{-2\epsilon}\,E^{E_7}_{(4+\epsilon)\Lambda_7} + \frac{8\pi^2}{3}\xi(5)\xi(4+2\epsilon)R^4\,E^{E_7}_{(2+\epsilon)\Lambda_7}\right).$$

To compute the logarithmic term one uses the property that the only divergent terms are

$$\mathcal{I}^{E_7}_{\Lambda_7}\left(\tfrac{\pi}{6}E^{SL(2)}_{-(3+2\epsilon)\Lambda_1}(\tau), 5\right) + \frac{4\pi^2\xi(1-2\epsilon)\xi(8+2\epsilon)}{9}R^{-2\epsilon}\,E^{E_7}_{(4+\epsilon)\Lambda_7}$$

$$= 40\xi(4)\left(\frac{\xi(7+2\epsilon)\xi(8+4\epsilon)\xi(9+2\epsilon)\xi(12+2\epsilon)}{\xi(4+2\epsilon)\xi(7+4\epsilon)}E^{E_7}_{(6+\epsilon)\Lambda_1} + \xi(1-2\epsilon)\xi(8+2\epsilon)R^{-2\epsilon}\,E^{E_7}_{(4+\epsilon)\Lambda_7}\right)$$

$$= 40\xi(8)\xi(9)\xi(12)\widehat{E}^{E_7}_{6\Lambda_1} + 40\xi(2)\xi(6)\xi(10)\widehat{E}^{E_7}_{5\Lambda_7} + \frac{5}{\pi}\log R\,\zeta(5)E^{E_7}_{5\Lambda_1} + \mathcal{O}(\epsilon), \qquad (5.52)$$

consistently with (5.50). The last constant term in $40\xi(2)\xi(6)\xi(d+4)\,R^{d+3}$ that comes from the function $\widehat{\mathcal{F}}^{(d)}_{(0,1)}$ for $d \leq 6$ now originates from the sixth layer of $\widehat{\mathcal{E}}^{(7),\mathrm{ExFT}}_{(0,1)}$ displayed in (E.33).

This analysis also lends support to our renormalised coupling (1.32) in the case $d = 7$.

## Acknowledgements

We are grateful to Charles Cosnier-Horeau and Rodolfo Russo for helpful discussions at early and very early stages of this project, respectively. We wish to thank Daniele Dorigoni, Steve Kudla and Stephen D. Miller for useful discussions related to parts of this paper. The authors gratefully acknowledge the Banff International Research Station during the 2017 workshop "Automorphic Forms, Mock Modular Forms and String Theory" and the Simons Center for Geometry and Physics at Stony Brook during the 2019 programme "Automorphic Structures in String Theory", for providing support and a stimulating atmosphere during parts of this project. GB thanks the Albert Einstein Institute, Potsdam, for hospitality during parts of this project. The work of GB was partly supported by the ANR grant Black-dS-String (ANR16-CE31-0004).

# A  Poincaré series representation of $\varphi_{\mathrm{KZ}}$

In this section, we provide evidence for the relation (1.21) expressing the Kawazumi–Zhang invariant $\varphi_{\mathrm{KZ}}$ as a Poincaré series seeded by its tropical limit. We first recall how both sides can be expressed as theta liftings for lattices of signature $(3,2)$ and $(2,1)$, following [48]. As a result, (1.21) would follow from a similar property (A.16) for Siegel–Narain theta series. We give evidence that (A.16) holds, by integrating both sides against a vector-valued Eisenstein series of weight $-1/2$, and invoking Langlands' functional relation for generic Eisenstein series of $SO(3,2) = Sp(4,\mathbb{R})/\mathbb{Z}_2$. Additional evidence for (1.21) comes from the analysis of constant terms in Sections 3 and 4 and in Appendix C.

## A.1  Theta series representation for real-analytic Siegel modular forms

The Siegel modular group $Sp(4,\mathbb{Z})$ is isomorphic to the automorphism group of the lattice $\mathbb{Z}^5$ with quadratic form $2(m_1 n^1 + m_2 n^2) + \frac{1}{2} b^2$ of signature $(3,2)$. Using this observation, we can obtain Siegel modular functions of $Sp(4,\mathbb{Z})$ from theta liftings of vector-valued modular forms under $SL(2,\mathbb{Z})$, generalising earlier constructions of the log-norm of the Igusa cusp form $\Psi_{10}$ [71] and of the genus-two Kawazumi–Zhang invariant [48]. For this purpose, we introduce the lattice partition functions for $i = 0,1$ (setting $z = x + \mathrm{i}y \in \mathbb{H}$ and $q = e^{2\pi \mathrm{i}z}$)

$$\Gamma_{3,2}^{(i)}(\Omega; z) = y \sum_{\substack{m_1, m_2, n^1, n^2 \in \mathbb{Z} \\ b \in 2\mathbb{Z}+i}} q^{\frac{1}{4}\vec{p}_L^2} \, \bar{q}^{\frac{1}{4}|p_R|^2} \,, \quad i = 0,1 \tag{A.1}$$

where

$$\begin{aligned}
p_{\mathrm{R}}^1 + \mathrm{i}p_{\mathrm{R}}^2 &= \frac{m_2 - \rho\, m_1 + \sigma n^1 + (\rho\sigma - v^2)\, n^2 - b\, v}{\sqrt{\rho_2\, \sigma_2 - v_2^2}}, \\
p_{\mathrm{L}}^1 + \mathrm{i}p_{\mathrm{L}}^2 &= \frac{m_2 - \rho\, m_1 + \bar{\sigma} n^1 + (\rho\bar{\sigma} - v^2)\, n^2 - b\, v + \frac{\mathrm{i}}{2}v_2^2(n^1 + \rho n^2)}{\sqrt{\rho_2\, \sigma_2 - v_2^2}} \\
p_{\mathrm{L}}^3 &= b + \mathrm{i}\frac{(n^1 + n^2\bar{\rho})v - (n^1 + n^2\rho)\bar{v}}{2\rho_2} \,,
\end{aligned} \tag{A.2}$$

such that $\vec{p}_L^2 - |p_R|^2 = 4m_i n^i + b^2$. Here,

$$\Omega = \begin{pmatrix} \rho & v \\ v & \sigma \end{pmatrix} \tag{A.3}$$

lives in the Siegel upper-half plane $\mathcal{H}_2$, and $(p_R, \bar{p}_R)$ and $\vec{p}_L = (p_L^1, p_L^2, p_L^3)$ are the projections of the lattice vector $\mathcal{Q} = (m_1, m_2, b, n^1, n^2)$ on the positive 2-plane and its orthogonal complement.

Given a weak Jacobi form $h(z, v)$ of weight $-1/2$ and index 1, we can take its theta series decomposition [72]

$$h(z, v) = h_0(z)\, \theta_3(2z, 2v) + h_1(z)\, \theta_2(2z, 2v) \tag{A.4}$$

and consider the modular integral

$$\mathcal{I}_{3,2}[h] = \int_{\mathcal{F}_1} \frac{\mathrm{d}x\mathrm{d}y}{y^2}\, \left[ \Gamma_{3,2}^{(0)}(\Omega; z)\, h_0(z) + \Gamma_{3,2}^{(1)}(\Omega; z)\, h_1(z) \right] \tag{A.5}$$

over the standard fundamental domain $\mathcal{F}_1 = \{\tau \in \mathcal{H}_1, |\tau| > 1, |\tau_1| < 1/2\}$ for $PSL(2, \mathbb{Z})$ (which consists of two copies of the fundamental domain $\mathcal{F}$ for $PGL(2, \mathbb{Z})$ defined below (1.14)). The integrand is invariant under $SL(2, \mathbb{Z}) \times Sp(4, \mathbb{Z})$, so the integral produces a Siegel modular form, possibly with singularities on rational quadratic divisors when $h$ has poles at the cusp. In the limit $\Omega \to i\infty$ (corresponding to the maximal non-separating degeneration in the language of genus-two Riemann surfaces, or the limit where one circle decompactifies in the language of torus compactifications), $\Gamma_{3,2}^{(i)}(\Omega; z)$ factorises into $\Gamma_{1,1}(V; z) \times \Gamma_{2,1}^{(i)}(\tau, z)$, where

$$\begin{aligned} \Gamma_{1,1}(V; z) &= V^{-1} \sum_{(p,q) \in \mathbb{Z}^2} e^{-\pi|p+qz|^2/(yV^2)} \\ \Gamma_{2,1}^{(i)}(\tau; z) &= y \sum_{a,c \in \mathbb{Z}, b \in 2\mathbb{Z}+i} q^{\frac{1}{4}|p_L|^2}\, \bar{q}^{\frac{1}{4}p_R^2}\,, \quad i = 0, 1\,. \end{aligned} \tag{A.6}$$

Here $V = 1/|\Omega_2|^{1/2}$ is the inverse radius of the large circle, $\tau$ is defined as in (1.14) and

$$p_R = \frac{a|\tau|^2 + b\tau_1 + c}{\tau_2}\,, \qquad p_L^1 + ip_L^2 = \frac{a\tau^2 + b\tau + c}{\tau_2} \tag{A.7}$$

such that $|p_L|^2 - p_R^2 = b^2 - 4ac$. In the decompactifying limit $V \to 0$, the dominant term in the modular integral (A.5) comes from the zero orbit $(p, q) = (0, 0)$, so $\mathcal{I}_{3,2}[h] \sim V^{-1} \mathcal{I}_{2,1}[h]$ where

$$\mathcal{I}_{2,1}[h] = \int_{\mathcal{F}_1} \frac{\mathrm{d}x\mathrm{d}y}{y^2}\, \left[ \Gamma_{2,1}^{(0)}(\tau; z)\, h_0(z) + \Gamma_{2,1}^{(1)}(\tau; z)\, h_1(z) \right]\,. \tag{A.8}$$

Thus, the leading tropical limit $\mathcal{I}_{2,1}$ of the Siegel modular form $\mathcal{I}_{3,2}$ is itself a theta lift. Sub-leading terms come from the terms with $(p, q) \neq (0, 0)$. For these terms, the integration domain can be unfolded to the strip $\mathbb{R}^+ \times [-\frac{1}{2}, \frac{1}{2}]$ at the expense of restricting to $q = 0$. The integral over $u$ picks up contributions from zero or negative Fourier modes of $h_i$, leading to powerlike or exponentially suppressed terms in $1/V$, respectively. The minimal non-separating degeneration limit $t \to \infty$ with $t = \tau_2/V$ keeping $\rho_2 = 1/(V\tau_2)$ fixed instead corresponds to the limit where the volume of $T^2$ becomes infinite, and can be extracted using similar orbit methods.

The Kawazumi–Zhang invariant $\varphi_{\mathrm{KZ}}$ is obtained by choosing [48]

$$(h_0, h_1) = -\frac{1}{2} D_{-5/2}(\tilde{h}_0, \tilde{h}_1) \tag{A.9}$$

where

$$\tilde{h}(z, v) = \tilde{h}_0(z)\,\theta_3(2\rho, 2v) + \tilde{h}_1(z)\,\theta_2(2z, 2v) = \frac{\theta_1^2(z, v)}{\eta^6(z)} \tag{A.10}$$

and $D_w = \frac{\mathrm{i}}{\pi}(\partial_z - \frac{\mathrm{i}w}{2\tau_2})$ is the Maaß raising operator, mapping modular forms of weight $w$ to modular forms of weight $w + 2$. Using the theta lift representation, it is straightforward to obtain the asymptotics of $\varphi_{\mathrm{KZ}}$ in the tropical limit $\Omega \to \mathrm{i}\infty$, and indeed the complete Fourier expansion,

$$
\begin{aligned}
\varphi_{\mathrm{KZ}} =\ & \frac{\pi}{6} |\Omega_2|^{1/2} A(\tau) + \frac{5\zeta(3)}{4\pi^2} |\Omega_2|^{-1} \\
& + 2 \sum_{M \in \mathcal{S}_+} \left( |M| + \frac{5}{16\pi^2 |\Omega_2|} (1 + 2\pi \mathrm{tr}[M\Omega_2]) \right) \sum_{k|M} k^{-3} \tilde{c}\left(\tfrac{4|M|}{k^2}\right) (e^{2\pi \mathrm{i} \mathrm{tr} M\Omega} + e^{-2\pi \mathrm{i} \mathrm{tr} M\bar{\Omega}})
\end{aligned} \tag{A.11}
$$

where $\mathcal{S}_+$ is defined below (3.57), $A(\tau)$ is the modular local function defined by (1.15) on the fundamental domain $\mathcal{F}$, and $\tilde{c}(n)$ are the Fourier coefficients of

$$-\frac{\theta_4(2\tau)}{\eta(4\tau)^6} = \sum_{n \geq -1} \tilde{c}(n) q^n \ . \tag{A.12}$$

In the minimal non-separating degeneration $t \to \infty$, one has instead

$$\varphi_{\mathrm{KZ}} = \frac{\pi}{6} t + \varphi_0 + \frac{\varphi_1}{t} + \mathcal{O}(e^{-\pi t}), \tag{A.13}$$

where

$$
\begin{aligned}
\varphi_0 =\ & \pi \rho_2 u_2^2 - \log\left| \frac{\vartheta_1(v, \rho)}{\eta(\rho)} \right| = \frac{1}{2} \mathcal{D}_{1,1}(\rho; v) \ , \\
\varphi_1 =\ & \frac{5}{16\pi^2 \rho_2}\, \mathcal{D}_{2,2}(\rho; v) + \frac{\pi}{36}\, E_{2\Lambda_1}^{SL(2)}(\rho) \ .
\end{aligned} \tag{A.14}
$$

Here, $\mathcal{D}_{a,b}$ is the Kronecker–Eisenstein series

$$\mathcal{D}_{a,b}(v, \rho) = \frac{(2\mathrm{i}\rho_2)^{a+b-1}}{2\pi \mathrm{i}} \sideset{}{'}\sum_{(m,n) \in \mathbb{Z}^2} \frac{e^{2\pi \mathrm{i}(nu_2 + mu_1)}}{(m\rho + n)^a (m\bar{\rho} + n)^b} \tag{A.15}$$

where $v = u_1 + u_2 \rho$. It may be worth noting that $\mathcal{D}_{1,1}$ coincides with the scalar propagator on the torus.

## A.2 Poincaré series from theta lifting

The identity (1.21) expressing the Kawazumi–Zhang invariant as a Poincaré series seeded by its tropical limit would follow from a similar property for the lattice theta series,

$$\Gamma_{3,2}^{(i)} = \lim_{\epsilon \to 0} \left[ \sum_{\gamma \in (GL(2,\mathbb{Z}) \ltimes \mathbb{Z}^3) \backslash Sp(4,\mathbb{Z})} \left( |\Omega_2|^{\frac{1}{2} + \epsilon} \Gamma_{2,1}^{(i)} \right) \Big|_\gamma \right], \tag{A.16}$$

where the limit $\epsilon \to 0$ should be taken after analytic continuation away from the region where the sum converges. While we do not know how to prove this relation, we shall test its consequence when integrating against the Eisenstein series $E(s, w; z)$ of weight $w = -\frac{1}{2}$ under the congruence subgroup $\Gamma_0(4) \subset SL(2, \mathbb{Z})$. The Eisenstein series is defined by

$$E(s, w; z) = \sum_{\gamma \in \Gamma_\infty \backslash \Gamma_0(4)} y^{s - \frac{w}{2}} \Big|_\gamma^w, \tag{A.17}$$

where the 'slash' notation corresponds to the action of $\gamma \in \Gamma_0(4)$ on the variable $z$ with an additional factor of automorphy $(cz + d)^{-w}$. Decomposing as in (A.4)

$$E(s, w; z) = E_0(s, w; 4z) + E_1(s, w; 4z) \tag{A.18}$$

and computing the integral (A.5) by unfolding, we get

$$\mathcal{I}_{3,2}\left[E(s - \tfrac{1}{4}, -\tfrac{1}{2})\right] = \frac{\Gamma(s)}{\pi^s} \sideset{}{'}\sum_{\substack{Q = (m_1, m_2, n^1, n^2, b) \in \mathbb{Z}^5 \\ b^2 + 4m_1 n^1 + 4m_2 n^2 = 0}} |p_R(Q)|^{-2s} = \xi(2s)\, E_{s\Lambda_2}^{Sp(4)} \tag{A.19}$$

which we recognise as the Siegel–Eisenstein series for $Sp(4, \mathbb{Z})$. Indeed, using the constant terms (for $w \in \mathbb{Z} + \frac{1}{2}$)

$$E(s, w; z) = y^{s - \frac{w}{2}} + \frac{4^{1-2s}(-1)^{\lfloor \frac{w}{2} - \frac{1}{4} \rfloor} \pi \Gamma(2s - 1)}{\Gamma(s + \frac{w}{2})\, \Gamma(s - \frac{w}{2})} \frac{\zeta(4s - 2)}{\zeta(4s - 1)} y^{1 - s - \frac{w}{2}} + \mathcal{O}(e^{-\pi y}) \tag{A.20}$$

and the orbit method, we find the constant terms

$$V^{-1} \mathcal{I}_{2,1}\left[E(s - \tfrac{1}{4}, -\tfrac{1}{2})\right] + \xi(2s)\, \xi(4s - 2) V^{-2s} + \frac{\xi(2s - 2)\, \xi(4s - 3)}{\xi(4s - 2)} V^{2s - 3}. \tag{A.21}$$

In the first term, the theta lift can be computed by unfolding,

$$\mathcal{I}_{2,1}\left[E(s - \tfrac{1}{4}, -\tfrac{1}{2})\right] = \frac{\Gamma(s - \frac{1}{2})}{\pi^{s - \frac{1}{2}}} \sideset{}{'}\sum_{\substack{(a,b,c) \in \mathbb{Z}^3 \\ b^2 - 4ac = 0}} p_R^{-(2s-1)}$$

$$= \frac{\Gamma(s - \frac{1}{2})}{\pi^{s - \frac{1}{2}}} \left[ \sideset{}{'}\sum_{\substack{(a,b) \in \mathbb{Z}^2 \\ a \neq 0,\, 4a | b^2}} \left[\frac{1}{a\tau_2} \left| a\tau + \frac{b}{2} \right|^2 \right]^{-(2s-1)} + \sum_{c \neq 0} \left(\frac{c}{\tau_2}\right)^{-(2s-1)} \right]$$

$$= V^{-1} \xi(2s - 1)\, E_{(2s-1)\Lambda_1}^{SL(2)}(\tau), \tag{A.22}$$

where in the last line, we solved the constraint $4a | b^2$ by setting $(a, b) = kp(p, q)$ with $\gcd(p, q) = 1$ and $k \geq 1$. In total, (A.21) reproduces the known constant terms of the Siegel–Eisenstein series $E_{s\Lambda_2}^{Sp(4)}$ [40, (3.13)].

The conjectural property (A.16) now predicts that

$$E_{s\Lambda_2}^{Sp(4)} = \frac{\xi(2s - 1)}{\xi(2s)} \lim_{\epsilon \to 0} \left[ \sum_{\gamma \in (GL(2,\mathbb{Z}) \ltimes \mathbb{Z}^3) \backslash Sp(4,\mathbb{Z})} \left( |\Omega_2|^{\frac{1}{2} + \epsilon}\, E_{(2s-1)\Lambda_1}^{SL(2)}(\tau) \right) \Big|_\gamma \right] \tag{A.23}$$

Expressing $E^{SL(2)}_{(2s-1)\Lambda_1}(\tau)$ as a sum over cosets, this is tantamount to

$$E^{Sp(4)}_{s\Lambda_2} \stackrel{?}{=} \frac{\xi(2s-1)}{\xi(2s)} \lim_{\epsilon \to 0} \left[ \sum_{\gamma \in B \cap Sp(4,\mathbb{Z}) \backslash Sp(4,\mathbb{Z})} \left( |\Omega_2|^{\frac{1}{2}+\epsilon} \tau_2^{2s-1} \right) \Big|_\gamma \right] \tag{A.24}$$

where $B$ is the Borel subgroup of $Sp(4,\mathbb{Z})$. The righthand side is proportional to the generic Langlands–Eisenstein series

$$E^{Sp(4)}_{(s_2-s_1)\Lambda_1+s_1\Lambda_2} = \sum_{\gamma \in B \cap Sp(4,\mathbb{Z}) \backslash Sp(4,\mathbb{Z})} \rho_2^{s_1} t^{s_2} \Big|_\gamma = \sum_{\gamma \in B \cap Sp(4,\mathbb{Z}) \backslash Sp(4,\mathbb{Z})} |\Omega_2|^{\frac{1}{2}(s_1+s_2)} \tau_2^{s_2-s_1} \Big|_\gamma \tag{A.25}$$

with $(s_1, s_2) = (1 - s + \epsilon, s + \epsilon)$. Using the functional equation satisfied by (A.25) under $(s_1, s_2) \mapsto (1 - s_1, s_2)$, and recalling (A.3), we find that the right-hand side of (A.24) is, in the limit $\epsilon \to 0$, equal to

$$\sum_{\gamma \in B \cap Sp(4,\mathbb{Z}) \backslash Sp(4,\mathbb{Z})} \left( |\Omega_2|^s \right) \Big|_\gamma \tag{A.26}$$

which is the standard definition of the Siegel–Eisenstein series $E^{Sp(4)}_{s\Lambda_2}$. This provides a strong consistency check on the conjecture (A.16), and therefore on its consequence (1.21).

## A.3  Poincaré series from 1/2-BPS state sums

If $(h_0, h_1)$ or equivalently $h(z) = h_0(4z) + h_1(4z)$ can be represented as a Poincaré series for $SL(2, \mathbb{Z})$, then we can evaluate the either of the integrals $\mathcal{I}_{3,2}[h]$ or $\mathcal{I}_{2,1}[h]$ by the unfolding method [73], and obtain a sum over lattice vectors of fixed norm, which can be reinterpreted as a Poincaré series for $Spin(3, 2, \mathbb{Z}) = Sp(4, \mathbb{Z})$ or for $O(2, 1, \mathbb{Z}) = PGL(2, \mathbb{Z})$. Let us assume that $h$ is proportional to the Niebur–Poincaré series $\mathcal{F}_4(s, \kappa, w; z)$. The integral then becomes

$$\mathcal{I}_{3,2}(s, \kappa; \Omega) = \Gamma(s + \tfrac{1}{4}) \sum_{\substack{(m_i, b, n^i) \in \mathbb{Z}^5 \\ 4m_i n^i + b^2 = \kappa}} (p_R^2/\kappa)^{-\frac{1}{4}-s} {}_2F_1\left(s + \tfrac{1}{4}, s + \tfrac{1}{4}; 2s; -\kappa/p_R^2\right) . \tag{A.27}$$

where

$$p_R = \frac{m_2 - \rho\, m_1 + \sigma n^1 + (\rho\sigma - v^2)\, n^2 - b\, v}{\sqrt{\rho_2\, \sigma_2 - v_2^2}} . \tag{A.28}$$

The summand is (away from the singular locus where $p_R = 0$ for some vector $\mathcal{Q}$) an eigenmode of $\Delta_{Sp(4)}$ with eigenvalue $\frac{1}{8}(4s + 1)(4s - 5)$. For $\kappa = 1$, all vectors are images of the vector $\mathcal{Q} = (m_i, b, n^i) = (0, 1, 0)$, whose stabiliser is $SL(2, \mathbb{Z})_\rho \times SL(2, \mathbb{Z})_\sigma$. Therefore, we can interpret (A.27) as

$$\mathcal{I}_{3,2}(s, \kappa = 1; \Omega) = \Gamma(s + \tfrac{1}{4}) \sum_{\gamma \in [SL(2,\mathbb{Z}) \times SL(2,\mathbb{Z})] \backslash Sp(4,\mathbb{Z})} M_s\left(\frac{|v|}{\sqrt{\rho_2 \sigma_2 - v_2^2}}\right) \Big|_\gamma \tag{A.29}$$

where

$$\begin{aligned} M_s(u) =&\, u^{-\frac{1}{2}-2s} {}_2F_1\left(s + \tfrac{1}{4}, s + \tfrac{1}{4}; 2s; -\frac{1}{u^2}\right) \\ =&\, (1 + u^2)^{-(s+\frac{1}{4})} {}_2F_1\left(s + \tfrac{1}{4}, s - \tfrac{1}{4}; 2s; -\frac{1}{1+u^2}\right) , \end{aligned} \tag{A.30}$$

where in the second equality we used Pfaff's identity $_2F_1(a, b, c; z) = (1-z)^{-a}{}_2F_1(a, c-b, c; \frac{z}{z-1})$. Note that $M_s(u)$ satisfies

$$\frac{1}{2}(1 + u^2)\partial_u^2 M_s(u) + \frac{1 + 4y^2}{2y}\partial_u M_s(u) = \left[2s(s-1) - \tfrac{5}{8}\right]M_s(u) \tag{A.31}$$

which ensures that the Poincaré series (A.29) is an eigenmode of $\Delta_{Sp(4)}$ with eigenvalue $2s(s-1) - \frac{5}{8}$. Similarly, we can write the tropical limit as a Poincaré series:

$$
\begin{aligned}
\mathcal{I}_{2,1}(s, \kappa; \tau) &= \Gamma(s - \tfrac{1}{4}) \sum_{\substack{(a,b,c)\in\mathbb{Z}^3 \\ b^2 - 4ac = \kappa}} (p_R^2/\kappa)^{\frac{1}{4} - s}{}_2F_1\left(s + \tfrac{1}{4}, s - \tfrac{1}{4}; 2s; -\kappa/p_R^2\right) \\
&= \Gamma(s - \tfrac{1}{4}) \sum_{\gamma \in SO(2)\backslash SO(2,1,\mathbb{Z})} m_s(\tau_1/\tau_2)|_\gamma
\end{aligned}
\tag{A.32}
$$

where $p_R = [a|\tau|^2 + b\tau_1 + c]/\tau_2$ and

$$m_s(u) = u^{-2s + \frac{1}{2}}{}_2F_1\left(s + \tfrac{1}{4}, s - \tfrac{1}{4}; 2s; -\frac{1}{u^2}\right) \tag{A.33}$$

Choosing $s = \frac{9}{4}, \kappa = 1$ and adjusting the normalisation, the Niebur–Poincaré series reduces to the weak holomorphic modular form (A.10) appearing in the theta lift representation of the Kawazumi–Zhang invariant or its tropical limit,

$$\tilde{h}(z) = \tilde{h}_0(4z) + \tilde{h}_1(4z) = -\frac{1}{\Gamma(\frac{9}{2})}\mathcal{F}_4(\tfrac{9}{4}, 1, -\tfrac{5}{2}; z)\ . \tag{A.34}$$

Using the identity

$$D_w \mathcal{F}_4(s, \kappa, w; z) = \kappa(2s + w)\mathcal{F}_4(s, \kappa, w + 2; z) \tag{A.35}$$

and setting $s = \frac{9}{4}$ in the previous formulae, we get

$$
\begin{aligned}
\varphi_{\mathrm{KZ}}(\Omega) &= \frac{1}{4\Gamma(9/2)} \int_{\mathcal{F}_1} \frac{\mathrm{d}x\,\mathrm{d}y}{y^2}\left[\Gamma_{3,2}^{(0)}(\Omega; z)\,\mathcal{F}_0(\tfrac{9}{4}, \tfrac{1}{4}, -\tfrac{5}{2}; z) + \Gamma_{3,2}^{(1)}(\Omega; z)\,\mathcal{F}_\infty(\tfrac{9}{4}, \tfrac{1}{4}, -\tfrac{5}{2}; z)\right] \\
&= \frac{1}{4\Gamma(9/2)}\mathcal{I}_{3,2}(9/4, 1/4; \Omega) \\
&= \frac{\Gamma(5/2)}{4\Gamma(9/2)} \sum_{\substack{(m_i, b, n^i)\in\mathbb{Z}^5 \\ 4m_i n^i + b^2 = 1}} M(|p_R|) = \frac{1}{35} \sum_{\gamma \in O(3,2,\mathbb{Z})/O(2,2,\mathbb{Z})} M\left(\frac{v}{\sqrt{\rho_2\sigma_2 - v_2^2}}\right)\Bigg|_\gamma
\end{aligned}
\tag{A.36}
$$

where

$$M(u) = u^{-5}{}_2F_1\left(\tfrac{5}{2}, \tfrac{5}{2}; \tfrac{9}{2}; -1/u^2\right) = \frac{35}{12}\left[3(2 + 5u^2)\mathrm{arcsinh}(1/u) - \frac{11 + 15u^2}{\sqrt{1 + u^2}}\right] \tag{A.37}$$

Similarly, for the tropical limit $A = \frac{6}{\pi}\varphi_{\text{KZ}}^{\text{tr}}$, we get

$$
\begin{aligned}
A(\tau) &= -\frac{3}{\pi}\int_{\mathcal{F}_1}\frac{\mathrm{d}x\mathrm{d}y}{y^2}\left[\Gamma_{2,1}^{(0)}(\tau;z)\,D_z\tilde{h}_0(z) + \Gamma_{2,1}^{(1)}(\tau;z)\,D_z\tilde{h}_1(z)\right] \\
&= -\frac{3}{\pi}\int_{\mathcal{F}_{\Gamma_0(4)}}\frac{\mathrm{d}x\mathrm{d}y}{y^2}\,\Gamma_{2,1}^{(0)}(\tau;4z)\,D_z\tilde{h}(z) \\
&= \frac{3}{2\pi^{1/2}\Gamma(\frac{9}{2})}\sum_{\substack{(a,b,c)\in\mathbb{Z}^3\\ b^2-4ac=1}} m(p_R) = \frac{8}{35\pi}\sum_{\gamma\in O(2,1,\mathbb{Z})/O(1,1,\mathbb{Z})} m\left(\frac{\tau_1}{\tau_2}\right)\Big|_\gamma
\end{aligned}
\tag{A.38}
$$

where

$$
m(u) = u^{-4}\,{}_2F_1\left(\tfrac{5}{2},2;\tfrac{9}{2};-1/u^2\right) = \frac{35}{12}\left(15u^2 + 4 - 3\left(3 + 5u^2\right)u\,\mathrm{arccot}(u)\right)\;.
\tag{A.39}
$$

Note that $m(u)$ is a bounded, continuous, even function of $u\in\mathbb{R}$, non-differentiable at $u = 0$, and decays as $1/|u|^4$ for $|u|\to\infty$. It is annihilated by the differential operator $\partial_u(1+u^2)\partial_u - 12$, which ensures that the Poincaré series (A.38) is annihilated by $\Delta_\tau - 12$ away from the locus $\tau_1 = 0$ and its images under $GL(2,\mathbb{Z})$.

# B  Integrating $A(\tau)$ against single and double Eisenstein series

In this appendix, we compute modular integrals of the local modular form $A(\tau)$ defined in (1.15), which we copy for convenience,

$$
A(\tau) = \frac{|\tau|^2 - \tau_1 + 1}{\tau_2} + \frac{5\tau_1(\tau_1 - 1)(|\tau|^2 - \tau_1)}{\tau_2^3}\;.
\tag{B.1}
$$

multiplied by either a standard non-holomorphic Eisenstein series $E_{s\Lambda_1}^{SL(2)}(\tau)$, or a 'double Eisenstein series' defined in (2.43), over the fundamental domain $\mathcal{F}$ for $PGL(2,\mathbb{Z})$ defined below (1.14). These results are used in the computation of the weak-coupling expansion in Section 3.2.

## B.1  Against a single Eisenstein series

Here we establish the formula (3.17), which we recall for convenience,

$$
\text{R.N.}\int_{\mathcal{F}}\frac{\mathrm{d}\tau_1\mathrm{d}\tau_2}{\tau_2^2}A(\tau)\,E_{s\Lambda_1}^{SL(2)}(\tau) = \frac{3\,[\xi(s)]^2}{[12 - s(s-1)]\xi(2s)}\;.
\tag{B.2}
$$

It will be convenient to unfold the integral to the domain $\mathcal{F}' = \{|\tau - \frac{1}{2}| > \frac{1}{2}, 0 < \tau_1 < 1\}$ which consists of the 6 images of $\mathcal{F}$ under the permutation group $S_3 \subset PGL(2,\mathbb{Z})$. Inside this domain, the two factors in the integrand are eigenmodes of the Laplacian [31, (3.12)],

$$
[\Delta_\tau - s(s-1)]\,E_{s\Lambda_1}^{SL(2)} = 0\;,
\tag{B.3}
$$

$$
(\Delta_\tau - 12)A = -12\tau_2\,\delta(\tau_1) - 12\tau_2\,\delta(1-\tau_1) - \frac{12\tau_2}{|\tau|^2}\delta\left(\frac{|\tau|^2 - \tau_1}{|\tau|^2}\right)
$$

We define the truncated fundamental domain $\mathcal{F}'(L)$ by removing the region $\tau_2 > L$ and its images under $S_3$. To avoid dealing with the delta functions, we regulate $\mathcal{F}'(L)$ by requiring $\delta < \tau_1 < 1 - \delta, |\tau - \frac{1}{2}| > \frac{1}{2} + \delta$ and we let $\delta \to 0$ at the end. Thus,

$$[s(s-1) - 12] \int_{\mathcal{F}'(\delta,L)} \frac{\mathrm{d}\tau_1 \mathrm{d}\tau_2}{\tau_2^2} A(\tau) E_{s\Lambda_1}^{SL(2)} = \int_{\partial\mathcal{F}'(\delta,L)} \star \left[ A \, \mathrm{d}E_{s\Lambda_1}^{SL(2)} - E_{s\Lambda_1}^{SL(2)} \, \mathrm{d}A \right] \qquad (B.4)$$

where $\star \mathrm{d}\tau_1 = \mathrm{d}\tau_2, \star \mathrm{d}\tau_2 = -\mathrm{d}\tau_1$. Due to $S_3$ symmetry, the three boundaries at $\tau_1 = 0, 1$ and $|\tau - \frac{1}{2}| = \frac{1}{2}$ produce identical contributions, while the the contribution from the boundary at $\tau_2 = L$ and its image is subtracted by the renormalisation prescription. The contribution from the boundary at $\tau_1 = 0$ can be computed by using

$$A(0, \tau_2) = \tau_2 + \frac{1}{\tau_2} \,, \quad \partial_{\tau_1} A|_{\tau_1 \to 0^+} = -\frac{6}{\tau_2} \,, \quad \lim_{\tau_1 \to 0} \partial_{\tau_1} E_{s\Lambda_1}^{SL(2)} = 0 \,. \qquad (B.5)$$

At $\tau_1 = 0$, $\tau_2$ runs from $L$ to $1/L$, hence

$$[s(s-1) - 12] \int_{\mathcal{F}'(\delta,L)} \frac{\mathrm{d}\tau_1 \mathrm{d}\tau_2}{\tau_2^2} A(\tau) E_{s\Lambda_1}^{SL(2)}(\tau) = -18 \int_{\frac{1}{L}}^{L} \frac{\mathrm{d}\tau_2}{\tau_2} E_{s\Lambda_1}^{SL(2)}(\mathrm{i}\tau_2) \,. \qquad (B.6)$$

The integral on the r.h.s. can be computed for $\mathrm{Re}[s] > 1$ by substituting $E_{s\Lambda_1}^{SL(2)} = \sum_{(c,d)=1} \frac{\tau_2^s}{|c\tau+d|^{2s}}$ and integrating term by term. Upon folding the integral and subtracting the divergence, we get

$$\lim_{L \to \infty} \left( \int_1^L \frac{\mathrm{d}\tau_2}{\tau_2} E_{s\Lambda_1}^{SL(2)}(\mathrm{i}\tau_2) - \frac{L^s}{s} \right) = \frac{2}{s\zeta(2s)} \sum_{n=1}^{\infty} n^{-2s} \sum_{m=1}^{\infty} {}_2F_1\left(\tfrac{s}{2}, s; \tfrac{s}{2} + 1; -\tfrac{m^2}{n^2}\right) \,, \qquad (B.7)$$

where the sum and the integral are absolutely convergent for $\mathrm{Re}[s] > 1$. Using the functional identity[32]

$${}_2F_1\left(\tfrac{s}{2}, s; \tfrac{s}{2} + 1; -x^2\right) + x^{-2s} {}_2F_1\left(\tfrac{s}{2}, s; \tfrac{s}{2} + 1; -\frac{1}{x^2}\right) = \frac{s\Gamma(\tfrac{s}{2})^2}{2\Gamma(s)} x^{-s} \,, \qquad (B.8)$$

and exchanging $m$ and $n$ one obtains

$$\frac{2}{s\zeta(2s)} \sum_{n=1}^{\infty} n^{-2s} \sum_{m=1}^{\infty} {}_2F_1\left(\tfrac{s}{2}, s; \tfrac{s}{2} + 1; -\tfrac{m^2}{n^2}\right) = \frac{\Gamma(\tfrac{s}{2})^2}{2\Gamma(s)\zeta(2s)} \sum_{n=1}^{\infty} \sum_{m=1}^{\infty} \frac{1}{(nm)^s} = \frac{\xi(s)^2}{2\xi(2s)} \,. \qquad (B.9)$$

consistently with the advertised formula (B.2).

It is worth noting that the integral (B.6) can be computed alternatively by inserting a power $\tau_2^\eta$ in the integrand, subtracting by hand the constant term from $E_{s\Lambda_1}^{SL(2)}(\mathrm{i}\tau_2)$, and extending the integral from $[1/L, L]$ to $\mathbb{R}^+$:

$$\int_0^{+\infty} \frac{\mathrm{d}\tau_2}{\tau_2^{1-\eta}} \left( E_{s\Lambda_1}^{SL(2)}(\mathrm{i}\tau_2) - \tau_2^s - \frac{\xi(2s-1)}{\xi(2s)} \tau_2^{1-s} \right) = L^\star \left( E_{s\Lambda_1}^{SL(2)}, \eta + \frac{1}{2} \right) \qquad (B.10)$$

---

[32]A special case of the general identity [74, Eq. (15.3.7)]

$${}_2F_1(a, b; c; z) = \frac{\Gamma(b-a)\Gamma(c)}{\Gamma(c-a)\Gamma(b)} \left(-\tfrac{1}{z}\right)^a {}_2F_1\left(a, a-c+1; a-b+1; \tfrac{1}{z}\right) + \frac{\Gamma(a-b)\Gamma(c)}{\Gamma(c-b)\Gamma(a)} \left(-\tfrac{1}{z}\right)^b {}_2F_1\left(b-c+1, b; -a+b+1; \tfrac{1}{z}\right) \,.$$

which gives the same result in the limit $\eta \to 0$ using (2.30). We therefore conclude that

$$\text{R.N.} \int_{\mathcal{F}'} \frac{\mathrm{d}\tau_1 \mathrm{d}\tau_2}{\tau_2^2} A(\tau) E_{s\Lambda_1}^{SL(2)}(\tau) = \frac{18}{12 - s(s-1)} \frac{\xi(s)^2}{\xi(2s)} \tag{B.11}$$

After dividing by 6 to get the integral over $\mathcal{F}$, we obtain (B.2).

## B.2 Against a double Eisenstein series

We now briefly consider the integral against the 'double Eisenstein series' defined in (2.43),

$$\widetilde{A}_s(U) \equiv \int_0^\infty \frac{\mathrm{d}V}{V^{1+2s}} \int_{\mathcal{F}} \frac{\mathrm{d}\tau_1 \mathrm{d}\tau_2}{\tau_2^2} A(\tau) \sum_{M \in \mathbb{Z}^{2 \times 2}}' e^{-\frac{\pi}{V} \mathcal{M}^2} \tag{B.12}$$

where, for an integer matrix $M = \begin{pmatrix} q_1 & p_1 \\ q_2 & p_2 \end{pmatrix}$,

$$\begin{aligned}
\mathcal{M}^2 &= \frac{1}{\tau_2 U_2} \text{Tr} \left[ \begin{pmatrix} p_1 & q_1 \\ p_2 & q_2 \end{pmatrix}^T \cdot \begin{pmatrix} 1 & -U_1 \\ -U_1 & |U|^2 \end{pmatrix} \cdot \begin{pmatrix} p_1 & q_1 \\ p_2 & q_2 \end{pmatrix} \cdot \begin{pmatrix} 1 & -\tau_1 \\ -\tau_1 & |\tau|^2 \end{pmatrix} \right] \\
&= \frac{1}{2\tau_2 U_2} \left( |p_1 - Up_2 - \tau(q_1 - Uq_2)|^2 + |p_1 - Up_2 - \bar{\tau}(q_1 - Uq_2)|^2 \right)
\end{aligned} \tag{B.13}$$

Using (B.3) and the fact that $\mathcal{M}^2$ degenerates to

$$\mathcal{M}^2 = \frac{1}{\tau_2 U_2} |p_1 - Up_2|^2 + \frac{\tau_2}{U_2} |q_1 - Uq_2|^2 \tag{B.14}$$

on the locus $\tau_1 = 0$, it is straightforward to check that the integral (B.12) satisfies the differential equation

$$\Delta \widetilde{A}_s(U) = 12 \widetilde{A}_s(U) - 6 \left( \xi(2s) E_{s\Lambda_1}^{SL(2)}(U) \right)^2 . \tag{B.15}$$

Using the same method as in [32, App. A], it is straightforward to show that the relevant solution to (B.15) can be represented as a sum of an Eisenstein series and a lattice sum-type series

$$\widetilde{A}_s(U) = 6\xi(2s)^2 \left[ \frac{E_{2s\Lambda_1}^{SL(2)}(U)}{2(2-s)(2s+3)} + \sum_{\gamma \in \mathcal{S}} (\det \gamma)^{-2s} h_s(U_1/U_2) \Big|_\gamma \right] \tag{B.16}$$

where[33]

$$\mathcal{S} = \{\pm 1\} \backslash \left\{ \begin{pmatrix} \alpha & \beta \\ \gamma & \delta \end{pmatrix} \in \mathbb{Z}^{2 \times 2} \cap GL^+(2, \mathbb{R}) \,\Big|\, \gcd(\alpha, \beta) = \gcd(\gamma, \delta) = 1 \right\} \tag{B.17}$$

and $h_s(u)$ is the unique smooth, decaying solution of

$$\left[ \partial_u((1+u^2)\partial_u) - 12 \right] h_s = -(1+u^2)^{-s} . \tag{B.18}$$

---

[33]In the expression (B.16), $GL^+(2, \mathbb{R})$ consists of positive determinant $GL(2, \mathbb{R})$ matrices and the action of $\begin{pmatrix} \alpha & \beta \\ \gamma & \delta \end{pmatrix} \in GL^+(2, \mathbb{R})$ on the upper half plane is $U \mapsto \frac{\alpha U + \beta}{\gamma U + \delta}$. The Laplacian on the upper half plane is also invariant under this action that extends the usual $SL(2, \mathbb{R})$ action.

This solution can be expressed for $s \notin \{1, 2\}$ as

$$h_s(u) = \frac{1 - 3s}{6(s-1)(s-2)} \, _2F_1\left(-\frac{3}{2}, s; \frac{1}{2}; -u^2\right) + \frac{su^2}{s-1} \, _2F_1\left(-\frac{1}{2}, s+1; \frac{3}{2}; -u^2\right)$$
$$+ \alpha(s) \left[\frac{4}{3} + 5u^2 + u(3 + 5u^2) \arctan(u)\right] \tag{B.19}$$

in terms of hypergeometric functions and the term in the second line is the unique homogeneous, even and smooth solution of (B.18). The latter can also be written as

$$\frac{4}{3} + 5u^2 + u(3 + 5u^2) \arctan(u) = \left[m(u) + \frac{35\pi}{8}|u|(3 + 5u^2)\right], \tag{B.20}$$

combining the non-smooth homogeneous solution $m(u)$ introduced in (A.39) and the independent non-smooth solution $|u|(3 + 5u^2)$. The numerical coefficient $\alpha(s)$ is fixed by requiring that $h_s(u)$ decays (as $1/|u|^2$) as $|u| \to \infty$ and is given explicitly by

$$\alpha(s) = \frac{\Gamma(\frac{3}{2} + s)}{3\sqrt{\pi}(s-1)(s-2)\Gamma(s)} . \tag{B.21}$$

For $s = 3/2$, we recover the solution in [32, (A.7)]

$$h_{3/2}(u) = \frac{7 + 44u^2 + 40u^4}{3\sqrt{1+u^2}} - \frac{16}{3\pi}\left(\frac{4}{3} + 5u^2 + u(3 + 5u^2)\arctan(u)\right) . \tag{B.22}$$

Similar closed algebraic forms arise when $s$ is half-integer, e.g.

$$h_{5/2}(u) = -\frac{13 + 102u^2 + 168u^4 + 80u^6}{9(1+u^2)^{3/2}} + \frac{32}{9\pi}\left(\frac{4}{3} + 5u^2 + u(3 + 5u^2)\arctan(u)\right) . \tag{B.23}$$

It is interesting to note that the representation (B.16) can be obtained directly by plugging in the Poincaré representation (A.38) of $A(\tau)$ into the integral (B.12), and unfolding the sum over $\gamma \in GL(2, \mathbb{Z})$. The first term in (B.16) comes from contributions of rank-one matrices $M$ while the second comes from non-degenerate matrices. The agreement with the second term in (B.16) relies on the conjectural identity for $A = \mathbb{I}_2$,

$$\frac{16}{35\pi} \int_0^\infty \frac{dV}{V^{1+2s}} \int_{\mathcal{H}_1} \frac{d\tau_1 d\tau_2}{\tau_2^2} e^{-\frac{2\pi}{V} - \frac{\pi|U-\tau|^2}{V\tau_2 U_2}} m(\tau_1/\tau_2) = 6[\pi^{-s}\Gamma(s)]^2 h_s(U_1/U_2) \tag{B.24}$$

which we have checked at the first few orders in a Taylor expansion around $U_1 = 0$ using Mathematica. Note that the factor $m(\tau_1/\tau_2)$ in the integrand, despite being annihilated by $\Delta_\tau - 12$, is not regular along the locus $\tau_1 = 0$ in $\mathcal{H}$, so that the reproducing kernel identity (2.20) does not apply. Indeed, upon applying it blindly, it would only produce the term proportional to the non-smooth $m(U_1/U_2)$ in (B.19) via (B.20).

Using the same method as in Section 3, it is straightforward to obtain the Fourier expansion

$$\widetilde{A}_s(U) = \frac{3\xi(2s)^2\, U_2^{2s}}{(2-s)(2s+3)} + \xi(2s)\xi(2s-1)U_2 + \frac{3\xi(2s-1)^2\, U_2^{2-2s}}{(s+1)(5-2s)} + \frac{\xi(5-2s)\xi(2s+3)}{6}U_2^{-3}$$

$$- \frac{2}{7}\sum_{N\in\mathbb{Z}}{}' \frac{(\sigma_{2s-1}(N))^2}{|N|^{2s-1}}B_{2s-1}(2\pi U_2|N|)$$

$$+ 2U_2 \sum_{\substack{M\in\mathbb{Z}^{2\times 2}\\ \det M\neq 0}} \int_{\mathcal{F}} \frac{\mathrm{d}\tau_1\mathrm{d}\tau_2}{\tau_2^2} A(\tau)\frac{|m_{11}+\tau m_{12}|^{2s-1}}{|m_{21}+\tau m_{22}|^{2s-1}}K_{2s-1}(2\pi U_2\tfrac{|m_{11}+\tau m_{12}||m_{21}+\tau m_{22}|}{\tau_2})e^{2\pi i\det M U_1} \quad \text{(B.25)}$$

where the function $B_s$ was defined in (3.30). It can be checked that this expansion is consistent with the Poisson equation (B.15).[34]

# C $\quad Spin(d,d)$ lattice sums

In this appendix, we analyse the two-loop/genus-two integrals introduced in (2.53) involving $Spin(d,d)$ lattice sums. This provides support for the conjectures in Sections 2 and 3 as well as in Appendix A.

## C.1 Large radius limit

We start with the genus-two modular integral (3.3), which we rewrite for convenience,

$$\mathcal{E}^{(d,2)}_{(0,1)} = 8\pi \,\text{R.N.} \int_{\mathcal{F}_2} \frac{\mathrm{d}^6\Omega}{|\Omega_2|^3}\varphi_{\text{KZ}}\,\Gamma_{d,d,2}\ . \quad \text{(C.1)}$$

Its asymptotics in the limit where one circle $S^1$ of radius $R$ inside $T^d$ decompactifies was discussed for generic $d$ in [2, (2.38)]:

$$\mathcal{E}^{(d,2)}_{(0,1)} = R^2\,\mathcal{E}^{(d-1,2)}_{(0,1)} + \frac{2\pi}{3}\xi(d-2)\,R^{d-1}\,\mathcal{E}^{(d-1,1)}_{(0,0)} + \frac{5}{\pi}\xi(d-6)\,R^{d-5}\,\mathcal{E}^{(d-1,1)}_{(1,0)} + \frac{16\pi^2\xi(d-2)^2}{(d+1)(6-d)}\,R^{2d-4} \quad \text{(C.2)}$$

and

$$\mathcal{E}^{(d-1,1)}_{(0,0)} = 4\pi\,\xi(d-2)\,E^{D_d}_{\frac{d-2}{2}\Lambda_1}\ , \quad \mathcal{E}^{(d-1,1)}_{(1,0)} = \frac{4\pi^4}{45}\,\xi(d+2)\,E^{D_d}_{\frac{d+2}{2}\Lambda_1} \quad \text{(C.3)}$$

Except for the last term proportional to $R^{2d-4}$, these constant terms can be obtained by using the orbit method: the term proportional to $R^2$ is the zero orbit contribution, while the terms proportional to $R^{d-1}$ and $R^{d-5}$ originate from the terms proportional to $t$ and $1/t$ in (A.13), the $\mathcal{O}(t^0)$ term giving a vanishing result after integrating over $v$. The orbit method fails to produce the complete expansion due to the logarithmic singularity of $\varphi_{\text{KZ}}$ at the separation limit, but one can recover the contribution in $R^{2d-4}$ by carefully extracting the contribution from this degeneration as in [75]. One can also determine this coefficient using the Poisson equation satisfied by the integral (C.1).

---

[34]Note that the only term of the Fourier expansion of $E^{SL(2)}_{2s\Lambda_1}(U)$ present in (B.16) that is not cancelled in (B.25) is the leading constant term proportional to $U_2^{2s}$.

We now consider the integral on the last line of (2.53),

$$\mathcal{I}_{S,d} = 8\pi \int_{\mathcal{G}} \frac{\mathrm{d}^3\Omega_2}{|\Omega_2|^{1-\epsilon}} \varphi_{\mathrm{KZ}}^{\mathrm{tr}} \, \theta_{\Lambda_d}^{D_d} \,, \qquad \theta_{\Lambda_d}^{D_d}(\Omega_2, \phi) = \sum_{\substack{Q_i \in S_+ \\ Q_i \gamma_{d-4} Q_j = 0}} e^{-\pi \Omega_2^{ij} v(Q_i) \cdot v(Q_j)} \,. \tag{C.4}$$

We shall see that the functional identity $\mathcal{E}_{(0,1)}^{(d,2)} = \widehat{\mathcal{I}}_{S,d}$ in (2.53) holds for the renormalised coupling

$$\widehat{\mathcal{I}}_{S,d} \equiv 8\pi \int_{\mathcal{G}} \frac{\mathrm{d}^3\Omega_2}{|\Omega_2|} \left( |\Omega_2|^{\epsilon} \varphi_{\mathrm{KZ}}^{\mathrm{tr}} - \frac{\pi}{36} \frac{E_{-3\Lambda_1}^{SL(2)}(\tau)}{V^{1+2\epsilon}} + \frac{\pi}{36} \frac{E_{-(3+2\epsilon)\Lambda_1}^{SL(2)}(\tau)}{V} \right) \theta_{\Lambda_d}^{D_d} \,, \tag{C.5}$$

as in (1.31).

In order to analyse the decompactification limit of (C.4), we decompose the sum in $\theta_{\Lambda_d}^{D_d}$. Under $Spin(d,d) \supset Spin(d-1,d-1) \times GL(1)$, the Weyl spinors $Q_i \in S_+$ decompose into two spinors $q_i \in S_+$, and $p_i \in S_-$ of opposite chiralities. The invariant quadratic form becomes

$$|v(Q)|^2 = R^{-1}|v(q+ap)|^2 + R|v(p)|^2 \tag{C.6}$$

while the constraints $Q_i \gamma_{d-4} Q_j = 0$ reduce to

$$q_i \gamma_{d-5} q_j = 0 \,, \quad p_i \gamma_{d-5} p_j = 0 \,, \quad q_i \gamma_{d-4} p_i = q_i \gamma_{d-6} p_i = 0 \,. \tag{C.7}$$

As in Section 3.2, we decompose the theta series into contributions where the components $(q_i, p_i)$ are gradually populated.

**The first layer**

The contribution from lattice spinors with $q_i \neq 0, p_i = 0$ gives

$$\mathcal{I}_{S,d}^{(1)} = 8\pi R^{2+2\epsilon} \int \frac{\mathrm{d}^3\Omega_2}{|\Omega_2|^{1-\epsilon}} \varphi_{\mathrm{KZ}}^{\mathrm{tr}}(\Omega_2) \sum_{\substack{q_i \in S_+ \\ q_i \gamma_{d-5} q_j = 0}}^{\prime} e^{-\pi \Omega_2^{ij} v(q_i) \cdot v(q_j)} \rightarrow R^2 \times \mathcal{I}_{S,d-1} \,, \tag{C.8}$$

where we can take the limit $\epsilon \to 0$ provided $\mathcal{I}_{S,d-1}$ is itself regular.

**The second layer**

For the layer with $p_i \neq 0$ but $p_1 \wedge p_2 = 0$, one has the Poincaré sum

$$\theta_{\Lambda_d}^{(2)}(\phi, \Omega_2) = \sum_{\gamma \in P_{d-1} \backslash D_{d-1}} \left( \sum_{n_i \in \mathbb{Z}}^{\prime} \sum_{q_i^a \in \mathbb{Z}^{d-1}} e^{-\pi \Omega_2^{ij} \left( R^{-1} y^2 \frac{d-3}{d-1} u_{ab}(q_i^a + a^a n_i)(q_j^b + a^b n_j) + R y^2 n_i n_j \right)} \right) \Bigg|_{\gamma}$$

$$= \frac{R^{d-1}}{|\Omega_2|^{\frac{d}{2}}} \sum_{\gamma \in P_{d-1} \backslash D_{d-1}} y^{-2(d-2)} \left( \sum_{n_i \in \mathbb{Z}}^{\prime} \sum_{q_a^i \in \mathbb{Z}^{d-1}} e^{-\pi \Omega_2^{ij} R y^2 n_i n_j - \pi \Omega_{2ij}^{-1} R y^{-2\frac{d-3}{d-1}} u^{ab} q_a^i q_b^j + 2\pi i n_i q_a^i a^a} \right) \Bigg|_{\gamma} \tag{C.9}$$

Constant terms originate from a) $q_a^i = 0$ and b) $n_i q_a^i = 0, q_a^i \neq 0$. The former requires to take into account both the dimensional regularisation $\epsilon \neq 0$ and the regularisation of the fundamental domain $\mathcal{F}_L$. One obtains after taking the limit $L \to \infty$

$$
\begin{aligned}
\mathcal{I}_{S,d}^{(2a)} &= 8\pi R^{2d-4-2\epsilon} \sum_{\gamma \in P_{d-1}\backslash D_{d-1}} (y^{-4\epsilon}) \Big|_\gamma \int \frac{\mathrm{d}^3\Omega_2}{|\Omega_2|^{\frac{d+2}{2}-\epsilon}} \varphi_{\mathrm{KZ}}^{\mathrm{tr}}(\Omega_2) \sideset{}{'}\sum_{n_i \in \mathbb{Z}} e^{-\pi \Omega_2^{ij} n_i n_j} \\
&= R^{2d-4-2\epsilon} \frac{16\pi^2 \xi(3-d+2\epsilon)^2}{(6-d+2\epsilon)(1+d-2\epsilon)} E_{2\epsilon\Lambda_{d-1}}^{D_{d-1}} \to R^{2d-4} \frac{16\pi^2 \xi(d-2)^2}{(6-d)(1+d)} , \qquad \text{(C.10)}
\end{aligned}
$$

using (3.17). The contributions b) are computed by unfolding the integration domain over $PGL(2,\mathbb{Z})$

$$
\begin{aligned}
16\pi R^{d-1} \int_0^\infty \frac{\mathrm{d}V}{V^{4-d+2\epsilon}} \int_0^\infty \frac{\mathrm{d}\tau_2}{\tau_2^2} \Bigg[ \int_{-\frac{1}{2}}^{\frac{1}{2}} \mathrm{d}\tau_1 \, A(\tau) \Bigg] & \\
\times \sum_{\gamma \in P_{d-1}\backslash D_{d-1}} \Bigg( y^{2(3-d)} \sum_{n\geq 1} \sideset{}{'}\sum_{q_a \in \mathbb{Z}^{d-1}} e^{-\frac{\pi}{V\tau_2} Ry^2 n^2 - \frac{\pi V}{\tau_2} Ry^2 \frac{3-d}{d-1} u^{ab} q_a q_b} \Bigg) \Bigg|_\gamma & \qquad \text{(C.11)}
\end{aligned}
$$

As in (3.19), this may be computed by inserting (3.27) in the square bracket. After changing variables to $\rho_2 = 1/(\tau_2 V), t = \tau_2/V$, The contribution from (3.20) to the integral gives

$$
\begin{aligned}
\mathcal{I}_{S,d}^{(2b)} &= \frac{4\pi^2 R^{d-1}}{3} \int_0^\infty \frac{\mathrm{d}t}{t^{\frac{d+1}{2}-\epsilon}} \int_0^\infty \frac{\mathrm{d}\rho_2}{\rho_2^{\frac{d-1}{2}-\epsilon}} \left( t + \frac{\rho_2^2}{6t} \right) \\
&\quad \times \sum_{\gamma \in P_{d-1}\backslash D_{d-1}} \Bigg( y^{2(3-d)} \sum_{n\geq 1} \sideset{}{'}\sum_{q_a \in \mathbb{Z}^{d-1}} e^{-\pi \rho_2 Ry^2 n^2 - \frac{\pi}{t} Ry^2 \frac{3-d}{d-1} u^{ab} q_a q_b} \Bigg) \Bigg|_\gamma \\
&= \frac{8\pi^2 R^{d-1}}{3} \xi(3-d+2\epsilon)\xi(d-3-2\epsilon) E_{(\frac{d-3}{2}-\epsilon)\Lambda_1 + 2\epsilon\Lambda_{d-1}}^{D_{d-1}} \\
&\quad + \frac{4\pi^2}{9} R^{d-5} \xi(7-d+2\epsilon) \xi(d+1-2\epsilon) E_{(\frac{d+1}{2}-\epsilon)\Lambda_1 + 2\epsilon\Lambda_{d-1}}^{D_{d-1}} \qquad \text{(C.12)}
\end{aligned}
$$

The terms on the last line are recognised as $R^{d-1}\mathcal{E}_{(0,0)}^{(d-1,1)}$ and $R^{d-5}\mathcal{E}_{(1,0)}^{(d-1,1)}$ in (C.2) with their respective coefficients. The last term in (3.27) gives additional non-perturbative contributions that would be overlooked by the naïve unfolding method. They are

$$
\begin{aligned}
\mathcal{I}_{S,d}^{(2c)} &= -\frac{2 \cdot 8^2 \pi^2}{21} R^d \int_0^\infty \frac{\mathrm{d}V}{V^{4-d+2\epsilon}} \int_0^1 \frac{\mathrm{d}\tau_2}{\tau_2^5} (1 + \tfrac{3}{2}\tau_2^2 + \tau_2^4)(1-\tau_2^2)^{\frac{3}{2}} \\
&\quad \times \sum_{\gamma \in P_{d-1}\backslash D_{d-1}} \Bigg( y^{3(2-d)} \sum_{\substack{a,c\geq 1 \\ \gcd(a,c)=1}} \sum_{n\geq 1} \sideset{}{'}\sum_{q_a \in \mathbb{Z}^{d-1}} e^{-\frac{\pi}{V\tau_2} Ry^2 (2acn^2) - \frac{\pi V}{\tau_2} Ry^2 \frac{3-d}{d-1} u^{ab}(2acq_a q_b)} \Bigg) \Bigg|_\gamma \\
&= -\frac{16\pi^2}{21} R^{d-1} \sideset{}{'}\sum_{\substack{q \in \mathit{II}_{d-1,d-1} \\ (q,q)=0}} \frac{\sigma_{d-3}(q)^2}{|v(q)|^{d-3}} B_{d-3}(2\pi R|v(q)|) , \qquad \text{(C.13)}
\end{aligned}
$$

where $B_s(x)$ was given in (3.31).

**The third layer**

The contribution from $p_1 \wedge p_2 \neq 0$ can be written as a Poincaré sum

$$
\theta_{\Lambda_d}^{(3)}(\phi, \Omega_2) = \sum_{\gamma \in P_{d-3} \backslash D_{d-1}} \left( \sum_{\substack{n_{i\hat{j}} \in \mathbb{Z}^2 \\ \det n \neq 0}} \sum_{\substack{q_i^a \in \mathbb{Z}^2 \\ p_\alpha \in \mathbb{Z}^{d-3}}} e^{-\pi \Omega_2^{ij} R^{-1} y \left( (R^2 n^2 + y^{\frac{2}{3-d}} \rho^{\alpha\beta}(p_\alpha + a_\alpha n)(p_\beta + a_\beta n)) v_{\hat{k}\hat{l}} \hat{n}_i{}^{\hat{k}} \hat{n}_j{}^{\hat{l}} \right)} \right.
$$
$$
\left. \times e^{-\pi \Omega_2^{ij} R^{-1} y u_{ab} (q_i^a + a_{\hat{k}}^a \hat{n}_i{}^{\hat{k}} + c_{\hat{k}}^{a\alpha} \hat{n}_i{}^{\hat{k}}(p_\alpha + a_\alpha n))(q_j^b + a_{\hat{l}}^b n_j{}^{\hat{l}} + c_{\hat{l}}^{b\beta} \hat{n}_j{}^{\hat{l}}(p_\beta + a_\beta n))} \right) \Bigg|_\gamma
$$

$$
= \frac{R^{\frac{d+1}{2}}}{|\Omega_2|} \sum_{\gamma \in P_{d-3} \backslash D_{d-1}} \left( \sum_{\substack{n_{i\hat{j}} \in \mathbb{Z}^2 \\ \det n \neq 0}} \sum_{\substack{q_a^i \in \mathbb{Z}^2 \\ p^\alpha \in \mathbb{Z}^{d-3}}} \frac{y^{-\frac{d-1}{2}}}{(\Omega_2^{ij} v_{\hat{k}\hat{l}} \hat{n}_i{}^{\hat{k}} \hat{n}_j{}^{\hat{l}})^{\frac{d-3}{2}}} e^{-\pi \Omega_2^{ij} R y v_{\hat{k}\hat{l}} n_i{}^{\hat{k}} n_j{}^{\hat{l}}} \right. \tag{C.14}
$$
$$
\left. \times e^{-\pi R \left( y^{-1} \Omega_{2ij}^{-1} u^{ab} q_a^i q_b^j + \frac{y^{\frac{5-d}{d-3}}}{\Omega_2^{ij} v_{\hat{k}\hat{l}} \hat{n}_i{}^{\hat{k}} \hat{n}_j{}^{\hat{l}}} \rho_{\alpha\beta}(p^\alpha - c_{\hat{k}}^{a\alpha} \hat{n}_i{}^{\hat{k}} q_a^i)(p^\beta - c_{\hat{l}}^{b\beta} \hat{n}_j{}^{\hat{l}} q_b^j) \right) + 2\pi i (q_a^i n_i{}^{\hat{j}} a_{\hat{j}}^a + np^\alpha a_\alpha)} \right) \Bigg|_\gamma .
$$

The constant term contribution is at $q_a^i = p^\alpha = 0$, since $n_i{}^{\hat{j}}$ is non-degenerate. After manipulating the integral over $V$ as in (3.42), one obtains the constant term

$$
\mathcal{I}_{S,d}^{(3a)} = 8\pi \frac{\xi(4\epsilon - d + 3)}{\xi(4\epsilon)} R^{d-1} \sum_{\gamma \in P_{d-3} \backslash D_{d-1}} \left( \sum_{\substack{n_i{}^{\hat{j}} \in \mathbb{Z}^2 \\ \det n \neq 0}} y^{-1} \int \frac{\mathrm{d}^3 \Omega_2}{|\Omega_2|^{2-\epsilon}} \varphi_{\mathrm{KZ}}^{\mathrm{tr}} e^{-\pi \Omega_2^{ij} R y v_{\hat{k}\hat{l}} n_i{}^{\hat{k}} n_j{}^{\hat{l}}} \right) \Bigg|_\gamma . \tag{C.15}
$$

The factor of $\xi(4\epsilon)$ in the denominator suggests that this contribution may vanish, but we shall see that the integral also diverges in $\xi(1 + 2\epsilon)$ so that there is a finite contribution. Nonetheless, we argue in Section C.3 that this terms drops out in the renormalised function (C.5) as a consequence of the tensorial differential equation. In particular, one has

$$
\begin{aligned}
\mathcal{I}_{S,d}^{(3a)} &= \frac{4\pi}{9} \frac{\xi(8)}{\xi(7)} R^{d-1-2\epsilon} \frac{\xi(4\epsilon - d + 3)}{\xi(4\epsilon)} \xi(2\epsilon + 3) \xi(2\epsilon - 4) E_{\frac{2\epsilon-3}{2}\Lambda_{d-3} + 4\Lambda_{d-2}}^{D_{d-1}} + \mathcal{O}(\epsilon) \\
&= \frac{4\pi}{9} \frac{\xi(8)}{\xi(7)} R^{d-1-2\epsilon} \frac{\xi(4\epsilon - d + 3)\xi(4\epsilon - 3)}{\xi(4\epsilon)\xi(4\epsilon - 2)} \frac{\xi(2 + 2\epsilon)\xi(1 + 2\epsilon)\xi(-5 + 2\epsilon)\xi(-6 + 2\epsilon)}{\xi(2\epsilon + 4)\xi(2\epsilon - 3)} E_{\frac{2\epsilon-3}{2}\Lambda_{d-5} + 4\Lambda_{d-2}}^{D_{d-1}} \\
&\underset{\epsilon \to 0}{=} -\frac{80\xi(2)\xi(6)\xi(8)\xi(d-2)}{\xi(3)} E_{-\frac{3}{2}\Lambda_{d-5} + 4\Lambda_{d-2}}^{D_{d-1}}
\end{aligned} \tag{C.16}
$$

for $d \geq 5$, where the function is $E_{4\Lambda_3}^{D_4}$ for $d = 5$, and zero for $d < 5$.

**Fourier coefficients**

The Fourier coefficients from (C.9) simplify to

$$
\mathcal{I}_{S,d}^{(2d)} = 8\pi R^{2d-4} \int \frac{\mathrm{d}^3 \Omega_2}{|\Omega_2|^{\frac{d+1}{2}}} \varphi_{\mathrm{KZ}}^{\mathrm{tr}}(\Omega_2) \sum_{n_i \in \mathbb{Z}}{}' \sum_{\substack{q^i \in II_{d-1,d-1} \\ (q^i, q^j) = 0 \\ n_i q^i \neq 0}}{}' e^{-\pi \Omega_2^{ij} n_i n_j - \pi \Omega_{2ij}^{-1} R^2 g(q^i, q^j) + 2\pi i n_i(q^i, a)} \tag{C.17}
$$

which can be computed as in Section 3.3. It is convenient to unfold the integral domain $\mathcal{G}$ to the set of positive matrices $\mathbb{R}^+ \times \mathcal{H}_1/\mathbb{Z}$ by fixing $n_i = (n,0)$ for $n > 0$. Setting $N = nq^1$, one can solve the constraint for $q^2$ in the $P_{d-1} \subset SO(d-1,d-1)$ parabolic decomposition associated to $N$ such that

$$
\begin{aligned}
\mathcal{I}_{S,d}^{(2d)} &= \frac{8\pi^2}{3} R^{2d-4} \int_0^\infty V^{d-4} dV \int_0^\infty \frac{d\tau_2}{\tau_2^2} \int_{-\frac{1}{2}}^{\frac{1}{2}} d\tau_1 \; A(\tau) \sum_{\gamma \in P_{d-1}\backslash SO(d\text{-}1,d\text{-}1)} {\sum_{N\in\mathbb{N}}}' \sum_{n|N} \qquad\qquad\text{(C.18)}\\
&\quad \times \sum_{\substack{j\in\mathbb{Z}\\ q\in\mathbb{Z}^{\frac{(d-1)(d-2)}{2}}\\ q\wedge q=0}} e^{-\pi\left(\frac{n^2}{V\tau_2}+V\tau_2 R^2\frac{y^2 N^2}{n^2}+\frac{V}{\tau_2}R^2(y^2(j+(\varsigma,q)-\frac{\tau_1}{n}N)^2+y^2\frac{d-5}{d-1}|v(q)|^2)\right)+2\pi i N a}\bigg|_\gamma \\[2mm]
&= \frac{8\pi^2}{3} R^{2d-5} \int_0^\infty V^{d-\frac{9}{2}} dV \int_0^\infty \frac{d\tau_2}{\tau_2^{\frac{3}{2}}} \sum_{\gamma\in P_{d\text{-}1}\backslash SO(d\text{-}1,d\text{-}1)} {\sum_{N\in\mathbb{N}}}' \sum_{n|N} \sum_{\tilde{\jmath}\in\mathbb{Z}} \int_{-\frac{1}{2}}^{\frac{1}{2}} d\tau_1 \; A(\tau) e^{-2\pi i \tau_1 \frac{N\tilde{\jmath}}{n}}\\
&\quad \times y^{-1} \sum_{\substack{q\in\mathbb{Z}^{\frac{(d-1)(d-2)}{2}}\\ q\wedge q=0}} e^{-\pi\left(\frac{n^2}{V\tau_2}+V\tau_2\frac{R^2 y^2 N^2}{n^2}+\frac{V}{\tau_2}R^2 y^2\frac{d-5}{d-1}|v(q)|^2+\frac{\tau_2}{VR^2 y^2}\tilde{\jmath}^2\right)+2\pi i(\tilde{\jmath}(q,\varsigma)+Na)}\bigg|_\gamma \;.
\end{aligned}
$$

Following the steps as in Section 3.3 and in particular (3.65), one computes that

$$
\begin{aligned}
\mathcal{I}_{S,d}^{(2d)} &= \frac{16\pi^2}{3} R^{\frac{d+1}{2}} {\sum_{\substack{N\in S_+\\ N\times N=0}}}' \Bigg( \sigma_{d-3}(N)\bigg( \frac{y_N^{\frac{9-3d}{2}+\frac{4(d-4)}{d-1}}}{\gcd(N)^{\frac{d-3}{2}}} \xi(d-4) E_{\frac{d-4}{2}\Lambda_2}^{SL(d-1)}(v_N) K_{\frac{d-3}{2}}\big(2\pi R\sqrt{g(N,N)}\big) \\
&\quad + \frac{3 y_N^{\frac{7-3d}{2}}}{\pi^2 R \gcd(N)^{\frac{d-1}{2}}} {\sum_{\substack{Q\in\mathbb{Z}^{\frac{(d-1)(d-2)}{2}}\\ Q\wedge Q=0}}}' \sigma_{d-4}(Q) e^{2\pi i(Q,\varsigma_N)} \frac{K_{\frac{d-2}{2}}(2\pi y_N^{-\frac{4}{d-1}}|v_N(Q)|)}{(y_N^{-\frac{4}{d-1}}|v_N(Q)|)^{\frac{d-2}{2}}} K_{\frac{d-5}{2}}\big(2\pi R\sqrt{g(N,N)}\big) \\
&\quad - \frac{15 y_N^{\frac{5-3d}{2}}}{2\pi^4 R^2 \gcd(N)^{\frac{d+1}{2}}} {\sum_{\substack{Q\in\mathbb{Z}^{\frac{(d-1)(d-2)}{2}}\\ Q\wedge Q=0}}}' \sigma_{d-4}(Q) e^{2\pi i(Q,\varsigma_N)} \frac{K_{\frac{d}{2}}(2\pi y_N^{-\frac{4}{d}}|v_N(Q)|)}{(y_N^{-\frac{4}{d}}|v_N(Q)|)^{\frac{d}{2}}} K_{\frac{d-7}{2}}\big(2\pi R\sqrt{g(N,N)}\big) \bigg) \\
&\quad + \frac{y_N^{\frac{5-3d}{2}+\frac{4d}{d-1}}}{6R^2} \frac{\sigma_{d-7}(N)}{\gcd(N)^{\frac{d-7}{2}}} \xi(d) E_{\frac{d}{2}\Lambda_2}^{SL(d-1)}(v_N) K_{\frac{d-7}{2}}\big(2\pi R\sqrt{g(N,N)}\big) \Bigg) e^{2\pi i(N,a)} \\
&\quad + \frac{8\pi^2}{3} R^{2d-5} \int_0^\infty \frac{d\nu}{\nu^{\frac{9}{2}-d}} \int_0^1 \frac{dt}{t^{\frac{3}{2}}} \sum_{\gamma\in P_{d\text{-}1}\backslash SO(d\text{-}1,d\text{-}1)} {\sum_{N\in\mathbb{N}}}' {\sum_{\substack{N_1\in\mathbb{Z}\\ Q\in\mathbb{Z}^{\frac{d(d-1)}{2}}\\ Q\wedge Q=0}}}' \sigma_{d-3}(N_1,Q)\sigma_{d-3}(N+N_1,Q) \\
&\quad \times y^{-1} e^{-\pi\left(\frac{2}{\nu t}+\nu t\frac{R^2 y^2 N^2}{2}+\frac{2\nu}{t}R^2 y^2\frac{d-5}{d-1}|v(Q)|^2\right)} F\big(t, R^2 y^2\nu, N, N_1+(\varsigma,Q)\big) e^{2\pi i N a_\gamma}\bigg|_\gamma \quad \text{(C.19)}
\end{aligned}
$$

where we kept the variable $y_N = \frac{\sqrt{g(N,N)}}{\gcd(N)}$ for simplicity, and the sum over $Q\in\mathbb{Z}^{\frac{(d-1)(d-2)}{2}}$ is a sum over characters of the unipotent stabilisers of the charge $N$, and $F$ is the function defined in (3.74). The leading term in $R$ factorises as an Eisenstein series over the Levi stabiliser of $N$,

while the full Fourier coefficient depends non-trivially on the whole parabolic stabiliser. The last term involving the integral and the function $F$ can be ascribed to instanton anti-instanton corrections, and is further exponentially suppressed.

The Fourier coefficients from (C.14) yield

$$
\mathcal{I}_{S,d}^{(3)} = 8\pi R^{d-1} \sum_{\gamma \in P_{d-3}\backslash D_{d-1}} \left( \sideset{}{'}\sum_{n_i{}^{\hat{j}} \in \mathbb{Z}^2} \int \frac{\mathrm{d}^3\Omega_2}{|\Omega_2|^2} \varphi_{\mathrm{KZ}}^{\mathrm{tr}}(\Omega_2) \frac{e^{-\pi \Omega_2^{ij} v_{\hat{k}\hat{l}} n_i{}^{\hat{k}} n_j{}^{\hat{l}}}}{y(\Omega_2^{ij} v_{\hat{k}\hat{l}} \hat{n}_i{}^{\hat{k}} \hat{n}_j{}^{\hat{l}})^{\frac{d-3}{2}}} \right.
$$

$$
\left. \times \sideset{}{'}\sum_{\substack{q_a^i \in \mathbb{Z}^2 \\ p^\alpha \in \mathbb{Z}^{d-3}}} e^{-\pi R^2 \left( \Omega_{2ij}^{-1} u^{ab} q_a^i q_b^j + \frac{y^{\frac{2}{d-3}}}{\Omega_2^{ij} v_{\hat{k}\hat{l}} \hat{n}_i{}^{\hat{k}} \hat{n}_j{}^{\hat{l}}} \rho_{\alpha\beta}(p^\alpha - c_{\hat{k}}^{a\alpha} \hat{n}_i{}^{\hat{k}} q_a^i)(p^\beta - c_{\hat{l}}^{b\beta} \hat{n}_j{}^{\hat{l}} q_b^j) \right) + 2\pi i (q_a^i n_i{}^{\hat{j}} a_{\hat{j}}^a + n p^\alpha a_\alpha)} \right) \Bigg|_\gamma \tag{C.20}
$$

Using (3.77), the integral gives in the saddle point approximation

$$
\int \frac{\mathrm{d}^3\Omega_2}{|\Omega_2|^2} \frac{\varphi_{\mathrm{KZ}}^{\mathrm{tr}}(\Omega_2)}{(\mathrm{Tr}\,\Omega_2 Y)^{\frac{d-2}{2}}} e^{-S(\Omega_2)} \sim \frac{\varphi_{\mathrm{KZ}}^{\mathrm{tr}}(X + \sqrt{|XY|}Y^{-1}) e^{-2\pi\sqrt{M + \mathrm{Tr}\,XY + 2\sqrt{|XY|}}}}{\sqrt{8|X|}(\mathrm{Tr}\,XY + 2\sqrt{|XY|})(M + \mathrm{Tr}\,XY + 2\sqrt{|XY|})^{\frac{2d-7}{4}}} \tag{C.21}
$$

For $P^\alpha = n p^\alpha$ and $Q_a^{\hat{i}} = n_j{}^{\hat{i}} q_a^j$, we obtain

$$
S(\Omega_2^\star) = 2\pi R \sqrt{y^{\frac{2}{d-2}} \rho_{\alpha\beta}(P^\alpha - c_{\hat{i}}^{a\alpha} Q_a^{\hat{i}})(P^\beta - c_{\hat{j}}^{b\beta} Q_b^{\hat{j}}) + v_{\hat{i}\hat{j}} u^{ab} Q_a^{\hat{i}} Q_b^{\hat{j}} + 2|\det Q|} \Bigg) \tag{C.22}
$$

which is recognised as the BPS mass for the vector $\mathcal{Q} \in II_{d,d}$ with a non-vanishing norm, such that

$$
S(\Omega_2^\star) = 2\pi R \sqrt{g(\mathcal{Q}, \mathcal{Q}) + (\mathcal{Q}, \mathcal{Q})} \,. \tag{C.23}
$$

Collecting all contributions, one finally obtains

$$
\mathcal{I}_{S,d} = R^2 \mathcal{I}_{S,d-1} + \frac{16\pi^2 \xi(3-d)^2}{(6-d)(1+d)} R^{2d-4}
$$

$$
+ \frac{8\pi^2}{3} \xi(d-2)\xi(d-3) R^{d-1} E_{\frac{d-3}{2}\Lambda_1}^{D_{d-1}} + \frac{4\pi^2}{9} \xi(d-6)\xi(d+1) R^{d-5} E_{\frac{d+1}{2}\Lambda_1}^{D_{d-1}}
$$

$$
- \frac{256\pi^2}{21} R^{d-1} \int_0^1 \frac{\mathrm{d}t}{t^5} (1 + \tfrac{3}{2}t^2 + t^4)(1 - t^2)^{\frac{3}{2}} \sideset{}{'}\sum_{\substack{q \in II_{d-1,d-1} \\ (q,q)=0}} \sigma_{d-3}(q)^2 \frac{K_{d-3}(\frac{4\pi}{t} R|v(q)|)}{|v(q)|^{d-3}}
$$

$$
+ 8\pi R^{2d-4} \int \frac{\mathrm{d}^3\Omega_2}{|\Omega_2|^{\frac{d+1}{2}}} \varphi_{\mathrm{KZ}}^{\mathrm{tr}}(\Omega_2) \sideset{}{'}\sum_{n_i \in \mathbb{Z}} \sideset{}{'}\sum_{\substack{q^i \in II_{d-1,d-1} \\ (q^i,q^j)=0}} e^{-\pi \Omega_2^{ij} n_i n_j - \pi \Omega_{2ij}^{-1} R^2 g(q^i,q^j) + 2\pi i n_i(q^i,a)}
$$

$$
+ 8\pi R^{d-1} \sum_{\gamma \in P_{d-3}\backslash D_{d-1}} \left( \sideset{}{'}\sum_{n_i{}^{\hat{j}} \in \mathbb{Z}^2} \int \frac{\mathrm{d}^3\Omega_2}{|\Omega_2|^2} \varphi_{\mathrm{KZ}}^{\mathrm{tr}}(\Omega_2) \frac{e^{-\pi \Omega_2^{ij} v_{\hat{k}\hat{l}} n_i{}^{\hat{k}} n_j{}^{\hat{l}}}}{y(\Omega_2^{ij} v_{\hat{k}\hat{l}} \hat{n}_i{}^{\hat{k}} \hat{n}_j{}^{\hat{l}})^{\frac{d-3}{2}}} \right.
$$

$$
\left. \times \sideset{}{'}\sum_{\substack{q_a^i \in \mathbb{Z}^2 \\ p^\alpha \in \mathbb{Z}^{d-3}}} e^{-\pi R^2 \left( \Omega_{2ij}^{-1} u^{ab} q_a^i q_b^j + \frac{y^{\frac{2}{d-3}}}{\Omega_2^{ij} v_{\hat{k}\hat{l}} \hat{n}_i{}^{\hat{k}} \hat{n}_j{}^{\hat{l}}} \rho_{\alpha\beta}(p^\alpha - c_{\hat{k}}^{a\alpha} \hat{n}_i{}^{\hat{k}} q_a^i)(p^\beta - c_{\hat{l}}^{b\beta} \hat{n}_j{}^{\hat{l}} q_b^j) \right) + 2\pi i (q_a^i n_i{}^{\hat{j}} a_{\hat{j}}^a + n p^\alpha a_\alpha)} \right) \Bigg|_\gamma
$$

$$\tag{C.24}$$

which is consistent with the identity (2.53). It is worth noting that the term proportional to $R^{2d-4}$ on the first line can be viewed as the contribution of the vector $q^i = 0$ in the integral on the fourth line, while the first term $R^2 \mathcal{I}_{S,d-1}$ can be viewed as the contribution from $n_i = 0$ in the same integral, upon using the identity

$$\int_{Sp(4,\mathbb{Z})\backslash\mathcal{H}_+(\mathbb{C})} \frac{\mathrm{d}^6\Omega}{|\Omega_2|^3}\varphi_{\mathrm{KZ}}(\Omega)\Gamma_{d,d,2} = \int_{GL(2,\mathbb{Z})\backslash\mathcal{H}_+(\mathbb{R})} \frac{\mathrm{d}^3\Omega_2}{|\Omega_2|}\varphi_{\mathrm{KZ}}^{\mathrm{tr}}(\Omega_2) \sum_{\substack{\chi_i\in S_+ \\ \chi_i\gamma_{d-4}\chi_j=0}} e^{-\pi\Omega_2^{ij}v(\chi_i)\cdot v(\chi_j)} \quad \text{(C.25)}$$

## C.2  Large volume limit

We now consider the large volume limit of the genus-two integral

$$\mathcal{E}_{(0,1)}^{(d,2)} = 8\pi\int_{\mathcal{F}_2} \frac{\mathrm{d}^6\Omega}{|\Omega_2|^3}\varphi_{\mathrm{KZ}}\,\Gamma_{d,d,2} = 8\pi\int_{\mathcal{G}} \frac{\mathrm{d}^3\Omega_2}{|\Omega_2|^{3-\frac{d}{2}-\epsilon}}\varphi_{\mathrm{KZ}}^{\mathrm{tr}}(\Omega_2)\,\theta_{\Lambda_1}^{D_d} \quad \text{(C.26)}$$

The latter may be computed either by the orbit method for the modular integral over $\mathcal{F}_2$, as in [75], or by decomposing the lattice sum $\theta_{\Lambda_1}^{D_d}$. We shall show that the two procedures give the same results, providing supporting evidence for the Poincaré series representation (1.21) which underlies the equality (C.26).

Applying the orbit method on the first expression in (C.26), we find constant terms coming from the rank-zero, rank-one and rank-two orbits, respectively,

$$\begin{aligned}
\mathcal{E}_{(0,1)}^{(d,2)} = 8\pi R^d \Bigg( & \int_{\mathcal{F}_2} \frac{\mathrm{d}^6\Omega}{|\Omega_2|^3}\varphi_{\mathrm{KZ}} \\
& + \int_0^\infty \frac{\mathrm{d}t}{t^3}\int_{\mathcal{F}} \frac{\mathrm{d}\rho_1\mathrm{d}\rho_2}{\rho_2^2}\int_{[0,1]^3}\mathrm{d}u_1\mathrm{d}u_2\mathrm{d}\sigma_1 \left(\frac{\pi}{6}t+\varphi_0+\frac{\varphi_1}{t}\right)\sideset{}{'}\sum_{n\in\mathbb{Z}^d} e^{-\pi\frac{R|v^{-\intercal}(n)|^2}{t}} \\
& + \int_{\mathcal{G}} \frac{\mathrm{d}^6\Omega}{|\Omega_2|^3}\left(\varphi_{\mathrm{KZ}}^{\mathrm{tr}}(\Omega_2)+\frac{5\zeta(3)}{4\pi^2}|\Omega_2|^{-1}\right)\sum_{\substack{n^i\in\mathbb{Z}^5 \\ \mathrm{rk}\,n=2}} e^{-\pi R\Omega_{2ij}^{-1}v^{-\intercal}(n^i)\cdot v^{-\intercal}(n^j)}\Bigg) \quad \text{(C.27)}
\end{aligned}$$

where $R^d = V_d^2$ and we replaced $\varphi_{\mathrm{KZ}}$ by its constant terms (A.13) and (A.11) in the Fourier–Jacobi and Fourier expansions, respectively. The first integral was evaluated in [36] using the Laplace eigenmode property of $\varphi_{\mathrm{KZ}}$,

$$\int_{\mathcal{F}_2} \frac{\mathrm{d}^6\Omega}{|\Omega_2|^3}\,\varphi_{\mathrm{KZ}} = \frac{\pi^3}{180} \quad \text{(C.28)}$$

In the rank-one contribution, the integral over $u_1, u_2$ annihilates $\varphi_0$ and replaces $\varphi_1$ by the Eisenstein series $\frac{5}{12}E_{2\Lambda_1}^{SL(2)}$, whose integral on $\mathcal{F}$ vanishes. In this way we arrive at the constant terms

$$\begin{aligned}
\mathcal{E}_{(0,1)}^{(d,2)} \sim\ & \frac{2\pi^4}{45}R^d + \frac{2\pi^4}{27}R^{d-1}E_{\Lambda_{d-1}}^{SL(d)} \\
& + 8\pi R^{d-2}\int \frac{\mathrm{d}^3\Omega_2}{|\Omega_2|^3}\varphi_{\mathrm{KZ}}^{\mathrm{tr}}(\Omega_2)\sideset{}{'}\sum_{n^i\in\mathbb{Z}^5} e^{-\pi\Omega_{2ij}^{-1}v^{-\intercal}(n^i)\cdot v^{-\intercal}(n^j)} + \frac{\zeta(3)\zeta(5)}{6\pi}R^{d-5}E_{\frac{5}{2}\Lambda_{d-2}}^{SL(d)}\,, \quad \text{(C.29)}
\end{aligned}$$

where we omitted in the third term the restriction of the sum to rank-two matrices, which would follow from (C.27). The additional sum over rank-one matrices arises due to the logarithmic divergence of the Kawazumi–Zhang invariant at the separating degeneration locus, similarly to the term proportional to $R^{2d-4}$ in (C.2) of the last section, and would be absent in the case of a regular theta lift (against a cuspidal form or a Siegel–Eisenstein series. Physically this third term comes from the two-loop ten-dimensional supergravity amplitude on $T^d$, which does include all Kaluza–Klein momenta and not only rank-two matrices.

The Fourier coefficients only get contributions from the rank-two orbit, but they are complicated and unilluminating, therefore we shall not display them.

Alternatively, one may compute the large volume limit by decomposing the constrained lattice sum in the vector representation,

$$
\begin{aligned}
\theta_{\Lambda_1}^{D_d} &= \sideset{}{'}\sum_{q_i \in \mathbb{Z}^d} e^{-\pi \Omega_2^{ij} R^{-1} v(q_i) \cdot v(q_j)} + \sideset{}{'}\sum_{p_i \in \mathbb{Z}^d} \sum_{\substack{q_i \in \mathbb{Z}^d \\ 2p_{(i} \cdot q_{j)} = 0}} e^{-\pi \Omega_2^{ij} \left( R^{-1} v(q_i + a p_i) \cdot v(q_j + a p_j) + R v^{-\intercal}(p_i) \cdot v^{-\intercal}(p_j) \right)} \\
&= \sideset{}{'}\sum_{q_i \in \mathbb{Z}^d} e^{-\pi \Omega_2^{ij} R^{-1} v(q_i) \cdot v(q_j)} \\
&\quad + \sum_{\gamma \in P_{d-1} \backslash SL(d)} \left( \sideset{}{'}\sum_{n_i \in \mathbb{Z}} \sum_{q^i \in \mathbb{Z}^{d-1}} \frac{R^{d-1}}{|\Omega_2|^{\frac{d-1}{2}} y^2} e^{-\pi \Omega_2^{ij} R y^2 n_i n_j - \pi \Omega_{2ij}^{-1} R y^{-\frac{2}{d-1}} v(q^i) \cdot v(q^j) + 2\pi i (n_i q^i, a)} \right) \Bigg|_\gamma \\
&\quad + \sum_{\gamma \in P_{d-2} \backslash SL(d)} \left( \sum_{\substack{n_i^{\hat{j}} \in \mathbb{Z}^{2 \times 2} \\ \det n \neq 0}} \sum_{\substack{\tilde{q}^i \in \mathbb{Z}^{2 \times (d-2)} \\ \tilde{p} \in \mathbb{Z}}} \frac{R^{d-\frac{3}{2}}}{|\Omega_2|^{\frac{d-2}{2}} y^{\frac{3}{2}} \sqrt{\Omega_2^{ij} \rho_{\hat{k}\hat{l}} \hat{n}_i^{\hat{k}} \hat{n}_j^{\hat{l}}}} e^{-\pi \Omega_2^{ij} R y \rho_{\hat{k}\hat{l}} n_i^{\hat{k}} n_j^{\hat{l}} + 2\pi i (n_i^{\hat{j}} \tilde{q}^i \cdot a_{\hat{j}} + n \tilde{p} a)} \right. \\
&\qquad\qquad \left. \times\; e^{-\pi \Omega_{2ij}^{-1} R y^{-\frac{2}{d-2}} v(\tilde{q}^i) \cdot v(\tilde{q}^j) - \frac{\pi}{\Omega_2^{ij} \rho_{\hat{k}\hat{l}} \hat{n}_i^{\hat{k}} \hat{n}_j^{\hat{l}}} R y (\tilde{p} - \hat{n}_i^{\hat{j}} \tilde{q}^i \cdot c_{\hat{j}})^2} \right) \Bigg|_\gamma .
\end{aligned}
\tag{C.30}
$$

Here we solved the constraints $p_{(i} \cdot q_{j)} = 0$ using the decompositions

$$
\overline{\boldsymbol{d}} = (\overline{\boldsymbol{d-2}})^{(-\frac{2}{d-2})} \oplus \boldsymbol{2}^{(1)} \ni p_i = (0, n_i^{\hat{j}}) ,
\tag{C.31}
$$

$$
\boldsymbol{d} = \boldsymbol{2}^{(-1)} \oplus (\boldsymbol{d-2})^{(\frac{2}{d-2})} \ni q_i = (\tilde{p}\, \hat{n}_i^{\hat{j}}, \tilde{q}_i) ,
\tag{C.32}
$$

where $\hat{n} = n/\gcd(n)$, and performed a Poisson resummation over $\tilde{q}_i \in \mathbb{Z}^{d-2}$ and $\tilde{p} \in \mathbb{Z}$. Inserting the decomposition (C.30) inside the last integral in (C.26), one obtains

$$
\begin{aligned}
\mathcal{E}^{(d,2)}_{(0,1)} \quad \sim \quad & 8\pi\, R^{d-2} \int_{\mathcal{G}} \frac{\mathrm{d}^3\Omega_2}{|\Omega_2|^{3-\frac{d}{2}}} \varphi^{\mathrm{tr}}_{\mathrm{KZ}}(\Omega_2) \sideset{}{'}\sum_{q_i\in\mathbb{Z}^d} e^{-\pi\Omega_2^{ij} v(q_i)\cdot v(q_j)} && \text{(C.33)} \\
& + \frac{8\pi^2}{3} R^{d-\epsilon}\, E^{SL(d)}_{\epsilon\Lambda_{d-2}}\, \xi(4-2\epsilon)\, \text{R.N.} \int_{\mathcal{F}} \frac{\mathrm{d}\tau_1\mathrm{d}\tau_2}{\tau_2^2} A(\tau)\, E^{SL(2)}_{(2-\epsilon)\Lambda_1}(\tau) \\
& + \frac{8\pi^2}{3} R^{d-1}\xi(-1-2\epsilon)\,\xi(1+2\epsilon)\, E^{SL(d)}_{\frac{1+2\epsilon}{2}\Lambda_{d-2}-2\epsilon\Lambda_{d-1}} \\
& + \frac{4\pi^2}{9} R^{d-5}\,\xi(3-2\epsilon)\,\xi(5+2\epsilon)\, E^{SL(d)}_{\frac{5+2\epsilon}{2}\Lambda_{d-2}-2\epsilon\Lambda_{d-1}} - \frac{16\pi^2}{21} R^{d-1} \sideset{}{'}\sum_{\substack{Q\in\wedge^2\mathbb{Z}^d \\ Q\times Q=0}} \frac{\sigma_2(Q)^2}{|v(Q)|} B_1(2\pi R|v(Q)|) \\
& + 8\pi \frac{\xi(4\epsilon-2)}{\xi(4\epsilon)} R^{d-1-2\epsilon} \sum_{\gamma\in P_{d-2}\backslash SL(d)} \left( y^{-1-2\epsilon} \int_{\mathcal{F}} \frac{\mathrm{d}^3\Omega_2}{|\Omega_2|^{2-\epsilon}} \varphi^{\mathrm{tr}}_{\mathrm{KZ}}(\Omega_2) \sum_{\substack{n_i{}^{\hat{j}}\in\mathbb{Z}^{2\times 2} \\ \det n\neq 0}} e^{-\pi\Omega_2^{ij}\rho_{\hat{k}\hat{l}} n_i{}^{\hat{k}} n_j{}^{\hat{l}}} \right)\Bigg|_\gamma
\end{aligned}
$$

where the second term comes from the contribution of rank-one charges with $q_i = 0$, and the third and fourth lines from rank-one charges with $q^i \neq 0, n_i q^i = 0$, which can be computed as in (3.24), (3.29), giving the two Eisenstein series above using

$$
\sum_{\gamma\in P_{d-1}\backslash SL(d)} \sideset{}{'}\sum_{q_a\in\mathbb{Z}^{d-1}} y^{4\epsilon} \left( y^{2\frac{d-2}{d-1}}|v(q)|^2 \right)^{-s-\epsilon} = 2\xi(2s+2\epsilon) E^{D_d}_{(s+\epsilon)\Lambda_{d-2}-2\epsilon\Lambda_{d-1}}\,, \tag{C.34}
$$

at $s = 1/2$ and $s = 5/2$, for the third and fourth terms respectively. The last line comes from the last line in (C.30) at $\tilde{q} = \tilde{p} = 0$ and generically vanishes at $\epsilon \to 0$ because of the overall $\frac{1}{\xi(4\epsilon)}$ factor. In addition, one checks using the Langlands constant term formula that for any $z \in \mathbb{C}$,

$$
\lim_{\epsilon\to 0}\left( \xi(1+2\epsilon)\, E^{SL(d)}_{\frac{1+2\epsilon}{2}\Lambda_{d-2}+z\epsilon\Lambda_{d-1}} \right) = \xi(2)\, E^{SL(d)}_{\Lambda_{d-1}} \tag{C.35}
$$

generalising the functional equation

$$
\xi(1+2\epsilon)\, E^{SL(d)}_{\frac{1+2\epsilon}{2}\Lambda_{d-2}} = \xi(2+2\epsilon)\, E^{SL(d)}_{\frac{2+2\epsilon}{2}\Lambda_{d-1}}\,. \tag{C.36}
$$

To identify the first term we use the identity

$$
\int_{\mathcal{G}} \frac{\mathrm{d}^3\Omega_2}{|\Omega_2|^{3-\frac{d}{2}}} \varphi^{\mathrm{tr}}_{\mathrm{KZ}}(\Omega_2) \sideset{}{'}\sum_{q_i\in\mathbb{Z}^d} e^{-\pi\Omega_2^{ij} v(q_i)\cdot v(q_j)} = \int_{\mathcal{G}} \frac{\mathrm{d}^3\Omega_2}{|\Omega_2|^3} \varphi^{\mathrm{tr}}_{\mathrm{KZ}}(\Omega_2) \sideset{}{'}\sum_{n^i\in\mathbb{Z}^d} e^{-\pi\Omega_{2ij}^{-1} v^{-\mathsf{T}}(n^i)\cdot v^{-\mathsf{T}}(n^j)}\,, \tag{C.37}
$$

that follows by Poisson summation using that the renormalised integral $\int_{\mathcal{G}} \frac{\mathrm{d}^3\Omega_2}{|\Omega_2|^{3-s}} \varphi^{\mathrm{tr}}_{\mathrm{KZ}}(\Omega_2)$ vanishes.

Putting these terms together, one therefore matches the expansion (C.27) in the limit $\epsilon \to 0$, up to the exponentially suppressed terms that are missed by the orbit method. This computation, valid for generic $d$, provides strong evidence for the Poincaré series representation (1.21).

It is worth noting that the term of order $R^d$ arises in two different ways in these two computations, leading to a rather remarkable identity for the integral of $\varphi_{\text{KZ}}$ over the fundamental domain of $Sp(4, \mathbb{Z})$,

$$\int_{\mathcal{F}_2} \frac{\mathrm{d}^6\Omega}{|\Omega_2|^3} \varphi_{\text{KZ}} = \frac{\pi^3}{270} \text{R.N.} \int_{\mathcal{F}} \frac{\mathrm{d}\tau_1 \mathrm{d}\tau_2}{\tau_2^2} A(\tau) E_{2\Lambda_1}^{SL(2)}(\tau) \tag{C.38}$$

This identity can presumably be established more directly by using the Rankin–Selberg method, i.e. computing the Petersson product between $\varphi_{\text{KZ}}$ and $E_{s\Lambda_1}^{Sp(4)}$ using the unfolding trick, and extracting the residue at $s = \frac{3}{2}$. However, there are regularisation issues which make this computation challenging.

In addition, there are non-perturbative corrections coming from the second line with $q^i \neq 0$ but $n_i q^i = 0$ through the extension of $\varphi_{\text{KZ}}^{\text{tr}}(\Omega_2)$ to $\mathcal{H}_+(\mathbb{R})$. The Fourier coefficients from the second line at $n_i q^i \neq 0$ can be computed after a change of variable in $\Omega_2 \to R^{-1} y^{-2} \Omega_2^{-1}$ and implementing the Poincaré sum at $\epsilon = 0$ as

$$8\pi R^d \int \frac{\mathrm{d}^3\Omega_2}{|\Omega_2|^{\frac{3}{2}}} \varphi_{\text{KZ}}^{\text{tr}}(\Omega_2) \sideset{}{'}\sum_{\substack{Q_i \in \mathbb{Z}^{\frac{d(d-1)}{2}} \\ Q_i \times Q_j = 0}} \sideset{}{'}\sum_{\substack{m^i \in \mathbb{Z} \\ m^i Q_i \neq 0}} e^{-\pi \Omega_{2\,ij}^{-1} m^i m^j - \pi \Omega_2^{ij} R^2 u(Q_i, Q_j) + 2\pi i (m^i Q_i, a)} \ . \tag{C.39}$$

For $d = 5$ it coincides with the last line in (4.14) with $d = 4$, in agreement with the functional equation (1.23). It can be simplified in the same way as in (4.15) for general $d$. The rank-two Fourier coefficients come from the last line in (C.30) with $(\tilde{q}^i, \tilde{p}) \neq 0$. One checks for $d = 5$ that they match the Fourier coefficients of (4.24) at $d = 5$, with a change of variable in $\Omega_2$ and upon identitfying $P_3 \subset SL(5)$ as $P_4 \subset E_4$. Under the assumption that the renormalised $\mathcal{I}_4^{(4)}$ of (4.32) indeed vanishes in the limit $\epsilon \to 0$, one obtains a perfect match of the two functions (1.6) and (1.32) at $d = 4$. This provides further evidence for the vanishing of the renormalised fourth layer contribution $\mathcal{I}_d^{(4)}$ in the decompactification limit.

## C.3   Vanishing of the third layer contribution

In Section 3.2, we relied on (3.44) to show that the third layer contribution to the weak coupling limit of the renormalised coupling (1.31) cancels out. To justify (3.44) we shall first establish that $\mathcal{I}_d^{(3a)}$ is an eigenfunction of the Laplace operator. The argument of this section will generalise straightforwardly to prove the similar result (C.16) for $\mathcal{I}_{S,d}^{(3a)}$.

For this purpose, one can write (3.43) as a Poincaré sum

$$\begin{aligned}
\mathcal{I}_d^{(3a)} &= \frac{8\pi^2}{3} g_D^{-\frac{24+8\epsilon}{8-d}+2+4\epsilon} \frac{\xi(4\epsilon-2)}{\xi(4\epsilon)} \sum_{\gamma \in P_{d-2}\backslash D_d} \left[ y^{-1-2\epsilon} \Big( \widetilde{A}_\epsilon(U) - \frac{3\xi(2\epsilon)^2}{6+\epsilon-2\epsilon^2} E_{2\epsilon\Lambda_1}^{SL(2)}(U) \Big) \right]\Big|_\gamma \\
&= \frac{8\pi^2}{3} g_D^{-\frac{24+8\epsilon}{8-d}+2+4\epsilon} \left( \frac{\xi(4\epsilon-2)}{\xi(4\epsilon)} \sum_{\gamma \in P_{d-2}\backslash D_d} \Big( y^{-1-2\epsilon} \widetilde{A}_\epsilon(U) \Big)\Big|_\gamma - \frac{\zeta(3)}{12} E_{\Lambda_d}^{D_d} + \mathcal{O}(\epsilon) \right) \tag{C.40}
\end{aligned}$$

where we used the functional relation

$$\xi(2\epsilon) \sum_{\gamma \in P_{d-2}\backslash D_d} \Big( y^{-1-2\epsilon} E_{2\epsilon\Lambda_1}^{SL(2)}(U) \Big)\Big|_\gamma = \xi(2\epsilon) E_{\frac{1}{2}\Lambda_{d-2}+\epsilon\Lambda_d}^{D_d} = \xi(2-2\epsilon) E_{\epsilon\Lambda_{d-2}+(1-\epsilon)\Lambda_d}^{D_d} \tag{C.41}$$

in the last line. Acting with the Laplace operator and integrating by parts, we find

$$(\Delta_{D_d} + \tfrac{(d-2)(1+2\epsilon)(d-2\epsilon)}{2} - 12)\left( \sum_{\gamma \in P_{d-2}\backslash D_d} \left[ y^{-1-2\epsilon}\Big( \widetilde{A}_\epsilon(U) - \frac{3\xi(2\epsilon)^2}{6+\epsilon-2\epsilon^2} E_{2\epsilon\Lambda_1}^{SL(2)}(U) \Big) \right]\Big|_\gamma \right)$$

$$= \sum_{\gamma \in P_{d-2}\backslash D_d} \left[ y^{-1-2\epsilon}(\Delta_U - 12)\Big( \widetilde{A}_\epsilon(U) - \frac{3\xi(2\epsilon)^2}{6+\epsilon-2\epsilon^2} E_{2\epsilon\Lambda_1}^{SL(2)}(U) \Big) \right]\Big|_\gamma$$

$$= -6\xi(2\epsilon)^2 \sum_{\gamma \in P_{d-2}\backslash D_d} \left[ y^{-1-2\epsilon}\Big( (E_{\epsilon\Lambda_1}^{SL(2)}(U))^2 - E_{2\epsilon\Lambda_1}^{SL(2)}(U) \Big) \right]\Big|_\gamma . \tag{C.42}$$

The right-hand-side of this differential equation is a Poincaré sum of a function with a finite limit at $\epsilon \to 0$, and so we expect $\mathcal{I}_d^{(3a)}$ (that includes an extra $\frac{1}{\xi(4\epsilon)}$) to satisfy a homogenous equation at $\epsilon = 0$. To study this, it will prove convenient to use the double lattice sum representation of the Poincaré sum

$$\sum_{\gamma \in P_{d-2}\backslash D_d} \left( y^{-1} \sum_{\substack{n_i{}^{\hat{j}} \in \mathbb{Z}^2 \\ \det(n_i{}^{\hat{j}}) \neq 0}} e^{-\pi \Omega_2^{ij} y v_{\hat{k}\hat{l}} n_i{}^{\hat{k}} n_j{}^{\hat{l}}} \right)\Bigg|_\gamma = \sum_{\substack{\mathcal{Q}_i \in S_+ \\ \mathcal{Q}_i \times \mathcal{Q}_j = 0 \\ \mathcal{Q}_1 \wedge \mathcal{Q}_2 \neq 0}} \left| \frac{\gcd(\mathcal{Q}_1 \wedge \mathcal{Q}_2)}{v(\mathcal{Q}_1 \wedge \mathcal{Q}_2)} \right| e^{-\pi \Omega_2^{ij} v(\mathcal{Q}_i) \cdot v(\mathcal{Q}_j)} \tag{C.43}$$

Using this representation one can rewrite the differential equation

$$(\Delta_{D_d} + \tfrac{(d-2)(1+2\epsilon)(d-2\epsilon)}{2} - 12)\left( \sum_{\gamma \in P_{d-2}\backslash D_d} \left[ y^{-1-2\epsilon}\Big( \widetilde{A}_\epsilon(U) - \frac{3\xi(2\epsilon)^2}{6+\epsilon-2\epsilon^2} E_{2\epsilon\Lambda_1}^{SL(2)}(U) \Big) \right]\Big|_\gamma \right)$$

$$= -\frac{3}{2}\pi^{-2\epsilon}\Gamma(\epsilon)^2 \sum_{\substack{\mathcal{Q}_i \in S_+ \\ \mathcal{Q}_i \times \mathcal{Q}_j = 0 \\ \mathcal{Q}_1 \wedge \mathcal{Q}_2 \neq 0}} \left| \frac{\gcd(\mathcal{Q}_1 \wedge \mathcal{Q}_2)}{v(\mathcal{Q}_1 \wedge \mathcal{Q}_2)} \right| \frac{1}{|v(\mathcal{Q}_1)|^{2\epsilon} |v(\mathcal{Q}_1)|^{2\epsilon}} \tag{C.44}$$

$$= -\frac{3}{2}\pi^{-2\epsilon}\Gamma(\epsilon)^2 \sum_{\gamma \in D_d/P_d} \sideset{}{'}\sum_{n_1 \in \mathbb{Z}} \sideset{}{'}\sum_{\substack{q_2 \in \wedge^2 \mathbb{Z}^d \\ q_2 \times q_2 = 0}} \sum_{n_2 \in \mathbb{Z}} \left| \frac{\gcd(q_2)}{y^{2\frac{d-2}{d}} v(q_2)} \right| \frac{1}{(y^2 n_1^2)^\epsilon} \frac{1}{(y^2(n_2+(q_2,a))^2 + y^{2\frac{d-4}{d}}|v(q_2)|^2)^\epsilon}\Bigg|_\gamma$$

$$= -6\xi(2\epsilon) \sum_{\gamma \in D_d/P_d} \left( \xi(2-2\epsilon) y^{-2+4\epsilon\frac{2-d}{d}} E_{\epsilon\Lambda_2}^{SL(d)} \right.$$

$$\left. + 2 y^{-2\frac{d-1}{d}+4\epsilon\frac{1-d}{d}} \sideset{}{'}\sum_{\substack{q \in \wedge^2 \mathbb{Z}^d \\ q \times q = 0}} \gcd(q)\sigma_{2\epsilon-1}(q) \frac{K_{\frac{1}{2}-\epsilon}(2\pi y^{-\frac{2}{d}}|v(q)|)}{|v(q)|^{\frac{1}{2}+\epsilon}} e^{2\pi i(q,a)} \right)\Bigg|_\gamma$$

The terms in the bracket are recognised as the Fourier expansion of the Eisenstein series $E_{\Lambda_d}^{D_d}$ with respect to the parabolic $P_d$, up to a constant term proportional to $E_{(1+\epsilon)\Lambda_1}^{SL(d)}$. Thus the previous result can be continued as

$$= -6\xi(2\epsilon) \sum_{\gamma \in D_d/P_d} \left( \xi(2) y^{-2\epsilon} E_{\Lambda_d}^{D_d} - \xi(2+2\epsilon) y^{-2(1+\epsilon)\frac{d-2}{d}} E_{(1+\epsilon)\Lambda_1}^{SL(d)} + \mathcal{O}(\epsilon) \right)$$

$$= -6\xi(2\epsilon)\xi(2)\big( E_{\Lambda_d}^{D_d} - E_{\Lambda_{d-1}}^{D_d} + \mathcal{O}(\epsilon) \big) = \mathcal{O}(\epsilon^0) , \tag{C.45}$$

where we use $E_{\Lambda_d}^{D_d} = E_{\Lambda_{d-1}}^{D_d}$ in the last step. After dividing out by $\xi(4\epsilon)$, the source term in the Laplace equation therefore vanishes.

Assuming that the source terms for higher order invariant differential operators vanish in the same way, we conclude that $\mathcal{I}_d^{(3a)}$ must be proportional to an $SO(d, d)$ Eisenstein series satisfying to the same differential equations as the one appearing in the same perturbative limit of the counterterm in (1.28). Since the counterterm in (1.28)

$$
\frac{2\pi^2}{9} \int_{\mathcal{G}} \frac{\mathrm{d}^3\Omega_2}{|\Omega_2|^{\frac{6-d-2\epsilon}{2}}} \frac{E_{-3\Lambda_1}^{SL(2)}(\tau)}{V} \theta_{\Lambda_{d+1}}^{(3)}(\phi, \Omega_2)
$$
$$
\sim \quad \frac{4\pi^2}{9} \frac{\xi(4\epsilon - 2)}{\xi(4\epsilon)} \xi(5 - 2\epsilon)\xi(3 + 2\epsilon) \frac{\xi(8)}{\xi(7)} g_D^{-\frac{24+8\epsilon}{8-d}+2+4\epsilon} E_{(-\frac{3}{2}+\epsilon)\Lambda_{d-2}+4\Lambda_d}^{D_d} \tag{C.46}
$$

(where $\theta_{\Lambda_{d+1}}^{(3)}$ corresponds to the third layer contribution (3.42)) satisfies by construction to the same differential equations as the function $\mathcal{I}_d(\phi, \epsilon)$ without the source terms, it follows that $\mathcal{I}_d^{(3a)}$ must be proportional to $E_{-\frac{3}{2}\Lambda_{d-4}+4\Lambda_d}^{D_d}$. We shall now argue that the coefficient of proportionality is such that this Eisenstein series cancels in the renormalised coupling (1.28).

To this aim, we compute the first non-trivial contribution to the double lattice sum (C.43) in the parabolic $P_d$. In this limit, one get a first contribution

$$
\sum_{\substack{n_i \in \mathbb{Z} \\ q_i \in \wedge^2 \mathbb{Z}^d \\ q_i \times q_j = 0 \\ q_1 \wedge q_2 = 0 \\ n_{[i}q_{j]} \neq 0}} \left| \frac{\gcd(2n_{[i}q_{j]})}{y^{2\frac{d-2}{d}} 2n_{[i}v(q_{j]})} \right| e^{-\pi\Omega_2^{ij}\left(y^2(n_i+(q_i,a))(n_j+(q_j,a))+y^{2\frac{d-4}{d}}v(q_i)\cdot v(q_j)\right)} \tag{C.47}
$$

$$
= \sum_{\gamma \in SL(2)/P_1} \sideset{}{'}\sum_{\substack{q \in \wedge^2 \mathbb{Z}^2 \\ q \times q = 0}} \sideset{}{'}\sum_{n_2 \in \mathbb{Z}} \sum_{n_1 \in \mathbb{Z}} \left| \frac{\gcd(q)}{y^{2\frac{d-2}{d}} v(q)} \right| e^{-\frac{\pi}{V}\left(\frac{(n_1+(q,a)+\tau_1 n_2)^2}{\tau_2}+\tau_2 n_2^2+\frac{1}{\tau_2}y^{2\frac{d-4}{d}}|v(q)|^2\right)} \Bigg|_\gamma
$$

$$
= \sum_{\gamma \in SL(2)/P_1} \sideset{}{'}\sum_{\substack{q \in \wedge^2 \mathbb{Z}^2 \\ q \times q = 0}} \sideset{}{'}\sum_{n_2 \in \mathbb{Z}} \sum_{\tilde{n}_1 \in \mathbb{Z}} \Bigg[ \left| \frac{\gcd(q)}{y^{2\frac{d-2}{d}} v(q)} \right| y^{-1}\sqrt{V\tau_2} e^{-\frac{\pi}{V}\left(\tau_2 y^2 n_2^2+\frac{1}{\tau_2}y^{2\frac{d-4}{d}}|v(q)|^2\right)-\pi V\tau_2 y^{-2}\tilde{n}_1^2}
$$

$$
\times e^{2\pi i \tilde{n}_1((q,a)+\tau_1 n_2)} \Bigg] \Bigg|_\gamma
$$

and the associated constant term is therefore

$$
\int_0^\infty \frac{\mathrm{d}V}{V^{\frac{1}{2}+2\epsilon}} \int_0^\infty \frac{\mathrm{d}\tau_2}{\tau_2^{\frac{3}{2}}} \int_{-\frac{1}{2}}^{\frac{1}{2}} \mathrm{d}\tau_1 A(\tau) \sideset{}{'}\sum_{\substack{q \in \wedge^2 \mathbb{Z}^2 \\ q \times q = 0}} \sideset{}{'}\sum_{n_2 \in \mathbb{Z}} \left| \frac{\gcd(q)}{y^{3-\frac{4}{d}} v(q)} \right| e^{-\frac{\pi}{V}\left(\tau_2 y^2 n_2^2+\frac{1}{\tau_2}y^{2\frac{d-4}{d}}|v(q)|^2\right)}
$$

$$
= 2\xi(2\epsilon)\xi(-1 + 2\epsilon) y^{-2-4\epsilon\frac{d-2}{d}} E_{\epsilon\Lambda_2}^{SL(d)} + \frac{\xi(-4+2\epsilon)\xi(3+2\epsilon)}{6} y^{2\frac{8-d}{d}-4\epsilon\frac{d-2}{d}} E_{(2+\epsilon)\Lambda_2}^{SL(d)} + \mathcal{O}(e^{-y^{-\frac{2}{d}}}). \tag{C.48}
$$

Comparing with a similar term in the expansion of $E_{-\frac{3}{2}\Lambda_{d-4}+4\Lambda_d}^{D_d}$ fixes the coefficient of the second term in (3.44) to match the one of (C.46) in (D.2). In contrast, the first term does not appear

in the expansion of $E^{D_d}_{-\frac{3}{2}\Lambda_{d-4}+4\Lambda_d}$, instead it is recognised as a constant term of the minimal Eisenstein series $E^{D_d}_{\Lambda_d}$. Indeed it is not a solution to the homogeneous Laplace equation, and we therefore expect that this term will cancel against another contribution at the next order in level expansion for the charges, including either $q_i \wedge q_j \neq 0$ or $p_i \in \wedge^4 \mathbb{Z}^d$.

We may also consider the tensorial differential equations (2.10) on the renormalised expression (1.28). Using the reduction formula for Whittaker coefficients of the series $E^{D_d}_{-\frac{3}{2}\Lambda_{d-4}+4\Lambda_d}$ [24, 76], one computes that it admits non-zero Whittaker vectors of type $A_2 A_1$ for $d \geq 5$. This implies that this function admits Fourier coefficients outside of the wavefront set determined by the tensorial equation (2.10), that allows at most for Whittaker vectors of type $A_2$. We conclude that the naïve pole subtraction prescription for $\mathcal{I}_d(\phi, \epsilon)$ and the counterterm (C.46) violate the tensorial equation (2.10), but the term proportional to $E^{D_d}_{-\frac{3}{2}\Lambda_{d-4}+4\Lambda_d}$ drops out in the renormalised function (1.28), such that it satisfies the required supersymmetry Ward identities.

# D  Integrating $\Xi^{E_{d+1}}_{\Lambda_{d+1}}(\tau, \phi, r)$ against an Eisenstein series

In order to determine the weak coupling and decompactification limit asymptotic expansions of the renomalised coupling (1.31), we shall repeat the computations of Sections 3 and 4 with $A(\tau)$ replaced by an Eisenstein series $E^{SL(2)}_{s\Lambda_1}$. Although these expansions can be easily computed by using Langlands's constant term formula, it is nevertheless instructive to obtain them in this way, since it will allow us to identify the constant terms that we could not compute directly using the method of Sections 3 and 4 as contributions of specific double cosets in the Weyl group. Since these contributions can be expressed as theta-lifts of $E^{SL(2)}_{s\Lambda_1}$ up to an overall factor of $\frac{1}{\xi(4\epsilon)}$, it is plausible that the analogous contributions for $A(\tau)$ will also vanish in the limit $\epsilon \to 0$, justifying our previous computations.

With these motivations in mind, let us consider the function

$$\mathcal{I}^{E_{d+1}}_{\Lambda_{d+1}}\big(E^{SL(2)}_{(4+\delta)\Lambda_1}, d + 2\epsilon - 2\big) = \int_{\mathcal{F}} \frac{\mathrm{d}\tau_1 \mathrm{d}\tau_2}{\tau_2^2} E^{SL(2)}_{(4+\delta)\Lambda_1} \Xi^{E_{d+1}}_{\Lambda_{d+1}}(\tau, \phi, d + 2\epsilon - 2) \qquad \text{(D.1)}$$

$$= \quad \xi(d + 2\epsilon - 6 - \delta)\xi(d + 2\epsilon + 1 + \delta)E^{E_{d+1}}_{\frac{d+2\epsilon-6-\delta}{2}\Lambda_d + (4+\delta)\Lambda_{d+1}}$$

$$\underset{d=4,5,6}{=} \frac{\xi(2s_{d+1} + \delta - 2\epsilon)\xi(2s_{d+1} + 3 - d + \delta - 2\epsilon)\xi(d + 1 + 2\epsilon + \delta)}{\xi(4 + \delta - 2\epsilon)} E^{E_{d+1}}_{(s_{d+1} + \frac{\delta}{2} - \epsilon)\Lambda_H + 2\epsilon\Lambda_{d+1}}$$

This function reproduces the last two terms in (1.31) upon setting either $\delta = 0$ first or $\epsilon = 0$ first and then writing $\delta = 2\epsilon$. Recall that $s_{d+1} = \frac{7}{2}, \frac{9}{2}, 6$ for $d = 4, 5, 6$.

## D.1  Weak coupling limit

We shall first write the result of Langlands constant term formula. We refer to [65, 24] for the precise statement of this formula in terms of double cosets in the Weyl group. We shall use the convention that $\Lambda_{d-k}$ stands for the trivial vanishing weight when $k = d$, and an Eisenstein series including a weight $\Lambda_{d-k}$ for $k > d$ vanishes. Using Langlands' functional relations between

Eisenstein series, one obtains the following formula valid for all $d \leq 6$

$$\xi(d + 2\epsilon - 6 - \delta)\xi(d + 2\epsilon + 1 + \delta)E^{E_{d+1}}_{\frac{d+2\epsilon-6-\delta}{2}\Lambda_d + (4+\delta)\Lambda_{d+1}} \tag{D.2}$$

$$\sim \ g_D^{-\frac{24+8\epsilon}{8-d}}\Bigg(\xi(d_\epsilon - 6 - \delta)\xi(d_\epsilon + 1 + \delta)g_D^4 E^{D_d}_{(4+\delta)\Lambda_1 + \frac{d_\epsilon-6-\delta}{2}\Lambda_2}$$

$$+ \xi(2 + \delta + 2\epsilon)\xi(-5 - \delta + 2\epsilon)g_D^{2\epsilon}\Big(g_D^{-1-\delta}E^{D_d}_{(-\frac{1+\delta}{2}-\epsilon)\Lambda_{d-1}+2\epsilon\Lambda_d} + \frac{\xi(7+2\delta)}{\xi(8+2\delta)}g_D^{6+\delta}E^{D_d}_{(\frac{6+\delta}{2}-\epsilon)\Lambda_{d-1}+2\epsilon\Lambda_d}\Big)$$

$$+ \frac{\xi(4\epsilon - 2)}{\xi(4\epsilon)}g_D^{2+4\epsilon}\xi(-4 - \delta + 2\epsilon)\xi(3 + \delta + 2\epsilon)E^{D_d}_{(-\frac{3+\delta}{2}+\epsilon)\Lambda_{d-2}+(4+\delta)\Lambda_d}$$

$$+ \frac{\xi(4\epsilon - 5)}{\xi(4\epsilon)}g_D^{1+6\epsilon}\Big(\frac{\xi(-8-\delta+2\epsilon)\xi(-6-\delta+2\epsilon)\xi(6+\delta+2\epsilon)}{\xi(-3-\delta+2\epsilon)}g_D^{-7-\delta}E^{D_d}_{(-\frac{6+\delta}{2}+\epsilon)\Lambda_{d-5}+(\frac{6+\delta}{2}+\epsilon)\Lambda_d}$$

$$+ \frac{\xi(-1+\delta+2\epsilon)\xi(1+\delta+2\epsilon)\xi(-1-\delta+2\epsilon)}{\xi(4+\delta+2\epsilon)}\frac{\xi(7+2\delta)}{\xi(8+2\delta)}g_D^{\delta}E^{D_d}_{(\frac{1+\delta}{2}+\epsilon)\Lambda_{d-5}+(-\frac{1+\delta}{2}+\epsilon)\Lambda_d}\Big)$$

$$+ \delta_{d,6}g_D^{2+4\epsilon}\Big(\frac{\xi(-11-\delta+2\epsilon)\xi(-8-\delta+2\epsilon)\xi(7+\delta+2\epsilon)}{\xi(-3-\delta+2\epsilon)}g_D^{-14-2\delta} + \frac{\xi(-4+\delta+2\epsilon)\xi(-1+\delta+2\epsilon)\xi(-\delta+2\epsilon)}{\xi(4+\delta+2\epsilon)}\frac{\xi(7+2\delta)}{\xi(8+2\delta)}g_D^{2\delta}\Big)E^{D_d}_{2\epsilon\Lambda_1}$$

$$+ \delta_{d,6}\frac{\xi(4\epsilon - 5)\xi(4\epsilon - 8)}{\xi(4\epsilon)\xi(4\epsilon - 4)}\frac{\xi(-7-\delta+2\epsilon)\xi(\delta+2\epsilon)\xi(-4-\delta+2\epsilon)\xi(3+\delta+2\epsilon)}{\xi(-3-\delta+2\epsilon)\xi(4+\delta+2\epsilon)}g_D^{-6+8\epsilon}E^{D_d}_{(4+\delta)\Lambda_1+(-\frac{4+\delta}{2}+\epsilon)\Lambda_2}\Bigg)\ .$$

This formula can be recast as a sum of contributions of the different layers of charges as in Section 3.2, with $A(\tau)$ replaced by the Eisenstein series $E^{SL(2)}_{(4+\delta)\Lambda_1}$,

$$\mathcal{I}^{E_{d+1}}_{\Lambda_{d+1}}\big(E^{SL(2)}_{(4+\delta)\Lambda_1}, d - 2 + 2\epsilon\big) \tag{D.3}$$

$$\sim \ g_D^{-\frac{24+8\epsilon}{8-d}}\Bigg(\xi(d + 2\epsilon - 6 - \delta)\xi(d + 2\epsilon + 1 + \delta)g_D^4\,\mathcal{I}^{D_d}_{\Lambda_1}\big(E^{SL(2)}_{(4+\delta)\Lambda_1}, d - 2 + 2\epsilon\big)$$

$$+ g_D^4 \int_0^\infty \frac{\mathrm{d}V}{V^{-1+2\epsilon}} \int_0^L \frac{\mathrm{d}\tau_2}{\tau_2^2} \int_{-\frac{1}{2}}^{\frac{1}{2}} \mathrm{d}\tau_1 E^{SL(2)}_{(4+\delta)\Lambda_1}$$

$$\times \sum_{\gamma \in P_d \backslash D_d} \left(y^{-4}\sum_{n\geq 1}{\sum_{q_a \in \mathbb{Z}^d}}' e^{-\frac{\pi}{V\tau_2}g_D^{-2}y^2 n^2 - \frac{\pi V}{\tau_2}y^{-\frac{4}{d}}u^{ab}q_a q_b}\right)\Bigg|_\gamma$$

$$+ \frac{\xi(4\epsilon - 2)}{\zeta(4\epsilon)}g_D^{2+4\epsilon}\int_{\mathcal{F}} \frac{\mathrm{d}^2\tau}{\tau_2^2} E^{SL(2)}_{(4+\delta)\Lambda_1} {\sum_{\substack{Q_i \in S_+ \\ Q_i \times Q_j = 0 \\ Q_1 \wedge Q_2 \neq 0}}}' \left|\frac{\gcd(Q_1 \wedge Q_2)}{v(Q_1 \wedge Q_2)}\right|\left(\frac{\tau_2}{v(Q_1 + \tau Q_2)\cdot v(Q_1 + \bar{\tau}Q_2)}\right)^{2\epsilon}$$

$$+ \frac{\xi(4\epsilon - 5)}{\xi(4\epsilon)}g_D^{1+6\epsilon}\Big(\frac{\xi(-8-\delta+2\epsilon)\xi(-6-\delta+2\epsilon)\xi(6+\delta+2\epsilon)}{\xi(-3-\delta+2\epsilon)}g_D^{-7-\delta}E^{D_d}_{(-\frac{6+\delta}{2}+\epsilon)\Lambda_{d-5}+(\frac{6+\delta}{2}+\epsilon)\Lambda_d}$$

$$+ \frac{\xi(-1+\delta+2\epsilon)\xi(1+\delta+2\epsilon)\xi(-1-\delta+2\epsilon)}{\xi(4+\delta+2\epsilon)}\frac{\xi(7+2\delta)}{\xi(8+2\delta)}g_D^{\delta}E^{D_d}_{(\frac{1+\delta}{2}+\epsilon)\Lambda_{d-5}+(-\frac{1+\delta}{2}+\epsilon)\Lambda_d}\Big)$$

$$+ \delta_{d,6}\int_{\mathcal{G}} \frac{\mathrm{d}^3\Omega_2}{|\Omega_2|^3}\Big(E^{Sp(4)}_{(4+\delta)\Lambda_1+\frac{2\epsilon-3-\delta}{2}\Lambda_2} - \frac{E^{SL(2)}_{(4+\delta)\Lambda_1}}{V^{1+2\epsilon}}\Big)\sum_{\gamma \in P_2 \backslash D_6}\sum_{\substack{p_\alpha \in S_+ \\ k \geq 1 \\ \gcd(k,p)=1 \\ m^{ij} \in \mathbb{Z}^2 \\ \det m \neq 0}}\frac{|\Omega_2|^2}{\tilde{y}_1^{3+\epsilon}\tilde{y}_2^{1+\epsilon}}e^{-\pi\sqrt{\frac{\tilde{y}_2}{\tilde{y}_1}}\Omega_{2ij}^{-1}v_{i\hat{j}}m^{i\hat{i}}m^{j\hat{j}}}\Bigg|_\gamma\Bigg)\ .$$

The first layer of charges, as in (3.13), gives the first line in both (D.2) and (D.3), while the second layer, as in (3.24), gives the second line in (D.2) and the second and third lines in (D.3). Note that for an Eisenstein series there are no exponentially suppressed contributions to the constant terms as they do arise for $A(\tau)$, see (3.29). The fourth layer of charges gives the fourth line in (D.3) as in (3.43), which can be identified with the third line in (D.2). For $d \leq 4$ this exhausts all terms. For $d = 5$ and 6, it follows by elimination that the fourth layer of charges gives the fourth and fifth lines in (D.2), that we have reproduced as such in (D.3). Although we have not been able to compute these latter using the double lattice sum, the overall factor of $\frac{\xi(4\epsilon-5)}{\xi(4\epsilon)}$ suggests that the total contribution from the fourth layer of charge to the abelian Fourier coefficients vanishes.

For $d = 6$, the same computation as in (3.59) gives the last line in (D.3). Using Langlands's constant term formula for the $Sp(4, \mathbb{R})$ Langlands–Eisenstein series

$$
E^{Sp(4)}_{(4+\delta)\Lambda_1 + \frac{2\epsilon-3-\delta}{2}\Lambda_2} \sim \frac{E^{SL(2)}_{(4+\delta)\Lambda_1}}{V^{1+2\epsilon}} + \frac{\xi(-4-\delta+2\epsilon)}{\xi(-3-\delta+2\epsilon)} \frac{E^{SL(2)}_{2\epsilon\Lambda_1}}{V^{5+\delta}} + \frac{\xi(7+2\delta)\xi(3+\delta+2\epsilon)}{\xi(8+2\delta)\xi(4+\delta+2\epsilon)} \frac{E^{SL(2)}_{2\epsilon\Lambda_1}}{V^{-2-\delta}}
$$
$$
+ \frac{\xi(4\epsilon-1)}{\xi(4\epsilon)} \frac{\xi(-4-\delta+2\epsilon)\xi(3+\delta+2\epsilon)}{\xi(-3-\delta+2\epsilon)\xi(4+\delta+2\epsilon)} \frac{E^{SL(2)}_{(4+\delta)\Lambda_1}}{V^{2-2\epsilon}} \quad (D.4)
$$

one obtains three contributions which, upon using the identification of the sum over coprime $p_\alpha$ and $k$ as a Poincaré sum over $P_1 \backslash SO(5,5)$ as in (3.59), give the two last lines in (D.2). We have not proved rigorously that one can indeed write the sum over $p_\alpha$ and $k$ of the lattice sum over $k_2 \mathit{\Pi}_{4,4}[k_1] \oplus \mathit{\Pi}_{2,2}[k_1 k_2^2]$ as a Poincaré sum over $P_1 \backslash SO(5,5)$ of a lattice sum over $\mathit{\Pi}_{6,6}$ that we used in (3.56), neither do we have a proof of the identities (3.57) and (3.58). The fact that the three constant terms match provides a strong consistency check that one has indeed

$$
\sum_{\substack{\gamma \in P_2 \backslash Sp(4,\mathbb{Z}) \\ \det C(\gamma) \neq 0}} \frac{E^{SL(2)}_{(4+\delta)\Lambda_1}}{V^{1+2\epsilon}}\Bigg|_\gamma \sim \frac{\xi(-4-\delta+2\epsilon)}{\xi(-3-\delta+2\epsilon)} \frac{E^{SL(2)}_{2\epsilon\Lambda_1}}{V^{5+\delta}} + \frac{\xi(7+2\delta)\xi(3+\delta+2\epsilon)}{\xi(8+2\delta)\xi(4+\delta+2\epsilon)} \frac{E^{SL(2)}_{2\epsilon\Lambda_1}}{V^{-2-\delta}}
$$
$$
+ \frac{\xi(4\epsilon-1)}{\xi(4\epsilon)} \frac{\xi(-4-\delta+2\epsilon)\xi(3+\delta+2\epsilon)}{\xi(-3-\delta+2\epsilon)\xi(4+\delta+2\epsilon)} \frac{E^{SL(2)}_{(4+\delta)\Lambda_1}}{V^{2-2\epsilon}} \,, \quad (D.5)
$$

in agreement with (3.57), (3.58) for an Einsenstein series and that one can indeed use (3.56).

Note that for generic $\delta$, the limit $\epsilon = 0$ is regular and produces the adjoint Eisenstein series constant terms (with $\delta$ replaced by $2\epsilon$ in order to match the notations in (1.31))

$$
\frac{\xi(2s_{d+1}+2\epsilon)\xi(2s_{d+1}+3-d+2\epsilon)\xi(d+1+2\epsilon)}{\xi(4+2\epsilon)} E^{E_{d+1}}_{(s_{d+1}+\epsilon)\Lambda_H} \quad (D.6)
$$
$$
\sim g_D^{-\frac{24}{8-d}} \Bigg( \xi(d-6-2\epsilon)\xi(d+1+2\epsilon) g_D^4 E^{D_d}_{(4+2\epsilon)\Lambda_1 + \frac{d-6-2\epsilon}{2}\Lambda_2}
$$
$$
+ \xi(2+2\epsilon)\xi(6+2\epsilon)\Big( g_D^{-1-2\epsilon} E^{D_d}_{-(\frac{1}{2}+\epsilon)\Lambda_{d-1}} + \frac{\xi(7+4\epsilon)}{\xi(8+4\epsilon)} g_D^{6+2\epsilon} E^{D_d}_{(3+\epsilon)\Lambda_{d-1}} \Big)
$$
$$
+ \delta_{d,6} g_D^2 \Big( \frac{\xi(12+2\epsilon)\xi(9+2\epsilon)\xi(7+2\epsilon)}{\xi(4+2\epsilon)} g_D^{-14-4\epsilon} + \frac{\xi(-4+2\epsilon)\xi(-1+2\epsilon)\xi(1+2\epsilon)}{\xi(4+2\epsilon)} \frac{\xi(7+4\epsilon)}{\xi(8+4\epsilon)} g_D^{4\epsilon} \Big) \Bigg) \,.
$$

## D.2 Decompactification limit

We shall first write the result of Langlands's constant term formula for $d \leq 7$, using again the convention that the weight $\Lambda_7$ vanishes for $d = 6$, and an Eisenstein series including a weight $\Lambda_7$ vanishes for $d < 6$. Applying the functional relations between Eisenstein series, one obtains the following formula valid for all $d \leq 7$

$$\xi(d + 2\epsilon - 6 - \delta)\xi(d + 2\epsilon + 1 + \delta) E^{E_{d+1}}_{\frac{d+2\epsilon-6-\delta}{2}\Lambda_d + (4+\delta)\Lambda_{d+1}} \tag{D.7}$$

$$\sim R^{\frac{12+4\epsilon}{8-d}} \left( \xi(d + 2\epsilon - 7 - \delta)\xi(d + 2\epsilon + \delta) E^{E_d}_{\frac{d+2\epsilon-7-\delta}{2}\Lambda_{d-1}+(4+\delta)\Lambda_d} \right.$$

$$+ \xi(d + 2\epsilon - 6 - \delta)\xi(d + 2\epsilon + 1 + \delta) R^{d+2\epsilon} \left( R^\delta E^{E_d}_{\frac{d+2\epsilon-6-\delta}{2}\Lambda_d} + \frac{\xi(7+2\delta)}{\xi(8+2\delta)} R^{-7-\delta} E^{E_d}_{\frac{d+2\epsilon+1+\delta}{2}\Lambda_d} \right)$$

$$+ \frac{\xi(4\epsilon-1)\xi(3+\delta+2\epsilon)\xi(5+\delta-2\epsilon)}{\xi(4\epsilon)\xi(4+\delta+2\epsilon)\xi(4+\delta-2\epsilon)} \xi(d - 5 - \delta - 2\epsilon)\xi(d + 2 + \delta - 2\epsilon)$$

$$\times \left( R^{d+1+\delta-2\epsilon} E^{E_d}_{\frac{4\epsilon-1}{2}\Lambda_1+\frac{d-5-\delta-2\epsilon}{2}\Lambda_d} + \frac{\xi(7+2\delta)}{\xi(8+2\delta)} R^{d-6-\delta-2\epsilon} E^{E_d}_{\frac{4\epsilon-1}{2}\Lambda_1+\frac{d+2+\delta-2\epsilon}{2}\Lambda_d} \right)$$

$$+ \frac{\xi(4\epsilon+1-d)}{\xi(4\epsilon)} \xi(6 - 2\epsilon + \delta)\xi(-1 - 2\epsilon - \delta) R^{d-1-4\epsilon} E^{E_d}_{(4+\delta)\Lambda_1 - \frac{3+\delta-2\epsilon}{2}\Lambda_3}$$

$$+ \frac{\xi(4\epsilon-5)\xi(4\epsilon-9)\xi(-1+\delta+2\epsilon)\xi(9+\delta-2\epsilon)}{\xi(4\epsilon)\xi(4\epsilon-4)\xi(4+\delta+2\epsilon)\xi(4+\delta-2\epsilon)} \xi(5 - \delta - 2\epsilon)\xi(12 + \delta - 2\epsilon) R^{d+3-6\epsilon}$$

$$\times \left( \frac{\xi(-d-10-\delta+6\epsilon)}{\xi(-16-\delta+6\epsilon)} R^{7+\delta} E^{E_d}_{\frac{5-\delta-2\epsilon}{2}\Lambda_6-\frac{9-\delta-6\epsilon}{2}\Lambda_7} + \frac{\xi(7+2\delta)}{\xi(8+2\delta)} \frac{\xi(-d-3+\delta+6\epsilon)}{\xi(-9+\delta+6\epsilon)} R^{-\delta} E^{E_d}_{\frac{12+\delta-2\epsilon}{2}\Lambda_6-\frac{16+\delta-6\epsilon}{2}\Lambda_7} \right)$$

$$+ \delta_{d,7} \left( \left( \frac{\xi(-16-\delta+2\epsilon)\xi(-12-\delta+2\epsilon)\xi(-8-\delta+2\epsilon)\xi(8+\delta+2\epsilon)}{\xi(-7-\delta+2\epsilon)\xi(-3-\delta+2\epsilon)} \frac{\xi(-17-\delta+6\epsilon)}{\xi(-16-\delta+6\epsilon)} R^{14+2\delta} \right. \right.$$

$$\left. + \frac{\xi(7+2\delta)}{\xi(8+2\delta)} \frac{\xi(\delta-9+2\epsilon)\xi(\delta-5+2\epsilon)\xi(\delta-1+2\epsilon)\xi(1-\delta+2\epsilon)}{\xi(\delta+2\epsilon)\xi(4+\delta+2\epsilon)} \frac{\xi(\delta-10+6\epsilon)}{\xi(\delta-9+6\epsilon)} \right) R^{10-4\epsilon} E^{E_7}_{2\epsilon\Lambda_7}$$

$$+ \frac{\xi(4\epsilon-9)\xi(4\epsilon-12)}{\xi(4\epsilon)\xi(4\epsilon-4)} \frac{\xi(-17-\delta+6\epsilon)\xi(\delta-10+6\epsilon)}{\xi(-16-\delta+6\epsilon)\xi(\delta-9+6\epsilon)} \frac{\xi(-8-\delta+2\epsilon)\xi(-4-\delta+2\epsilon)\xi(\delta-1+2\epsilon)\xi(3+\delta+2\epsilon)}{\xi(4+\delta-2\epsilon)\xi(4+\delta+2\epsilon)} R^{18-8\epsilon} E^{E_7}_{-\frac{4+\delta-2\epsilon}{2}\Lambda_6+(4+\delta)\Lambda_7}$$

$$+ \frac{\xi(4\epsilon-9)\xi(4\epsilon-13)\xi(4\epsilon-17)}{\xi(4\epsilon)\xi(4\epsilon-4)\xi(4\epsilon-8)} \frac{\xi(-17-\delta+6\epsilon)\xi(\delta-10+6\epsilon)}{\xi(-16-\delta+6\epsilon)\xi(\delta-9+6\epsilon)} R^{20-10\epsilon} \left( \frac{\xi(17+\delta-2\epsilon)\xi(13+\delta-2\epsilon)\xi(9+\delta-2\epsilon)\xi(8+\delta+2\epsilon)}{\xi(4+\delta-2\epsilon)\xi(-7+\delta+2\epsilon)} R^\delta E^{E_7}_{\frac{8+\delta+2\epsilon}{2}\Lambda_7} \right.$$

$$\left. \left. \left. + \frac{\xi(7+2\delta)}{\xi(8+2\delta)} \frac{\xi(2+\delta-2\epsilon)\xi(1-\delta+2\epsilon)\xi(\delta-9+2\epsilon)\xi(\delta-5+2\epsilon)}{\xi(4+\delta+2\epsilon)\xi(\delta+2\epsilon)} R^{-7-\delta} E^{E_7}_{\frac{1-\delta+2\epsilon}{2}\Lambda_7} \right) \right) \right).$$

For $d \leq 6$, the last five lines drop out and this formula can be rewritten as a sum of contributions of the various layers of charges in Section 4.2 as[35]

$$\mathcal{I}^{E_{d+1}}_{\Lambda_{d+1}} \left( E^{SL(2)}_{(4+\delta)\Lambda_1}, d_\epsilon - 2 \right) \sim R^{\frac{12+4\epsilon}{8-d}} \left( \mathcal{I}^{E_d}_{\Lambda_d} \left( E^{SL(2)}_{(4+\delta)\Lambda_1}, d_\epsilon - 3 \right) \right.$$

$$+ \int_0^\infty \frac{\mathrm{d}V}{V^{d_\epsilon-2}} \int_0^L \frac{\mathrm{d}\tau_2}{\tau_2^2} \int_{-\frac{1}{2}}^{\frac{1}{2}} \mathrm{d}\tau_1 \, E^{SL(2)}_{(4+\delta)\Lambda_1} \sum_{n=1}^\infty \sideset{}{'}\sum_{\substack{Q \in M^{E_d}_{\Lambda_d} \\ Q \times Q = 0}} e^{-\frac{\pi V}{\tau_2} R^2 n^2 - \frac{\pi}{V\tau_2}|Z(Q)|^2}$$

---

[35]Note that the general theory of Fourier coefficients for Eisenstein series induced from cusp forms predicts precisely the structure of $L$-functions appearing in (D.8), suggesting that this formula should hold for any Hecke eigenfunction.

$$+ \frac{\xi(4\epsilon-1)L^\star(E^{SL(2)}_{(4+\delta)\Lambda_1}, -\frac{1}{2}+2\epsilon)}{\xi(4\epsilon)L^\star(E^{SL(2)}_{(4+\delta)\Lambda_1}, \frac{1}{2}+2\epsilon)} \int_0^\infty \frac{dV}{V^{d-1-2\epsilon}} \int_0^L \frac{d\tau_2}{\tau_2^2} \int_{-\frac{1}{2}}^{\frac{1}{2}} d\tau_1\, E^{SL(2)}_{(4+\delta)\Lambda_1}$$

$$\times \sum_{\gamma \in P_1 \backslash E_d} \left( y^{3-d-2\epsilon} \sum_{n=1}^\infty \underset{(q,q)=0}{\sideset{}{'}\sum_{q \in II_{d-1,d-1}}} e^{-\frac{\pi V}{\tau_2}R^2 y n^2 - \frac{\pi}{V\tau_2}g(q,q)} \right)\Bigg|_\gamma$$

$$+ \frac{\xi(4\epsilon+1-d)L^\star(E^{SL(2)}_{(4+\delta)\Lambda_1}, -\frac{3}{2}+2\epsilon)}{\xi(4\epsilon)L^\star(E^{SL(2)}_{(4+\delta)\Lambda_1}, \frac{1}{2}+2\epsilon)} R^{d-1-4\epsilon} \mathcal{I}^{E_d}_{\Lambda_1}\left(E^{SL(2)}_{(4+\delta)\Lambda_1}, 1+2\epsilon\right)$$

$$+ \delta_{d,6} \frac{\xi(4\epsilon-5)\xi(4\epsilon-9)L^\star(E^{SL(2)}_{(4+\delta)\Lambda_1}, -\frac{9}{2}+2\epsilon)}{\xi(4\epsilon)\xi(4\epsilon-4)L^\star(E^{SL(2)}_{(4+\delta)\Lambda_1}, \frac{1}{2}+2\epsilon)} R^{5-4\epsilon}$$

$$\times \int_0^\infty \frac{dV}{V^{9-2\epsilon}} \int_0^L \frac{d\tau_2}{\tau_2^2} \int_{-\frac{1}{2}}^{\frac{1}{2}} d\tau_1\, E^{SL(2)}_{(4+\delta)\Lambda_1} \sum_{n=1}^\infty \underset{Q \times Q=0}{\sideset{}{'}\sum_{Q \in M^{E_d}_{\Lambda_d}}} e^{-\frac{\pi V}{\tau_2}R^2 n^2 - \frac{\pi}{V\tau_2}|Z(Q)|^2} \Bigg). \qquad \text{(D.8)}$$

The first layer of charges does not contribute for an Eisenstein series because the regularised integral over $\mathcal{F}_L$ of the product of two Eisenstein series vanishes [77]. The first line of (D.7) is reproduced from the first line of (D.8) that comes from the second layer of charges $\mathcal{I}^{(2a)}_d$ with $Q_i = 0$ in (3.15), while $\mathcal{I}^{(2b)}_d$ gives the second line in (D.8) that reproduces the second line in (D.7). Eq. (4.24) might suggest that the third layer of charges does not contribute to the constant terms, but the use of (4.20) is only valid at $\epsilon = 0$ and there is a non-zero contribution at $\epsilon \neq 0$. Using the Langlands's constant term formula for the $Sp(4)$ Siegel–Eisenstein series in the Fourier–Jacobi expansion $P_1 \backslash Sp(4)$

$$E^{Sp(4)}_{(4+\delta)\Lambda_1 + \frac{2\epsilon-3-\delta}{2}\Lambda_2} \sim t^{\frac{5+\delta}{2}+\epsilon} E^{SL(2)}_{\frac{2\epsilon-3-\delta}{2}\Lambda_1} + \frac{\xi(7+2\delta)}{\xi(8+2\delta)} t^{-\frac{2+\delta}{2}+\epsilon} E^{SL(2)}_{\frac{4+\delta+2\epsilon}{2}\Lambda_1}$$
$$+ \frac{\xi(4\epsilon-1)}{\xi(4\epsilon)} \frac{\xi(-4-\delta+2\epsilon)\xi(3+\delta+2\epsilon)}{\xi(-3-\delta+2\epsilon)\xi(4+\delta+2\epsilon)} \left( t^{\frac{6+\delta}{2}-\epsilon} E^{SL(2)}_{-\frac{2+\delta+2\epsilon}{2}\Lambda_1} + \frac{\xi(7+2\delta)}{\xi(8+2\delta)} t^{-\frac{1+\delta}{2}-\epsilon} E^{SL(2)}_{\frac{5+\delta-2\epsilon}{2}\Lambda_1} \right) \quad \text{(D.9)}$$

one obtains

$$\mathcal{I}^{(3)}_d[E^{SL(2)}_{(4+\delta)\Lambda_1}] \qquad\qquad\qquad\qquad\qquad\qquad\qquad\qquad\qquad\qquad\qquad\qquad\qquad \text{(D.10)}$$

$$= \frac{1}{2} R^{2\frac{d+2\epsilon-2}{8-d}} \sum_{\gamma \in P_1 \backslash E_d} \left( y^{2-d-2\epsilon} \int_{\mathcal{F}_2} \frac{d^6\Omega}{|\Omega_2|^3} E^{Sp(4)}_{(4+\delta)\Lambda_1 + \frac{2\epsilon-3-\delta}{2}\Lambda_2} \Gamma_{d,d,2}(\Omega, R y^{\frac{1}{2}}) \right)\Bigg|_\gamma$$

$$- \frac{1}{2} R^{\frac{12+4\epsilon}{8-d}} \sum_{\gamma \in P_1 \backslash E_d} \left( y^{3-d_\epsilon} \int_{\mathcal{G}} \frac{d^3\Omega_2}{|\Omega_2|^{\frac{6-d}{2}-\epsilon}} E^{SL(2)}_{(4+\delta)\Lambda_1} \underset{(q_i,q_j)=0}{\sum_{\substack{n^i \in \mathbb{Z} \\ q_i \in II_{d-1,d-1}}}} e^{-\pi \Omega_{2ij}^{-1} R^2 y n^i n^j - \pi \Omega_2^{ij} g(q_i,q_j) + 2\pi i(n^i q_i, a)} \right)\Bigg|_\gamma$$

$$\sim \frac{\xi(4\epsilon-1)}{\xi(4\epsilon)} \frac{\xi(-4-\delta+2\epsilon)\xi(3+\delta+2\epsilon)}{\xi(-3-\delta+2\epsilon)\xi(4+\delta+2\epsilon)} R^{\frac{12+4\epsilon}{8-d}} \int_0^\infty \frac{dV}{V^{d-1-2\epsilon}} \int_0^L \frac{d\tau_2}{\tau_2^2} \int_{-\frac{1}{2}}^{\frac{1}{2}} d\tau_1 \left( \tau_2^{4+\delta} + \frac{\xi(7+2\delta)}{\xi(8+2\delta)} \tau_2^{-3-\delta} \right)$$

$$\times \sum_{\gamma \in P_1 \backslash E_d} \left( y^{3-d-2\epsilon} \sum_{n=1}^\infty \underset{(q,q)=0}{\sideset{}{'}\sum_{q \in II_{d-1,d-1}}} e^{-\frac{\pi V}{\tau_2}R^2 y n^2 - \frac{\pi}{V\tau_2}g(q,q)} \right)\Bigg|_\gamma$$

$$\sim \frac{\xi(4\epsilon-1)\xi(3+\delta+2\epsilon)\xi(5+\delta-2\epsilon)}{\xi(4\epsilon)\xi(4+\delta+2\epsilon)\xi(4+\delta-2\epsilon)}\xi(d-5-\delta-2\epsilon)\xi(d+2+\delta-2\epsilon)R^{\frac{12+4\epsilon}{8-d}}$$

$$\times\left(R^{d+1+\delta-2\epsilon}E^{E_d}_{\frac{4\epsilon-1}{2}\Lambda_1+\frac{d-5-\delta-2\epsilon}{2}\Lambda_d}+\frac{\xi(7+2\delta)}{\xi(8+2\delta)}R^{d-6-\delta-2\epsilon}E^{E_d}_{\frac{4\epsilon-1}{2}\Lambda_1+\frac{d+2+\delta-2\epsilon}{2}\Lambda_d}\right)$$

such that the third layer of charges gives the second and third lines in (D.8) that gives the third and fourth lines in (D.7). Consistently with (4.20), these two terms appear with a factor of $\frac{\xi(4\epsilon-1)}{\xi(4\epsilon)}$ that vanishes at $\epsilon=0$. We expect that the integral of $A(\tau)$ will give the same result from (3.58) such that this contribution vanishes in the renormalised function (1.31).

The fourth layer of charges gives the fifth line in (D.8) using (4.33), where the ratio of $L$-functions (2.30) comes from the presence of the factor $|v(Q_1\wedge Q_2)|^{-2}$ in (4.33) that shifts the weight $s$ in the Eisenstein series but not in the parameter of the $L$-function in (2.27). This term reproduces the fifth line in (D.7). By elimination, the last two lines in (D.8), which reproduces the sixth and seventh lines in (D.7), must come from the fifth layer of charges that only exists in $d\geq 6$.

The same analysis holds for $d=7$ for the first five layers of charges as we show in Appendix E. The sixth layer of charges that only appears for $d=7$ can be computed as in (E.27), (E.33) to give

$$\frac{1}{2}\int_{\mathcal{G}}\frac{\mathrm{d}^3\Omega_2}{|\Omega_2|^{\frac{1}{2}}}\int_{\mathbb{Z}^3\backslash\mathbb{R}^3}\mathrm{d}^3\Omega_1\left(E^{Sp(4)}_{(4+\delta)\Lambda_1+\frac{2\epsilon\cdot 3-\delta}{2}\Lambda_2}-\frac{E^{SL(2)}_{(4+\delta)\Lambda_1}}{V^{1+2\epsilon}}\right)\tag{D.11}$$

$$\times\sum_{\substack{m^i\in\mathbb{Z}^2\\ \det m\neq 0}}\sum_{q_i\in II_{5,5}}\tilde{R}^2\frac{e^{-\pi\Omega_{2ij}^{-1}\tilde{R}\tilde{v}^{i\hat{j}}m_{\hat{i}}^{~i}m_{\hat{j}}^{~j}-\pi\Omega_2^{ij}\tilde{u}_{ab}q_i^aq_j^b-\pi i\Omega_1^{ij}\eta_{ab}q_i^aq_j^b+2\pi i m_{\hat{j}}^{~i}q_i^a\tilde{a}_a^{\hat{j}}}}{\left(\left(y+\frac{v(\tilde{\ell},\tilde{\ell})}{R^2}\right)^2+\left(y+\frac{v(\tilde{\ell},\tilde{\ell})}{R^2}\right)\frac{y^{\frac{1}{2}}}{R^2}u(\tilde{p},\tilde{p})+\frac{1}{4}\frac{y}{R^4}u(\tilde{p}\gamma\tilde{p},\tilde{p}\gamma\tilde{p})\right)^{\frac{5}{2}+\epsilon}}$$

$$\sim\left(\frac{\xi(-16-\delta+2\epsilon)\xi(-12-\delta+2\epsilon)\xi(-8-\delta+2\epsilon)\xi(8+\delta+2\epsilon)}{\xi(-7-\delta+2\epsilon)\xi(-3-\delta+2\epsilon)}\frac{\xi(-17-\delta+6\epsilon)}{\xi(-16-\delta+6\epsilon)}R^{14+2\delta}\right.$$

$$\left.+\frac{\xi(7+2\delta)}{\xi(8+2\delta)}\frac{\xi(\delta-9+2\epsilon)\xi(\delta-5+2\epsilon)\xi(\delta-1+2\epsilon)\xi(1-\delta+2\epsilon)}{\xi(\delta+2\epsilon)\xi(4+\delta+2\epsilon)}\frac{\xi(\delta-10+6\epsilon)}{\xi(\delta-9+6\epsilon)}\right)R^{10-4\epsilon}E^{E_7}_{2\epsilon\Lambda_7}$$

$$+\frac{\xi(4\epsilon-9)\xi(4\epsilon-12)}{\xi(4\epsilon)\xi(4\epsilon-4)}\frac{\xi(-17-\delta+6\epsilon)\xi(\delta-10+6\epsilon)}{\xi(-16-\delta+6\epsilon)\xi(\delta-9+6\epsilon)}\frac{\xi(-8-\delta+2\epsilon)\xi(-4-\delta+2\epsilon)\xi(\delta-1+2\epsilon)\xi(3+\delta+2\epsilon)}{\xi(4+\delta-2\epsilon)\xi(4+\delta+2\epsilon)}R^{18-8\epsilon}E^{E_7}_{-\frac{4+\delta-2\epsilon}{2}\Lambda_6+(4+\delta)\Lambda_7}$$

By elimination one then concludes that the last seventh layer of charges contributes the last two lines in (D.7) for $d=7$.

In the limit $\epsilon\to 0$ at generic $\delta$ (a posteriori set to $\delta=2\epsilon$) one obtains from (D.7) the constant terms of the adjoint Eisenstein series

$$\xi(d-6-2\epsilon)\xi(d+1+2\epsilon)E^{E_{d+1}}_{\frac{d-6-2\epsilon}{2}\Lambda_d+(4+2\epsilon)\Lambda_{d+1}}\tag{D.12}$$

$$=\frac{\xi(2s_{d+1}+2\epsilon)\xi(2s_{d+1}+3-d-\delta_{d,7}+2\epsilon)\xi(d+1+\delta_{d,7}+2\epsilon)}{\xi(4+2\epsilon)}E^{E_{d+1}}_{(s_{d+1}+\epsilon)\Lambda_H}$$

$$\sim R^{\frac{12}{8-d}}\left(\frac{\xi(2s_d+2\epsilon)\xi(2s_d+4-d+2\epsilon)\xi(d+2\epsilon)}{\xi(4+2\epsilon)}E^{E_d}_{(s_d+\epsilon)\Lambda_H}\right.$$

$$+\xi(d-6-2\epsilon)\xi(d+1+2\epsilon)\left(R^{d+2\epsilon}E^{E_d}_{\frac{d-6-2\epsilon}{2}\Lambda_d}+\frac{\xi(7+4\epsilon)}{\xi(8+4\epsilon)}R^{d-7-2\epsilon}E^{E_d}_{\frac{d+1+2\epsilon}{2}\Lambda_d}\right)$$

$$\left.+\delta_{d,7}\left(\frac{\xi(18+2\epsilon)\xi(13+2\epsilon)\xi(9+2\epsilon)}{\xi(4+2\epsilon)}R^{24+4\epsilon}+\frac{\xi(7+4\epsilon)}{\xi(8+4\epsilon)}\frac{\xi(2\epsilon-10)\xi(2\epsilon-5)\xi(2\epsilon-1)}{\xi(4+2\epsilon)}R^{10-4\epsilon}\right)\right).$$

## D.3 Comments on layers with vanishing contribution

We have claimed in Section 5 that all the constant terms in the weak coupling and the large radius limit with an overall factor of $\frac{1}{\xi(4\epsilon)}$ vanish for the renormalised function (1.28) at $\epsilon \to 0$. We further argued that the whole layer of charges generating them, including contributions to the Fourier coefficients, vanishes in the limit $\epsilon \to 0$. In this section, we shall discuss the corresponding terms for the Eisenstein series (D.1).

For $d = 4, 5, 6$, there are additional poles in $\frac{1}{\epsilon}$ when one first sets $\delta = 0$, such that Formulae (D.6) and (D.13) are not valid at $\delta = 0$. We must therefore be more careful in the analysis of the contributions in $\frac{1}{\xi(4\epsilon)}$. We shall first discuss the constant terms and then the Fourier coefficients.

### Decompactification limit

Let us first discuss the constant terms in the decompactification limit (D.7). The term

$$\frac{\xi(4\epsilon-1)\xi(3+2\epsilon)\xi(5-2\epsilon)}{\xi(4\epsilon)\xi(4+2\epsilon)\xi(4-2\epsilon)}\xi(d-5-2\epsilon)\xi(d+2-2\epsilon)\frac{\xi(7)}{\xi(8)}R^{d-6-2\epsilon}E^{E_d}_{\frac{4\epsilon-1}{2}\Lambda_1+\frac{d+2-2\epsilon}{2}\Lambda_d} \tag{D.13}$$

has a finite limit at $\epsilon \to 0$ in $d = 4, 5, 6$ and

$$\frac{\xi(4\epsilon-1)\xi(3+2\epsilon)\xi(5-2\epsilon)}{\xi(4\epsilon)\xi(4+2\epsilon)\xi(4-2\epsilon)}\xi(d-5-2\epsilon)\xi(d+2-2\epsilon)R^{d+1-2\epsilon}E^{E_d}_{\frac{4\epsilon-1}{2}\Lambda_1+\frac{d-5-2\epsilon}{2}\Lambda_d} \tag{D.14}$$

is also finite in $d = 5$. Assuming the conjectured expansion (3.58) is correct, these terms cancel in (1.28). Next, the term

$$\frac{\xi(4\epsilon+1-d)}{\xi(4\epsilon)}\xi(6-2\epsilon)\xi(-1-2\epsilon)R^{d-1-4\epsilon}\,E^{E_d}_{-3\Lambda_1+(2+\epsilon)\Lambda_3} \tag{D.15}$$

also admits a finite limit at $\epsilon \to 0$ in $d = 4, 5$, thanks to the functional identity

$$E^{E_d}_{-3\Lambda_1+(2+\epsilon)\Lambda_3} \underset{d=4,5}{=} \frac{\xi(1+2\epsilon)}{\xi(4+2\epsilon)}E^{E_d}_{(-\frac{3}{2}+\epsilon)\Lambda_1+(d-4+\epsilon)\Lambda_2}\,, \tag{D.16}$$

and the finiteness of $E^{E_d}_{-\frac{3}{2}\Lambda_1+(d-4)\Lambda_2}$.

By the same reasoning as in Section C.3, we expect that the leading contribution from $\mathcal{I}^{E_{d+1}}_{\Lambda_{d+1}}(A, d-2+2\epsilon)$ will include the same $L$-function factors as for the Eisenstein series in (D.8), such that

$$\sum_{\substack{\mathcal{Q}_i \in M^{E_d}_{\Lambda_1} \\ \mathcal{Q}_i \times \mathcal{Q}_j = 0 \\ \mathcal{Q}_1 \wedge \mathcal{Q}_2 \neq 0}} \left|\frac{\gcd(\mathcal{Q}_1 \wedge \mathcal{Q}_2)}{v(\mathcal{Q}_1 \wedge \mathcal{Q}_2)}\right|^2 \int_{\mathcal{G}} \frac{]\mathrm{d}^3\Omega_2}{|\Omega_2|^{2-\epsilon}}A(\tau)e^{-\pi\Omega_2^{ij}G(\mathcal{Q}_i,\mathcal{Q}_j)} \tag{D.17}$$

$$= \frac{\xi(6)\xi(2)}{\xi(4)^2}\mathcal{I}^{E_d}_{\Lambda_1}(A, 1+2\epsilon) + \mathcal{O}(\epsilon^0) = \frac{\xi(6)\xi(2)}{3}E^{E_d}_{-3\Lambda_1+(2+\epsilon)\Lambda_3} + \mathcal{O}(\epsilon^0)\,,$$

reproducing (4.35). We checked explicitly in the decompactification limit that the last equality holds for $d = 4$.

For $d = 6$ we moreover have a finite contribution from the fifth layer of charges,

$$\frac{\xi(4\epsilon-5)\xi(4\epsilon-9)\xi(-1+2\epsilon)\xi(9-2\epsilon)}{\xi(4\epsilon)\xi(4\epsilon-4)\xi(4+2\epsilon)\xi(4-2\epsilon)}\xi(5-2\epsilon)\xi(12-2\epsilon)R^{9-6\epsilon}E^{E_6}_{(6-\epsilon)\Lambda_6} \underset{\epsilon\to 0}{\to} \frac{2\xi(2)\xi(6)\xi(10)}{\xi(4)}R^9\,. \tag{D.18}$$

This contribution comes from the constant term in $\tau_2^{-3}$ of the Eisenstein series $E^{SL(2)}_{4\Lambda_1}(\tau)$ that also appears in $A(\tau)$, so it is expected to cancel in (1.28).

**Weak coupling limit**

Turning to the weak coupling limit (D.2) , we have already seen that the contribution

$$\frac{\xi(4\epsilon - 5)}{\xi(4\epsilon)} g_D^{1+6\epsilon} \frac{\xi(-8+2\epsilon)\xi(-6+2\epsilon)\xi(6+2\epsilon)}{\xi(-3+2\epsilon)} g_D^{-7} E_{(-3+\epsilon)\Lambda_{d-5}+(3+\epsilon)\Lambda_d}^{D_d} \qquad (D.19)$$

of the third layer of charges has a finite limit in $d = 4, 5, 6$, but we argued in Appendix C.3 that it cancels in (1.28). The contributions from the fourth layer of charges for $d = 5, 6$ do not vanish at $\delta = 0$ in the limit $\epsilon \to 0$. The two terms contribute for $d = 5$ and only the first for $d = 6$. We expect them to cancel in (1.28). The contribution from the fifth layer of charges gives a finite contribution in $d = 6$

$$\frac{\xi(4\epsilon - 5)\xi(4\epsilon - 8)}{\xi(4\epsilon)\xi(4\epsilon - 4)} \frac{\xi(-7+2\epsilon)\xi(2\epsilon)\xi(-4+2\epsilon)\xi(3+2\epsilon)}{\xi(-3+2\epsilon)\xi(4+2\epsilon)} g_D^{-6+8\epsilon} E_{4\Lambda_1+(-2+\epsilon)\Lambda_2}^{D_d} \qquad (D.20)$$

which cancels in (1.28) provided the expansion (3.58) is correct.

**Borel Fourier coefficients**

We want now to argue that the Fourier coefficients associated to the layers of charges that give constant terms with an overall factor of $\frac{1}{\xi(4\epsilon)}$, also include a similar factor and generically vanish. For this one can use a reduction formula for abelian Fourier coefficients in the Borel decomposition, the so-called (degenerate) Whittaker coefficients or Whittaker vectors [76, 24].

The abelian Fourier coefficients of the $SL(2)$ Eisenstein series can be written as

$$W_{s\Lambda_1}^{A_1}(n\alpha_1) \equiv \int_0^1 d\tau_1 e^{-2\pi i n \tau_1} E_{s\Lambda_1}^{SL(2)} = \frac{W_s(n)}{\xi(2s)} = \frac{1}{\xi(2s)} \sqrt{\tau_2} \frac{\sigma_{2s-1}(|n|)}{|n|^{s-\frac{1}{2}}} K_{s-\frac{1}{2}}(2\pi|n|\tau_2) . \quad (D.21)$$

Similarly for $SL(r + 1)$ Eisenstein series, the generic abelian Fourier coefficients in the Borel decomposition[36] take the form

$$W_{\sum s_k \Lambda_k}^{A_r}(\sum_k n_k \alpha_k) \equiv \int_U da\, e^{-2\pi i \sum_k n_k a_k} E_{\sum s_k \Lambda_k}^{SL(r+1)} = \frac{W_{\{s_k\}}(n_k)}{\prod_{k=0}^{r-1} \prod_{j=1}^{r-k} \xi(2\sum_{i=j}^{k+j} s_i - k)} , \quad (D.22)$$

for $\{s_k\}$ such that none of the $\xi$ arguments vanish, i.e. $\prod_{k=0}^{r-1} \prod_{j=1}^{r-k}(2\sum_{i=j}^{k+j} s_i - k) \neq 0$. The functions $W_{\{s_k\}}(n_k)$ are Eulerian functions [37] on the Cartan torus and are regular for all $s_k$. In particular the reduction of the wavefront set at special values of $\{s_k\}$ is a consequence of the vanishing $\xi$ factors only. The abelian Fourier coefficients of an arbitrary Eisenstein series over a reductive group $G$

$$W_{\sum s_k \Lambda_k}^G(\sum_k n_k \alpha_k) \equiv \int_U da\, e^{-2\pi i \sum_k n_k a_k} E_{\sum s_k \Lambda_k}^G \qquad (D.23)$$

can be written in a similar way. It can however happen that the 'instanton charges' $n_k$ on the simple roots are not all non-zero, in which case one is therefore computing a *degenerate* Whittaker

---

[36]The product of $\xi$ functions in the denominator is due to the product over all positive roots of $SL(r + 1)$, see also [78].

[37]i.e. they can be written as infinite products of $p$-adic Whittaker vectors for all primes $p$, including a special function contribution from the 'archimedean prime at infinity'.

coefficient. The resulting expression is then not necessarily Eulerian but can be given by a sum of different terms in a way described by Weyl cosets according to a reduction formula [24]. If the subset of non-zero $n_k$ corresponds to a subgroup $SL(r_1+1) \times SL(r_2+1) \times \cdots \times SL(r_N+1)$ of $G$, the corresponding Whittaker coefficient is said to be of Bala–Carter type $A_{r_1} A_{r_2} \ldots A_{r_N}$. It is generally given by a sum over Weyl elements acting on $\sum s_k \Lambda_k$ and subsequent projection to the subgroup $SL(r_1+1) \times SL(r_2+1) \times \cdots \times SL(r_N+1)$ generating products of terms of the generic type (D.22) with coefficients depending on the $s_k$.

We shall now analyse some of the Whittaker vectors for the Eisenstein series (D.1). We will only display the $\xi$ factors and will schematically write $f_k$ for some products of functions $W_{\{s_k\}}(n_k)$.

- For $D_5$, using the reduction formula one computes the Whittaker vector of type $A_2 A_1 A_1$

$$W^{D_5}_{(\epsilon-1)\Lambda_3+4\Lambda_5}(n_1\alpha_1 + n_2\alpha_2 + m\alpha_4 + p\alpha_5) = \frac{f_1 W_{2+\epsilon}(m) + \frac{\xi(7)}{\xi(8)} f_2 W_{\epsilon-\frac{3}{2}}(m)}{\xi(4\epsilon)\xi(4+2\epsilon)\xi(4-2\epsilon)\xi(5+2\epsilon)\xi(3-2\epsilon)} , \quad \text{(D.24)}$$

where, at the identity in the Cartan torus,

$$f_1 = W_{\epsilon-2,2+\epsilon}(n_1\alpha_1 + n_2\alpha_2)W_{\epsilon-2}(p) , \quad f_2 = W_{\frac{3}{2}+\epsilon,\epsilon-\frac{3}{2}}(n_1\alpha_1 + n_2\alpha_2)W_{\frac{3}{2}+\epsilon}(p) . \quad \text{(D.25)}$$

One recognises the structure of the terms in the third lines of (D.7) that have the same factor of $\frac{1}{\xi(4\epsilon)\xi(4+2\epsilon)\xi(4-2\epsilon)}$, suggesting that they come from the third layer of charges in the decompactification limit. One can understand that the type $A_2 A_1 A_1$ corresponds to generic Fourier coefficients in the decompactification limit. In this case the Fourier coefficients in the $\overline{\mathbf{10}}$ of $P_5$

$$\mathfrak{so}(5,5) \cong \ldots \oplus (\mathfrak{gl}_1 \oplus \mathfrak{sl}_5)^{(0)} \oplus \overline{\mathbf{10}}^{(2)} \quad \text{(D.26)}$$

supported on the simple root $p\alpha_5$, of type $A_1$ have a Levi stabiliser $\mathfrak{sl}_3 \oplus \mathfrak{sl}_2$, so the generic Fourier coefficient that can be related to a Whittaker vector corresponds to a Fourier coefficient of the generic $SL(3) \times SL(2)$ Levi functions that are by definition of type $A_2 A_1$, leading to a total Bala–Carter type $A_2 A_1 A_1$. The Whittaker vectors of type $A_2 A_1$ have a structure similar to (D.24) where $W_{2+\epsilon}(m)$ and $W_{\epsilon-\frac{3}{2}}(m)$ are replaced by the constant terms (at the identity) of the corresponding $SL(2)$ Eisenstein series $E^{SL(2)}_{(2+\epsilon)\Lambda_1}$ and $E^{SL(2)}_{(\epsilon-3/2)\Lambda_1}$, together with one additional new contribution

$$W^{D_5}_{(\epsilon-1)\Lambda_3+4\Lambda_5}(n_1\alpha_1 + n_2\alpha_2 + p\alpha_5) = \frac{f_1(\xi(4+2\epsilon) + \xi(3+2\epsilon)) + \frac{\xi(7)}{\xi(8)} f_2(\xi(4-2\epsilon) + \xi(5-2\epsilon))}{\xi(4\epsilon)\xi(4+2\epsilon)\xi(4-2\epsilon)\xi(5+2\epsilon)\xi(3-2\epsilon)}$$
$$+ \frac{\xi(2+2\epsilon)\xi(6-2\epsilon)W_{4,\epsilon-\frac{3}{2}}(n_1\alpha_1 + n_2\alpha_2)W_{2\epsilon-1}(p)}{\xi(4\epsilon)\xi(4+2\epsilon)\xi(4-2\epsilon)\xi(5+2\epsilon)\xi(3-2\epsilon)\xi(8)} . \quad \text{(D.27)}$$

The new contribution has a factor of $W_{2\epsilon-1}(p)$ associated to the Eisenstein series $E^{SL(2)}_{(2\epsilon-1)\Lambda_1}$, whose corresponding constant term includes a factor $\xi(4\epsilon-3)$, and is understood to correspond to the fourth layer of charges in the decompactification limit. One may check that for $\epsilon \to 0$, the $A_2$ Whittaker vectors collapse to the Eulerian Whittaker vectors of the adjoint series, so that all Fourier coefficients associated to the third and the fourth layer of charges indeed vanish at $\epsilon = 0$.

- For $E_6$, using the reduction formula one computes the Whittaker vector of type $A_2A_1A_1$

$$W^{E_6}_{(\epsilon-\frac{1}{2})\Lambda_5+4\Lambda_6}(n_1\alpha_1 + m\alpha_2 + n_2\alpha_3 + p\alpha_6) = \frac{f_1 W_{2+\epsilon}(m) + \frac{\xi(7)}{\xi(8)}f_2 W_{\epsilon-\frac{3}{2}}(m)}{\xi(4\epsilon)\xi(4+2\epsilon)\xi(4-2\epsilon)\xi(6+2\epsilon)\xi(2-2\epsilon)} \quad \text{(D.28)}$$

that can similarly be attributed to the third layer of charges, and does vanish in the limit $\epsilon \to 0$. One finds for type $A_2A_1$

$$\begin{aligned}W^{E_6}_{(\epsilon-\frac{1}{2})\Lambda_5+4\Lambda_6}(n_1\alpha_1 + n_2\alpha_3 + p\alpha_6) &= \frac{f_1(\xi(4+2\epsilon)+\xi(3+2\epsilon)) + \frac{\xi(7)}{\xi(8)}f_2(\xi(4-2\epsilon)+\xi(5-2\epsilon))}{\xi(4\epsilon)\xi(4+2\epsilon)\xi(4-2\epsilon)\xi(6+2\epsilon)\xi(2-2\epsilon)} \\ &+ \frac{\xi(2+2\epsilon)\xi(6-2\epsilon)W_{4,\epsilon-\frac{3}{2}}(n_1\alpha_1+n_2\alpha_2)W_{2\epsilon-\frac{3}{2}}(p)}{\xi(4\epsilon)\xi(4+2\epsilon)\xi(4-2\epsilon)\xi(6+2\epsilon)\xi(2-2\epsilon)} \\ &+ \frac{\xi(7-2\epsilon)f_3 + \xi(1+2\epsilon)\frac{\xi(7)}{\xi(8)}f_4}{\xi(4\epsilon)\xi(4+2\epsilon)\xi(4-2\epsilon)\xi(6+2\epsilon)\xi(2-2\epsilon)} \ . \end{aligned} \quad \text{(D.29)}$$

Again, one can attribute the first line to the third layer of charges, the second line to the fourth layer of charges and the third line to the third layer of charges. All these contributions vanish in the limit $\epsilon \to 0$, but the last term associated to the third layer of charges. One finds also a Whittaker vector of type $A_2A_1$ that does not vanish at $\epsilon \to 0$ ,

$$W^{E_6}_{(\epsilon-\frac{1}{2})\Lambda_5+4\Lambda_6}(n_1\alpha_1 + m\alpha_2 + n_2\alpha_3) = \frac{\xi(1+2\epsilon)\frac{\xi(7)}{\xi(8)}f_5}{\xi(4\epsilon)\xi(4+2\epsilon)\xi(4-2\epsilon)\xi(6+2\epsilon)\xi(2-2\epsilon)} + \mathcal{O}(\epsilon) \ . \quad \text{(D.30)}$$

This Fourier coefficient can be identified as a $A_2A_1$ type Fourier coefficient of $E^{E_5}_{\frac{4\epsilon-1}{2}\Lambda_1 + \frac{7+\delta-2\epsilon}{2}\Lambda_5}$ in (D.7) and is therefore associated to the third layer of charges. This shows that the wavefront set of $E^{E_6}_{(\epsilon-\frac{1}{2})\Lambda_5+4\Lambda_6}\big|_{\epsilon^0}$ is of type $A_2A_1$ and not $A_2$, and therefore this function cannot be a solution to the tensorial differential equation (2.10). In order for the renormalised function (1.28) to satisfy this equation, this contribution must cancel against the Fourier coefficients of the theta lift of $A(\tau)$.

- For $E_7$ the Eisenstein series $\xi(2\epsilon)E^{E_{d+1}}_{\epsilon\Lambda_6+4\Lambda_7}$ is of Bala–Carter type $A_3A_1$ (with wavefront set associated to the smallest nilpotent orbit of that type). The corresponding Whittaker vector is

$$\xi(2\epsilon)\xi(7+2\epsilon)W^{E_7}_{\epsilon\Lambda_6+4\Lambda_7}(n_1\alpha_2 + n_2\alpha_4 + n_3\alpha_5 + p\alpha_7) = \frac{\frac{1}{\xi(8)}f}{\xi(4\epsilon)\xi(4\epsilon-4)\xi(4+2\epsilon)\xi(4-2\epsilon)} \quad \text{(D.31)}$$

consistently with the fifth and last layer of charges contribution in (D.7), that includes the same denominator.

Turning to the $A_2A_1A_1$ type, we find

$$\begin{aligned}&\xi(2\epsilon)\xi(7+2\epsilon)W^{E_7}_{\epsilon\Lambda_6+4\Lambda_7}(n_1\alpha_1 + m\alpha_2 + n_2\alpha_3 + p\alpha_7) \\ &= \frac{f_1 + \frac{\xi(7)}{\xi(8)}f_2}{\xi(4\epsilon)\xi(4+2\epsilon)\xi(4-2\epsilon)} + \frac{\xi(4\epsilon-5)(\tilde{f}_1 + \frac{\xi(7)}{\xi(8)}\tilde{f}_2)}{\xi(4\epsilon)\xi(4\epsilon-4)\xi(4+2\epsilon)\xi(4-2\epsilon)} \ , \end{aligned} \quad \text{(D.32)}$$

where the first term comes from the third layer of charges as in (D.24) and (D.29), while the second comes from the fifth layer of charges. One finds again that the $A_2 A_1$ Whittaker vector

$$
\begin{aligned}
&\xi(2\epsilon)\xi(7+2\epsilon)W^{E_7}_{\epsilon\Lambda_6+4\Lambda_7}(n_1\alpha_1+n_2\alpha_3+p\alpha_7) \\
&= \frac{\xi(4\epsilon-5)}{\xi(4\epsilon)\xi(4\epsilon-4)\xi(4+2\epsilon)\xi(4-2\epsilon)} \cdot \frac{1}{\xi(8)}\Big(\xi(2\epsilon)\xi(7)(f_3+f_4)+\xi(1+2\epsilon)\xi(7-2\epsilon)f_4(\epsilon)\Big) \\
&\quad + \frac{1}{\xi(4\epsilon)\xi(4\epsilon-4)\xi(4+2\epsilon)\xi(4-2\epsilon)} \cdot \frac{\xi(7)}{\xi(8)}\xi(2\epsilon-5)\xi(1+2\epsilon)f_3(\epsilon)+\mathcal{O}(\epsilon) \quad \text{(D.33)}
\end{aligned}
$$

vanishes at $\epsilon \to 0$. For $E_7$ all the $A_2 A_1$ type Whittaker vectors are in the same Weyl orbit and therefore vanish in the limit $\epsilon \to 0$. For $W^{E_7}_{\epsilon\Lambda_6+4\Lambda_7}(n_1\alpha_1+m\alpha_2+n_2\alpha_3)$ the contribution from (D.30) coming from the second layer of charges in (D.7) cancels agains the same contribution from $E^{E_6}_{\frac{4\epsilon-1}{2}\Lambda_1+(4-\epsilon)\Lambda_6}$ coming from the third layer of charges in (D.7). We conclude that $\xi(2\epsilon)E^{E_{d+1}}_{\epsilon\Lambda_6+4\Lambda_7}\big|_{\epsilon^0}$ is of Bala–Carter type $A_2$, and must therefore satisfy the tensorial equation (2.10). We also checked that all the Whittaker vectors of Bala–Carter type $A_1 A_1 A_1 A_1$ vanish at $\epsilon \to 0$ and the ones of type $A_2$ collapse to the ones of the adjoint Eisenstein series (i.e. all terms proportional to $\frac{1}{\xi(4\epsilon)}$ cancel in the limit $\epsilon \to 0$).

To summarise, we have found that for $d=4$ and $d=6$, all the Whittaker vectors of Bala–Cater type exceeding $A_2$ vanish in the limit $\epsilon \to 0$ and the ones of type $A_2$ collapse to the Whittaker vectors of the adjoint Eisenstein series (D.6), while for $d=5$ we found that some Fourier coefficients of Bala–Carter type $A_2 A_1$ originating from the third layer of charges remain in the limit. We take this as further evidence for the fact that for all $d$, the fourth and fifth layers of charges do not contribute to the Fourier coefficients of the renormalised coupling (1.28) in the decompactification limit.

# E   Decompactification limit for $E_8$

In this appendix, we discuss the $d=7$ case considered in Section 5.4 in more detail. We first explain how to rewrite the charge sum in the double theta series (1.18) in this case. We extract the constant terms and abelian Fourier coefficients from the new layers that have no counter part for $d<7$. In particular we extract the summation measure for 1/8-BPS instantons in the decompactification limit, which is related to the index of BPS black holes in four dimensions.

We consider the lattice sum (1.18)

$$
\theta^{E_8}_{\Lambda_8}(\phi,\Omega_2) = \sum_{\substack{\mathcal{Q}_i\in M^{E_8}_{\Lambda_8}\\ \mathcal{Q}_i\times\mathcal{Q}_j=0}}' e^{-\pi\Omega_2^{ij}G(\mathcal{Q}_i,\mathcal{Q}_j)} , \tag{E.1}
$$

where $M^{E_8}_{\Lambda_8}$ is the lattice in the adjoint representation invariant under the Chevalley group $E_8(\mathbb{Z})$. Under the grading

$$
\mathfrak{e}_{8(8)} \cong \mathbf{1}^{(-2)} \oplus \mathbf{56}^{(-1)} \oplus \big(\mathfrak{gl}_1 \oplus \mathfrak{e}_{7(7)}\big)^{(0)} \oplus \mathbf{56}^{(1)} \oplus \mathbf{1}^{(2)} , \tag{E.2}
$$

one defines $\mathcal{Q} = (n, \Upsilon, \ell + Q, \Gamma, m) \in M_{\Lambda_8}^{E_8}$ with $n, m \in \mathbb{Z}$, $\Upsilon, \Gamma \in M_{\Lambda_7}^{E_7} \cong \mathbb{Z}^{56}$ and $Q \in \mathfrak{e}_7$ and $\ell \in \mathbb{Z}/2$ such that $Q + \ell$ acts on $M_{\Lambda_7}^{E_7}$ as a $\mathbb{Z}^{56 \times 56}$ matrix [50, §4.1]. For $\ell$ integer, $Q \in M_{\Lambda_1}^{E_7}$. The $Spin(16)$ invariant bilinear form is

$$
\begin{aligned}
G(\mathcal{Q}, \mathcal{Q}) = & \; R^{-4}\big(m + \langle a, \Gamma + b\Upsilon \rangle + 2b\ell + b^2 n + \tfrac{1}{2}\langle a, Q \cdot a \rangle + \tfrac{1}{4}\langle \Upsilon, \Delta'(a) \rangle + \tfrac{1}{4}n\Delta(a)\big)^2 \\
& + R^{-2}\big| Z\big(\Gamma + Q \cdot a + \ell a + \tfrac{1}{8}\Delta'(a, a, \Upsilon) + \tfrac{1}{2}a\langle a, \Upsilon \rangle + \tfrac{1}{4}n\Delta'(a) + b(\Upsilon + an)\big)\big|^2 \\
& + \big| V\big(Q + 2a \times \Upsilon + a \times an\big)\big|^2 + \big(\ell + \tfrac{1}{2}\langle a, \Upsilon \rangle + bn\big)^2 \\
& + R^2\big|Z(\Upsilon + an)\big|^2 + R^4 n^2 \; ,
\end{aligned}
\tag{E.3}
$$

where the axions $a \in \mathbb{R}^{56}, b \in \mathbb{R}$ parametrise the Heisenberg unipotent subgroup $\mathbb{R}^{56+1} \subset P_8$, $R \in \mathbb{R}^+$ the $GL(1)^+$ subgroup and the $SU(8)$ invariant norms $|V(Q)|$ and $|Z(\Gamma)|$ depend on $E_7/SU(8)$. Altogether they parametrise $P_8/SU(8) \cong E_8/Spin(16)$. Recall that $\Delta'(\Gamma) \in M_{\Lambda_7}^{E_7}$ is the gradient of the quartic invariant $\Delta(\Gamma) \in \mathbb{Z}$ and $\Delta'(\Gamma_1, \Gamma_2, \Gamma_3) \in M_{\Lambda_7}^{E_7}$ is the corresponding symmetric trilinear map. The $1/2$ BPS constraint $\mathcal{Q}_i \times \mathcal{Q}_j = 0$ is satisfied if and only if the symmetric product $\mathcal{Q}_i \otimes \mathcal{Q}_j|_{\Lambda_1} = 0$, and the highest weight $\Lambda_1$ module decomposes under (E.2) as

$$
\mathbf{3875} \cong \mathbf{133}^{(-2)} \oplus \big(\mathbf{912} \oplus \mathbf{56}\big)^{(-1)} \oplus \big(\mathbf{1539} \oplus \mathbf{133} \oplus \mathbf{1}\big)^{(0)} \oplus \big(\mathbf{912} \oplus \mathbf{56}\big)^{(1)} \oplus \mathbf{133}^{(2)} \; .
\tag{E.4}
$$

The five components of (E.4) can be written explicitly as [50]

$$
\begin{aligned}
i) & \quad \Upsilon \times \Upsilon = nQ \; , \\
ii) & \quad \tfrac{1}{3}Q \cdot \Upsilon = n\Gamma - \ell\Upsilon \; , \qquad 2\Upsilon \times (Q \cdot J) + \tfrac{2}{3}J \times (Q \cdot \Upsilon) = \langle J, \Upsilon \rangle Q \; , \quad \forall J \in M_{\Lambda_7}^{E_7} \; , \\
iii) & \quad Q^2 \cdot J = (3\ell^2 - 3mn + \tfrac{1}{2}\langle \Upsilon, \Gamma \rangle)J + 2\Upsilon\langle \Gamma, J \rangle - 2\Gamma\langle \Upsilon, J \rangle \; , \quad \forall J \in M_{\Lambda_7}^{E_7} \; , \quad \Upsilon \times \Gamma = \ell Q \; , \\
iv) & \quad \tfrac{1}{3}Q \cdot \Gamma = \ell\Gamma - m\Upsilon \; , \qquad 2\Gamma \times (Q \cdot J) + \tfrac{2}{3}J \times (Q \cdot \Gamma) = \langle J, \Gamma \rangle Q \; , \quad \forall J \in M_{\Lambda_7}^{E_7} \; , \\
v) & \quad \Gamma \times \Gamma = mQ \; ,
\end{aligned}
\tag{E.5}
$$

We consider the computation of $\theta_{\Lambda_8}^{E_8}$ layer by layer as in Section 4.2.

## E.1 Constant terms

### 1) The first, second, third and fourth layers

For the first four layers of charges, i.e. with $\Upsilon_i = n_i = 0$, one finds that $\ell_i = 0$ from the constraint and the computation is identical to the one carried out in Section 4.2. All the corresponding results in Section 4.2 apply to the case $d = 7$.

### 2) The fifth layer

Let us now consider $\Upsilon_i \neq 0$ and $n_i = 0$. First we shall discuss the case in which $\Upsilon_i$ are linearly dependent, so one can consider the $E_6$ grading:

$$
\begin{aligned}
\mathfrak{e}_{7(7)} & \cong \; \mathbf{27}^{(-2)} \oplus \big(\mathfrak{gl}_1 \oplus \mathfrak{e}_{6(6)}\big)^{(0)} \oplus \overline{\mathbf{27}}^{(2)} \; , \\
\mathbf{56} & \cong \; \mathbf{1}^{(-3)} \oplus \overline{\mathbf{27}}^{(-1)} \oplus \mathbf{27}^{(1)} \oplus \mathbf{1}^{(3)} \; , \\
\mathbf{912} & \cong \; \mathbf{78}^{(-3)} \oplus \big(\mathbf{351} \oplus \overline{\mathbf{27}}\big)^{(-1)} \oplus \big(\overline{\mathbf{351}} \oplus \mathbf{27}\big)^{(1)} \oplus \mathbf{78}^{(3)} \; ,
\end{aligned}
$$

$$\mathbf{1539} \cong \mathbf{27}^{(-4)} \oplus \left(\overline{\mathbf{351}} \oplus \mathbf{27}\right)^{(-2)} \oplus \left(\mathbf{1} \oplus \mathbf{78} \oplus \mathbf{650}\right)^{(0)} \oplus \left(\mathbf{351} \oplus \overline{\mathbf{27}}\right)^{(2)} \oplus \overline{\mathbf{27}}^{(4)} , \quad \text{(E.6)}$$

such that $\Upsilon_i = (0, 0, 0, n_i) \in \mathbf{1}^{(3)}$. Using the $\mathbf{912}$ constraint one obtains that $Q_i = (0, \varkappa_i + 0, \tilde{p}_i)$, such that $\varkappa_i \in \mathbb{Z}$. The $\mathbf{56}$ constraint then gives $\varkappa_i = -\ell_i$. Then the condition $Q^2 + 4\Gamma \wedge \Upsilon$ in the $\mathbf{1539}$ enforces that $\Gamma_i = (p_i^0, p_i, q_i, q_{i0})$ with the additional constraints

$$n_{(i} q_{j)} = -\tilde{p}_i \times \tilde{p}_j , \qquad n_{(i} p_{j)} = 2\varkappa_{(i} \tilde{p}_{j)} , \qquad n_{(i} p_{j)}^0 = -4\varkappa_i \varkappa_j , \qquad \text{(E.7)}$$

where $\varkappa_i = -\ell_i$ is in $\mathbb{Z}/2$. Then the constraint $\Upsilon \times \Gamma = \ell Q$ gives

$$\ell_{(i} \tilde{p}_{j)} = -\frac{1}{2} n_{(i} p_{j)} , \qquad n_{(i} p_{j)}^0 = 4\ell_{(i} \varkappa_{j)} , \qquad \text{(E.8)}$$

the constraint $Q\Gamma|_{\mathbf{912}} = 0$ gives

$$2\varkappa_{(i} p_{j)} + \tilde{p}_{(i} p_{j)}^0 = 0 , \quad p_{(i} \times \tilde{p}_{j)} = 2\varkappa_{(i} q_{j)} , \quad 4q_{(i} \times (\tilde{p}_{j)} \times y) = \tilde{p}_{(i} \operatorname{tr} q_{j)} y + \frac{1}{3} y \operatorname{tr} \tilde{p}_{(i} q_{j)} , \quad \text{(E.9)}$$

whereas the $\mathbf{56}$ component of $iv)$ gives

$$(\varkappa_{(i} - \ell_{(i)}) q_{0j)} + \tfrac{1}{3} \operatorname{tr} \tilde{p}_{(i} q_{j)} = -m_{(i} n_{j)} \quad (\varkappa_{(i} - 3\ell_{(i)}) q_{j)} = 2\tilde{p}_{(i} \times p_{j)} , \quad (\varkappa_{(i} + 3\ell_{(i)}) p_{j)} = p_{(i}^0 \tilde{p}_{j)} . \quad \text{(E.10)}$$

Finally $v)$ gives

$$p_i \times p_j = -p_{(i}^0 q_{j)} , \quad 2m_{(i} \varkappa_{j)} = 3p_{(i}^0 q_{0j)} - \operatorname{tr} p_{(i} q_{j)} , \quad 4q_{(i} \times (p_{j)} \times y) = p_{(i} \operatorname{tr} q_{j)} y + \frac{1}{3} y \operatorname{tr} p_{(i} q_{j)} \quad \text{(E.11)}$$

and

$$q_i \times q_j - p_{(i}^0 p_{j)} = n_{(i} \tilde{p}_{j)} . \qquad \text{(E.12)}$$

For $\ell_i = 0$ the solution is the same as for the fifth layer of charges in $E_7$. We are not able to extract the constant terms from the fifth layer, but the Langlands constant term formula suggests that they will involve a factor of $\frac{1}{\xi(4\epsilon)\xi(4\epsilon-4)}$ and vanish in the limit $\epsilon \to 0$, along with the corresponding abelian Fourier coefficients.

## 2) The sixth layer

We shall now consider $\Upsilon_i \neq 0$ and linearly independent with $n_i = 0$. In this case one can consider the $SO(5,5)$ grading:

$$\begin{aligned}
\mathfrak{e}_7 &\cong \mathbf{10}^{(-2)} \oplus (\mathbf{2} \otimes \mathbf{16})^{(-1)} \oplus \left(\mathfrak{gl}_1 \oplus \mathfrak{sl}_2 \oplus \mathfrak{so}(5,5)\right)^{(0)} \oplus (\mathbf{2} \otimes \overline{\mathbf{16}})^{(1)} \oplus \mathbf{10}^{(2)} , \\
\mathbf{56} &\cong \mathbf{2}^{(-2)} \oplus \overline{\mathbf{16}}^{(-1)} \oplus (\mathbf{2} \otimes \mathbf{10})^{(0)} \oplus \mathbf{16}^{(1)} \oplus \mathbf{2}^{(2)} , \\
\mathbf{912} &\cong \cdots \oplus \left(\mathbf{2} \otimes \mathbf{120} \oplus 2 \times \mathbf{2} \otimes \mathbf{10}\right)^{(0)} \oplus \left(\mathbf{3} \otimes \mathbf{16} \oplus \overline{\mathbf{144}} \oplus \mathbf{16}\right)^{(1)} \oplus (\mathbf{2} \otimes \mathbf{45} \oplus \mathbf{2})^{(2)} \oplus \overline{\mathbf{16}}^{(3)} , \\
\mathbf{1539} &\cong \cdots \oplus (\mathbf{120} \oplus \mathbf{3} \otimes \mathbf{10} \oplus \mathbf{10})^{(2)} \oplus (\mathbf{2} \otimes \mathbf{16})^{(3)} \oplus \mathbf{1}^{(4)} , \qquad \text{(E.13)}
\end{aligned}$$

with $\Upsilon_i = (0, 0, 0, 0, n_i{}^{\hat{\jmath}}) \in \mathbf{2}^{(2)}$. The condition $Q\Upsilon|_{\mathbf{912}} = 0$ then implies that

$$Q = \left(0, 0, n_i{}^{\hat{\jmath}} \frac{\ell_{\hat{k}}}{r} + \tfrac{1}{2} \delta_{\hat{k}}^{\hat{\jmath}} n_i{}^{\hat{l}} \frac{\ell_{\hat{l}}}{r}, 0, n_i{}^{\hat{\jmath}} \frac{p}{k}, q_i\right) \in (\mathbf{2} \otimes \mathbf{2})^{(0)} \oplus (\mathbf{2} \otimes \overline{\mathbf{16}})^{(1)} \oplus \mathbf{10}^{(2)} , \qquad \text{(E.14)}$$

so that $\ell = \frac{1}{2} n_i{}^{\hat{j}} \ell_{\hat{j}}/r$. The condition $Q^2 + 4\Gamma \wedge \Upsilon$ to vanish in the **1539** implies that

$$\Gamma_i = \left( n_i{}^{\hat{j}} \frac{\ell_{\hat{i}}}{r} \frac{\ell_{\hat{j}}}{r}, n_i{}^{\hat{j}} \frac{\ell_{\hat{j}}}{r} \frac{p}{k}, \frac{1}{2} n_i{}^{\hat{j}} \frac{p}{k} \gamma \frac{p}{k} + \varepsilon^{\hat{j}\hat{k}} \frac{\ell_{\hat{k}}}{r} q_i, \not{q}_i \frac{p}{k}, m_i{}^{\hat{j}} \right) \tag{E.15}$$

with the constraint

$$2\varepsilon_{\hat{k}\hat{l}} n_{(i}{}^{\hat{k}} m_{j)}{}^{\hat{l}} = (q_i, q_j) \ . \tag{E.16}$$

The only constraint that is not yet satisfied is $\Gamma_i \times \Gamma_j = m_{(i} Q_{j)}$ that enforces

$$m_i = \frac{1}{2} \frac{p}{k} \not{q}_i \frac{p}{k} + m_i{}^{\hat{j}} \frac{\ell_{\hat{j}}}{r} \ . \tag{E.17}$$

Here we defined $k$ coprime to $p \in \overline{S}_+$ and $r$ coprime to $\ell_{\hat{i}} \in \mathbb{Z}^2$ such that they divide $n_i{}^{\hat{j}}$ and all the other necessary quantities for the charges to be integer valued.

One can then interpret the sum over $(k, p)$ as a Poincaré sum over $P_1 \backslash E_6$, and the sum over $(r, \ell_{\hat{i}})$ as a Poincaré sum over $P_1 \backslash SL(3)$, for the maximal pair $SL(3) \times E_6 \subset E_8$, and manipulate the sum over $n_i{}^{\hat{j}}, q_i, m_i{}^{\hat{j}}$ using the orbit method for an auxiliary genus two theta lift. One can understand this in two steps. One can first rewrite the set of charges at $\ell_{\hat{i}} = 0$ in the $P_7 \subset E_8$ decomposition in which the sum over $(k, p)$ can be interpreted as a sum over $P_1 \backslash E_6$

$$\mathfrak{e}_8 \cong \mathbf{2}^{(-3)} \oplus \mathbf{27}^{(-2)} \oplus (\mathbf{2} \otimes \overline{\mathbf{27}})^{(-1)} \oplus (\mathfrak{gl}_1 \oplus \mathfrak{sl}_2 \oplus \mathfrak{e}_6)^{(0)} \oplus (\mathbf{2} \otimes \mathbf{27})^{(1)} \oplus \overline{\mathbf{27}}^{(2)} \oplus \mathbf{2}^{(3)}$$

$$(n_i{}^{\hat{j}}, n_i{}^{\hat{j}} \frac{p}{k}, n_i{}^{\hat{j}} \frac{p\gamma p}{2k^2}) \in (\mathbf{2} \otimes \mathbf{27})^{(1)}$$

$$(q_i, \not{q}_i \frac{p}{k}, \frac{p\not{q}_i p}{2k^2}) \in \overline{\mathbf{27}}^{(2)}$$

$$m_i{}^{\hat{j}} \in \mathbf{2}^{(3)} \ . \tag{E.18}$$

The set of charges of the sixth layer, at $\ell_{\hat{i}} = 0$, span the three first degrees in the decomposition above, where the doublet of non-collinear charges in the $(\mathbf{2} \otimes \mathbf{27})$ is in the $E_6$ orbit of $(n_i{}^{\hat{j}}, 0, 0)$. One recognises the sum over $(k, p)$ as the Poincaré sum over $P_1 \backslash E_6$ of the solution to $\mathcal{Q}_i \times \mathcal{Q}_j = 0$ at $p = \ell_{\hat{i}} = 0$. Similarly, the sum over non-trivial $(r, \ell_{\hat{i}})$ can be interpreted as a sum over $P_1 \backslash SL(3)$ in the decomposition

$$\mathfrak{e}_8 \cong \cdots \oplus (\mathfrak{gl}_1 \oplus \mathfrak{sl}_3 \oplus \mathfrak{so}(5,5))^{(0)} \oplus (\mathbf{3} \otimes \mathbf{16})^{(1)} \oplus (\overline{\mathbf{3}} \otimes \mathbf{10})^{(2)} \oplus \overline{\mathbf{16}}^{(3)} \oplus \mathbf{3}^{(4)}$$

$$(n_i{}^{\hat{j}}, n_i{}^{\hat{j}} \frac{\ell_{\hat{k}}}{r} + \delta_{\hat{k}}^{\hat{j}} n_i{}^{\hat{l}} \frac{\ell_{\hat{l}}}{r}, n_i{}^{\hat{j}} \frac{\ell_{\hat{j}} \ell_{\hat{k}}}{r^2}) \in (\mathfrak{sl}_3)^{(0)}$$

$$(n_i{}^{\hat{j}} \frac{p}{k}, n_i{}^{\hat{j}} \frac{\ell_{\hat{j}}}{r} \frac{p}{k}) \in (\mathbf{3} \otimes \mathbf{16})^{(1)}$$

$$(q_i, \varepsilon^{\hat{j}\hat{k}} \frac{\ell_{\hat{k}}}{r} q_i + \frac{p\gamma p}{2k^2}) \in (\overline{\mathbf{3}} \otimes \mathbf{10})^{(2)}$$

$$\not{q}_i \frac{p}{k} \in \overline{\mathbf{16}}^{(3)}$$

$$(m_i{}^{\hat{j}}, m_i{}^{\hat{j}} \frac{\ell_{\hat{j}}}{r} + \frac{p\not{q}_i p}{2k^2}) \in \mathbf{3}^{(4)} \ . \tag{E.19}$$

Now, the set of charges of the sixth layer span the five first degrees in the decomposition above, where the doublet of non-collinear charges in $\mathfrak{sl}_3$ is in the $SL(3, \mathbb{Z})$ orbit of $(n_i{}^{\hat{j}}, 0, 0)$. One

recognises the sum over $(r, \ell_{\hat{\imath}})$ as the Poincaré sum over $P_1 \backslash SL(3)$ of the solution to $\mathcal{Q}_i \times \mathcal{Q}_j = 0$ at $\ell_{\hat{\imath}} = 0$.

With this interpretation as a $P_1 \backslash E_6 \times P_1 \backslash SL(3)$ Poincaré sum in mind, we rewrite the invariant bilinear form as

$$G(\mathcal{Q}_i, \mathcal{Q}_j) = (R^{-2} y v_{\hat{\imath}\hat{\jmath}} + R^{-4} \tilde{\ell}_{\hat{\imath}} \tilde{\ell}_{\hat{\jmath}}) \big( m_i{}^{\hat{\imath}} + (\tilde{a}^{\hat{\imath}}, q_i + \tfrac{1}{2} \tilde{a}_{\hat{k}} n_i{}^{\hat{k}}) + \tilde{b} n_i{}^{\hat{\imath}} \big) \big( m_j{}^{\hat{\jmath}} + (\tilde{a}^{\hat{\jmath}}, q_j + \tfrac{1}{2} \tilde{a}_{\hat{\imath}}) n_j{}^{\hat{l}} + \tilde{b} n_j{}^{\hat{\jmath}} \big)$$

$$+ \left( (y + R^{-2} v(\tilde{\ell}, \tilde{\ell})) u_{ab} + \frac{y^{\frac{1}{2}}}{R^2} \tilde{p} \gamma_a u \gamma_b \tilde{p} + \tfrac{1}{4} \frac{(\tilde{p}\gamma_a \tilde{p})(\tilde{p}\gamma_b \tilde{p})}{R^4 + \frac{R^2}{y} v(\tilde{\ell}, \tilde{\ell})} \right) (q_i^a + \tilde{a}_{\hat{\imath}}^a n_i{}^{\hat{\imath}})(q_j^b + \tilde{a}_{\hat{\jmath}}^b n_j{}^{\hat{\jmath}})$$

$$+ \left( R^2 y + v(\tilde{\ell}, \tilde{\ell}) + y^{\frac{1}{2}} u(\tilde{p}, \tilde{p}) + \tfrac{1}{4} \frac{u(\tilde{p}\gamma \tilde{p}, \tilde{p}\gamma \tilde{p})}{R^2 + \frac{1}{y} v(\tilde{\ell}, \tilde{\ell})} \right) (v_{\hat{\imath}\hat{\jmath}} + \frac{1}{R^2 y} \tilde{\ell}_{\hat{\imath}} \tilde{\ell}_{\hat{\jmath}}) n_i{}^{\hat{\imath}} n_j{}^{\hat{\jmath}} \quad \text{(E.20)}$$

where we introduced for short

$$\tilde{\ell}_{\hat{\imath}} = \frac{\ell_{\hat{\imath}}}{r} + a_{\hat{\imath}}, \qquad \tilde{p} = \frac{p}{k} + a + c^{\hat{\imath}} \left( \frac{\ell_{\hat{\imath}}}{r} + a_{\hat{\imath}} \right),$$

$$\tilde{a}_{\hat{\imath}}^a = a_{\hat{\imath}}^a + c_{\hat{\imath}} \gamma^a \left( \frac{p}{k} + a + \tfrac{1}{2} c^{\hat{\jmath}} \left( \frac{\ell_{\hat{\jmath}}}{r} + a_{\hat{\jmath}} \right) \right) + c^a \left( \frac{\ell_{\hat{\imath}}}{r} + a_{\hat{\imath}} \right) + \tfrac{1}{2} \varepsilon_{\hat{\imath}\hat{\jmath}} v^{\hat{\jmath}\hat{k}} \left( \frac{\ell_{\hat{k}}}{r} + a_{\hat{k}} \right) \frac{\tilde{p}\gamma^a \tilde{p}}{R^2 y + v(\tilde{\ell}, \tilde{\ell})},$$

$$\tilde{b} = b + \tfrac{1}{2} \bar{a} \left( \frac{p}{k} + a + \tfrac{1}{2} c^{\hat{\imath}} \left( \frac{\ell_{\hat{\imath}}}{r} + a_{\hat{\imath}} \right) \right) + \tfrac{1}{2} \bar{a}^{\hat{\imath}} \left( \frac{\ell_{\hat{\imath}}}{r} + a_{\hat{\imath}} \right) + \tfrac{1}{2} \tilde{p} \not{c} \tilde{p} + \dots \quad \text{(E.21)}$$

The factors of $R^2 + \frac{1}{y} v(\tilde{\ell}, \tilde{\ell})$ in the denumerator in (E.20) comes from completing the squares in

$$R^{-2} y v_{\hat{\imath}\hat{\jmath}} m_i{}^{\hat{\imath}} m_j{}^{\hat{\jmath}} + R^{-4} (m_i{}^{\hat{\imath}} \tilde{\ell}_{\hat{\imath}} + \tfrac{1}{2} \tilde{p} \not{q}_i \tilde{p})(m_j{}^{\hat{\jmath}} \tilde{\ell}_{\hat{\jmath}} + \tfrac{1}{2} \tilde{p} \not{q}_j \tilde{p})$$

$$= (R^{-2} y v_{\hat{\imath}\hat{\jmath}} + R^{-4} \tilde{\ell}_{\hat{\imath}} \tilde{\ell}_{\hat{\jmath}}) \left( m_i{}^{\hat{\imath}} + \tfrac{1}{2} v^{\hat{\imath}\hat{k}} \tilde{\ell}_{\hat{k}} \frac{\tilde{p} \not{q}_i \tilde{p}}{R^2 y + v(\tilde{\ell}, \tilde{\ell})} \right) \left( m_j{}^{\hat{\jmath}} + \tfrac{1}{2} v^{\hat{\jmath}\hat{l}} \tilde{\ell}_{\hat{l}} \frac{\tilde{p} \not{q}_j \tilde{p}}{R^2 y + v(\tilde{\ell}, \tilde{\ell})} \right)$$

$$+ \tfrac{1}{4} \frac{(\tilde{p}\gamma_a \tilde{p})(\tilde{p}\gamma_b \tilde{p})}{R^4 + \frac{R^2}{y} v(\tilde{\ell}, \tilde{\ell})} q_i^a q_j^b \quad \text{(E.22)}$$

and

$$y u_{ab} q_i^a q_j^b + R^{-2} v_{\hat{\imath}\hat{\jmath}} u_{ab} (\varepsilon^{\hat{\imath}\hat{k}} \tilde{\ell}_{\hat{k}} q_i^a + \tfrac{1}{2} n_i{}^{\hat{\imath}} \tilde{p}\gamma^a \tilde{p})(\varepsilon^{\hat{\jmath}\hat{l}} \tilde{\ell}_{\hat{l}} q_j^b + \tfrac{1}{2} n_j{}^{\hat{\jmath}} \tilde{p}\gamma^b \tilde{p})$$

$$= (y + R^{-2} v(\tilde{\ell}, \tilde{\ell})) u_{ab} \left( q_i^a + \tfrac{1}{2} n_i{}^{\hat{\imath}} v_{\hat{\imath}\hat{k}} \varepsilon^{\hat{k}\hat{p}} \tilde{\ell}_{\hat{p}} \frac{\tilde{p}\gamma^a \tilde{p}}{R^2 y + v(\tilde{\ell}, \tilde{\ell})} \right) \left( q_j^b + \tfrac{1}{2} n_j{}^{\hat{\jmath}} v_{\hat{\jmath}\hat{l}} \varepsilon^{\hat{l}\hat{q}} \tilde{\ell}_{\hat{q}} \frac{\tilde{p}\gamma^b \tilde{p}}{R^2 y + v(\tilde{\ell}, \tilde{\ell})} \right)$$

$$+ \tfrac{1}{4} \frac{u(\tilde{p}\gamma \tilde{p}, \tilde{p}\gamma \tilde{p})}{R^2 + \frac{1}{y} v(\tilde{\ell}, \tilde{\ell})} (v_{\hat{\imath}\hat{\jmath}} + \frac{1}{R^2 y} \tilde{\ell}_{\hat{\imath}} \tilde{\ell}_{\hat{\jmath}}) n_i{}^{\hat{\imath}} n_j{}^{\hat{\jmath}} \quad \text{(E.23)}$$

and repeatedly using the $Spin(5,5)$ identity [79]

$$\gamma^a \tilde{p} \tilde{p} \gamma_a = -\tfrac{1}{2} (\tilde{p}\gamma^a \tilde{p}) \gamma_a \quad \Rightarrow \quad \gamma^a \tilde{p} (\tilde{p}\gamma_a \tilde{p}) = 0. \quad \text{(E.24)}$$

$G(Q_i, Q_j)$ in (E.20) is then recognised, up to a scale factor, as the the metric on the $SO(7,7)$ lattice $II_{7,7}$

$$\tilde{g}(Q_i, Q_j) = \tilde{R}^{-1} \tilde{v}_{\hat{\imath}\hat{\jmath}} \big( m_i{}^{\hat{\imath}} + (\tilde{a}^{\hat{\imath}}, q_i + \tfrac{1}{2} \tilde{a}_{\hat{k}} n_i{}^{\hat{k}}) + \tilde{b} n_i{}^{\hat{\imath}} \big) \big( m_j{}^{\hat{\jmath}} + (\tilde{a}^{\hat{\jmath}}, q_j + \tfrac{1}{2} \tilde{a}_{\hat{\imath}}) n_j{}^{\hat{l}} + \tilde{b} n_j{}^{\hat{\jmath}} \big)$$

$$+ \tilde{u}_{ab} (q_i^a + \tilde{a}_{\hat{\imath}}^a n_i{}^{\hat{\imath}})(q_j^b + \tilde{a}_{\hat{\jmath}}^b n_j{}^{\hat{\jmath}}) + \tilde{R} \tilde{v}_{\hat{\imath}\hat{\jmath}} n_i{}^{\hat{\imath}} n_j{}^{\hat{\jmath}}, \quad \text{(E.25)}$$

with

$$
\begin{aligned}
G(\mathcal{Q}_i, \mathcal{Q}_j) &= \sqrt{\left(y + \tfrac{\upsilon(\tilde{\ell},\tilde{\ell})}{R^2}\right)^2 + \left(y + \tfrac{\upsilon(\tilde{\ell},\tilde{\ell})}{R^2}\right)\frac{y^{\frac{1}{2}}}{R^2}u(\tilde{p},\tilde{p}) + \tfrac{1}{4}\frac{y}{R^4}u(\tilde{p}\gamma\tilde{p},\tilde{p}\gamma\tilde{p})} \ \ \tilde{g}(Q_i, Q_j) \\
\tilde{R} &= R^2\sqrt{1 + \frac{\upsilon(\tilde{\ell},\tilde{\ell})}{R^2 y} + \frac{u(\tilde{p},\tilde{p})}{R^2 y^{\frac{1}{2}}} + \tfrac{1}{4}\frac{u(\tilde{p}\gamma\tilde{p},\tilde{p}\gamma\tilde{p})}{R^4 y + R^2\upsilon(\tilde{\ell},\tilde{\ell})}} \\
\tilde{\upsilon}_{\hat{\imath}\hat{\jmath}} &= \frac{\upsilon_{\hat{\imath}\hat{\jmath}} + \frac{\tilde{\ell}_i\tilde{\ell}_j}{R^2 y}}{\sqrt{1 + \frac{\upsilon(\tilde{\ell},\tilde{\ell})}{R^2 y}}} \\
\tilde{u}_{ab} &= \frac{\left(1 + \frac{\upsilon(\tilde{\ell},\tilde{\ell})}{R^2 y}\right)u_{ab} + \frac{1}{R^2 y}\tilde{p}\gamma_a u\gamma_b\tilde{p} + \tfrac{1}{4}\frac{(\tilde{p}\gamma_a\tilde{p})(\tilde{p}\gamma_b\tilde{p})}{R^4 y + R^2\upsilon(\tilde{\ell},\tilde{\ell})}}{\sqrt{\left(1 + \frac{\upsilon(\tilde{\ell},\tilde{\ell})}{R^2 y}\right)^2 + \left(1 + \frac{\upsilon(\tilde{\ell},\tilde{\ell})}{R^2 y}\right)\frac{u(\tilde{p},\tilde{p})}{R^2 y^{\frac{1}{2}}} + \tfrac{1}{4}\frac{u(\tilde{p}\gamma\tilde{p},\tilde{p}\gamma\tilde{p})}{R^4 y}}}
\end{aligned}
\tag{E.26}
$$

where one checks that $\tilde{u}_{ab}$ is indeed an orthogonal symmetric matrix using (E.24).

We can now use the orbit method for the genus-two Siegel–Narain theta series on the lattice $II_{7,7}$ to compute the sum

$$
\int_{\mathcal{G}}\frac{\mathrm{d}^3\Omega_2}{|\Omega_2|^3}\int_{\mathbb{Z}^3\backslash\mathbb{R}^3}\mathrm{d}^3\Omega_1\ |\Omega_2|^\epsilon\varphi_{\mathrm{KZ}}^{\mathrm{tr}}(\Omega_2)\,|\Omega_2|^{\frac{7}{2}}\sum_{\substack{n_i^{\hat{\jmath}}\in\mathbb{Z}^2 \\ \det n\neq 0 \ m_i^{\hat{\jmath}}\in\mathbb{Z}^2}}\sum_{q_{ia}\in II_{5,5}}e^{-\pi\Omega_2^{ij}G(\mathcal{Q}_i,\mathcal{Q}_j)+\pi\mathrm{i}\Omega_1^{ij}(2\varepsilon_{\hat{\imath}\hat{\jmath}}m_i{}^{\hat{\imath}}n_j{}^{\hat{\jmath}}-(q_i,q_j))}
$$

$$
= \int_{\mathcal{G}}\frac{\mathrm{d}^3\Omega_2}{|\Omega_2|^{\frac{1}{2}}}\int_{\mathbb{Z}^3\backslash\mathbb{R}^3}\mathrm{d}^3\Omega_1\sum_{\substack{\gamma\in P_2\backslash Sp(4,\mathbb{Z}) \\ \det C(\gamma)\neq 0}}\left(|\Omega_2|^\epsilon\varphi_{\mathrm{KZ}}^{\mathrm{tr}}(\Omega_2)\right)\big|_\gamma\sum_{\substack{m^i\in\mathbb{Z}^2 \\ \det m\neq 0}}\sum_{q_i\in II_{5,5}}
$$

$$
\times\tilde{R}^2\frac{e^{-\pi\Omega_{2ij}^{-1}\tilde{R}\tilde{\upsilon}^{\hat{\imath}\hat{\jmath}}m_i{}^i m_j{}^j-\pi\Omega_2^{ij}\tilde{u}_{ab}q_i^a q_j^b-\pi\mathrm{i}\Omega_1^{ij}\eta_{ab}q_i^a q_j^b+2\pi\mathrm{i}m_j{}^i q_i^a\tilde{a}_a^{\hat{\jmath}}}}{\left((y+\frac{\upsilon(\tilde{\ell},\tilde{\ell})}{R^2})^2+(y+\frac{\upsilon(\tilde{\ell},\tilde{\ell})}{R^2})\frac{y^{\frac{1}{2}}}{R^2}u(\tilde{p},\tilde{p})+\tfrac{1}{4}\frac{y}{R^4}u(\tilde{p}\gamma\tilde{p},\tilde{p}\gamma\tilde{p})\right)^{\frac{5}{2}+\epsilon}} + \dots
\tag{E.27}
$$

where the ellipsis denotes non-abelian Fourier coefficients. The constant term at $q_i = 0$ can be computed using the interpretation of the sum over $k$ and $p$ as the principal layer of the Poincaré sum $P_1\backslash E_6$ and the sum over $r$ and $\ell$ as the principal layer of the Poincaré sum over $P_1\backslash SL(3)$.

In order to carry out the sum over $k$ and $p$ bellow we shall use that the sum over $P_1\backslash E_6$ can be interpreted as a weak coupling limit with $g_5 = (R^2 y^{\frac{1}{2}} + y^{-\frac{1}{2}}\upsilon(\tilde{\ell},\tilde{\ell}))^{-\frac{1}{2}}$ such that

$$
\sum_{k=1}^{\infty}\sum_{\substack{p\in S_+ \\ \frac{p\gamma p}{2k}\in II_{5,5} \\ (k,p)=1}}\frac{1}{\left(1 + \frac{\upsilon(\tilde{\ell},\tilde{\ell})}{R^2 y} + \frac{u(\frac{p}{k}+a,\frac{p}{k}+a)}{R^2 y^{\frac{1}{2}}} + \tfrac{1}{4}\frac{u[(\frac{p}{k}+a)\gamma(\frac{p}{k}+a),(\frac{p}{k}+a)\gamma(\frac{p}{k}+a)]}{R^4 y + R^2\upsilon(\tilde{\ell},\tilde{\ell})}\right)^s}
$$

$$
= \frac{\xi(2s-8)\xi(2s-11)}{\xi(2s)\xi(2s-3)}y^4 R^{16}\left(1 + \frac{\upsilon(\tilde{\ell},\tilde{\ell})}{R^2 y}\right)^{8-s} + \mathcal{O}(e^{-\pi\sqrt{R^2 y^{\frac{1}{2}} + y^{-\frac{1}{2}}\upsilon(\tilde{\ell},\tilde{\ell})}})
\tag{E.28}
$$

up to the exponentially suppressed Fourier coefficients in $e^{2\pi\mathrm{i}(q,a)}$, by recognising

$$
E_{s\Lambda_6}^{E_6} = \frac{1}{2\zeta(2s)}\sum_{\substack{k\in\mathbb{Z} \\ p\in S_+ \\ q\in II_{5,5} \\ 2nq=p\gamma p}}\frac{1}{\left(g_5^{-\frac{8}{3}}k^2 + g_5^{-\frac{2}{3}}u(p+ak,p+ak) + g_5^{\frac{4}{3}}u(q+(a\gamma p)+\tfrac{1}{2}(a\gamma a)k,q+(a\gamma p)+\tfrac{1}{2}(a\gamma a)k)\right)^s} \ .
\tag{E.29}
$$

Similarly, to carry out the sum over $\ell$ and $r$ one can recognise the unrestricted sum over $\ell$ and $r$ as the $SL(3)$ Eisenstein series

$$E_{(s-\epsilon)\Lambda_1+2\epsilon\Lambda_2}^{SL(3)} = \frac{1}{2\zeta(2\epsilon)} \sideset{}{'}\sum_{\substack{r\in\mathbb{Z}\\ \ell\in\mathbb{Z}^2}} \frac{1}{(y_3^{-1}v(\ell+ar,\ell+ar)+y_3^2 r^2)^s} E_{2\epsilon\Lambda_1}^{SL(2)}\left(\frac{v+\frac{(\frac{\ell}{r}+a)(\frac{\ell}{r}+a)}{y_3^3}}{\sqrt{1+\frac{v(\frac{\ell}{r}+a,\frac{\ell}{r}+a)}{y_3^3}}}\right) \quad (\text{E}.30)$$

such that the restricted sum with $r \neq 0$ can be recognised as its last constant term using Langlands constant term formula, giving

$$\sum_{r=1}^{\infty} \sum_{\substack{\ell\in\mathbb{Z}^2\\ (r,\ell)=1}} \frac{1}{(1+\frac{v(\frac{\ell}{r}+a,\frac{\ell}{r}+a)}{R^2 y})^s} E_{2\epsilon\Lambda_1}^{SL(2)}\left(\frac{v+\frac{(\frac{\ell}{r}+a)(\frac{\ell}{r}+a)}{R^2 y}}{\sqrt{1+\frac{v(\frac{\ell}{r}+a,\frac{\ell}{r}+a)}{R^2 y}}}\right)$$

$$= \frac{\xi(2s-2\epsilon-1)\xi(2s+2\epsilon-2)}{\xi(2s-2\epsilon)\xi(2s+2\epsilon-1)} R^2 y E_{2\epsilon\Lambda_1}^{SL(2)}(v) + \mathcal{O}(e^{-\pi R y^{\frac{1}{2}}}) . \quad (\text{E}.31)$$

One determines that this is the unique constant term coming out of the principal layer by computing the scaling in $R^2 y$ from the homogeneity of the Fourier transform.

Using (E.27) and (3.58), one can compute in this way the constant term contribution

$$\mathcal{I}_7^{(6a)} = \frac{10\zeta(3)}{\pi} \sum_{\gamma\in P_7\backslash E_8} \sum_{(r,\ell)} \sum_{k=1}^{\infty} \sum_{\substack{p\in S_+\\ \frac{p\gamma p}{2k}\in II_{5,5}\\ (k,p)=1}}$$

$$\times \sum_{\substack{m_i\in\mathbb{Z}^2\\ \det m\neq 0\\ m(\ell/r)\in\mathbb{Z}^2}} \int_{\mathcal{G}} \frac{d^3\Omega_2}{|\Omega_2|^{\frac{3}{2}}} \frac{\tilde{R}^2 E_{2\epsilon\Lambda_1}^{SL(2)}(\tau) e^{-\pi\Omega_2^{ij}\tilde{R}\tilde{v}_{\hat{i}\hat{j}}m_i{}^{\hat{i}}m_j{}^{\hat{j}}}}{\left((y+\frac{v(\tilde{\ell},\tilde{\ell})}{R^2})^2+(y+\frac{v(\tilde{\ell},\tilde{\ell})}{R^2})\frac{y^{\frac{1}{2}}}{R^2}u(\tilde{p},\tilde{p})+\frac{1}{4}\frac{y}{R^4}u(\tilde{p}\gamma\tilde{p},\tilde{p}\gamma\tilde{p})\right)^{\frac{5}{2}+\epsilon}}$$

$$= \frac{20\zeta(3)}{\pi} R^4 \sum_{\gamma\in P_7\backslash E_8} \sum_{(r,\ell)} \sum_{k=1}^{\infty} \sum_{\substack{p\in S_+\\ \frac{p\gamma p}{2k}\in II_{5,5}\\ (k,p)=1}} \frac{\frac{\xi(-2\epsilon)\xi(2\epsilon-1)}{y(y+\frac{v(\tilde{\ell},\tilde{\ell})}{R^2})} E_{2\epsilon\Lambda_1}^{SL(2)}\left(\frac{v+\frac{(\frac{\ell}{r}+a)(\frac{\ell}{r}+a)}{R^2 y}}{\sqrt{1+\frac{v(\frac{\ell}{r}+a,\frac{\ell}{r}+a)}{R^2 y}}}\right)}{\left((y+\frac{v(\tilde{\ell},\tilde{\ell})}{R^2})^2+(y+\frac{v(\tilde{\ell},\tilde{\ell})}{R^2})\frac{y^{\frac{1}{2}}}{R^2}u(\tilde{p},\tilde{p})+\frac{1}{4}\frac{y}{R^4}u(\tilde{p}\gamma\tilde{p},\tilde{p}\gamma\tilde{p})\right)^{\frac{3}{2}+\epsilon}}$$

$$= \frac{20\zeta(3)}{\pi} R^{20} \sum_{\gamma\in P_7\backslash E_8} \sum_{(r,\ell)} \sum_{k=1}^{\infty} \frac{\xi(-2\epsilon)\xi(2\epsilon-1)\xi(2\epsilon-5)\xi(2\epsilon-8) E_{2\epsilon\Lambda_1}^{SL(2)}\left(\frac{v+\frac{(\frac{\ell}{r}+a)(\frac{\ell}{r}+a)}{R^2 y}}{\sqrt{1+\frac{v(\frac{\ell}{r}+a,\frac{\ell}{r}+a)}{R^2 y}}}\right)}{\xi(2\epsilon)\xi(3+2\epsilon)y^{1+2\epsilon}(1+\frac{v(\frac{\ell}{r}+a,\frac{\ell}{r}+a)}{R^2 y})^{2\epsilon-4}}$$

$$= \frac{20\zeta(3)}{\pi} R^{22} \sum_{\gamma\in P_7\backslash E_8} \frac{\xi(2\epsilon-9)\xi(6\epsilon-10)\xi(-2\epsilon)\xi(2\epsilon-1)\xi(2\epsilon-5)\xi(2\epsilon-8) E_{2\epsilon\Lambda_1}^{SL(2)}(v)}{\xi(2\epsilon-8)\xi(6\epsilon-9)\xi(2\epsilon)\xi(3+2\epsilon)y^{2\epsilon}}$$

$$= \frac{20\zeta(3)}{\pi} R^{22} \frac{\xi(2\epsilon-9)\xi(6\epsilon-10)\xi(-2\epsilon)\xi(2\epsilon-1)\xi(2\epsilon-5)\xi(2\epsilon-8)}{\xi(2\epsilon-8)\xi(6\epsilon-9)\xi(2\epsilon)\xi(3+2\epsilon)} E_{2\epsilon\Lambda_7}^{E_7}$$

$$\underset{\epsilon\to 0}{=} -40\xi(2)\xi(6)\xi(11)R^{22} . \quad (\text{E}.32)$$

In the third equality we carried out the sum over $p$ and $k$ using (E.28), and in the fourth equality the sum over $\ell$ and $r$ using (E.31). This is the term that appears in the decompactification limit (4.1), except for the sign. The sign will be resolved in considering the renormalised coupling (1.31). Indeed, this contribution drops out in (1.28) because the constant term $\frac{5\zeta(3)}{4\pi^2}V^2 E_{2\epsilon\Lambda_1}^{SL(2)}(\tau)$ in (3.58) also appears in the constant terms of the Siegel–Eisenstein series (D.4) at $\delta = 0$. After these cancellations, the only remaining contribution in (1.28) is the one from the adjoint Eisenstein series coming from the Siegel–Eisenstein series constant term $\frac{5\zeta(3)}{4\pi^2}V^{2+2\epsilon}$ that gives instead

$$
\frac{2\pi^2\xi(3+2\epsilon)}{9\xi(4+2\epsilon)} \sum_{\gamma\in P_7\backslash E_8} \sum_{(r,\ell)} \sum_{k=1}^{\infty} \sum_{\substack{p\in S_+ \\ \frac{p\gamma p}{2k}\in II_{5,5} \\ (k,p)=1}}
$$

$$
\times \sum_{\substack{m_i\in\mathbb{Z}^2 \\ \det m\neq 0 \\ m(\ell/r)\in\mathbb{Z}^2}} \int_{\mathcal{G}} \frac{\mathrm{d}^3\Omega_2}{|\Omega_2|^{\frac{3}{2}}} \frac{|\Omega_2|^\epsilon \tilde{R}^2 e^{-\pi\Omega_2^{ij}\tilde{R}\tilde{v}_{\hat{i}\hat{j}}m_i{}^{\hat{i}}m_j{}^{\hat{j}}}}{\left((y+\frac{v(\tilde{\ell},\tilde{\ell})}{R^2})^2 + (y+\frac{v(\tilde{\ell},\tilde{\ell})}{R^2})\frac{y^{\frac{1}{2}}}{R^2}u(\tilde{p},\tilde{p}) + \frac{1}{4}\frac{y}{R^4}u(\tilde{p}\gamma\tilde{p},\tilde{p}\gamma\tilde{p})\right)^{\frac{5}{2}}}
$$

$$
= \frac{4\pi^2\xi(3+2\epsilon)}{9\xi(4+2\epsilon)} \sum_{\gamma\in P_7\backslash E_8} \sum_{(r,\ell)} \sum_{k=1}^{\infty} \sum_{\substack{p\in S_+ \\ \frac{p\gamma p}{2k}\in II_{5,5} \\ (k,p)=1}} \frac{R^{4-4\epsilon}\frac{\xi(2\epsilon)\xi(2\epsilon-1)}{y^5(1+\frac{v(\tilde{\ell},\tilde{\ell})}{R^2 y})^{\frac{5}{2}}}}{\left(1+\frac{v(\tilde{\ell},\tilde{\ell})}{R^2 y}+\frac{u(\tilde{p},\tilde{p})}{R^2 y^{\frac{1}{2}}}+\frac{1}{4}\frac{u(\tilde{p}\gamma\tilde{p},\tilde{p}\gamma\tilde{p})}{R^4 y+v(\tilde{\ell},\tilde{\ell})}\right)^{\frac{3}{2}+\epsilon}}
$$

$$
= \frac{4\pi^2\xi(3+2\epsilon)}{9\xi(4+2\epsilon)} R^{20-4\epsilon} \sum_{\gamma\in P_7\backslash E_8} \sum_{(r,\ell)} \sum_{k=1}^{\infty} \frac{\xi(2\epsilon)\xi(2\epsilon-1)\xi(2\epsilon-5)\xi(2\epsilon-8)}{\xi(2\epsilon)\xi(3+2\epsilon)y(1+\frac{v(\frac{\ell}{r}+a,\frac{\ell}{r}+a)}{R^2 y})^{\epsilon-4}}
$$

$$
= \frac{4\pi^2\xi(3+2\epsilon)}{9\xi(4+2\epsilon)} R^{22-4\epsilon} \sum_{\gamma\in P_7\backslash E_8} \frac{\xi(2\epsilon-9)\xi(2\epsilon-10)\xi(2\epsilon)\xi(2\epsilon-1)\xi(2\epsilon-5)\xi(2\epsilon-8)}{\xi(2\epsilon-8)\xi(2\epsilon-9)\xi(2\epsilon)\xi(3+2\epsilon)}
$$

$$
= \frac{4\pi^2}{9} R^{22-4\epsilon}\frac{\xi(2\epsilon-10)\xi(2\epsilon-1)\xi(2\epsilon-5)}{\xi(4+2\epsilon)} \underset{\epsilon\to 0}{=} 40\xi(2)\xi(6)\xi(11)R^{22} \ . \tag{E.33}
$$

Note that in both cases we have used formal identities for divergent sum or integrals. In the first sum for $\mathcal{I}_7^{(5a)}$, we integrated the logarithmically divergent integral over $V$ by analytic continuation of $\frac{\mathrm{d}V}{V^{1+\tilde{\epsilon}}}$ at $\tilde{\epsilon} = 0$. For the second sum we have the formal Poincaré sum of 1 over $P_7\backslash E_8$, which we consider equal to one by analytic continuation of the Eisenstein series $E_{\tilde{\epsilon}\Lambda_7}^{E_8}$. We encounter these divergences because we have neglected the cut-off $L$ on the fundamental domain $\mathcal{F}$ in the computation, in particular when we used the orbit method in (E.27). We expect that a proper handling of the cut-off $L$ in the orbit method should be equivalent to introducing such parameter $\tilde{\epsilon}$ as in (3.21). Although this computation is not rigorous, the fact that the same method reproduces correctly three of the constant terms of the two-parameter Eisenstein series in (D.7) provides a strong consistency check of our result.

For $d = 7$ both the counterterm and the three-loop contribution are finite, so one may wonder why one needs the renormalised coupling to get the right answer. The point is that $\mathcal{I}_{\Lambda_8}^{E_8}\big(E_{(4+\delta)\Lambda_1}^{SL(2)}, 5+2\epsilon\big)$ includes a non-analytic factor in $\frac{\xi(\delta-2\epsilon)}{\xi(\delta+2\epsilon)} \sim \frac{\delta+2\epsilon}{\delta-2\epsilon}$ near $(\delta,\epsilon) = (0,0)$, such that the finite value at $(\delta,\epsilon) = (0,0)$ depends on direction in which it is approached in $\mathbb{C}^2$.

## E.2 Abelian Fourier coefficients

We now consider the abelian Fourier coefficients coming from the sixth layer, since the contributions from the other layers were already discussed in Section 4.2. Combining the results of the last section, and using the same method as in Section 3.3 for the weak coupling limit in $D = 4$, one concludes that they take the form

$$
\begin{aligned}
\mathcal{I}_7^{(6c)} &= \sum_{\gamma \in P_6 \backslash E_7} \Bigg( \frac{R^4}{y^5} \sum_{\substack{Q \in \mathbb{Z}^2 \otimes II_{5,5} \\ \Delta(Q) \geq 1}} \sum_{\substack{A \in \mathbb{Z}^{2 \times 2}/GL(2,\mathbb{Z}) \\ A^{-1}Q \in \mathbb{Z}^2 \otimes II_{5,5}}} \sum_{d | A^{-1}Q \cdot Q^\intercal A^{-\intercal}} d^{-3} \tilde{c}\big(\tfrac{\Delta(Q)}{|A|^2 d^2}\big) \sum_{\substack{r \geq 1 \\ \ell \in \mathbb{Z}^2 \\ \gcd(r,\ell)=1 \\ r | A^{-1}Q}} \frac{1}{\big(1 + \frac{1}{R^2 y} v(\tilde{\ell}, \tilde{\ell})\big)^{\frac{5}{2}}} \\
&\times \sum_{\substack{k \geq 1 \\ p \in S_+ \\ \gcd(k,p)=1 \\ \frac{(p\gamma p)}{2k} \in \mathbb{Z} \\ \frac{A^{-1}Qp}{k} \in S_- \\ \frac{A^{-1}pQp}{2k^2} \in \mathbb{Z}}} \int_{\mathcal{H}_+} \frac{\mathrm{d}^3\Omega_2}{|\Omega_2|} \bigg( \frac{4\pi \Delta(Q)}{L(\tilde{p}, \tilde{\ell})} + \frac{5}{\pi |\Omega_2|} \Big( \frac{1}{L(\tilde{p}, \tilde{\ell})^3} + \frac{\pi}{L(\tilde{p}, \tilde{\ell})^2} \mathrm{tr}[\Omega_2 Q \cdot Q^\intercal] \Big) \bigg) \\
&\times e^{-\pi \mathrm{tr}[\Omega_2(L(\tilde{p}, \tilde{\ell}) Q \cdot Q^\intercal + (1 + \frac{1}{R^2 y} v(\tilde{\ell}, \tilde{\ell})) Q u Q^\intercal + \frac{1}{R^2 y^{\frac{1}{2}}} \tilde{p}^\intercal Q u Q^\intercal \tilde{p} + \frac{1}{4} \frac{\tilde{p} Q \tilde{p} \tilde{p} Q^\intercal \tilde{p}}{R^4 y + R^2 v(\tilde{\ell}, \tilde{\ell})})] - \frac{\pi R^2}{\sqrt{1 + \frac{1}{R^2 y} v(\tilde{\ell}, \tilde{\ell})}} \mathrm{tr}[\Omega_2^{-1} \varepsilon(v + \frac{1}{R^2 y} \tilde{\ell}\tilde{\ell}^\intercal)\varepsilon^\intercal]} \\
&\times e^{2\pi \mathrm{i}\big(Q, a_{\hat{\imath}} + c_{\hat{\imath}} \gamma(\frac{p}{k} + a + \frac{1}{2} c^{\hat{\jmath}}(\frac{\ell_{\hat{\jmath}}}{r} + a_{\hat{\jmath}})) + c(\frac{\ell_{\hat{\imath}}}{r} + a_{\hat{\imath}})\big)} \Bigg) \Bigg|_\gamma
\end{aligned}
\tag{E.34}
$$

where

$$
\begin{aligned}
\Delta(Q) &= \det[\eta^{ab} Q_{\hat{\imath}a} Q_{\hat{\jmath}b}] , \tag{E.35} \\
L(\tilde{p}, \tilde{\ell}) &= \sqrt{1 + \frac{1}{R^2 y} v(\tilde{\ell}, \tilde{\ell}) + \frac{1}{R^2 y^{\frac{1}{2}}} u(\tilde{p}, \tilde{p}) + \frac{1}{4} \frac{u(\tilde{p}\gamma\tilde{p}, \tilde{p}\gamma\tilde{p})}{R^4 y + R^2 v(\tilde{\ell}, \tilde{\ell})}} .
\end{aligned}
$$

To exhibit the Fourier expansion, we still need to decompose the sum over $(k, p)$ into $p \bmod k$ and the integral part $p'$, and to use the Poisson formula on the sum over $p'$. We define the function

$$
f_Q\big(\tfrac{\tilde{p}}{R y^{\frac{1}{4}}}, \tfrac{\tilde{\ell}}{R y^{\frac{1}{2}}}\big) = \int_{\mathcal{H}_+} \frac{\mathrm{d}^3\Omega_2}{|\Omega_2|} \bigg( \frac{4\pi \Delta(Q)}{L(\tilde{p}, \tilde{\ell})} + \frac{5}{\pi |\Omega_2|} \Big( \frac{1}{L(\tilde{p}, \tilde{\ell})^3} + \frac{\pi}{L(\tilde{p}, \tilde{\ell})^2} \mathrm{tr}[\Omega_2 Q \cdot Q^\intercal] \Big) \bigg)
\tag{E.36}
$$
$$
\times e^{-\pi \mathrm{tr}[\Omega_2(L(\tilde{p}, \tilde{\ell}) Q \cdot Q^\intercal + (1 + \frac{1}{R^2 y} v(\tilde{\ell}, \tilde{\ell})) Q u Q^\intercal + \frac{1}{R^2 y^{\frac{1}{2}}} \tilde{p}^\intercal Q u Q^\intercal \tilde{p} + \frac{1}{4} \frac{\tilde{p} Q \tilde{p} \tilde{p} Q^\intercal \tilde{p}}{R^4 y + R^2 v(\tilde{\ell}, \tilde{\ell})})] - \frac{\pi R^2}{\sqrt{1 + \frac{1}{R^2 y} v(\tilde{\ell}, \tilde{\ell})}} \mathrm{tr}[\Omega_2^{-1} \varepsilon(v + \frac{1}{R^2 y} \ell\ell^\intercal)\varepsilon^\intercal]}
$$

which can be evaluated in terms of matrix variate Bessel functions [80, 50] if so desired, and its Fourier transform

$$
\tilde{f}_Q(\chi, \lambda) = \int \mathrm{d}^2\tilde{\ell} \int \mathrm{d}^{16}\tilde{p} \, f_Q(\tilde{p}, \tilde{\ell}) e^{2\pi \mathrm{i}(\chi, \tilde{p}) + 2\pi \mathrm{i}(\lambda, \ell)} ,
\tag{E.37}
$$

where we have rescaled variables such that $\tilde{f}_Q(\chi, \lambda)$ does not depend on $y$. While we do not have an explicit formula for $\tilde{f}_Q(\chi, \lambda)$, we note that the integral is absolutely convergent. The

generic Fourier coefficients can be written as[38]

$$
\mathcal{I}_7^{(6c)} = R^{22} \sum_{\gamma \in P_6 \backslash E_7} \Bigg( \sum_{\substack{Q \in \mathbb{Z}^2 \otimes II_{5,5} \\ \Delta(Q) \geq 1}} \sum_{\substack{\chi \in S_+ \\ \lambda \in \mathbb{Z}^2}} \sum_{\substack{A \in \mathbb{Z}^{2\times2}/GL(2,\mathbb{Z}) \\ A^{-1}Q \in \mathbb{Z}^2 \otimes II_{5,5}}} \sum_{d|A^{-1}Q \cdot Q^\intercal A^{-\intercal}} d^{-3} \tilde{c}\big(\tfrac{\Delta(Q)}{|A|^2 d^2}\big)
$$

$$
\times \sum_{\substack{k \geq 1 \\ p \in S_+ \bmod kS_+ \\ \frac{(p\gamma p)}{2k} \in II_{5,5} \\ \frac{A^{-1}Qp}{k} \in S_- \\ \frac{A^{-1}{}_p Qp}{2k^2} \in \mathbb{Z}}} e^{2\pi i(\frac{p}{k},\chi)} \sum_{\substack{r \geq 1 \\ \ell \in \mathbb{Z}^2 \bmod r\mathbb{Z}^2 \\ r|A^{-1}Q}} e^{2\pi i(\frac{\ell}{r},\lambda)}
$$

$$
\times \tilde{f}_Q \big( R y^{\frac{1}{4}} (\chi - Q^{\hat{i}} c_i), R y^{\frac{1}{2}} (\lambda^{\hat{i}} - c^{\hat{i}} \chi + \tfrac{1}{2} c_{\hat{j}} Q^{\hat{j}} c^{\hat{i}} - Q_a^{\hat{i}} c^a)\big) e^{2\pi i(Q,a) + 2\pi i(\chi,a) + 2\pi i(\ell,a)} \Bigg)\Bigg|_\gamma . \quad \text{(E.38)}
$$

where the coefficients $\tilde{c}(n)$ were defined in (A.12). As expected for a generic Fourier coefficient saturating the Gelfand–Kirillov dimension of the automorphic representation, these Fourier coefficients decompose into a measure factor

$$
\mu_{P_6}(Q,\chi,\lambda) = \sum_{\substack{A \in \mathbb{Z}^{2\times2}/GL(2,\mathbb{Z}) \\ A^{-1}Q \in \mathbb{Z}^2 \otimes II_{5,5}}} \sum_{d|A^{-1}Q \cdot Q^\intercal A^{-\intercal}} d^{-3} \tilde{c}\big(\tfrac{\Delta(Q)}{|A|^2 d^2}\big) \sum_{\substack{k \geq 1 \\ p \in S_+ \bmod kS_+ \\ \frac{(p\gamma p)}{2k} \in II_{5,5} \\ \frac{A^{-1}Qp}{k} \in S_- \\ \frac{A^{-1}{}_p Qp}{2k^2} \in \mathbb{Z}}} e^{2\pi i(\frac{p}{k},\chi)} \sum_{\substack{r \geq 1 \\ \ell \in \mathbb{Z}^2 \bmod r\mathbb{Z}^2 \\ r|A^{-1}Q}} e^{2\pi i(\frac{\ell}{r},\lambda)}
$$

$$
\tag{E.39}
$$

and a real part given by the function $R^{22} \tilde{f}_Q$ of $R$ and the Levi factor $v$ acting on the charge $\Gamma = (0,0,Q,\chi,\lambda)$ only. Note indeed that the dependence of the function in $y$ and the axions $c^{\hat{i}}$, $c_a$ is manifestly covariant under $P_6$. The main complication in this formula is the Poincaré sum over $P_6 \backslash E_7$. One must still determine the set of $(Q,\chi,\lambda)$ mapping to the same charge $\Gamma \in M_{\Lambda_7}^{E_7}$ under the Poincaré sum.

The computation simplifies drastically if the charge $\Gamma$ is projective according to the definition given in [69]. Any primitive charge $\Gamma \in M_{\Lambda_7}^{E_7}$ (with $\gcd(\Gamma) = 1$) can be rotated by $E_7(\mathbb{Z})$ to a doublet of vectors $(Q_1, Q_2) \in II_{2,2}$ (corresponding to the so-called STU truncation with a single magnetic charge $p^0 = 1$)

$$
Q_1 = e_{1+} + q_1 e_{1-} , \qquad Q_2 = q_2 e_{2+} + q_3 e_{2-} + q_0 e_{1-} , \tag{E.40}
$$

for a specific basis of light-like vectors $e_{i\pm}$ normalised such that

$$
(e_{i\pm}, e_{j\pm}) = 0 , \qquad (e_{i+}, e_{j-}) = \delta_{ij} . \tag{E.41}
$$

A primitive charge is moreover projective if and only if (where $q_{I+3} = q_I$)

$$
\gcd(q_0, q_I, q_{I+1} q_{I+2}) = 1 \qquad \text{for } I = 1, 2, 3 . \tag{E.42}
$$

---

[38]The condition $d|Q \cdot Q^\intercal$ is a shorthand notation for $d|(Q_1,Q_1)/2, (Q_2,Q_2)/2, (Q_1,Q_2)$.

If $\Delta(\Gamma) = 1 \bmod 4$ (i.e. $q_0$ odd), a charge is projective if and only if $\gcd(\frac{1}{2}\Delta'(\Gamma)) = 1$ [69], with

$$\frac{1}{2}\Delta'(Q_1, Q_2) = \begin{pmatrix} -q_0 e_{1+} + q_0 q_1 e_{1-} + 2q_1 q_2 e_{2+} + 2q_1 q_3 e_{2-} \\ -2q_2 q_3 e_{1+} + (q_0^2 - 2q_1 q_2 q_3)e_{1-} + q_0 q_2 e_{2+} + q_0 q_3 e_{2-} \end{pmatrix}. \tag{E.43}$$

Considering the representative (E.40), one finds that (E.40) for $i = 1$ gives that $\gcd(Q_{\hat{\imath}} \cdot Q_{\hat{\jmath}}) = 1$ and (E.40) for $i = 2$ implies $\gcd(Q_1 \wedge Q_2) = 1$. Since $\gcd(Q) = 1$, it follows that the only matrix that divides $(Q_1, Q_2)$ is the identity $A = \mathbb{1}$, and the only integer dividing the norms are $d = 1$ and $k = 1$. In this case there is no sum over $p$ and the measure reduces to $\tilde{c}(\Delta(Q)) = \tilde{c}(\Delta(Q, \chi, \lambda))$ where the first $\Delta$ is the quartic invariant of $SO(5,5)$, while the second is the one of $E_7$. Since the measure factor is the same for all representatives, the Poincaré sum will not modify the measure in this case, and one recovers the expected index of BPS black holes in four dimensions determined in [51–53].

A slightly more general orbit of charges is defined by primitive charges with $\gcd(\Gamma \times \Gamma)' = 1$, where $(\Gamma \times \Gamma)'$ includes all components of $\Gamma \times \Gamma$, except for the possibly half-integer $E_6$ singlet, i.e. for the representative (E.40)

$$\Gamma \times \Gamma = \left\{ q_0, q_I, q_{I+1}q_{I+2}, \tfrac{3}{2}q_0 \mid I = 1, 2, 3 \right\}, \qquad (\Gamma \times \Gamma)' = \left\{ q_0, q_I, q_{I+1}q_{I+2} \mid I = 1, 2, 3 \right\}. \tag{E.44}$$

Note that the condition $\gcd(\Gamma \times \Gamma)' = 1$ is $E_7(\mathbb{Z})$ invariant [69]. The helicity supertrace counting 1/8-BPS states with such charges was determined in [81] as

$$\Omega_{14}(\Gamma) \underset{\substack{\gcd(\Gamma)=1 \\ \gcd(\Gamma \times \Gamma)'=1}}{=} \sum_{d \mid \Gamma \wedge \frac{1}{4}\Delta'(\Gamma)} d\, \tilde{c}\left(\tfrac{\Delta(\Gamma)}{d^2}\right). \tag{E.45}$$

For a charge $\Gamma = (0, 0, Q, 0, 0)$, it can be written in a way similar to (E.39), namely

$$\Omega_{14}(Q) \underset{\substack{\gcd(\Gamma)=1 \\ \gcd(\Gamma \times \Gamma)'=1}}{=} \sum_{\substack{A \in \mathbb{Z}^{2 \times 2}/GL(2,\mathbb{Z}) \\ A^{-1}Q \in \mathbb{Z}^2 \otimes II_{5,5}}} \sum_{d \mid A^{-1}Q \cdot Q^{\mathsf{T}} A^{-\mathsf{T}}} |A|\, d\, \tilde{c}\left(\tfrac{\Delta(Q)}{|A|^2 d^2}\right). \tag{E.46}$$

Indeed, the set of matrices modulo $GL(2, \mathbb{Z})$ that divides $(Q_1, Q_2)$ is restricted in this case to diagonal matrices parametrised by one integer $k$ such that $k$ divides $Q_2$. They are the same as the integers dividing

$$Q_1 \wedge Q_2 = (q_2, q_3, q_0, q_1 q_2, q_1 q_3). \tag{E.47}$$

The second condition on $d$ dividing $A^{-1}Q \cdot Q^{\mathsf{T}} A^{-\mathsf{T}}$ is that it divides $(q_1, \frac{q_0}{k}, \frac{q_2 q_3}{k^2})$, but since $q_1$ is coprime to $\gcd(q_2, q_3, q_0, q_1 q_2, q_1 q_3)$ by the assumption that $\gcd(\Gamma \times \Gamma)' = 1$, $d$ must be coprime to $k$ and divide $\gcd(q_2, q_3, q_0, q_1 q_2, q_1 q_3)$. The sum over $d$ is then over the integers dividing $(q_1, q_0, q_2 q_3)$, independently of $k$ dividing $(q_2, q_3, q_0, q_1 q_2, q_1 q_3)$. This sum is then the same as the one over all the integers $d' = dk$ dividing $\Gamma \wedge \frac{1}{4}\Delta'(\Gamma)$ in (E.45).

It is reasonable to expect that upon taking into account the different representatives of the same charge $\Gamma = (0, 0, Q, 0, 0)$ under the Poincaré sum $P_6 \backslash E_7$, the measure (E.39) will be modified to (E.46). However, the latter is not invariant under triality (permutations of $I = 1, 2, 3$) for more general charges, so that it depends on the chosen representative charge (E.40) in general and it is therefore too naïve to hope that the Poincaré sum over $P_6 \backslash E_7$ gives simply (E.46) out of (E.39) for $\gcd(\Gamma \times \Gamma)' \neq 1$.

# F $\quad \nabla^6 \mathcal{R}^4$ in $D \geq 8$

In this section, we briefly discuss the explicit form of the $\nabla^6 \mathcal{R}^4$ coupling at small $d \leq 2$, in relation to earlier proposals in the literature.

## F.1 $\quad D = 10$ type IIB

In [31] it was proposed that the exact $\nabla^6 \mathcal{R}^4$ coupling in ten-dimensional type IIB string theory is given by the two-loop amplitude in 11D supergravity compactified on $T^2$, with metric $g_{ij} = \frac{\sqrt{\det g}}{U_2} \begin{pmatrix} 1 & U_1 \\ U_1 & |U|^2 \end{pmatrix}$, where $U$ is identified with the type IIB axiodilaton. In the notation of the present paper, this amounts to

$$
\begin{aligned}
\mathcal{E}^{(0)}_{(0,1)}(U) &= \frac{8\pi^2}{3} \int_{\mathcal{G}} \frac{\mathrm{d}V}{V^4} \frac{\mathrm{d}\tau_1 \mathrm{d}\tau_2}{\tau_2^2} A(\tau) \sum_{M \in \mathbb{Z}^{2\times 2}}' e^{-\frac{2\pi}{V} \det M - \frac{\pi}{V\tau_2 U_2} \left| (1,U) M \left( \begin{smallmatrix} -\tau \\ 1 \end{smallmatrix} \right) \right|^2} \\
&= \frac{8\pi^2}{3} \widetilde{A}_{3/2}(U)
\end{aligned} \tag{F.1}
$$

where $\widetilde{A}_s$ was introduced in (2.43). The weak coupling expansion can be obtained from (B.25) and reproduces the known perturbative terms, as well as the instanton and anti-instanton effects which were inferred in [32] by solving the Poisson equation (B.15).

## F.2 $\quad D = 9$

The exact $\nabla^6 \mathcal{R}^4$ coupling in $D = 9$ was proposed in [22] to be given by

$$
\mathcal{E}^{(1)}_{(0,1)} = \nu^{-\frac{6}{7}} \mathcal{E}^{(0)}_{(0,1)} + \frac{4}{3}\zeta(2)\zeta(3)\, \nu^{\frac{1}{7}} E^{SL(2)}_{\frac{3}{2}\Lambda_1} + \frac{8}{5}\zeta(2)^2 \nu^{\frac{8}{7}} + \frac{4}{63}\zeta(2)\zeta(5)\left( \nu^{\frac{15}{7}} E^{SL(2)}_{\frac{5}{2}\Lambda_1} + \nu^{-\frac{20}{7}} \right) \tag{F.2}
$$

where

$$
\nu = \left( \frac{r}{\ell_s} \right)^{7/4} \sqrt{g_9}, \quad U = C_0 + \mathrm{i}\frac{\sqrt{r/\ell_s}}{g_9} \tag{F.3}
$$

and $\mathcal{E}^{(0)}_{(0,1)}(U)$ is the function (F.1) which governs the $\nabla^6 \mathcal{R}^4$ term in ten-dimensional type IIB string theory. In our formalism, the last two terms come from the 1-loop exceptional field theory amplitude

$$
\mathcal{F}^{(1)}_{(0,1)} = \frac{4\zeta(2)\zeta(5)}{63} \left[ \nu^{\frac{15}{7}} \widehat{E}^{SL(2)}_{\frac{5}{2}\Lambda_1} + \nu^{-\frac{20}{7}} \right] \tag{F.4}
$$

while the two-loop amplitude in exceptional field theory accounts for the term

$$
\mathcal{E}^{(1),\text{ExFT}}_{(0,1)} = \nu^{-\frac{6}{7}} \mathcal{E}^{(0)}_{(0,1)} + \frac{8}{5}\zeta(2)^2 \nu^{\frac{8}{7}}. \tag{F.5}
$$

The remaining contribution $\frac{4}{3}\zeta(2)\zeta(3)\, \nu^{\frac{1}{7}} E^{SL(2)}_{\frac{3}{2}\Lambda_1}$ does not appear as a 1/2-BPS particle state sum and instead resembles a string multiplet state sum.

**F.3  $D = 8$**

The exact $\nabla^6 \mathcal{R}^4$ coupling in $D = 8$ was proposed in [22], using results from [33], as

$$
\begin{aligned}
\mathcal{E}_{(0,1)}^{(2)} &= \mathcal{E}_{(0,1)}^{SL(3)} + \mathcal{E}_{(0,1)}^{SL(2)} + \frac{4}{3}\zeta(2)\zeta(3)\,\widehat{E}_{\frac{3}{2}\Lambda_1}^{SL(3)}\,\widehat{E}_{\Lambda_1}^{SL(2)} + \frac{\pi\zeta(3)}{18}\widehat{E}_{\frac{3}{2}\Lambda_1}^{SL(3)} + \frac{2\pi}{9}\zeta(2)\widehat{E}_{\Lambda_1}^{SL(2)} + \frac{\zeta(2)}{9} \\
&\quad + \frac{4\zeta(6)}{27}\,E_{-\frac{3}{2}\Lambda_1}^{SL(3)}\,E_{3\Lambda_1}^{SL(2)}
\end{aligned}
\tag{F.6}
$$

where $\mathcal{E}_{(0,1)}^{SL(2)}$ and $\mathcal{E}_{(0,1)}^{SL(3)}$ are solutions to Poisson-type equations

$$
\begin{aligned}
(\Delta_U - 12)\,\mathcal{E}_{(0,1)}^{SL(2)} &= -\left(4\zeta(2)\widehat{E}_{\Lambda_1}^{SL(2)}\right)^2 , \\
(\Delta_{SL(3)} - 12)\mathcal{E}_{(0,1)}^{SL(3)} &= -\left(2\zeta(3)\widehat{E}_{\frac{3}{2}\Lambda_1}^{SL(3)}\right)^2 .
\end{aligned}
\tag{F.7}
$$

with suitable asymptotics. The last term is recognised as the homogeneous solution (1.11),

$$
\mathcal{F}_{(0,1)}^{(2)} = \frac{4\zeta(6)}{27}\,E_{-\frac{3}{2}\Lambda_1}^{SL(3)}\,E_{3\Lambda_1}^{SL(2)} ,
\tag{F.8}
$$

To see the origin of the other terms, note that the particle multiplet transforms as $(\bar{\mathbf{3}}, \mathbf{2})$ under $SL(3) \times SL(2)$. The double lattice sum therefore decomposes into

$$
\sideset{}{'}\sum_{\substack{\Gamma_i \in \mathbb{Z}^{2\times 3} \\ \Gamma_i \times \Gamma_j = 0}} e^{-\pi\Omega_2^{ij}G(\Gamma_i,\Gamma_j)}
\tag{F.9}
$$

$$
= \left( \sum_{\substack{q_i \in \mathbb{Z}^2 \\ q_i \wedge q_j \neq 0}} \sum_{\gamma \in P_2 \backslash SL(3)} e^{-\pi\Omega_2^{ij}y_2^2 G(q_i,q_i)}\Big|_\gamma + \sum_{\substack{p_i \in \mathbb{Z}^3 \\ p_i \wedge p_j \neq 0}} \sum_{\gamma \in P_1 \backslash SL(2)} e^{-\pi\Omega_2^{ij}y_1^2 G(p_i,p_i)}\Big|_\gamma \right.
$$

$$
\left. + \sideset{}{'}\sum_{n_i \in \mathbb{Z}^2} \sum_{\gamma \in P_2 \backslash SL(3) \times P_1 \backslash SL(2)} e^{-\pi\Omega_2^{ij}y_1^2 y_2^2 n_i n_j}\Big|_\gamma \right)
$$

The first two terms can be further decomposed into an unconstrained sum minus the sum over collinear charges that can be computed using (B.25) as

$$
8\pi \int_{\mathcal{G}} \frac{\mathrm{d}^3\Omega_2}{|\Omega_2|^{2-\epsilon}}\varphi_{\mathrm{KZ}}^{\mathrm{tr}} \sum_{\substack{q_i \in \mathbb{Z}^2 \\ q_i \wedge q_j \neq 0}} e^{-\pi\Omega_2^{ij}G(q_i,q_i)} = \frac{8\pi^2}{3}\widetilde{A}_\epsilon - \frac{16\pi^2 \xi(2\epsilon)^2}{(4-2\epsilon)(3+2\epsilon)}E_{2\epsilon\Lambda_1}^{SL(2)} ,
\tag{F.10}
$$

$$
8\pi \int_{\mathcal{G}} \frac{\mathrm{d}^3\Omega_2}{|\Omega_2|^{2-\epsilon}}\varphi_{\mathrm{KZ}}^{\mathrm{tr}} \sum_{\substack{p_i \in \mathbb{Z}^3 \\ p_i \wedge p_j \neq 0}} e^{-\pi\Omega_2^{ij}G(p_i,p_i)} = \frac{8\pi^2}{3}\sum_{\gamma \in P_1 \backslash SL(3)} \widetilde{A}_\epsilon\Big|_\gamma - \frac{16\pi^2 \xi(2\epsilon)^2}{(4-2\epsilon)(3+2\epsilon)}E_{2\epsilon\Lambda_2}^{SL(3)} .
$$

These combinations are finite as $\epsilon \to 0$, as can be checked using (B.25) for the first, and

$$8\pi g_8^{-\frac{2\epsilon}{3}} \int_{\mathcal{G}} \frac{\mathrm{d}^3\Omega_2}{|\Omega_2|^{2-\epsilon}} \varphi_{\mathrm{KZ}}^{\mathrm{tr}} \sum_{q_i \in \mathbb{Z}^3}' e^{-\pi\Omega_2^{ij} G(p_i,p_i)} = \frac{8\pi^2}{3} g_8^{-2\epsilon} \widetilde{A}_\epsilon(T) + \frac{16\pi^2 \xi(2\epsilon)^2}{(6-2\epsilon)(1+2\epsilon)} g_8^{-4+2\epsilon}$$

$$+ \frac{8\pi^2}{3} \xi(2-2\epsilon)\xi(3-2\epsilon) g_8^{-2} E_{(1-\epsilon)\Lambda_1}^{SL(2)} + \frac{4\pi^2}{9} \xi(2+2\epsilon)\xi(6-2\epsilon) g_8^2 E_{(3-\epsilon)\Lambda_1}^{SL(2)} + \dots$$

$$+ \frac{16\pi^2}{3} g_8^{-1-\epsilon} \sum_{N\in\mathbb{Z}^2}' \left( \frac{\sigma_{2-2\epsilon}(N)}{\gcd(N)^{-2\epsilon}} \xi(2\epsilon) \frac{K_{1-\epsilon}\left(\frac{2\pi}{g_8}\frac{|N_1+TN_2|}{\sqrt{T_2}}\right)}{\left(\frac{|N_1+TN_2|}{\sqrt{T_2}}\right)^{1+\epsilon}} \right.$$

$$\left. + \frac{g_8^2}{6} \frac{\sigma_{2+2\epsilon}(N)}{\gcd(N)^6} \xi(5-2\epsilon) \frac{K_{1+\epsilon}\left(\frac{2\pi}{g_8}\frac{|N_1+TN_2|}{\sqrt{T_2}}\right)}{\left(\frac{|N_1+TN_2|}{\sqrt{T_2}}\right)^{\epsilon-5}} + \dots \right) e^{2\pi\mathrm{i}(N,a)} , \quad \text{(F.11)}$$

for the second. This expansion in turn follows from (3.13), (3.18), (3.22) and (3.66) up to exponentially suppressed terms (represented by the dots) that are finite at $\epsilon \to 0$. Therefore one can set $\epsilon = 0$ in the first term

$$\int_{\mathcal{G}} \frac{\mathrm{d}^3\Omega_2}{|\Omega_2|^{2-\epsilon}} \sum_{\substack{q_i \in \mathbb{Z}^2 \\ q_i\wedge q_j\neq 0}} \sum_{\gamma\in P_2\backslash SL(3)} e^{-\pi\Omega_2^{ij} y_2^2 G(q_i,q_i)}\Big|_\gamma = E_{2\epsilon\Lambda_2}^{SL(3)} \int_{\mathcal{G}} \frac{\mathrm{d}^3\Omega_2}{|\Omega_2|^{2-\epsilon}} \varphi_{\mathrm{KZ}}^{\mathrm{tr}} \sum_{\substack{q_i \in \mathbb{Z}^2 \\ q_i\wedge q_j\neq 0}} e^{-\pi\Omega_2^{ij} G(q_i,q_i)}$$

$$\underset{\epsilon\to 0}{=} \int_{\mathcal{G}} \frac{\mathrm{d}^3\Omega_2}{|\Omega_2|^2} \varphi_{\mathrm{KZ}}^{\mathrm{tr}} \sum_{\substack{q_i \in \mathbb{Z}^2 \\ q_i\wedge q_j\neq 0}} e^{-\pi\Omega_2^{ij} G(q_i,q_i)} , \quad \text{(F.12)}$$

and the second

$$\int_{\mathcal{G}} \frac{\mathrm{d}^3\Omega_2}{|\Omega_2|^{2-\epsilon}} \sum_{\substack{p_i \in \mathbb{Z}^3 \\ p_i\wedge p_j\neq 0}} \sum_{\gamma\in P_1\backslash SL(2)} e^{-\pi\Omega_2^{ij} y_1^2 G(p_i,p_i)}\Big|_\gamma = E_{2\epsilon\Lambda_1}^{SL(2)} \int_{\mathcal{G}} \frac{\mathrm{d}^3\Omega_2}{|\Omega_2|^{2-\epsilon}} \varphi_{\mathrm{KZ}}^{\mathrm{tr}} \sum_{\substack{p_i \in \mathbb{Z}^3 \\ p_i\wedge p_j\neq 0}} e^{-\pi\Omega_2^{ij} G(p_i,p_i)}$$

$$\underset{\epsilon\to 0}{=} \int_{\mathcal{G}} \frac{\mathrm{d}^3\Omega_2}{|\Omega_2|^2} \varphi_{\mathrm{KZ}}^{\mathrm{tr}} \sum_{\substack{p_i \in \mathbb{Z}^3 \\ p_i\wedge p_j\neq 0}} e^{-\pi\Omega_2^{ij} G(q_i,q_i)} . \quad \text{(F.13)}$$

As for the mixed term, we get, after integrating over the volume factor and using (B.2),

$$8\pi \int_{\mathcal{G}} \frac{\mathrm{d}^3\Omega_2}{|\Omega_2|^{2-\epsilon}} \varphi_{\mathrm{KZ}}^{\mathrm{tr}} \sum_{n_i \in \mathbb{Z}^2}' \sum_{\gamma\in P_2\backslash SL(3)\times P_1\backslash SL(2)} e^{-\pi\Omega_2^{ij} y_1^2 y_2^2 n_i n_j}\Big|_\gamma = \frac{16\pi^2 \xi(2\epsilon)^2}{(4-2\epsilon)(3+2\epsilon)} E_{2\epsilon\Lambda_1}^{SL(2)} E_{2\epsilon\Lambda_2}^{SL(3)} .$$

$$\text{(F.14)}$$

As in (5.37), this function is divergent and one needs to take into account the contribution from the supergravity amplitude and the $\mathcal{R}^4$ form factor associated to the partly massless contribution, giving[39]

$$\frac{16\pi^2 \xi(2\epsilon)^2}{(4-2\epsilon)(3+2\epsilon)} E_{2\epsilon\Lambda_1}^{SL(2)} E_{2\epsilon\Lambda_2}^{SL(3)} + \frac{\pi}{3} \frac{\Gamma(\epsilon)}{(\pi\mu^2)^\epsilon} \mathcal{E}_{(0,0),\epsilon}^{(2)} + \frac{\pi^2}{3} \left( \frac{\Gamma(\epsilon)^2}{\pi^{2\epsilon}} + \frac{1}{6}\frac{\Gamma(\epsilon)}{\pi^\epsilon} + \mathcal{O}(\epsilon^0) \right) \mu^{-4\epsilon}$$

$$\underset{\epsilon\to 0}{\sim} \frac{4}{3}\zeta(2)\zeta(3) \widehat{E}_{\frac{3}{2}\Lambda_1}^{SL(3)} \widehat{E}_{\Lambda_1}^{SL(2)} + \frac{\pi^2}{3} \partial_s^2 \left( E_{s\Lambda_1}^{SL(2)} + E_{s\Lambda_1}^{SL(3)} \right)\big|_{s=0} + \frac{\pi}{18} \mathcal{E}_{(0,0)}^{(2)} + \frac{11\pi^2}{108}$$

$$+ \frac{4\pi^2}{3} \log(2\pi\mu)^2 - \frac{2\pi}{3} \log(2\pi\mu) \left( \mathcal{E}_{(0,0)}^{(2)} + \frac{\pi}{3} \right) . \quad \text{(F.15)}$$

---

[39]Observe that the first term by itself produces $\frac{8}{3}\zeta(2)\zeta(3) \widehat{E}_{\frac{3}{2}\Lambda_1}^{SL(3)} \widehat{E}_{\Lambda_1}^{SL(2)}$, which is twice the correct result.

One can then identify the automorphic forms $\mathcal{E}_{(0,1)}^{SL(2)}$ and $\mathcal{E}_{(0,1)}^{SL(3)}$ introduced above as

$$
\begin{aligned}
\mathcal{E}_{(0,1)}^{SL(2)} &= \left( \frac{8\pi^2}{3} \widetilde{A}_\epsilon - \frac{16\pi^2 \xi(2\epsilon)^2}{(4-2\epsilon)(3+2\epsilon)} E_{2\epsilon\Lambda_1}^{SL(2)} \right)\Big|_{\epsilon=0} + \frac{\pi^2}{3} \partial_s^2 E_{s\Lambda_1}^{SL(2)}\big|_{s=0} + \frac{\pi}{9}\zeta(2)\widehat{E}_{\Lambda_1}^{SL(2)} + \frac{13\pi^2}{216}\,, \\
\mathcal{E}_{(0,1)}^{SL(3)} &= 8\pi \int_{\mathcal{G}} \frac{\mathrm{d}^3\Omega_2}{|\Omega_2|^2} \varphi_{\mathrm{KZ}}^{\mathrm{tr}} \sum_{\substack{q_i \in \mathbb{Z}^3 \\ p_i \wedge p_j \neq 0}} e^{-\pi\Omega_2^{ij}G(p_i,p_i)} + \frac{\pi^2}{3}\partial_s^2 E_{s\Lambda_1}^{SL(2)}\big|_{s=0} + \frac{\pi}{9}\zeta(3)\widehat{E}_{\frac{3}{2}\Lambda_1}^{SL(3)} + \frac{13\pi^2}{216}\,.
\end{aligned}
$$

$$(\text{F.16})$$

One checks using (B.25) that they have indeed the same constant terms as [2, (B.25)]. We conclude that summing all contributions we reproduce the expected coupling $\mathcal{E}_{(0,1)}^{(2)}$ in (F.6) with

$$
\begin{aligned}
& 8\pi \int_{\mathcal{G}} \frac{\mathrm{d}^3\Omega_2}{|\Omega_2|^{2-\epsilon}} \varphi_{\mathrm{KZ}}^{\mathrm{tr}} \sideset{}{'}\sum_{\substack{\Gamma_i \in \mathbb{Z}^{2\times 3} \\ \Gamma_i \times \Gamma_j = 0}} e^{-\pi\Omega_2^{ij}G(\Gamma_i,\Gamma_j)} + \mathcal{F}_{(0,1)}^{(2)} \\
& \quad + \frac{\pi}{3}\frac{\Gamma(\epsilon)}{(\pi\mu^2)^\epsilon}\mathcal{E}_{(0,0),\epsilon}^{(2)} + \frac{\pi}{3}\left( \frac{\Gamma(\epsilon)^2}{\pi^{2\epsilon}} + \frac{1}{6}\frac{\Gamma(\epsilon)}{\pi^\epsilon} + \mathcal{O}(\epsilon^0) \right)\mu^{-4\epsilon} + \frac{\pi}{18}\mathcal{E}_{(0,0)}^{(2)} + \frac{2\zeta(2)}{9} \\
& = \mathcal{E}_{(0,1)}^{(2)} + \frac{4\pi^2}{3}\log(2\pi\mu)^2 - \frac{2\pi}{3}\log(2\pi\mu)\left( \mathcal{E}_{(0,0)}^{(2)} + \frac{\pi}{3} \right)\,,
\end{aligned}
$$

$$(\text{F.17})$$

where the last two terms in the second line are scheme dependent terms which can be reabsorbed in the definition of the infrared cutoff $\mu$ of the non-local component of the amplitude.

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
