# Peer review of "/8-BPS Couplings and Exceptional Automorphic Functions"

_SciPost Physics_

## Round 1 · Referee Report · Anonymous (Referee 1) · 2020-3-4

Strengths

1-delivers long sought-after result, resolves puzzles
2-amazing structural clarity, good explanations
3-very good introduction in a complex field, with a lot of references

Weaknesses

1-complexity of the content, if you are not an expert working in this particular field, you will have a lot of work reading the article. It is, however, a very well written article, so suffering is because of complexity of the topic exclusively.

Report

The article ,,1/8 BPS couplings and exceptional automorphic functions'' by
Guillaume Bossard, Axel Kleinschmidt and Boris Pioline is an interesting and
important contribution towards the determination of coeffiecients of various
couplings in the low-energy effective action of type II string theories. Of
central interest in the current article is the $\nabla^6{\cal R)^4$ coupling.

Results for the calculation of the coefficients of various couplings in
type-II-actions can be obtained by either performing a calculation in
exceptional field theory or as a direct string calculation for the appropriate
genus. While both calculations lead to automorphic forms eventually, they
theories differ by the particle content allowed to run in the ,,loop''
effectively: while particles in the exceptional field theory are confined to
1/2 BPS states, string theory allows in addition for 1/4, 1/8 BPS states as
well as non-BPS ones.

In earlier work of two of the authors, it was already suggested, that a
particular two-loop EFT calculation would deliver the result for the
$\nabla^6{\cal R)^4$ coupling up to some yet to be determined
Langlands-Eisenstein series. This gap is closed with the current article by
calculating the missing terms and interpreting their appearances (and
cancellations) between different loop orders in a string calculation.

Most of the insight relating EFT calculation to the (covariantized) string
answer rests on considering a particular representation of the Kawazumi-Zhang
invariant, which can be represented as a Poincar\'e series over its tropical
limit. This representation allows to express the string integral in terms of a
constrained lattice sum over pairs of vectors in the string multiplet lattice.
Similarly, the EFT calculation can be brought into a form of an integral over a
constrained sum over pairs of vectors/spinors in a particle multiplet lattice.
The comparison can then be performed recognizing identities between the two
lattice sums mentioned before: the central formula to consider. These lattice
identities provide the link between the two formulations and allow the careful
investigation and disection of various contributions in the sections to follow.

In short: the EFT calculation is able to provide certain subparts of the string
calculation, whose weak-coupling and decompactification (one of the
compactifications becomes large) limits allow to quantify parts of the
remaining ingredients to the final answer. In particular do the results allow
to canonically attribute properly regulated expressions to particular
coefficients in a U-duality compatible way resulting in the final formula
(1.32) (which is commonly called non-perterbative to underline the U-duality
properties).

Section 2 is devoted to the explanation (and partial derivation) of the central
lattice sum identity mentioned in the previous paragraphs. The two sides of the
identities are investigated considering appropriate Laplace identities to be
satisfied by the EFT lattice sum and the string lattice sum. Furthermore,
tensorial differential equations extending the Laplace equations are considered
and interpreted in the language of nilpotent orbits, which in turn can be used
for classification of Fourier coefficients of autormorphic forms. The equality
of the two lattice sums is then finally established by integrating over a
suitable Maa\ss eigenform and evaluating the result using functional relations
for the resulting Langlands-Eisenstein series. The discussion is complemented
by a subsection about convergence of the used expressions and a subsection
extending the lattice sums on the string side of the central formula (1.23) to
spinor representations.

In sections 3 and 4 (which are of equal structure), the weak coupling limit and
the decompactification limit of the EFT expression are considered. The analysis
relies on splitting the appropriate double theta series into different so-called
layers, which correspond to a decomposition of the lattice sum into suitable
subsets of charges, each of which allows for evaluation separately.
Calculations here are rather involved, in particular if it comes to regulating
the integrals. Several details of the calculation are referred to various
appendices, which is absolutely necessary: even after doing so the exposition
is clear but very dense.

In section 5, the various results are combined, various further expansions are
determined and regulations left out in previous sections are provided. Again,
the section is divided into considerations originating in the weak-coupling
limit and the decompactification limit. For subsection 5.2 (weak-coupling
limit), the overview of what has been done hides in a paragraph on page 63,
which I would have loved to read at the beginning of this subsection. The
natural appearance of the logarithmic term already mentioned in the
introduction is properly derived and explained afterwards. For subsection 5.3 a
similar discussion is performed, which culminates in the final (yet careful)
claim (1.31) is indeed the correct (non-perturbative) coefficient in front of
the $\nabla^6{\cal R}^4$ term.

%%%%%%%%%%%%%%%%%%%%%%%%%%%%%%%%%%%%%%%%%%%%%%%%%%%%%%%%%%%%%%%%%%%%%%%%%%%%%

Refereeing this article was an ardous task. While definitely rather close to the
research field of the referee, the actual tools used in this particular
subfield are very special and require a rather thorough knowledge of
automorphic forms and the underlying group theory/representation theory.
Moreover, it is many particular identities, special cases, nifty special
functions, particular special cases in various dimensions, which the referee
had to look up or doublecheck. I did so in most of the cases.

The first section of the article provided an excellent guideline towards a
rather steep entrance in the field. Having followed the recent developments
loosely, it was easily possible to assemble all previous information and
results consistently, mostly based on other publications of (subsets of) the
authors. The balance between restating previous results, explanations
of their origin on the one-hand side and the justification and motivation for
the investigations to follow as well as corresponding results on the other hand
is very carefully kept. Still, the complexity of arguments as well as the tools
used is rather high. In particular, a lot of work presented in the article is
not explained but referenced: this makes it sometimes difficult to follow and
cumbersome to check calculations.

Careful reading of the article required a couple of days, which I enjoyed to
the very end of the main part. The outcome -- beyond the recommendation for
publication -- is a couple of comments at the end of this report. In addition,
I have been consulting the appendices several times, however, did not check
every single argument carefully.

The article is very well written, of overall amazing structural clarity (with a
couple of points for improvement in section 5, see above). The article is
almost free of typos and great care has been taken to format long equations in
a nicely reeadible and accessible way. It is impossible within the timeframe
for refereeing such an article to check all calculations: this would amount to
several weeks of work. However, I checked some calculations explicitly. Part of
this (cross-) checking work is also done by the authors already: any hint and
crosscheck from U-duality and other results from different calculations is
carefully taken into account, explained and (positively) checked.

Thus, the result is certainly correct and constitutes the probably final step
towards the determination of the coefficient of the $\nabla^6{\cal R)^4$ term.
In total, I strongly recommend this article for publication in SciPost and
would recommend to consider some small corrections related to the suggestions
at the end of this report.

One thing I would like to mention finally is the fact, that for each of the
statements the authors have taken great care in stating whether it is proven,
conjectured, a claim. Several calculations, in particular related to analytic
continuation when regulating are probably correct, but not yet proven: I
appreciate that these uncertainties are clearly indicated (and discussed).

Requested changes

Here is a list of things, which might help to improve readibility of the article. The list is neither weighted nor complete or should pose requests: it is a mere collection of suggestions having apppeared to the referee during reading:

  1. General %%%%%%
  2. maybe a new paragraph in the abstract (before ,,Here'') would divide between previous results and new results from the current article.
  3. in the table of contents, formul\ae{} are not bold face
  4. there are a couple of occurrences of ,,2-loop'' in the text (e.g. page 12), deviating from the general strategy of writing ,,two-loop''
  5. pages 12 and 63 and in general: zero-th -> zeroth

  6. Introduction %%%%%%%%%%%%%%%

  7. page 1, first paragraph, ,,Combined with the ... from supersymmetry'': something is missing in this sentence.
  8. page 1: allows one to determine -> allows to determine
  9. the hat in eqn. (1.6) is not explained (only later an explanation shows up on page 12)
  10. the standard decomposition of \tau into \tau_1 and \tau_2 should be provided at the bottom of page 6 and before eqns.(1.14) and (1.15).
  11. the argument of {\cal I}_d in eqn.(1.17) contains \phi, which is only defined in the text above eqn.(1.1). Maybe one could repeat it here.
  12. I would suggest to make the importance of eqn. (1.23) more clear when stating it. It is one of the most referred equations in the following.
  13. page 12: contribution -> contributions
  14. page 12: cancel each others -> cancel each other

  15. From particle to ... %%%%%%%%%%%%%%%%

  16. page 14: below eqn.(2.2): I hope I didn't miss on a definition, but when D=10-d, then D+2=12-d and not 8-d.
  17. same paragraph: is there a good reference for ,,convenient convention'' and maybe as well for ,,top degree space'' and the other group theory concepts mentioned in the following paragraphs?
  18. eqn.(2.10) has been mentioned in the introduction already. Maybe one could refer back to the introduction here? I am not sure about a general strategy, as this occurs several times later.
  19. page 17: the space after a.k.a. appears to be too large.
  20. page 19, eqn. 2.19: H^{(U)}_N is probably the Hecke operator. (It is mentioned after 2.23, though in a different notation. ).
  21. page 19, eqn. 2.21: please define K_{s-\frac{1}{2}} (Bessel function). It shows up at several places below and I wouldn't consider that common knowledge.
  22. page 21: after eqn.(2.30) you mention the symmetry of the (finally) Riemann zeta function. Why should the rhs be invariant und this symmetry? Please explain.
  23. page 21: The statement right above subsection 2.5 appears strange as it stands: one better doesn't put d=3 in the expressions for the maximal weights, as this would lead to $\Lambda_2=0$. However, in the footnote on the same page the replacements appear to be correct: what did I miss?
  24. I would like to suggest a reference for relation (2.35): maybe 1511.04265 is good here, maybe also for (2.36).

  25. Weak .... %%%%%%%%%%%%

  26. page 25: at one point (probably in the introduction) the notion of ,,non-perturbative'' should be related to U-duality. It is probably clear to everybody in the field, but still.
  27. page 25: standard constant formulas -> standard constant term formulas ?
  28. page 25: it follows from this, ... -> it follows from the above equation, ....
  29. page 26: in your recapitulation you left out the argument $L$ in eqn.(3.7) as well as in the text. But in the next sentence you write about the $L$-dependent counterterm. I would suggest to stick to the original way of writing the integral.
  30. page 36: a specific choice of triality assignments ... here I would have loved a reference.
  31. page 39: the calculation of the Fourier coefficients is really tough to read. However, I wouldn't know how to improve, unfortunately.
  32. please, provide a reference for the matrix variate Bessel functions above (3.88)

  33. Decompactification limit %%%%%%%%%%%%%%%%%%%%%%%%%%%

  34. in the first paragraph: does the enhanced symmetry for d=5 actually lead to a crosscheck or a simplification?
  35. Frequently you use the phrase ,,One recognizes''. You do, for sure. I did sometimes, in many situations I had to look it up. It would pave the way to rather use something along the lines: as shown in ....

  36. Regularisation and divergences %%%%%%%%%%%%%%%%%%%%%%%%%%%%%%%%%

  37. page 58: after eqn (5.2): what is J_3? Again a Bessel-type function?
  38. page 60, eqn.(5.17) needs a full stop.
  39. page 63: second paragraph: cancel -> cancels

  • validity: top
  • significance: high
  • originality: high
  • clarity: top
  • formatting: perfect
  • grammar: excellent

Author:  Boris Pioline  on 2020-03-20  [id 772]

(in reply to Report 1 on 2020-03-04)
Category:
answer to question

We are grateful to the referee for taking up the daunting task of reviewing this admittedly arduous paper, for his/her kind remarks and for numerous suggestions for improving the manuscript. We understand the subsection on Fourier coefficients is very dense, but we have not found a better way to present it so we have kept its original form. We have implemented most of the referee's suggestions, but since they are minor, it is probably not useful to list them all.

One more significant change is that we have spelled out the definition of the Hecke operator above Eq. (2.19), and corrected a misprint in its action on the Fourier modes in the line below 2.24 (a factor of $d^{-1}$ was missing). We have also dropped the comment below eq 2.30, which was incorrect as stated and unnecessary.

To answer one of the referee's query, $J_3$ below eq 5.2 denotes the projection of the angular momentum along the z-axis in the rest frame of the particle, i.e. the helicity. We did not think that it was necessary to spell it out.

---

## Editorial Decision

resubmitted